# Divide and Learn: Multi-Objective Combinatorial Optimization at Scale

Esha Singh [1]  Dongxia Wu [2]  Chien-Yi Yang [1]  Tajana Rosing [1]  Rose Yu [1 3]  Yi-An Ma [1 3]

## Abstract

Multi-objective combinatorial optimization seeks Pareto-optimal solutions over exponentially large discrete spaces, yet existing methods sacrifice generality, scalability, or theoretical guarantees. We reformulate it as an online learning problem over a decomposed decision space, solving position-wise bandit subproblems via adaptive expert-guided sequential construction. This formulation admits sublinear regret bounds depending on subproblem dimensionality rather than combinatorial action space size, an exponential-to-polynomial reduction over standard combinatorial bandits. On standard benchmarks, our method achieves 80–98% of specialized solvers' performance while achieving two to three orders of magnitude improvement in sample and computational efficiency over Bayesian optimization methods. On real-world hardware-software co-design for AI accelerators with expensive simulations, we outperform competing methods under fixed evaluation budgets. The advantage grows with problem scale and objective count, establishing bandit optimization over decomposed decision spaces as a principled alternative to surrogate modeling or offline training for multi-objective optimization.

## 1. Introduction

Multi-objective combinatorial optimization (MOCO) requires balancing multiple competing objectives over combinatorial spaces. Oftentimes, each evaluation can cost hours of simulation, hence exhaustive search becomes impossible and gradient descent inapplicable. Yet this is precisely the setting practitioners face in hardware-software co-design, protein engineering, and drug discovery (Yang et al., 2025;

Krishnan et al., 2023; Romero et al., 2013; Brown et al., 2019): approximating Pareto-optimal (i.e., solutions where no objective can improve without degrading another) trade-offs over discrete or mixed-integer variables, with no closed-form gradients and no cheap evaluations.

Existing approaches to MOCO include surrogate-based methods such as Multi-objective Bayesian optimization which has advanced significantly for continuous domains (Couckuyt et al., 2014; Zhao et al., 2019; Knowles, 2006) but struggle in combinatorial settings. Discrete spaces introduce distinct challenges: search spaces grow exponentially ($|\mathcal{X}| = O(k^n)$ for $n$ decision variables with $k$ values each) (Baptista & Poloczek, 2018; Oh et al., 2019), smooth structure is absent for surrogate modeling (Garrido-Merchán & Hernández-Lobato, 2020), and correlation patterns between configurations are difficult to capture with standard kernels (Saves et al., 2023). Classical evolutionary methods (Deb et al., 2002; Zhang & Li, 2007) and, more recently, neural methods that learn search policies via deep reinforcement learning (Bello et al., 2016; Kool et al., 2019) have emerged. Multi-objective extensions (Lin et al., 2022) achieve strong generalization across problem instances but demand millions of training samples and substantial computational overhead. Any practical algorithm must therefore balance three competing demands: (i) computational scalability as the search space grows, (ii) sample efficiency under expensive evaluation budgets, and (iii) tractable per-iteration complexity. Achieving this balance remains hard: surrogate methods minimize evaluations but incur cubic model overhead, evolutionary heuristics offer generality but lack convergence guarantees, and problem-specific heuristics (Helsgaun, 2000; Tinós et al., 2018) exploit structure effectively but do not generalize across domains. No existing approach simultaneously offers domain generality, sample efficiency under expensive evaluations, and formal convergence guarantees.

We propose an alternative framing: MOCO as a sequential decision problem under uncertainty, where feedback from each evaluation informs subsequent search. This formulation requires no offline training, learns directly from expensive evaluations, and admits formal regret guarantees. The key insight is that tractable structure can be imposed on exponential search spaces, enabling principled exploration where exhaustive search is infeasible.

[1]Department of Computer Science and Engineering, University of California, San Diego, La Jolla, CA 92093, USA [2]Stanford University, Stanford, CA 94305, USA [3]Halıcıoğlu Data Science Institute, University of California, San Diego. Correspondence to: Esha Singh <e3singh@ucsd.edu>.

*Proceedings of the 43rd International Conference on Machine Learning*, Seoul, South Korea. PMLR 306, 2026. Copyright 2026 by the author(s).

**Contributions**  We make the following contributions

- We propose DIVIDE & LEARN (D&L), a novel *position-wise bandits with multiple experts* framework, for online multi-objective combinatorial optimization, that transforms intractable exponential search into efficient sequential decision-making by exploiting problem structure and adapting to observed rewards without offline training or surrogate models.

- We establish regret bounds of $O(d\sqrt{T \log T})$ that depend only on subproblem size $d \ll n$ (with $n$ decision variables and $T$ iterations), rather than on the size of the combinatorial action space $|\mathcal{X}| \in \{2^n, n!\}$, yielding an exponential-to-polynomial reduction over standard combinatorial bandit approaches.

- We validate on standard MOCO benchmarks with search spaces up to $10^{60}$ configurations and a real-world hardware-software co-design problem with extreme evaluation costs. D&L attains 80–98% of specialized solvers' performance, matches offline-trained neural solvers without their training overhead, and uses 90% less compute than Bayesian optimization, while yielding more diverse Pareto fronts. These advantages grow at larger scales and objective counts.

## 2. Related Work

**Surrogates, Evolutionary, and Neural Methods.** Multi-objective Bayesian optimization uses Gaussian Process (GP) surrogates and scalarizations such as ParEGO (Knowles, 2006) or EHVI (Daulton et al., 2021), but scales cubically in evaluations and requires intractable acquisition optimization over combinatorial domains. Evolutionary approaches such as MOEA/D (Zhang & Li, 2007) decompose objectives via weight vectors and rely on heuristic operators without finite-time guarantees in black-box combinatorial settings. Neural combinatorial optimization methods amortize inference through offline training (Kool et al., 2019; Lin et al., 2022; Chen et al., 2023) but require large training corpora and may degrade under distribution shift.

**Combinatorial Bandits.**  Combinatorial bandit methods (Chen et al., 2013; Kveton et al., 2015) attain sublinear regret under *semi-bandit feedback*, observing per-component rewards rather than aggregate solution quality. Linear bandits (Dani et al., 2008) handle full-bandit feedback but impose a linear reward structure, and multi-objective (MO) extensions (Öner et al., 2018) inherit both restrictions. These assumptions fail in black-box MOCO, where evaluating a solution yields only aggregate objective values and rewards may depend non-linearly on global interactions.

**Online learning for black-box MOCO.**  The problem class is classical multi-objective combinatorial optimization: structured discrete decision spaces (routing, configuration selection, subset selection) with vector-valued objectives inducing a Pareto front. What changes is the information setting. Each query reveals only the aggregate per-objective values (one scalar per objective), with no per-component decomposition, produced by simulators, hardware measurements, or other expensive black-box evaluators (e.g., hardware/software co-design via circuit-level simulators, as in our HW-SF experiments § 6.4). This setting faces exponential action spaces, non-differentiable objectives, and costly evaluations, precluding enumeration, gradient-based optimization, and exhaustive sampling.

Any sample-efficient algorithm must therefore select queries adaptively based on past observations, formalized as the explore–exploit structure of online learning. Yet existing combinatorial and MO bandit formulations rely on semi-bandit feedback, linear reward structure, or an offline approximation oracle for the inner optimization, enabling tighter estimators but limiting applicability here. To the best of our knowledge, there are no regret-theoretic results for black-box MOCO under full-bandit (per-objective aggregate) feedback. D&L bridges this gap by decomposing combinatorial decisions into coordinated local components learned online from aggregate feedback, achieving finite-time regret guarantees with lightweight coordination.

## 3. Preliminaries

**Problem Statement.** We study multi-objective optimization over combinatorial domains:

$$\min_{x \in \mathcal{X}} \boldsymbol{f}(x) = [f_1(x), \ldots, f_m(x)]^\top \tag{1}$$

where each objective $f_i : \mathcal{X} \to \mathbb{R}$ is a black-box, expensive-to-evaluate, non-convex, and possibly non-smooth function. The domain $\mathcal{X}$ is a combinatorial or mixed-discrete space comprising permutations, binary vectors, integer or categorical assignments, or combinations thereof (see Appendix C.1.1). Consequently, the search space scales as $|\mathcal{X}| = O(n!)$ or $O(2^n)$ depending on problem structure.

**Pareto Optimality.** A solution $x \in \mathcal{X}$ is *Pareto optimal* if no $x' \in \mathcal{X}$ satisfies $f_i(x') \leq f_i(x)$ for all $i \in \{1, \ldots, m\}$ with at least one strict inequality. The Pareto set $\mathcal{P}^* \subset \mathcal{X}$ comprises all Pareto optimal solutions: $\mathcal{P}^* = \{x \in \mathcal{X} : \nexists x' \in \mathcal{X}, \ \boldsymbol{f}(x') \preceq \boldsymbol{f}(x) \wedge \boldsymbol{f}(x') \neq \boldsymbol{f}(x)\}$ where $\preceq$ denotes component-wise inequality (Pareto dominance); image $\boldsymbol{f}(\mathcal{P}^*)$ in objective space is the *Pareto front*.

**Acquisition Functions.** In sequential optimization, acquisition functions balance exploration and exploitation. Upper Confidence Bound (UCB) (Auer et al., 2002a) selects actions $a$ maximizing $\hat{V}(a) + c\sqrt{\log t / N(a)}$—estimated value plus an uncertainty bonus that shrinks with visits.

D&L adapts this principle to combinatorial domains via position-wise bandits (Section 4.2.2).

# 4. The D&L Algorithm

## 4.1. Online Learning Formulation

We reformulate multiobjective combinatorial optimization as an online learning problem over $T$ iterations (Cesa-Bianchi & Lugosi, 2006). At each iteration $t$, the learner selects a solution $x_t \in \mathcal{X}$ and observes a scalar reward $r_t(x_t) = \phi(\mathbf{f}(x_t)) + \epsilon_t$, where $\mathbf{f}(x) = (f_1(x), \ldots, f_m(x))$ is the multi-objective vector, $\phi(\cdot)$ is a scalarization [1] (e.g., weighted sum), and $\epsilon_t$ is stochastic noise (Ehrgott, 2005). Furthermore, the reward function $r_t(\cdot)$ is non-convex and observed with bounded, zero-mean stochastic noise, i.e., $\mathbb{E}\left[\epsilon_t \mid x_1, r_1, \ldots, x_{t-1}, r_{t-1}\right] = 0$ and $|\epsilon_t| \leq \sigma$ almost surely. While D&L is empirically robust to non-stationarity, our analysis focuses on this stochastic setting without convexity or linearity assumptions. Crucially, unlike standard multi-armed or semi-bandit settings, the learner observes only a single scalar reward for the entire selected solution—*full-bandit feedback* (Combes et al., 2015)—not per-position or per-arm observations. This is among the most challenging feedback models for combinatorial optimization.

A solution $x = (x_1, \ldots, x_n)$ assigns action $x_i \in \mathcal{A}_i$ to each position $i \in [n]$. Each position is treated as a multi-armed bandit with arm set $\mathcal{A}_i$. The exponential cardinality $|\mathcal{X}| = O(2^n)$ or $O(n!)$ combined with bandit feedback renders direct application of online learning algorithms intractable. Our objective is to minimize cumulative regret against the best fixed solution in hindsight for the induced scalar reward:

$$R(T) = \sum_{t=1}^{T} r_t(x^*) - \sum_{t=1}^{T} r_t(x_t), \quad x^* = \arg\max_{x \in \mathcal{X}} \mathbb{E}[r_t(x)]$$

To approximate the Pareto front, we execute D&L with multiple scalarizations; under finite evaluation budgets, each run yields a regret-bounded approximation. To enable tractable regret analysis, we impose the following weak regularity conditions on the problem. We require the following (formal statements in Appendix E.2):

**A.1 (Decomposability)** $\mathcal{X}$ decomposes into $K$ overlapping subproblems $\{\mathcal{X}^k\}_{k=1}^{K}$ with overlap set $\mathcal{O}$.

**A.2 (Discrete Lipschitz / Bounded Differences)** $|f_j(x) - f_j(x')| \leq L \cdot \delta(x, x')$ for problem-specific metric $\delta$. This bounds global sensitivity under arbitrary solution perturbations.

**A.3 (Bounded Coupling)** Modifying subproblem $k$ induces

---

**Algorithm 1** D&L: Position-wise Bandits with Multiple Experts

**Require:** Size $n$, objectives $\mathbf{f} : \mathcal{X} \to \mathbb{R}^m$, subproblems $\mathcal{S} = \{S_1, \ldots, S_K\}$, weights $\mathcal{W}$, iters $T$, $\hat{V}$: values, $N$: counts, $W$: EXP3 weights, $\tilde{L}$: FTRL losses
**Output:** Pareto archive $\mathcal{P}$
1: **for** $w \in \mathcal{W}$ **do**  ▷ Each scalarization
2:    $\hat{V}, N, W, \tilde{L} \leftarrow \mathbf{0}_{n \times |\mathcal{A}|}$; $\lambda_0 \leftarrow \mathbf{0}_n$; $r^* \leftarrow -\infty$
3:    $x_{0,i} \leftarrow \text{SELECTACTION}(i) \ \forall i \in [n]$  ▷ ALG. 2: Expert-guided init
4:    **for** $t = 1, \ldots, T$ **do**
5:       $x_t \leftarrow x_{t-1}$  ▷ Carry over current iterate
6:       **for** $k = 1, \ldots, K$ **do** $x_t \leftarrow \text{LOCALREFINE}(x_t, S_k)$  ▷ ALG. 3
7:       **end for**
8:       $r_t \leftarrow \sum_i w_i f_i(x_t)$  ▷ Scalarized reward
9:       $\text{UPDATEEXPERTS}(x_t, r_t, \hat{V}, N, W, \tilde{L})$  ▷ ALG. 5
10:      $\lambda_t \leftarrow \text{DUALUPDATE}(\lambda_{t-1}, x_t, \mathcal{S})$  ▷ ALG. 4
11:      **if** $r_t > r^*$ **then** $x^* \leftarrow x_t$; $r^* \leftarrow r_t$  ▷ Track best
12:      **end if**
13:    **end for**
14:    $\mathcal{P} \leftarrow \mathcal{P} \cup \{(x^*, \mathbf{f}(x^*))\}$  ▷ Add to Pareto archive
15: **end for**

---

bounded non-local effects: $|f_j(x') - f_j(x)| \leq L \cdot \delta(x, x') + C \cdot |S_k \cap \mathcal{O}|$, where $S_k \subseteq [n]$ is index set of subproblem $k$. This only requires that cross-subproblem interactions scale linearly with overlap, not additive decomposability of $f$.

**A.4 (Bounded Range)** Objectives satisfy $f_j(x) \in [0, B]$ for all $x \in \mathcal{X}$ and $j \in [m]$.

Given these conditions, we now describe how D&L achieves sublinear regret through problem decomposition and an ensemble of no-regret algorithms (Freund & Schapire, 1997; Arora et al., 2012). At each position, one algorithm—termed an *expert*—is stochastically selected to propose an action based on learned action-quality estimates, enabling heterogeneous exploration across decision coordinates. All experts update from every observed reward, decoupling strategy selection from reward assumptions. The selection distribution adapts to local estimation uncertainty, favoring exploration where confidence is low and exploitation as estimates converge.

Since maintaining statistics over $|\mathcal{X}|$ is intractable, experts share parameters at the position-action level, requiring $O(n \cdot |\mathcal{A}|)$ memory. The shared parameters are the minimal statistics each expert requires (Section 4.2.2). When a reward is observed, all position-action pairs in the selected solution update simultaneously. Combined with subproblem decomposition (Section 4.2.1), this enables information transfer: observations in one subproblem improve decisions wherever the same variables appear.

## 4.2. Algorithm Overview

D&L addresses the intractability of combinatorial bandits through three coordinated mechanisms, see Algorithm 1, Figure 1. First, we *decompose* the solution space into $K$ overlapping subproblems, each involving $O(n/K)$ variables, reducing effective action space from $O(2^n)$ to $O(K \cdot 2^{n/K})$. *Lagrangian relaxation* (DUALUPDATE) reconciles overlaps through dual multipliers, ensuring

---

[1]Scalarization defines the reward signal $r_t = \phi(\mathbf{f}(x_t))$, not the optimization procedure, D&L is scalarization-agnostic. See B.1

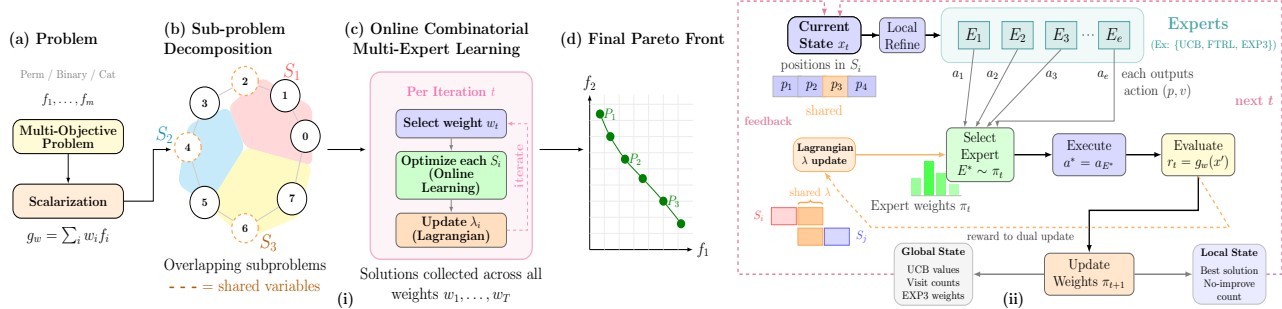

*Figure 1.* **Overview of D&L.** (i) Any multi-objective problem is scalarized & decomposed into overlapping subproblems with shared variables (dashed nodes in (b)). Bandit-based action selection optimizes each subproblem independently; Lagrangian duality enforces cross-subproblem consistency (ii) **Online Combinatorial Multi-expert learning:** For each position in a subproblem, an expert (this work: UCB, FTRL, EXP3 or TS, EXP3, see Section 4.2.2) is sampled via the mixture distribution $\pi_t$ to propose an action. All experts update shared position-action statistics, amortizing exploration across exponential solution spaces. Lagrangian multipliers coordinate overlaps.

cross-subproblem consistency without destroying separability. Second, *multi-expert learning* (SELECTACTION) proposes actions at each position by sampling from experts sharing global statistics; LOCALREFINE captures within-subproblem interactions via zeroth-order perturbations. Third, all experts update statistics (UPDATEEXPERTS) from observed reward $r^{(t)}$, decoupling selection from learning. We detail each component in the following subsections.

### 4.2.1. DECOMPOSITION STRATEGY

The exponential action space makes maintaining per-solution statistics intractable. This is the fundamental bottleneck for bandit algorithms in combinatorial domains without finite arm assumptions. Our key insight is that while the solution space is exponential, the *decision structure* is factored: a solution comprises $n$ position-action assignments optimizable in coordinated groups. We therefore decompose in ***decision-space***, partitioning variables rather than objectives. Our decomposition maintains constant subproblem size ($d$) as $n$ grows while preserving regret bounds, combining scalability with theoretical grounding.

We partition the $n$ variables into $K$ overlapping subproblems $\{S_1, \ldots, S_K\}$, each involving $d = O(n/K)$ variables. This achieves two reductions: (1) per-subproblem action space shrinks from $2^n$ to $2^d$, and (2) effective dimensionality reduces from $n$ to $d$, improving sample efficiency. Overlap regions $\mathcal{O}_{pq} = \mathcal{S}_p \cap \mathcal{S}_q$ enable information flow across partitions. However, overlapping variables must satisfy consistency constraints $x_i^{(p)} = x_i^{(q)}$ for all $i \in \mathcal{O}_{pq}$. Without coordination, independent subproblem optimization yields conflicting assignments.

**Definition 4.1** (**Metric-Based Decomposition**). Given a problem-specific distance $D : [n] \times [n] \to \mathbb{R}_+$, we construct subproblems via two strategies.

The first is a *sliding window*, $S_k = \{(k-1)(s - o_t) + 1, \ldots, \min(n, (k-1)(s - o_t) + s)\}$, with diminishing over-

lap $o_t = \lfloor o_0 \cdot t^{-\alpha} \rfloor$. The second is a *k-nearest neighbors*[2] construction, $S_c = \arg\min_{|S|=s} \sum_{j \in S} D_{c,j}$. When overlap is time-varying, $S_k^{(t)}$ denotes the subproblem at iteration $t$. See Figure 26 and Appendices E.18 and E.7.

*Remark* 4.2 (**Domain Agnosticity**). The decomposition operates on indices $[n]$, not solution values and the same construction applies to permutations (using Kendall tau distance Kendall (1938)), binary vectors (Hamming distance), and other combinatorial domains.

**Proposition 4.3** (**Bounded Coupling**). *The metric-based decomposition (Definition 4.1) satisfies Assumption A.3. Let $\rho := \max_i |\{k : i \in S_k\}|$ denote the maximum overlap multiplicity. For any modification on subproblem $S_k$:*

$$|f_j(x') - f_j(x)| \le L \cdot \Delta_s + C \cdot (\rho - 1) \cdot s$$

*where $\Delta_s := \max_{x,x' \in \mathcal{X}^k} \delta(x, x')$ denotes the diameter of a subproblem of size $s$ under the problem-specific metric, and $\rho$ is the maximum overlap multiplicity.*

Proposition 4.3 establishes the foundation for regret decomposition by showing that subproblem-level optimization induces bounded objective variation.

**The Conflict Problem** The overlap regions $\mathcal{O}_{pq}$ in Definition 4.1 are intentional: shared variables enable learning to transfer across partitions. However, early in learning, different subproblems may favor incompatible actions for shared positions, producing inconsistent solutions whose rewards cannot be reliably attributed and whose errors can cascade across the partition structure. We control these effects using two complementary mechanisms. *Diminishing overlap* ($o_t = \lfloor o_0 t^{-\alpha} \rfloor$, $\alpha = 0.5$) progressively shrinks the shared region, guaranteeing $\sum_t o_t = O(\sqrt{T})$ (Proof E.28); this limits where conflicts can occur but does not prevent

---

[2]$D$ encodes problem locality: geographic distance (TSP), value-weight similarity (Knapsack). Grouping nearby indices reduces cross-subproblem dependencies.

them. *Lagrangian relaxation* is adaptive: it penalizes disagreement precisely at positions where conflicts arise and vanishes as estimates converge. Neither mechanism alone suffices. Lagrangian penalties over large overlaps can incur linear cost, while diminishing overlap without active resolution leaves conflicts unresolved (see D.7).

**Coordinating Overlapping Subproblems via Lagrangian Relaxation** To resolve conflicts without destroying separability we relax the hard consistency constraints $x_i^{(p)} = x_i^{(q)}$ into soft penalties. We maintain dual variables $\lambda_i \geq 0$ for each overlapping position $i \in \mathcal{O}$ & penalize a *soft violation measure* $\xi_i^{(t)}$ that quantifies coordination risk with shared value estimates $\hat{V}$, visit counts $N$:

$$\mathcal{L}(x, \lambda) = r_t(x) - \sum_{i \in \mathcal{O}} \lambda_i \cdot \xi_i^{(t)}$$

$$\xi_i^{(t)} = (k_i - 1) \cdot \mathrm{Var}(\hat{V}_{i,\cdot}^{(t)}) \cdot \left(1 - \frac{N_{i,x_i^{(t)}}}{\sum_a N_{i,a}}\right)$$

where $k_i$ counts subproblems containing position $i$, the variance captures disagreement potential, and the visit ratio reflects assignment confidence. Three limiting cases confirm the intuition. When $k_i = 1$ (the position sits in a single subproblem), $\xi_i^{(t)} = 0$, since no coordination is needed. When value estimates have converged ($\mathrm{Var}(\hat{V}_{i,\cdot}^{(t)}) \to 0$), penalties vanish. When the played action is well-explored (visit ratio approaches one), confidence is high and risk is low. Crucially, $\xi_i^{(t)}$ upper-bounds the expected disagreement probability across overlapping subproblems (Lemma E.38), permitting Lagrangian relaxation when hard violations are unobservable under full-bandit feedback. For fixed $\lambda$, this decomposes across subproblems, preserving separability. We update $\lambda_i$ via accelerated mirror descent (Beck & Teboulle, 2003) on the resulting convex dual objective, with step size $\alpha_t = O(1/\sqrt{t})$. As learning progresses, $\mathrm{Var}(\hat{V}_{i,\cdot}) \to 0$ implies $\xi_i^{(t)} \to 0$, naturally diminishing penalties (see § E.6.3).

**Theorem 4.4** (Coupling Error Bound). *Under Assumptions A.1–A.4 and Lagrangian coordination with accelerated mirror descent, expected cumulative coupling error $C_t$ satisfies:*

$$\mathbb{E}\left[\sum_{t=1}^T C_t\right] = O(K^2 Q R_{\max} \sqrt{T})$$

*where $K$ is the number of subproblems, $Q = \max_{p,q} |\mathcal{O}_{pq}|$ is maximum overlap size, and $R_{\max}$ is maximum per-position regret. See Theorem E.13 for the full proof.*

This $O(\sqrt{T})$ rate ensures coordination costs diminish sublinearly—the **"price of coordination."** The bound reveals a tradeoff: larger $K$ reduces per-subproblem regret but increases coordination overhead ($\tilde{O}(K)$ under sparse local overlap). Balancing yields an asymptotically optimal decomposition $K^* = O(\sqrt{n/(QR_{\max})})$. With coordination established, we now turn to how each subproblem is solved.

### 4.2.2. MULTI-EXPERT LEARNING FOR SUBPROBLEMS

We employ a *multi-expert* strategy instantiating three no-regret algorithms with complementary strengths. The first expert (UCB) provides optimistic action selection, balancing exploration and exploitation under stochastic rewards. The second expert (EXP3) is adversarially robust and addresses the core challenge of bandit feedback: estimating rewards for unchosen actions via importance weighting by inverse selection probabilities (Auer et al., 2002b), which can suffer high variance when probabilities are small (Bubeck & Cesa-Bianchi, 2012). Under full-bandit feedback, we use a clipped importance-weighted estimator for each position–action pair $(i, a)$ with shared denominator $p_{i,a}(t) = W_{i,a}/\|W_i\|_1$. Clipping controls variance with bounded cumulative bias $O(dB \log d/\eta)$ that diminishes as probability concentrates on optimal actions, yielding $O(d\sqrt{T \log T})$ regret (Corollary 5.3). The unbiased alternative uses the mixture-marginal denominator $\pi_{t,i}(a) = \sum_e p_e^{(t,i)} \pi_{t,i}^{(e)}(a)$ at the cost of additional per-expert tracking. Both choices yield equivalent rates (§ E.6.4). In our setting, EXP3 operates at the position-action level, correcting for the probability of selecting each action under the expert mixture. The third expert (FTRL) (Hazan, 2016a) optimizes a distribution over actions on the probability simplex $\Delta_{|\mathcal{A}|}$, where expected loss becomes linear, using cumulative clipped importance-weighted losses with confidence-bonus regularization for variance reduction and stability. (see Fig 1-ii). Thus, at each position, one expert is sampled via mixture weights $\rho$ to select actions, but *all experts update parameters* after observing rewards, ensuring robustness across reward structures. Unlike standard expert advice, selection adapts to local estimation uncertainty rather than tracking expert performance (Ablation D.3). Parameters $\hat{V}, N, W, \tilde{L}$ indexed by position–action pairs $(i, a)$ are shared globally across subproblems, enabling information transfer through overlapping positions.

**Expert Selection and Coordination** Algorithm 2 details the selection procedure. The dual variables $\lambda_i$ from Lagrangian coordination track coordination uncertainty at overlapping positions without altering action preferences within a position. As estimates converge, conflicts naturally diminish (Theorem 4.4), coupling bandit learning with constraint satisfaction: experts[3] maximize reward while respecting cross-subproblem consistency.

**Local Refinement and Guarantees.** Expert-driven selection operates at the position level, constructing solutions from per-position statistics. However, this coarse granularity cannot capture fine-grained interactions within subprob-

---

[3]Note, the multi-expert set $\{\mathrm{UCB}, \mathrm{EXP3}, \mathrm{FTRL}\}$ is flexible—any no-regret algorithms with complementary strengths may be substituted.

lems. Between expert-driven phases (every $c$ iterations), *gradient-free local search* refines solutions via zeroth-order perturbations (Flaxman et al., 2005) within subproblems, exploiting local structure that per-position statistics miss. Critically, all expert parameters update at every iteration, including during local search, ensuring no samples are wasted. See proof in Lemma E.4 and Algorithm 3 for details.

---

**Algorithm 2** SELECTACTION: Multi-Expert Action Selection

---

**Require:** Position $i$, actions $\mathcal{A}_i$, Global:$(\hat{V}, N, W, \tilde{L})$; HP: $(\rho_0, c, \gamma)$
**Output:** Selected action $a \in \mathcal{A}_i$
1: $u_i \leftarrow \left(1 + \log(1 + \frac{1}{|\mathcal{A}_i|} \sum_{a'} N_{i,a'})\right)^{-1}$ ▷ $\in (0,1]$; decays slowly with visits
2: $\rho \leftarrow [(1-\rho_0) \cdot u_i/2, \ (1-\rho_0)(1-u_i/2), \ \rho_0]; \ \ e \sim \rho, e \in [3]$ ▷ Uncertainty-adaptive
3: **if** $e = 1$ **then** ▷ **Expert 1**: UCB-based selection
4: $\quad a \leftarrow \arg\max_{a' \in \mathcal{A}_i} [\hat{V}_{i,a'} + c\sqrt{\log t/N_{i,a'}}]$
5: **else if** $e = 2$ **then** ▷ **Expert 2**: EXP3-based selection
6: $\quad p_{a'} \leftarrow W_{i,a'}/\|W_i\|_1; \quad a \sim$ Categorical($p$)
7: **else** ▷ **Expert 3**: FTRL-based selection
8: $\quad a \leftarrow \arg\max_{a' \in \mathcal{A}_i} [-\tilde{L}_{i,a'} + \gamma\sqrt{N_{i,a'}}]$
9: **end if**

---

## 5. Regret Bounds

We establish regret guarantees for D&L through a novel decomposition that separates subproblem learning from coordination costs.

**Definition 5.1** (**Decomposed Regret**). Under Assumption A.3 (Bounded Coupling), surrogate subproblem objectives $\{g_k\}_{k=1}^K$ can be constructed (induced by the decomposition) such that the instantaneous regret $\ell_t := r_t(x^*) - r_t(x_t)$ admits the upper bound:

$$\ell_t \ \leq \ \underbrace{\sum_{k=1}^K \left[g_k((x^*)^{(k)}) - g_k(x_t^{(k)})\right]}_{\text{Subproblem Regret } S_t} + \underbrace{C_t}_{\text{Coupling Error}} \quad (2)$$

where $(x^*)^{(k)}$ and $x_t^{(k)}$ denote the restriction of $x^*$ and $x_t$ to subproblem $k$, and the coupling error $C_t$ captures non-local interactions across overlapping positions $\mathcal{O}$.

The surrogates $g_k$ are constructed by fixing variables outside subproblem $k$ to $x^*$; we do *not* assume $f$ decomposes additively. Assumption A.3 guarantees that modifying subproblem $k$ affects $f$ by at most $L \cdot \delta(x, x') + C \cdot |S_k \cap \mathcal{O}|$, which suffices for (2) and separates subproblem regret from coupling error. Exhaustive definition in E.6.

**Theorem 5.2** (**Regret Decomposition**). *Let Assumptions A.1–A.4 hold. Under the decomposed regret structure of Definition 5.1, the average regret of* D&L *satisfies:*

$$R_{\text{avg}}(T) \leq R_{\text{sub}}^{\text{avg}}(T) + R_{\text{overlap}}^{\text{avg}}(T) + R_{\text{local}}^{\text{avg}}(T) + C, \quad (3)$$

*where $R_{\text{sub}}^{\text{avg}}(T)$ measures the online-learning regret accumulated within each subproblem by the expert ensemble,*

*and admits the bound,*

$$R_{\text{sub}}^{\text{avg}}(T) \ \leq \ \frac{1}{K}\sum_{k=1}^K \sum_{e=1}^E p_e \, R_{\text{alg}}^{(k,e)}(T)$$

*i.e., the regret $R_{\text{alg}}^{(k,e)}(T)$ of each expert $e$ on subproblem $k$, weighted by its selection probability $p_e$ and averaged across the $K$ subproblems. The term $R_{\text{overlap}}^{\text{avg}}(T)$ captures the averaged Lagrangian coordination cost (Theorem 4.4), and $R_{\text{local}}^{\text{avg}}(T)$ accounts for gradient-free local refinement. The normalization factor $1/K$ defines the average regret, $R_{\text{avg}}(T) := \frac{1}{K}R_{\text{total}}(T)$, and isolates per-subproblem learning.*

*Proof Sketch.* By Definition 5.1, the instantaneous regret decomposes as $\ell_t \leq S_t + C_t$, the subproblem regret $S_t$ and the coupling error $C_t$. Summing over $T$ rounds and applying the $1/K$ normalization, we bound each separately. For $S_t$, each subproblem contributes two sources of error: (i) the expert regret accumulated within each subproblem, and, (ii) the residual from gradient-free local refinement. Assumption A.3 permits constructing surrogate objectives $g_k$ without requiring additive decomposition of $f$ (see § E.7), and each subproblem $k$ is updated over $T_k$ rounds with $\sum_k T_k = T$. For (i), the expert term inherits each base expert's no-regret guarantee; combining via the selection probabilities $p_e$ and applying Cauchy–Schwarz ($\sum_k \sqrt{T_k} \leq \sqrt{KT}$) together with the $1/K$ normalization controls the expert regret. For (ii), the local-refinement contributes $O(LD\sqrt{dT/K})$ via standard zeroth-order analysis (§ E.5). The coupling error $C_t$ is bounded by Theorem 4.4, giving $O(KQR_{\max}\sqrt{T})$. Full proof deferred to § E.4.

**Corollary 5.3** (**Explicit Regret Bound**). *Let* D&L *use experts $\{\text{UCB}, \text{EXP3}, \text{FTRL}\}$ with mixture weights $(p_1, p_2, p_3)$. With $K$ subproblems of size $d = n/K$, overlap size $Q$, Lipschitz constant $L$, and domain diameter $\Delta$, the expected average regret satisfies:*

$$\mathbb{E}[R_{\text{avg}}(T)] \ \leq \ \underbrace{O\left(d\sqrt{T\log T}\right)}_{\text{Subproblem learning}} + \underbrace{O\left(KQR_{\max}\sqrt{T}\right)}_{\text{Overlap coordination}}$$
$$+ \underbrace{O\left(L\Delta\sqrt{dT/K}\right)}_{\text{Local refinement}}.$$

$$(4)$$

*At optimal decomposition $K^\star = O\left(\sqrt{n/(QR_{\max})}\right)$, the bound simplifies to $O\left(\sqrt{nQR_{\max}T\log T}\right)$.*

The subproblem learning term combines contributions from each expert: UCB contributes $O(d\sqrt{T\log T/K})$, EXP3 contributes $O(dB\sqrt{T\log d}/C_{\text{clip}})$, which absorbs the clipped-estimator bias, and FTRL contributes $O(dB\sqrt{T\log T}/(C_{\text{clip}}\sqrt{K}))$. In the worst case across mixture weights, the dominant rate $O(d\sqrt{T\log T})$ therefore

arises from EXP3 (using $\log d \leq \log T$), which benefits less from a $1/K$ averaging gain but provides robustness to adversarial and misspecified rewards. See Theorem E.1 and Corollary E.3 for the per-expert bounds.

**Interpretation.** The bound reveals three key insights (1) *subproblem-independent scaling*, where the dominant term depends on subproblem size $d = n/K$ rather than full problem size $n$, yielding an exponential-to-polynomial reduction over naive combinatorial bandits with $O(\sqrt{T|\mathcal{X}|})$ regret, where $|\mathcal{X}|$ can be as large as $n!$ or $2^n$ (2) the *"price of coordination"* $O(KQ\sqrt{T})$ is sublinear in $T$, so the average coordination cost per round vanishes as $T \to \infty$ (3) the optimal decomposition $K^* = O(\sqrt{n/(QR_{\max})})$ balances subproblem complexity against coordination overhead.

**Comparison to prior work.** Stochastic linear bandits (Dani et al., 2008) achieve $O(n^{3/2}\sqrt{T})$ under full-bandit feedback. Classical combinatorial methods (Chen et al., 2013; Kveton et al., 2015) achieve $O(\sqrt{nkT\log T})$ under semi-bandit feedback ($n$ base arms with $k$ selected per round). In contrast, D&L achieves $O(d\sqrt{T\log T})$ under full-bandit feedback without linearity, where $d = n/K$ is subproblem size. Naive combinatorial bandits incur $O(\sqrt{T|\mathcal{X}|})$ regret where $|\mathcal{X}|$ may be $2^n$ or $n!$. D&L's structured decomposition reduces this to exponential dependence to polynomial in $n$. The multi-expert and coordination mechanisms ensure robustness without inflating the rate (see Appendix A for extended discussion).

# 6. Experiments

## 6.1. Datasets

For task of multi-objective optimization (MOO) (Xue et al., 2024), our evaluation spans two complementary benchmarks differing in structure, scale, and evaluation cost.

**Multi-Objective Combinatorial Optimization (MOCO).** We consider multi-objective variants of classical NP-hard problems: Traveling Salesman Problem (TSP; up to 100 cities, $n! \approx 10^{157}$), Knapsack (up to 200 items, $2^n \approx 10^{60}$) (Zitzler & Thiele, 1999), and Capacitated Vehicle Routing Problem (CVRP; up to 100 customers) Jozefowiez et al. (2008), covering permutation, binary, and constrained combinatorial domains. These benchmarks enable direct comparison with state-of-the-art across diverse Pareto geometries and exponentially scaling search spaces.

**Hardware-Software Co-Design.** This real-world four-objective problem requires expensive simulation of neural network accelerators (Dutta et al., 2022) over a 20-dimensional mixed-integer design space ($\sim (2.2 \times 10^5)$ configurations). Each evaluation incurs substantial computational cost ($O(\text{hours})$) and intrinsic stochasticity from hardware-level variance, presenting a challenging regime

for sample-efficient optimization under noisy, expensive black-box feedback with unknown Pareto geometry.

Together, these benchmarks validate performance across search spaces ranging from $10^5$ to $10^{157}$ configurations, encompassing binary, permutation, and mixed-integer domains with heterogeneous evaluation costs.

## 6.2. Experimental Details

All BO baselines use GP surrogates with Matérn-5/2 kernels and ARD (Rasmussen & Williams, 2006); acquisition functions use Sobol QMC sampling (Sobol, 1967) with 64–128 MC samples. Objectives are normalized to $[0, 1]$ using bounds computed from initial samples, with minimization transformed to maximization orientation for acquisition functions. All methods share identical initialization (50–100 random feasible solutions depending on problem size). Since MOCO evaluations are inexpensive, we tune each baseline individually and report *best-achievable* hypervolume. We report two variants of D&L. Both variants include EXP3 as the second expert. The default D&L pairs it with UCB and FTRL whereas D&L-TS pairs it with Thompson Sampling. Weighted-sum scalarization is used for MOCO benchmarks, Augmented Tchebycheff for HW-SF (Appendix B.1). Full hyperparameters in Appendix C.5. For evaluation, we report the *hypervolume ratio* $\text{HV}'_r(F) = \text{HV}r(F)/\prod i = 1^M |r_i - z_i|$ (Pareto front $F$, reference point $r$, ideal point $z$) ensuring comparability across problem scales and instance sizes. We adopt standardized reference and ideal points, see Table 5. Each method is evaluated over 200 randomly generated problem instances, and we report mean hypervolume ratio with standard error.

## 6.3. Baselines

We compare D&L against baselines spanning five algorithmic paradigms. NSGA-II (Deb et al., 2002) is a heuristic-based evolutionary method employing non-dominated sorting with crowding distance, testing whether D&L can match performance achieved through problem-specific operators. For Bayesian optimization, MOBO-qParEGO (Knowles, 2006) decomposes via random Chebyshev scalarizations, testing whether explicit decomposition outperforms bandit-based weight selection. MOBO-qNEHVI (Daulton et al., 2021) maximizes expected hypervolume improvement, testing whether modeling objective correlations improves sample efficiency. BOPR (Daulton et al., 2022) handles discrete structures through probabilistic reparameterization. As the most directly comparable baseline, we extend BOPR to multi-objective settings to contrast gradient-based acquisition with our approach. PPLS/D-C (Shi et al., 2022) decomposes objective space into sub-regions with archive cooperation, testing parallel local search against learning-based methods. WS-LKH (40 wt.) and WS-DP (101 wt.)

*Table 1.* Condensed results across benchmarks (200 instances each). HV ratio (↑, normalized to $[0, 1]$), number of non-dominated solutions (NDS, ↑), and compute cost in TFLOPs ($\mathcal{F}$, ↓). Methods span five algorithmic families: problem-specific heuristics (**WS**, **PPLS/D-C**), pretrained neural (**PMOCO***, matched compute budget), evolutionary (**NSGA-II**), Bayesian optimization (**Best BO**, best of qNEHVI/qParEGO/PR per instance), and training-free black-box (**D&L**, **D&L-TS**, ours). The dashed vertical line separates problem-specific solvers (left) from general-purpose methods (right). Pretrained PMOCO results and other complete results in Table 11.

| | | WS | | | PPLS/D-C | | | PMOCO* | | | NSGA-II | | | Best BO | | | D&L | | | D&L-TS | | |
|---|---|---|---|---|---|---|---|---|---|---|---|---|---|---|---|---|---|---|---|---|---|---|
| | | HV | NDS | $\mathcal{F}$ | HV | NDS | $\mathcal{F}$ | HV | NDS | $\mathcal{F}$ | HV | NDS | $\mathcal{F}$ | HV | NDS | $\mathcal{F}$ | HV | NDS | $\mathcal{F}$ | HV | NDS | $\mathcal{F}$ |
| Bi-TSP | 20 | **.625** | 21 | **.001** | .626 | 71 | .03 | .360 | 8 | 68 | .573 | 225 | .15 | .536 | 10 | 4.4 | .585 | 45 | .01 | .600 | 102 | .01 |
| | 50 | **.629** | 26 | .06 | .628 | 213 | .07 | .330 | 6 | 734 | .347 | 20 | .15 | .472 | 23 | 51 | .530 | 62 | **.05** | .540 | 147 | .07 |
| | 100 | **.690** | 50 | 1.0 | .630 | 533 | .15 | .309 | 6 | 685 | .294 | 51 | .31 | .104 | 6 | 2462 | .400 | 48 | **.13** | .470 | 114 | .16 |
| Bi-KP | 50 | .356 | 17 | **.003** | **.357** | 25 | .01 | .340 | 40 | 362 | .219 | 68 | .31 | .301 | 28 | 41 | .350 | 45 | .13 | .347 | 71 | .07 |
| | 100 | .440 | 23 | **.02** | **.447** | 49 | .02 | .370 | 34 | 816 | .221 | 24 | .31 | .314 | 7 | 19 | .338 | 62 | .28 | .385 | 56 | .13 |
| | 200 | .360 | 30 | .08 | **.362** | 63 | **.07** | .186 | 2 | 583 | .280 | 13 | .32 | .266 | 13 | 9.9 | .260 | 21 | .39 | .300 | 73 | .22 |
| Tri-TSP | 20 | **.467** | 46 | .008 | .388 | 35 | **.01** | .430 | 40 | 51 | .411 | 200 | 1.5 | .320 | 6 | 1.7 | .430 | 108 | .11 | .420 | 218 | **.02** |
| | 50 | **.429** | 189 | .34 | .262 | 93 | **.04** | .390 | 25 | 236 | .173 | 198 | 1.5 | .062 | 10 | 87 | .262 | 172 | .26 | .282 | 494 | .09 |
| | 100 | **.488** | 200 | 5.2 | .241 | 158 | **.16** | .300 | 40 | 654 | .135 | 200 | 6.1 | .030 | 31 | 1408 | .210 | 223 | .26 | .234 | 310 | .24 |
| Bi-CVRP | 20 | NA | NA | NA | **.480** | 7 | .10 | .360 | 8 | 398 | .430 | 81 | .15 | .352 | 13 | 2.5 | .450 | 22 | **.01** | .460 | 36 | .04 |
| | 50 | NA | NA | NA | **.450** | 21 | .12 | .330 | 6 | 571 | .310 | 9 | 1.0 | .140 | 7 | 14 | .350 | 37 | .12 | .370 | 50 | **.10** |
| | 100 | NA | NA | NA | **.430** | 25 | .14 | .300 | 6 | 2000 | .310 | 14 | 3.5 | .110 | 10 | 17 | .253 | 17 | .31 | .300 | 42 | .19 |

**Bold** = best overall; underline = best among general-purpose methods (excl. WS, PPLS/D-C).

apply weighted-sum scalarization (Miettinen, 1999) with LKH (Jaszkiewicz, 2002) and dynamic programming respectively, providing near-optimal references. PMOCO (Lin et al., 2022) employs preference-conditioned hypernetworks trained via reinforcement learning. Despite the pretraining requirement, we include it as a representative neural method. For fair comparison, we tune each baseline individually and report best achievable hypervolume. Problem-specific operators are employed where applicable (Lacomme et al., 2004; Prins, 2004). See § C.3.

### 6.4. Experimental Results

We evaluate D&L across combinatorial benchmarks (Tables 1 and hardware-software co-design (Table 2), comparing against problem-specific solvers, neural methods, and general-purpose MOCO algorithms.

#### 6.4.1. Multi-Objective Combinatorial Optimization

**Expected performance hierarchy.** Problem-specific solvers (WS-LKH/DP, PPLS/D-C) exploit domain structure through tailored heuristics and establish quality upper bounds. Neural methods (PMOCO) amortize computation through offline training on problem distributions, approaching specialized performance at inference. NSGA-II, while general-purpose, relies on problem-specific crossover and mutation operators for combinatorial search. In contrast, D&L operates as a pure black-box method, requiring only objective evaluations without domain operators or pretraining. Our goal is to approach the quality of specialized

methods under this more restrictive setting.

**Solution quality.** D&L achieves the highest hypervolume among black-box methods across most benchmarks, reaching up to 80–98% of specialized solvers performance while outperforming Bayesian baselines by 20–90%. For fair comparison with neural methods, we match PMOCO* training instances to our evaluation budget. Under this setting, our approach exceeds offline training by an average of $\sim 30\%$ across bi-objective benchmarks, with largest margins on BiTSP ($\sim 70\%$). D&L's advantage grows with scale as MOBO methods hypervolume degrades 70–85% from small to large instances.

**Scaling to higher objectives.** In tri-objective settings, Bayesian methods collapse under the expanded Pareto front dimensionality. D&L's bandit-based decomposition scales gracefully, discovering 2–3× more non-dominated solutions than the strongest baselines with competitive HV.

**Computational efficiency.** To isolate algorithmic efficiency from hardware effects, we report analytical FLOPs in Tables 1; wall-clock times appear in the Appendix. FLOP ratios closely track runtime ratios across components (see C.6, Fig. 2). D&L reduces computation by 90-99% compared to Bayesian baselines, yielding 10–30× wall-clock speedups (see Tables 7-10). This efficiency stems from closed-form acquisition updates that bypass expensive hypervolume computation and GP inference.

#### 6.4.2. Hardware-Software Co-Design

This benchmark presents a contrasting regime: a compact search space but expensive evaluations requiring neural ac-

celerator simulation. We fix the budget to 150 evaluations to test sample efficiency under costly queries. D&L achieves the highest hypervolume, improving ∼22% over baselines on average (Table 2). This demonstrates that bandit-based exploration remains competitive in sample-limited regimes where GP surrogates should excel. Notably, qNEHVI struggles with the four-objective structure, where hypervolume acquisition cost scales exponentially with objective count.

**Ablations.** Bandit-based decomposition scales gracefully on large-scale, many-objective problems where baselines degrade. Appendix D validates key design choices. A progressive ablation isolates each component's contribution (D.4). Overlapping subproblem decomposition outperforms monolithic optimization (D.1), with D&L exceeding specialized solvers on BiTSP-20 (Fig. 3). The multi-experts framework (UCB, EXP3, etc.) outperforms any single expert (D.3). The scalarization choice is non-critical, with common scalarizations within 2% HV on Bi-KP (D.10). D&L extends to single-objective CO with gains growing in $n$ (D.8). Adaptive perturbation improves local search escape and sample diversity analysis confirms broad trajectory coverage (D.5). Thompson sampling variants consistently outperform UCB, reflecting superior exploration-exploitation balance in combinatorial spaces as seen in Tables 1.

*Table 2.* Results (averaged over 10 seeds) on Hardware-Software Co-design (4 objectives).

| Method | Hypervolume |
|---|---|
| NSGA-II | $0.291 \pm 0.040$ |
| MOBO-qNEHVI | $0.287 \pm 0.015$ |
| MOBO-qParEGO | $0.34 \pm 0.012$ |
| **D&L** | $\mathbf{0.36 \pm 0.01}$ |
| **D&L-TS** | $\mathbf{0.372 \pm 0.006}$ |

## 7. Conclusion

We presented D&L[4], a decomposition-based multi-expert framework for combinatorial optimization under full-bandit feedback. The framework is modular: any no-regret experts suffice, and decomposition strategies can be tailored to problem structure. The key requirement is bounded coupling, which holds when problems exhibit locality (geographic proximity for TSP, temporal adjacency for scheduling). Problems with dense global interactions may benefit less. Our ablations reveal a tradeoff between subproblem count $K$ and overlap $Q$. Balancing at $K^* = O(\sqrt{n/(QR_{\max})})$ keeps coordination overhead sublinear. Empirically, D&L matches or exceeds specialized solvers on MOCO benchmarks in 2-10× less time, scales to tri-objective settings where Bayesian methods degrade, and remains competi-

tive on expensive black-box problems, all without problem-specific operators or pretraining. Current limitations include fixed mixture weights $\rho$ (adaptive selection could improve constants) and bounded rewards (sub-Gaussian extensions remain open). The results establish decomposed bandit optimization as a principled, sample-efficient alternative to surrogate modeling and offline training for MOCO.

## Impact Statement

This paper develops improved algorithms for multi-objective combinatorial optimization. Such methods have broad applicability to resource allocation, logistics, and scheduling problems where multiple competing objectives must be balanced. We do not foresee specific negative societal consequences beyond those general to advances in optimization. The method is objective-agnostic, and practitioners deploying it on consequential systems should treat the choice of objectives as part of the deliverable, not a fixed input.

## Acknowledgement

This project is partially supported by the NSF award CCF-2112665 (TILOS). It is also supported in part by the CDC-RFA-FT-23-0069 from the CDC's Center for Forecasting and Outbreak Analytics. Additional support comes from the U.S. Army Research Office under Army-ECASE award W911NF-07-R-0003-03, the U.S. Department of Energy Office of Science, the ARPA-H-SOL-24-101 program, the IARPA HAYSTAC Program, DARPA YFA, and NSF Grants #2205093, #2146343, #2134274, and #2441832.

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

# Appendix

*Table 3.* **Proof Roadmap.** Each main-text result mapped to its full proof with a one-line summary of the key technique. All references are clickable.

| Main-Text Result | Page | Key Idea | Full Proof |
|---|---|---|---|
| *Foundations* | | | |
| Assumptions A.1–A.4 | 3 | Decomposability, discrete Lipschitz, bounded coupling, bounded range | §E.2 |
| Prop. 4.3 (Bounded Coupling) | 4 | Metric decomposition satisfies A.3; change in $f$ under a subproblem-local modification is bounded: $\|f_j(x')-f_j(x)\| \le L\,D(s) + C(\rho-1)s$ | Prop. E.25 in §E.7; supporting Lems. E.20, E.21, E.23, E.24 |
| *Coordination* | | | |
| Thm. 4.4 (Coupling Error) | 5 | Lagrangian mirror descent on soft violation $\xi_i^{(t)}$; cumulative error $O(K^2 Q R_{\max}\sqrt{T})$ | Thm. E.13 in §E.6.3 |
| Lem. E.38 (Disagreement) | 95 | Pr[disagree] controlled by non-concentration $2(1-q_{i,\max}^{(t)})$ of the shared selection distribution | Lem. E.38 in §E.8.4 |
| Mirror descent dual gap (used in Thm. 4.4) | 91 | Entropic mirror descent on concave dual; cumulative gap $O(d\lambda_{\max}L\sqrt{T})$ | Lem. E.37 in §E.8.4 |
| *Per-Expert Regret (discussed in §4.2.2, bounds used in §5)* | | | |
| UCB subproblem regret (§4.2.2) | 65 | Position–value UCB under A.5'; gap-free bound $O(d\sqrt{A_{\max}T_k \log T})$ per subproblem | Lem. E.33, Thm. E.8 |
| EXP3 regret, clipped (§4.2.2) | 75 | Clipped importance weighting with bounded bias $O(dB \log d/\eta)$; per-subproblem rate $O(dB\sqrt{T \log d}/C_{\text{clip}})$ | Lem. E.15, Thm. E.16 |
| FTRL subproblem regret (§4.2.2) | 79 | Probability-simplex lifting makes the loss linear; confidence-bonus FTRL under clipped IS gives $O\big((dB/C)\sqrt{T_k \log T}\big)$ per subproblem | Thm. E.17 |
| Local refinement regret (§4.2.2, Alg. 3) | 62 | Zeroth-order perturbations; $O(LD\sqrt{dT_k})$ via projected OGD with one-point estimator | Lem. E.4 in §E.5 |
| *Main Guarantees* | | | |
| Def. 5.1 (Decomposed Regret) | 6 | Surrogate $g_k$ fixes variables outside $S_k$ to $x^*$; instantaneous regret upper bounded as $r_t \le S_t + C_t$ | Def. E.6 in §E.6 |
| Thm. E.1 (Regret Decomposition) | 57 | Sum $K$ subproblems, Cauchy–Schwarz ($\sum_k \sqrt{T_k} \le \sqrt{KT}$), add coordination + local terms | Thm. E.2 in §E.4 |
| **Cor. 5.3 (Explicit Bound)** | 6 | Combine expert bounds; dominant $O(d\sqrt{T \log T})$ from the bandit experts; optimal $K^* = O(\sqrt{n/(QR_{\max})})$ | Cor. E.3 in §E.4 |
| *Decomposition Validity* | | | |
| Decomposition construction (Def. 4.1, Fig. 26) | 4 | Sliding-window & $k$-NN partitions; coverage and overlap properties | Def. E.18, §E.7 |
| Diminishing overlap (§4.2.1) | 83 | $o_t = \lfloor o_0 t^{-\alpha}\rfloor$; cumulative coupling error $\sum_t C_t = O(\sqrt{T})$ for $\alpha = 0.5$ | Thm. E.28 in §E.7 |

## A. Related Work

**Multi-Objective Bayesian Optimization.** Surrogate-based methods have achieved strong sample efficiency in continuous multi-objective optimization. ParEGO (Knowles, 2006) decomposes via random scalarizations, while EHVI-based methods (Couckuyt et al., 2014; Daulton et al., 2021) directly maximize hypervolume improvement. However, these approaches face fundamental limitations in combinatorial domains: GP surrogates scale cubically in observations and require careful kernel design for discrete inputs (Garrido-Merchán & Hernández-Lobato, 2020), while acquisition optimization over exponential spaces ($|\mathcal{X}| = O(n!)$ or $O(2^n)$) is intractable. BOPR (Daulton et al., 2022) addresses discrete domains through

*Table 4.* **Ablation Guide.** Each ablation study mapped to its experimental validation, key finding, and supporting figures. All references are clickable.

| Ablation Study | Section | Key Finding | Figures & Tables |
|---|---|---|---|
| Decomposition size & overlap | §D.1 | $d \approx n/2$ with ~44% overlap balances quality–efficiency; redundancy $>10\times$ harms runtime | Figs. 3, 4, 5, 6, 7 |
| UCB vs. Thompson Sampling (Expert 1) | §D.2 | Thompson consistently outperforms UCB in HV and Pareto front diversity | Table 1 |
| FTRL necessity (Expert 3) | §D.3.2 | Moderate FTRL rate (30–50%) optimal: lowest cumulative regret and highest stability (mean/std $\approx$9.8, UCB+FTRL) at both scales; lower variance under distribution shift | Figs. 8, 9, 10 |
| EXP3 configuration (Expert 2) | §D.3 | EXP3 ratio $\leq$50% optimal; higher ratios cause premature convergence | Figs. 11, 12; Table 15 |
| Component contribution (progressive) | §D.4 | Marginal contribution of each component, HV increases monotonically per component. | Fig. 13 |
| Sample diversity | §D.5 | $100\times$ faster than NSGA-II with 97% coverage; highest structural diversity | Figs. 14, 15, 16, 17; Tables 17, 18 |
| Hyperparameter sensitivity | §D.6 | HV varies $<$2% across all tested ranges; smooth parameter landscape | Fig. 19 |
| Lagrangian coordination | §D.7 | 95% variance reduction (BiTSP-20); prevents catastrophic failures on conflicting seeds | Figs. 20, 21, 22 |
| Single-objective tasks | §D.8 | D&L framework generalizes beyond MOO | Table 20 |
| Decomposition strategy | §D.9 | Metric-based grouping beats index-based sliding window, gap widening with scale | Figs. 23 |
| Scalarization sensitivity | §D.10 | Robustness across scalarization choices. | Figs. 24, 25 |

probabilistic reparameterization but remains limited by surrogate overhead. In mixed-integer spaces, these methods become computationally prohibitive due to combinatorial explosion.

**Decomposition-based MOCO.** MOEA/D (Zhang & Li, 2007) is the seminal decomposition framework for multi-objective optimization, and superficially resembles our approach: both use weight vectors to scalarize objectives into subproblems. However, the methods differ fundamentally in *what* is decomposed, *how* subproblems are solved, and *whether* guarantees exist. MOEA/D decomposes in **objective space**: each subproblem corresponds to a different weight vector, and solutions are generated via evolutionary operators (crossover and mutation) with implicit coordination through mating restriction among neighboring weights. In contrast, D&L decomposes in **decision space**: each subproblem corresponds to a group of decision variables, and solutions are constructed via bandit-based action selection with explicit Lagrangian coordination. The extensive literature on MOEA/D variants (Qi et al., 2013; Jiang et al., 2011; Cheng et al., 2016; Wang et al., 2016; Trivedi et al., 2017) primarily focuses on adaptive weight vector generation and improved scalarization mechanisms. These approaches introduce effective heuristic refinements that improve empirical performance, but generally do not provide global regret, complexity, or approximation guarantees. Separate theoretical studies analyze the search dynamics and runtime behavior of MOEA/D under restrictive assumptions or simplified settings, such as specific scalarization schemes, variation operators, or objective structures (Huang et al., 2021; Li et al., 2015), but these results constitute *post-hoc* analyses of an existing heuristic framework. In contrast, D&L is *designed* from first principles as an online learning algorithm, yielding $O(d\sqrt{T \log T})$ regret bounds as a direct consequence of its formulation.

**Neural Combinatorial Optimization.** Neural combinatorial optimization (NCO) methods learn solution construction policies, most commonly via policy-gradient reinforcement learning. Attention-based models trained with REINFORCE (Bello et al., 2016; Kool et al., 2019) achieve strong performance on single-objective CO. Recent multi-objective extensions include Pareto set learning with preference-conditioned models (PMOCO) (Lin et al., 2022), diversity-enhanced neural heuristics (NHDE) (Chen et al., 2023), and related neural combinatorial optimization approaches that amortize search across instances (Khalil et al., 2017). While these approaches can amortize inference at test time, they typically require substantial offline training data and computational resources to generalize across instance distributions, and may require retraining or fine-tuning when objectives, constraints, or instance families shift (Angioni et al., 2025). In contrast, D&L requires no

pretraining and adapts online using bandit feedback alone, making it naturally suited to settings with expensive evaluations, limited data budgets, or rapidly varying instance structure.

**Relationship to Reinforcement Learning.** Multi-armed bandits can be viewed as a special case of reinforcement learning: a one-state (stateless) MDP with no state transitions and immediate rewards (Sutton & Barto, 2018). D&L fits this regime—each round selects a complete solution and observes a single scalar reward—so the learning problem is fundamentally *credit assignment without temporal dynamics*. This contrasts with neural CO methods, which typically cast solution construction as a sequential decision process and train a parameterized policy over action prefixes using policy gradients (often REINFORCE with a rollout baseline), i.e., an actor-only or actor-critic-style training loop. Technically, D&L differs from actor-critic RL in three ways: (i) **no value-function learning**: there is no learned critic estimating long-horizon returns because the objective is evaluated on the constructed solution (one-step return); (ii) **no differentiable policy optimization**: actions are selected by index/weight updates (UCB confidence bounds, EXP3/FTRL multiplicative weights) rather than gradients through a neural policy; and (iii) **structured action space exploitation**: D&L decomposes the decision into coupled subproblems and coordinates them via Lagrangian multipliers, analogous in spirit to factorization ideas but without modeling transition dynamics. Thus, D&L is best interpreted as an online learning/bandit formulation for solution construction (with regret guarantees), whereas NCO uses offline RL to amortize a policy across many instances, typically without finite-time regret guarantees.

**Combinatorial Bandits.** Online convex optimization (Hazan, 2016a) and multi-armed bandits (Auer et al., 2002a;b) provide principled frameworks for sequential decision-making under uncertainty. The combinatorial multi-objective MAB (COMO-MAB) (Öner et al., 2018) is most related to our setting, achieving $O(NL^3 \log T)$ Pareto regret. However, COMO-MAB assumes *linear* reward structure (action rewards are linear combinations of base-arm rewards), *semi-bandit* feedback (observing per-arm rewards), finitely many base arms, and stationary i.i.d. rewards—assumptions substantially stronger than ours. In classical stochastic combinatorial MAB methods, CUCB (Chen et al., 2013) and subsequent work (Kveton et al., 2015) achieve gap-free regret bounds on the order of $\tilde{O}(\sqrt{KLT})$; adversarial variants (Cesa-Bianchi & Lugosi, 2012; Audibert et al., 2014) address non-stochastic settings with similar bounds. Both lines require semi-bandit feedback, which is unavailable when only aggregate solution quality is observed. Our formulation goes beyond these settings in three key respects: (i) combinatorial (often exponentially large) action spaces built from finite base arms, (ii) full-bandit feedback rather than semi-bandit, and (iii) nonlinear scalarized rewards rather than linear additive reward models. Recent work on adversarial combinatorial optimization (Ito et al., 2019; Audibert et al., 2014) shows that correlation among losses can significantly impact achievable regret; our decision-space decomposition explicitly mitigates such dependencies by isolating subproblems, yielding regret that scales with subproblem dimension $d$ rather than full problem size $n$.

**Exploration-Exploitation Tradeoffs.** Balancing exploration and exploitation is central to optimization across domains. This tradeoff can be exploited in adaptive optimizers for deep neural networks, to design methods that navigate loss landscapes toward generalizing solutions (Singh et al., 2025). Here, we address the analogous challenge in combinatorial optimization: exploring exponential search spaces efficiently while exploiting promising regions. Our multi-expert framework instantiates this tradeoff through complementary no-regret algorithms, UCB for optimistic exploration (Auer et al., 2002a), Exp3 for adversarial robustness (Auer et al., 2002b), and FTRL for regularized stability (Hazan, 2016a), coordinated via Lagrangian decomposition (Fisher, 1981).

**Multi-expert and algorithm selection.** Bandits have been used for hyper-heuristic operator selection in evolutionary algorithms (Lagos et al., 2023; Fialho et al., 2010), where a bandit chooses among crossover or mutation operators. This differs fundamentally from D&L: we use bandits for *solution construction* (position-wise action selection), not operator selection. The multi-expert framework relates to EXP4 (Auer et al., 2002b) (bandits with expert advice), but EXP4 maintains a meta-layer weighting over experts. D&L instead samples directly from complementary algorithms $\{UCB, EXP3, FTRL\}$ and updates *all* experts after each observation, inheriting the best expert's guarantee without the $O(\sqrt{N})$ cost of $N$ experts. Unlike contextual bandits where contexts are exogenous, D&L exploits *endogenous* structure through overlapping subproblem decompositions, creating a fundamentally different information architecture. This hybrid framework, synthesizing bandit-style learning with structured decomposition enables tractable optimization over exponential spaces while maintaining sublinear regret guarantees (Section E.4).

To our knowledge, no prior work combines: (i) decision-space decomposition with regret guarantees, (ii) multi-expert bandit construction under full-bandit feedback, and (iii) Lagrangian coordination with provable $O(\sqrt{T})$ coupling error. The

superficial similarity to MOEA/D (both use "decomposition") obscures a fundamentally different problem formulation: MOEA/D is an evolutionary heuristic decomposing objectives; D&L is an online learning algorithm decomposing decisions.

## B. Why Weighted-Sum methods are not feasible?

Weighted-sum approaches, while prevalent in multi-objective optimization, suffer from fundamental limitations that restrict their applicability to general combinatorial problems. First, existing weighted-sum methods like WS-LKH and PPLS/D rely on problem-specific heuristics (e.g., Lin-Kernighan for TSP, branch-and-bound for knapsack) that are hand-crafted for particular combinatorial structures. These algorithms achieve state-of-the-art performance precisely because they exploit domain-specific properties for e.g. TSP's geometric structure, knapsack's fractional relaxation, these do not transfer to other problem classes. Developing such specialized solvers for every new combinatorial domain is computationally prohibitive and requires extensive domain expertise. Second, the weighted-sum scalarization inherently restricts exploration to the convex hull of the Pareto front. For minimization problems, only solutions on convex portions of the Pareto front can be found, regardless of the weights chosen. Non-convex regions, which often contain practically important trade-offs, remain inaccessible. This limitation is particularly severe in combinatorial spaces where Pareto fronts are typically highly non-convex due to the discrete nature of solutions. Third, weighted-sum methods require solving the scalarized problem to optimality for each weight vector, which becomes intractable for large-scale combinatorial problems without domain-specific algorithms. In contrast, our online learning framework provides a domain-agnostic approach that explores the entire Pareto front through adaptive decomposition and multi-expert strategies, without requiring problem-specific heuristics or convexity assumptions.

### B.1. Scalarization in D&L

Our framework is agnostic to the choice of scalarizing function and the bandit-based decomposition operates on any scalar objective derived from the multi-objective vector. We briefly review the two scalarizations used in our experiments.

**Weighted-sum scalarization** combines objectives linearly:

$$g_{\text{ws}}(x \mid w) = \sum_{i=1}^{m} w_i f_i(x), \quad w \in \Delta^{m-1} \tag{5}$$

where $\Delta^{m-1}$ denotes the $(m-1)$-simplex. This is the simplest scalarization but can only recover Pareto-optimal points on the convex hull of the front.

**Chebyshev (Tchebycheff) scalarization** uses the weighted infinity-norm:

$$g_{\text{tch}}(x \mid w, z^*) = \max_{i \in \{1,\ldots,m\}} \left\{ w_i \left| f_i(x) - z_i^* \right| \right\} \tag{6}$$

where $z^* = (z_1^*, \ldots, z_m^*)$ is a reference point (typically the ideal point). Unlike weighted-sum, Chebyshev scalarization can recover non-convex Pareto regions.

**Choice in experiments.** For combinatorial benchmarks (TSP, knapsack, CVRP), we use weighted-sum the simplest scalarization and demonstrating that strong performance stems from our decomposition and bandit framework rather than sophisticated scalarization. For hardware-software co-design, we use Chebyshev scalarization to handle non-convex trade-offs common in four-objective design spaces. Switching scalarizations is trivial as only the scalar objective function changes while the bandit-based subproblem optimization remains unchanged. The key contribution is the online learning framework over decomposed subproblems, not the scalarization itself.

## C. Additional Experimental details

### C.1. Datasets

#### C.1.1. MULTI-OBJECTIVE COMBINATORIAL OPTIMIZATION

Multi-objective combinatorial optimization (MOCO) commonly exists in industries, such as transportation, manufacturing, energy, and telecommunication (Chen et al., 2023b). We consider three typical MOCO problems that are commonly studied: multi-objective traveling salesman problem (MO-TSP) (Paquete & Stützle, 2003; Florios & Mavrotas, 2014), multi-objective knapsack problem (MO-KP), and multi-objective capacitated vehicle routing problem (MO-CVRP).

**MO-TSP** In the $m$-objective TSP (Lust & Teghem, 2010) with $n$ cities, each city has $m$ different 2D coordinate sets, where coordinates are sampled uniformly from $[0, 1]^2$. Each objective $f_i$ minimizes the total Euclidean distance of a tour using the $i$-th coordinate set. We evaluate both bi-objective (BiTSP) and tri-objective (TriTSP) variants, with solution space $|\mathcal{X}| = n!$. The objectives minimized are the tour lengths $f_i(\mathbf{x}) = \sum_{j=1}^{n} d_i(x_j, x_{j+1})$, where $d_i(\cdot, \cdot)$ denotes the Euclidean distance using the $i$-th coordinate set and $x_{n+1} = x_1$.

**MO-KP** The $m$-objective knapsack problem (Bazgan et al., 2009) consists of $n$ items with weights $w_j \sim \mathcal{U}(0, 1)$ and $m$ value sets where $v_{ij} \sim \mathcal{U}(0, 1)$ represents the value of item $j$ for objective $i$. The goal is to maximize $f_i(\mathbf{x}) = \sum_{j=1}^{n} v_{ij} x_j$ for each objective while satisfying the capacity constraint $\sum_{j=1}^{n} w_j x_j \leq C$. The solution space has cardinality $|\mathcal{X}| = 2^n$. The knapsack capacity is set to $C \in \{12.5, 25, 25\}$ for problem sizes $n \in \{50, 100, 200\}$ respectively.

**MO-CVRP** The bi-objective CVRP (Lacomme et al., 2004; Prins, 2004) consists of $n$ customers and one depot, with all node coordinates sampled uniformly from $[0, 1]^2$. The depot is positioned at $(0.5, 0.5)$. Each customer $j$ has an integer demand $d_j \sim \mathcal{U}\{1, ..., 9\}$. The two objectives are: (1) minimizing the total distance of all routes $f_1(\mathbf{x}) = \sum_{r \in \mathbf{x}} \text{length}(r)$, and (2) minimizing the makespan $f_2(\mathbf{x}) = \max_{r \in \mathbf{x}} \text{length}(r)$, where $r$ denotes a route. Following convention in (Kool et al., 2019; Lin et al., 2022) we consider the instances with different sizes where each vehicle has capacity $D \in \{30, 40, 50\}$ for problem sizes $n \in \{20, 50, 100\}$ respectively.

## C.2. Real-World Application: Hardware-Software Co-design

HDnn-PIM (Dutta et al., 2022) is an edge AI accelerator design that supports the HDnn algorithm. Designing HDnn-PIM requires choosing both software (model parameters, etc.) and hardware (systolic array size, etc.) design parameters. The choice of the design parameters then contributes to various design metrics, including energy, delay, area (PPA), and accuracy, in a black-box manner, which is only attainable through expensive simulation. Additionally, the architecture consists of components with stochastic nature (PIM) that results into noisy evaluation. The goal of the hardware-software co-design problem is to minimize PPA and maximize accuracy within trade-offs through hardware and software design parameter choices.

## C.3. Baseline Algorithm Details

**WS-LKH/DP (Weighted Sum Scalarization).** We implement weighted sum scalarization, a classical *a priori* multi-objective optimization technique that converts a vector-valued objective into a series of single-objective subproblems via linear aggregation (Miettinen, 1999). Although weighted sum scalarization cannot recover unsupported Pareto-optimal solutions in non-convex fronts (see discussion B), it remains a widely used baseline due to its simplicity and interpretability. For MO-TSP, we employ WS-LKH, combining weighted sum scalarization with two-opt local search inspired by the Lin-Kernighan-Helsgaun algorithm (Helsgaun, 2000; Tinós et al., 2018). Given $\lambda \in [0, 1]$ sampled uniformly across $K$ weights, each scalarized instance uses combined distance matrix $D_\lambda = \lambda D_1 + (1 - \lambda) D_2$ for bi-objective problems (extended to three weights summing to 1 for tri-objective). The algorithm generates random initial tours and applies iterative 2-opt improvements until local optimality. For MO-KP, we implement WS-DP, solving each scalarized knapsack exactly via dynamic programming. Item weights are scaled by factor 30 to enable integer-based DP while maintaining precision. For each $\lambda$, it solves $\max_x \sum_i [\lambda v_{i1} + (1 - \lambda) v_{i2}] x_i$ subject to capacity constraints using standard DP recurrence with backtracking for solution reconstruction. Both methods aggregate solutions across all weight combinations and retain only the non-dominated set. We use $K = 100$ for knapsack and vary $K$ by instance size for TSP, including extreme weights to ensure Pareto front boundary coverage.

**PPLS/D-C (Parallel Pareto Local Search with Decomposition).** We implement PPLS/D-C (Shi et al., 2022), a parallel local search that decomposes the objective space into subregions explored concurrently with periodic archive cooperation. The space is partitioned into $P$ regions, each defined by weight vector $\boldsymbol{\lambda}_p$ inducing preference via weighted sum scalarization. Each process maintains a local archive and explores its region using problem-specific operators: 2-opt for TSP, bit-flip and swap moves for knapsack with feasibility checking, and intra-route 2-opt plus inter-route relocation for CVRP. Cooperation occurs every $\tau$ iterations: processes import compatible solutions from the shared archive and export their top-ranked local solutions. A master process monitors coverage and dynamically reassigns regions to sparse areas of the Pareto front. Initialization combines constructive heuristics (nearest-neighbor for TSP, greedy by efficiency for knapsack) with random solutions. The final output is the non-dominated subset of the merged archive.

**MOBO-NSGA-II (Evolutionary Multi-Objective Optimization).** We implement NSGA-II (Deb et al., 2002) as our evolutionary baseline, which balances convergence and diversity through non-dominated sorting and crowding distance. The algorithm maintains a population that evolves through selection, crossover, and mutation, with environmental selection based on Pareto dominance rank and crowding distance to balance convergence and diversity. Non-dominated sorting partitions the population into fronts $F_1, F_2, \ldots$ where solutions in $F_k$ are dominated only by solutions in earlier fronts. Crowding distance measures local density in objective space, computed as the average side length of the cuboid formed by neighboring solutions. Binary tournament selection favors lower rank, with crowding distance as tiebreaker. Following established practices (Jaszkiewicz, 2002; Ishibuchi et al., 2015), we employ specialized operators for each problem: (1) TSP uses Order Crossover (OX) (Davis, 1985) preserving relative city ordering, with swap mutation; (2) Knapsack uses uniform crossover with greedy repair—infeasible offspring have items removed by increasing value-to-weight ratio until feasible, then items are greedily added back by decreasing efficiency (Ishibuchi et al., 2015); (3) CVRP implements the (Savelsbergh, 1985; Lacomme et al., 2004; Prins, 2004) operator suite including route-based crossover, cross-exchange mutation, and capacity-aware repair ensuring customer coverage.

**MOBO-qNEHVI (Direct Hypervolume Optimization).** qNEHVI (Daulton et al., 2021) is a state-of-the-art multi-objective Bayesian optimization method that directly maximizes expected hypervolume improvement rather than relying on scalarization. Given reference point $\mathbf{r}$ and current Pareto approximation $\mathcal{P}$, the acquisition is $\alpha_{\text{qNEHVI}}(\mathbf{x}) = \mathbb{E}[\text{HVI}(\mathbf{y}(\mathbf{x}) \mid \mathcal{P}, \mathbf{r})]$, estimated via Monte Carlo sampling from the GP posterior using Sobol quasi-random sequences. Originally proposed for continuous domains, we adapt it to combinatorial optimization following the same strategy as qParEGO: Kendall kernels for TSP permutations, Matérn kernels for knapsack, corresponding solution encodings, and NSGA-II with problem-specific operators for acquisition optimization. For computational tractability, we limit the baseline set to 20 diverse non-dominated solutions when $|\mathcal{P}|$ is large. Reference points are set conservatively beyond worst observed values to ensure positive hypervolume contributions.

**MOBO-qParEGO (Bayesian Optimization with Random Scalarization).** qParEGO (Knowles, 2006) is a Bayesian optimization method that decomposes multi-objective optimization into randomly scalarized subproblems; we use the batch variant qNParEGO available in BoTorch (Balandat et al., 2020). Each iteration samples $q$ weight vectors from Dirichlet($\alpha\mathbf{1}_M$) with $\alpha < 1$ to encourage diverse trade-off exploration. The augmented Chebyshev scalarization is $s_{\boldsymbol{\lambda}}(\mathbf{y}) = \max_j \lambda_j \tilde{y}_j + \rho \sum_j \lambda_j \tilde{y}_j$, where $\tilde{y}_j$ denotes the normalized objective and $\rho > 0$ ensures Pareto-optimal solutions are optimal for some weight. Each objective is modeled independently using Gaussian Process surrogates with input normalization and output standardization for numerical stability. Since the original method targets continuous domains, our adaptation to combinatorial problems involves: (1) problem-specific kernels, Kendall kernels (Kendall, 1938) for TSP capturing permutation similarity via pairwise concordance, and Matérn-5/2 with ARD for knapsack; (2) solution encoding—position-based for permutations, direct binary for knapsack; and (3) evolutionary optimization of qNoisyExpectedImprovement (Letham et al., 2018) with problem-specific operators (Order Crossover for TSP, uniform crossover with greedy repair for knapsack) replacing gradient-based continuous optimization.

**BOPR (Probabilistic Reparameterization).** BOPR (Daulton et al., 2022) is a Bayesian optimization method that natively handles discrete structures through probabilistic reparameterization, enabling gradient-based acquisition optimization over combinatorial domains. Unlike qParEGO and qNEHVI which require adaptation from continuous settings, BOPR directly addresses discrete optimization—we implement the original method faithfully for our benchmark problems. The method parameterizes distributions over solutions using continuous parameters $\boldsymbol{\theta}$ and optimizes expected objective values via the REINFORCE likelihood-ratio estimator (Williams, 1992). For TSP, we use the Plackett-Luce distribution with Gumbel-Max sampling: $\pi = \text{argsort}(\boldsymbol{\theta}/\tau + \mathbf{g})$ where $g_i \sim \text{Gumbel}(0,1)$ and $\tau$ controls exploration. For knapsack, we use independent Bernoulli distributions with $p_i = \sigma(\theta_i/\tau)$. The gradient is estimated as $\nabla_{\boldsymbol{\theta}}\mathbb{E}[\alpha(x)] \approx \frac{1}{M}\sum_m (\alpha(x^{(m)}) - b)\nabla_{\boldsymbol{\theta}} \log p_{\boldsymbol{\theta}}(x^{(m)})$, where $b$ is a moving-average baseline for variance reduction. Our extensions include edge-incidence encoding for TSP to improve GP modeling and qLogNEHVI as acquisition for numerical stability, with optimization via Adam across multiple random restarts.

**PMOCO (Neural Multi-Objective Combinatorial Optimization).** PMOCO (Lin et al., 2022) is a deep reinforcement learning approach that learns to approximate the entire Pareto set using a single preference-conditioned model. Among neural MOCO methods, we select PMOCO due to its widespread adoption and strong reported performance across standard benchmarks. The architecture builds upon POMO (Kwon et al., 2021), employing an attention-based encoder-decoder with instance normalization, extended to multi-objective settings via a hyper-network that generates decoder parameters

conditioned on preference vectors. Training uses reinforcement learning with Tchebycheff scalarization rewards across batches of randomly generated problem instances and uniformly distributed weight vectors $\{\boldsymbol{\lambda}_1, \ldots, \boldsymbol{\lambda}_K\}$, enabling a single trained model to produce approximate Pareto-optimal solutions for any trade-off preference via a single forward pass. Unlike other baselines which perform test-time optimization, PMOCO requires pre-training on a distribution of problem instances (e.g., TSP with cities sampled uniformly in $[0, 1]^2$). This amortized approach offers fast inference but assumes access to a training distribution matching the test instances—a fundamentally different setting from our work, which optimizes each instance from scratch without pre-training. We use the authors' official implementation with pre-trained models.

*Fair Comparison with Neural Methods* However, the standard PMOCO training protocol requires substantial data: approximately 20 million training instances (200 epochs $\times$ 100,000 instances per epoch) to achieve competitive performance. This creates a fundamental asymmetry when comparing against Bayesian or bandit-based methods, which require no training phase but instead perform iterative evaluations directly on the test instance. To enable a fair comparison in terms of sample efficiency, we constrain PMOCO's training budget to match the number of function evaluations used by our method on a single problem instance. Specifically, let $N_{\text{eval}}$ denote the total number of objective function evaluations performed by our method when solving one problem instance. We then train PMOCO with a maximum of $N_{\text{eval}}$ training samples:

$$\text{PMOCO training budget} = N_{\text{eval}} = \sum_{i=1}^{K} T_i \cdot E_i \qquad (7)$$

where $K$ is the number of weight vectors, $T_i$ is the number of iterations for weight vector $i$, and $E_i$ is the number of evaluations per iteration. In our experiments, $N_{\text{eval}} \approx 622,000$ evaluations per instance.

This comparison reveals the sample efficiency of different algorithmic paradigms: neural methods like PMOCO require millions of training samples to learn generalizable patterns, whereas our bandit-based approach achieves competitive performance using orders of magnitude fewer samples by directly optimizing on the target instance. Table 1 reports the hypervolume achieved by each method under matched sample budgets. This budget constrained evaluation is denoted by $PMOCO^*$ and we use approximately same scalarization weights as our method for fair comparison i.e $\approx 40$ weights.

## C.4. Evaluation Methodology

**Hypervolume** Given a Pareto front approximation $F = \{f^{(1)}, \ldots, f^{(|F|)}\}$ in $M$-dimensional objective space and a reference point $r \in \mathbb{R}^M$, the hypervolume indicator measures the $M$-dimensional Lebesgue measure of the region dominated by $F$ and bounded by $r$:

$$\text{HV}_r(F) = \lambda_M \left( \bigcup_{f \in F} \{p \in \mathbb{R}^M : f \prec p \prec r\} \right) \qquad (8)$$

where $\lambda_M$ denotes the Lebesgue measure and $f \prec p$ indicates that $f$ dominates $p$. For minimization problems, solution $f$ dominates point $p$ if $f_i \leq p_i$ for all $i \in \{1, \ldots, M\}$ with strict inequality in at least one objective; for maximization, the inequalities are reversed.

**Hypervolume Ratio** To enable scale-invariant comparison across problem sizes and objective magnitudes, we report the normalized hypervolume ratio:

$$\text{HV}'_r(F) = \frac{\text{HV}_r(F)}{\prod_{i=1}^{M} |r_i - z_i|} \qquad (9)$$

where $z \in \mathbb{R}^M$ is the ideal point representing the best achievable value in each objective. This normalization divides the raw hypervolume by the volume of the bounding box defined by $r$ and $z$, yielding values in $[0, 1]$ comparable across problem scales. Note, this is across all baselines consistently including PMOCO.

**Reference and Ideal Points.** For minimization problems (TSP, CVRP), the ideal point is set to the origin $z = \mathbf{0}$ and the reference point $r$ represents a worst-case upper bound exceeding all observed objective values. For maximization problems (Knapsack), the ideal point represents maximum achievable values and the reference point is set below all Pareto-optimal solutions to ensure positive hypervolume contributions. Standardized values are listed in Table 5 (Chen et al., 2023; Lin et al., 2022).

**Statistical Testing** Each method is evaluated on 200 randomly generated problem instances per configuration. We report mean hypervolume ratio with standard error and 95% confidence intervals computed via Student's $t$-distribution. For

pairwise algorithm comparison, we apply the Wilcoxon signed-rank test with significance level $\alpha = 0.05$, appropriate for paired samples on identical problem instances.

*Table 5.* Benchmark problems with standardized reference and ideal points for hypervolume ratio computation.

| PROBLEM | $n$ | REFERENCE POINT $r$ | IDEAL POINT $z$ |
|---|---|---|---|
| BITSP | 20 | $(20, 20)$ | $(0, 0)$ |
|  | 50 | $(35, 35)$ | $(0, 0)$ |
|  | 100 | $(65, 65)$ | $(0, 0)$ |
| TRITSP | 20 | $(20, 20, 20)$ | $(0, 0, 0)$ |
|  | 50 | $(35, 35, 35)$ | $(0, 0, 0)$ |
|  | 100 | $(65, 65, 65)$ | $(0, 0, 0)$ |
| BIKP | 50 | $(5, 5)$ | $(30, 30)$ |
|  | 100 | $(20, 20)$ | $(50, 50)$ |
|  | 200 | $(30, 30)$ | $(75, 75)$ |
| BICVRP | 20 | $(30, 4)$ | $(0, 0)$ |
|  | 50 | $(45, 4)$ | $(0, 0)$ |
|  | 100 | $(80, 4)$ | $(0, 0)$ |

**Note:** For BiCVRP50/100, BOPR and qParEGO struggle to find solutions with makespan $< 4$. We relaxed the reference point to $(45, 5)$ for both methods and $(80, 5)$ for BOPR with BICVRP100. This difficulty arises from the *permutation-to-route mapping discontinuity*: (i) small perturbations in the customer permutation can drastically change route compositions due to capacity-constrained splitting, causing large jumps in makespan; (ii) the GP surrogate's continuous encoding cannot capture route structure, making it difficult to learn the relationship between solutions and makespan; and (iii) makespan depends on the single longest route (a min-max objective), which is inherently harder to optimize than aggregate objectives like total distance. While problem-specific operators (e.g., NSGA-II with route-level crossover and relocation mutations) can achieve makespan $< 4$, we use generic operators for BOPR and qParEGO to ensure a fair comparison across non-heuristic methods. Because the hypervolume ratio normalizes by the bounding-box volume $\prod_i |r_i - z_i|$, the relaxed reference point partially rescales the affected BOPR/qParEGO cells onto a comparable $[0, 1]$ scale.

### C.5. Hyperparameter and Tuning used for reporting results

Table 6 summarizes the hyperparameters used across problem scales. These settings are shared by both the UCB and Thompson Sampling (TS) variants, with two adjustments for TS: temperature decay of 0.995 (vs. 0.98) and FTRL disabled. All other parameters remain identical. For decomposition size and overlap, we adopt the recommended settings from Section D.1: $d = n/2$ (50% of problem size) with rounding to the values above. Small deviations (e.g. at $n{=}20$, $n{=}100$) fall within the tolerant region identified in the harm-detection analysis and do not materially affect performance. Initial overlap is set to $\approx 44\%$ of $d$ and decays via the diminishing schedule. These values are guidelines rather than strict requirements and as our ablation demonstrates, the algorithm tolerates a range of configurations (e.g. diminishing-overlap) provided computational redundancy remains below the $2.5\times$ threshold. We provide a sensitivity analysis in Section D.6.

### C.6. Extended Results

#### C.6.1. HARDWARE AND COMPUTE

Experiments used heterogeneous hardware: NVIDIA A100 GPUs for most benchmarks, RTX 4090/L40S GPUs for Bi-CVRP (due to resource constraints), and a local macOS CPU for our method. To ensure thorough and fair computational analysis despite this variability, we report: (i) *algorithmic FLOPs* as the primary metric for fair, hardware-agnostic comparison (more discussion below), and (ii) *wall-clock time* for practical execution context (reported in Tables 7-10).

*Table 6.* Hyperparameter settings across problem scales. Small: 20 cities (TSP/CVRP) or 50 items (KP); Medium: 50 cities or 100 items; Large: 100 cities or 200 items.

| Hyperparameter | Small | Medium | Large |
|---|---|---|---|
| *Multiplicative Weights / UCB* | | | |
| Learning rate $\eta$ | 0.5 | 0.5 | 0.5 |
| UCB coefficient | 3.0 | 3.0 | 3.0 |
| Initial temperature | 1.0 | 1.0 | 1.0 |
| Temperature decay | 0.98 | 0.98 | 0.98 |
| *Decomposition* | | | |
| Decomposition size | 15 | 25 | 35 |
| Initial overlap | 6 | 11 | 18 |
| Overlap decay rate $\beta$ | 0.1 | 0.1 | 0.1 |
| Diminishing overlap | ✓ | ✓ | ✓ |
| *Lagrangian Dual* | | | |
| Dual step size $\alpha$ | 1.0 | 1.0 | 1.0 |
| Accelerated dual | ✓ | ✓ | ✓ |
| Use FTRL | ✓ | ✓ | ✓ |
| *Search Control* | | | |
| Max iterations | 100 | 150 | 200 |
| Number of rounds | 5 | 10 | 20 |
| Patience | 10 | 20 | 50 |
| Hybrid ratio | 0.5 | 0.5 | 0.5 |
| *Additional Components* | | | |
| Weight vectors | 20 | 20/25 | 20/35 |

*Note:* ✓ indicates the feature is enabled.

### C.6.2. ALGORITHMIC COMPUTATIONAL COST ESTIMATION

We report computational cost using *algorithmic operation counting*, a standard technique in optimization and machine learning for estimating hardware-independent computational complexity. Rather than attempting to measure executed hardware instructions, we count the number of floating-point operations implied by the algorithmic structure of each method.

**Algorithmic Operation Count.** For each method, we decompose execution into major algorithmic components (e.g., surrogate model fitting, acquisition evaluation, evolutionary search, solution evaluation, and Pareto updates). Within each component, we analytically count primitive arithmetic operations (addition, multiplication, comparison, exponentiation, etc.) required by the corresponding algorithmic kernels. This yields a symbolic operation count that depends only on problem size, objective dimension, and algorithmic hyperparameters. Formally, let $\mathcal{A}$ denote an algorithm composed of components $\{C_i\}$. The total algorithmic cost is computed as

$$\mathrm{AOC}(\mathcal{A}) = \sum_i \mathrm{AOC}(C_i),$$

where $\mathrm{AOC}(C_i)$ is the number of arithmetic operations implied by component $C_i$.

**Gaussian Process Models.** For Gaussian process (GP) surrogates, we follow standard complexity analyses that count linear algebra operations rather than hardware instructions. Exact GP training incurs $O(n^3)$ operations due to Cholesky factorization, while sparse GP variants reduce this to $O(nm^2)$ with $m$ inducing points. Posterior prediction costs are accounted for separately. Backward passes are incorporated using a calibrated multiplicative factor applied to forward kernel costs.

**Multi-Objective Acquisition Functions.** For multi-objective Bayesian optimization, we analytically count the cost of acquisition evaluation, including Monte Carlo sampling, hypervolume box decomposition, and per-sample hypervolume contribution checks. These costs scale with the number of objectives $M$, Monte Carlo samples $S$, and Pareto front size $|\mathcal{P}|$, consistent with prior analyses of expected hypervolume improvement.

**Evolutionary Optimization.**    For evolutionary optimizers such as NSGA-II, we count operations for non-dominated sorting, crowding distance computation, selection, crossover, and mutation. These counts follow the standard complexity results of $O(MN^2)$ for non-dominated sorting and $O(MN \log N)$ for crowding distance, with explicit constants retained.

**Problem-Specific Evaluation.**    Finally, problem-dependent costs (e.g., TSP tour evaluation, knapsack feasibility repair, or CVRP route evaluation) are explicitly counted based on the arithmetic operations required to evaluate objective functions and enforce constraints.

The resulting operation counts are *algorithmic* and hardware-independent. They do not correspond to executed GPU or CPU instructions and should not be interpreted as exact hardware FLOPs. Instead, they provide a standardized and reproducible measure of relative computational cost across methods and problem scales.

*Table 7.* Bi-objective TSP (200 instances). Wall-clock time on [GPU/CPU spec].

| Method | Bi-TSP 20 | | | Bi-TSP 50 | | | Bi-TSP 100 | | |
|---|---|---|---|---|---|---|---|---|---|
| | HV↑ | —NDS—↑ | Time↓ | HV↑ | —NDS—↑ | Time↓ | HV↑ | —NDS—↑ | Time↓ |
| *Problem-specific* | | | | | | | | | |
| WS-LKH | **0.625** | 21 | 6m | **0.629** | 26 | 50m | **0.69** | 50 | 50h |
| PPLS/D-C | **0.626** | 71 | 26m | 0.628 | 213 | 2.8h | 0.63 | 533 | 8h |
| *Neural MOCO* | | | | | | | | | |
| PMOCO (101 wt.) | 0.509 | 69 | 15h† | 0.550 | 76 | 25h† | 0.67 | 86 | 100h† |
| PMOCO (40 wt.) | 0.511 | 22 | 15h† | 0.50 | 28 | 25h† | 0.55 | 31 | 100h† |
| PMOCO*(40 wt.) | 0.324 | 2 | 6m | 0.436 | 7 | 21m | 0.47 | 2 | 2h |
| *General MOCO* | | | | | | | | | |
| NSGA-II | 0.573 | 225 | 4.3h | 0.347 | 20 | 4.2h | 0.294 | 51 | 8.5h |
| MOBO-qNEHVI | 0.352 | 18 | 10h | 0.126 | 16 | 15h | 0.083 | 17 | 16h |
| MOBO-qParEGO | 0.373 | 4 | 15h | 0.137 | 7 | 34h | 0.104 | 6 | 54h |
| PR | 0.536 | 10 | 1.2h | 0.472 | 23 | 38h | 0.07 | 11 | 88h |
| **Ours** | 0.585 | 45 | **10m** | 0.530 | 62 | **1h** | 0.40 | 48 | **1.8h** |
| **Ours-TS** | **0.60** | 102 | 12m | **0.54** | 147 | 2.2h | **0.47** | 114 | 5h |

†Training time. *Same computational budget as Ours.

*Table 8.* Bi-objective Knapsack (200 instances). Wall-clock time on [GPU/CPU spec].

| Method | Bi-KP 50 | | | Bi-KP 100 | | | Bi-KP 200 | | |
|---|---|---|---|---|---|---|---|---|---|
| | HV↑ | —NDS—↑ | Time↓ | HV↑ | —NDS—↑ | Time↓ | HV↑ | —NDS—↑ | Time↓ |
| *Problem-specific* | | | | | | | | | |
| WS-DP | **0.356** | 17 | 7m | **0.440** | 23 | 1h | **0.36** | 30 | 2h |
| PPLS/D-C | **0.357** | 25 | 36m | 0.447 | 49 | 1h | 0.362 | 63 | 1.4h |
| *Neural MOCO* | | | | | | | | | |
| PMOCO | 0.355 | 55 | 13h† | 0.443 | 42 | 82h† | 0.354 | 52 | 40h |
| PMOCO*(40 wt.) | 0.34 | 40 | 10m | 0.37 | 34 | 2h | 0.186 | 2 | 1h |
| *General MOCO* | | | | | | | | | |
| NSGA-II | 0.219 | 68 | 8h | 0.221 | 24 | 9h | 0.28 | 13 | 9h |
| qNEHVI | 0.301 | 28 | 17h | 0.25 | 18 | 33h | 0.10 | 9 | 51h |
| qParEGO | 0.29 | 8 | 14h | 0.314 | 7 | 38h | 0.21 | 8 | 55h |
| PR | 0.264 | 6 | 3h | 0.304 | 10 | 4.8h | 0.266 | 13 | 11h |
| **Ours** | **0.35** | 45 | **43m** | 0.338 | 62 | **2h** | 0.26 | 21 | 7.5 |
| **Ours-TS** | 0.347 | 71 | 113m | **0.385** | 56 | 3h | **0.30** | 73 | 10h |

†Training time.

### C.6.3. ALGORITHMIC COST VS. WALL-CLOCK TIME

Algorithmic operation counts and wall-clock runtime capture complementary aspects of computational cost. Algorithmic counts reflect the logical number of arithmetic operations implied by an algorithm and are independent of hardware, software libraries, and implementation details. In contrast, wall-clock time measures actual execution time on a specific machine and is influenced by factors such as parallelism, memory access, kernel fusion, and hardware acceleration.

As a result, there is no fixed conversion between algorithmic operation counts and wall-clock time. However, for a fixed

*Table 9.* Tri-objective TSP (200 instances). Wall-clock time on [GPU/CPU spec].

| Method | Tri-TSP 20 | | | Tri-TSP 50 | | | Tri-TSP 100 | | |
|---|---|---|---|---|---|---|---|---|---|
| | HV↑ | —NDS—↑ | Time↓ | HV↑ | —NDS—↑ | Time↓ | HV↑ | —NDS—↑ | Time↓ |
| *Problem-specific* | | | | | | | | | |
| WS-LKH | **0.467** | 46 | **9.3m** | **0.429** | 189 | – | **0.488** | 200 | – |
| PPLS/D-C | 0.388 | 35 | 1.3h | 0.262 | 93 | 2.5h | 0.241 | 158 | 4h |
| *Neural MOCO* | | | | | | | | | |
| PMOCO | 0.460 | 105 | 11h | 0.420 | 105 | 23h | 0.44 | 104 | 66h |
| PMOCO* | 0.44 | 40 | 10m | 0.37 | 103 | 1h | 0.40 | 105 | 5h |
| *General MOCO* | | | | | | | | | |
| NSGA-II | 0.411 | 200 | 31h | 0.173 | 198 | 32h | 0.135 | 200 | 92h |
| qNEHVI | 0.23 | 68 | 40h | 0.052 | 69 | 42h | 0.024 | 107 | 48h |
| qParEGO | 0.32 | 6 | 33h | 0.062 | 10 | 42h | 0.03 | 31 | 94h |
| PR | 0.296 | 45 | 2.5h | 0.04 | 30 | 63h | 0.02 | 39 | 80h |
| **Ours** | **0.43** | 108 | **10m** | 0.262 | 172 | **1.5h** | 0.21 | 223 | 7h |
| **Ours-TS** | 0.42 | 218 | 25m | **0.282** | 494 | 2.8h | **0.234** | 310 | **6h** |

[†]: Training time. HV↑: higher is better, —NDS—↑: higher is better, Time↓: lower is better

*Table 10.* Bi-objective CVRP (200 instances). Wall-clock time on [GPU/CPU spec].

| Method | Bi-CVRP 20 | | | Bi-CVRP 50 | | | Bi-CVRP 100 | | |
|---|---|---|---|---|---|---|---|---|---|
| | HV↑ | —NDS—↑ | Time↓ | HV↑ | —NDS—↑ | Time↓ | HV↑ | —NDS—↑ | Time↓ |
| *Problem-specific* | | | | | | | | | |
| PPLS/D-C | **0.48** | 7 | 27m | **0.45** | 21 | 33m | **0.43** | 25 | 1.7h |
| *Neural MOCO* | | | | | | | | | |
| PMOCO (40 wt.) | 0.36 | 8 | 7h | 0.346 | 6 | 11h | 0.35 | 7 | 26h |
| PMOCO* (40 wt.) | 0.36 | 8 | 5m | 0.33 | 6 | 20m | 0.309 | 6 | 30m |
| *General MOCO* | | | | | | | | | |
| NSGA-II | 0.43 | 81 | 7h | 0.31 | 9 | 26h | 0.31 | 14 | 100h |
| qNEHVI | 0.352 | 13 | 13h | 0.14 | 7 | 31h | 0.11 | 10 | 50h |
| qParEGO | 0.21 | 4 | 33h | 0.066 | 3 | 45h | 0.11 | 9 | 58h |
| PR | 0.22 | 5 | 31h | 0.052 | 8 | 55h | 0.08 | 4 | 80h |
| **Ours** | **0.45** | 22 | **15m** | 0.35 | 37 | **3h** | 0.253 | 17 | **6h** |
| **Ours-TS** | **0.46** | 36 | 44m | **0.37** | 50 | 4h | **0.30** | 42 | 8h |

[†]: Training time. HV↑: higher is better, —NDS—↑: higher is better, Time↓: lower is better

*Table 11.* Results across benchmarks (200 instances each), reporting Hypervolume (HV, ↑), number of non-dominated solutions (NDS, ↑), computational cost in Tera-FLOPs ($\mathcal{F}$, ↓). [†]Cost includes training; *evaluated under matched compute budget, NA: no WS solver for CVRP. The vertical line separates specialized problem-specific solvers (left) from general-purpose methods (right).

| | | WS-* | | | PPLS/D-C | | | PMOCO[†] | | | PMOCO* | | | NSGA-II | | | qNEHVI | | | qParEGO | | | PR | | | D&L | | | D&L-TS | | |
|---|---|---|---|---|---|---|---|---|---|---|---|---|---|---|---|---|---|---|---|---|---|---|---|---|---|---|---|---|---|---|---|
| | | HV | NDS | $\mathcal{F}$ | HV | NDS | $\mathcal{F}$ | HV | NDS | $\mathcal{F}$ | HV | NDS | $\mathcal{F}$ | HV | NDS | $\mathcal{F}$ | HV | NDS | $\mathcal{F}$ | HV | NDS | $\mathcal{F}$ | HV | NDS | $\mathcal{F}$ | HV | NDS | $\mathcal{F}$ | HV | NDS | $\mathcal{F}$ |
| Bi-TSP | 20 | **.625** | 21 | **.001** | **.626** | 71 | .03 | .509 | 69 | 5e3 | .36 | 8 | 68 | .573 | 225 | .15 | .352 | 18 | 24 | .373 | 4 | 4.6 | .536 | 10 | 4.4 | .585 | 45 | .01 | **.60** | 102 | .01 |
| | 50 | **.629** | 26 | .06 | .628 | 213 | .07 | .550 | 76 | 24e3 | .33 | 6 | 734 | .347 | 20 | .15 | .126 | 16 | 29 | .137 | 7 | 274 | .472 | 23 | 51 | .530 | 62 | **.05** | **.54** | 147 | .07 |
| | 100 | **.69** | 50 | 1.0 | .63 | 533 | .15 | .67 | 86 | 105e3 | .309 | 6 | 685 | .294 | 51 | .31 | .083 | 17 | 37 | .104 | 6 | 2462 | .07 | 11 | 357 | .40 | 48 | **.13** | **.47** | 114 | .16 |
| Bi-KP | 50 | **.356** | 17 | **.003** | **.357** | 25 | .01 | .355 | 55 | 10e3 | .34 | 40 | 362 | .219 | 68 | .31 | .301 | 28 | 41 | .29 | 8 | 4.4 | .264 | 6 | 2.0 | **.35** | 45 | .13 | .347 | 71 | **.07** |
| | 100 | .440 | 23 | **.02** | **.447** | 49 | .02 | .443 | 42 | 22e3 | .37 | 34 | 816 | .221 | 24 | .31 | .25 | 18 | 131 | .314 | 7 | 19 | .304 | 10 | 3.7 | .338 | 62 | .28 | **.385** | 56 | **.13** |
| | 200 | .36 | 30 | .08 | **.362** | 63 | .07 | .354 | 52 | 47e3 | .186 | 28 | 583 | .28 | 13 | .31 | .10 | 9 | 590 | .21 | 8 | 300 | .266 | 13 | 9.9 | .26 | 21 | .39 | **.30** | 73 | .22 |
| Tri-TSP | 20 | **.467** | 46 | .008 | .388 | 35 | .01 | .460 | 105 | 5e3 | .43 | 40 | 51 | .411 | 200 | 1.5 | .23 | 68 | 3.9 | .32 | 6 | 1.7 | .296 | 45 | 5.1 | **.43** | 108 | .11 | .42 | 218 | **.02** |
| | 50 | **.429** | 189 | .34 | .262 | 93 | **.04** | .420 | 105 | 24e3 | .39 | 25 | 236 | .173 | 198 | 1.5 | .052 | 69 | 4.9 | .062 | 10 | 87 | .04 | 30 | 83 | .262 | 172 | .26 | **.282** | 494 | .09 |
| | 100 | **.488** | 200 | 5.2 | .241 | 158 | **.16** | .44 | 104 | 104e3 | **.30** | 40 | 654 | .135 | 200 | 6.1 | .024 | 107 | 6.4 | .03 | 31 | 1408 | .02 | 39 | 308 | .21 | 223 | .26 | **.234** | 310 | .24 |
| Bi-CVRP | 20 | NA | NA | NA | **.48** | 7 | .10 | .36 | 8 | 6e3 | .36 | 8 | 398 | .43 | 81 | .15 | .352 | 13 | 2.5 | .21 | 4 | 1.8 | .22 | 5 | 3.0 | **.45** | 22 | **.01** | .46 | 36 | .04 |
| | 50 | NA | NA | NA | **.45** | 21 | .12 | .346 | 6 | 28e3 | .33 | 6 | 571 | .31 | 9 | 1.0 | .14 | 7 | 14 | .06 | 3 | 14 | .052 | 8 | 1.3 | .35 | 37 | **.12** | **.37** | 50 | .10 |
| | 100 | NA | NA | NA | **.43** | 25 | .14 | .35 | 7 | 126e3 | .30 | 6 | 2e3 | .31 | 14 | 3.5 | .11 | 10 | 17 | .11 | 9 | 128 | .08 | 4 | 1 | .253 | 17 | .31 | **.30** | 42 | **.19** |

hardware and implementation, components with larger algorithmic costs are expected to dominate runtime. We therefore validate our analytical cost model by comparing *relative* operation counts against observed runtime ratios across algorithmic components, rather than attempting to match absolute times.

**FLOP Model Validation.** To validate that our analytical FLOP estimates accurately reflect actual computational costs, we compare predicted and measured time distributions across algorithm components. For each method, we decompose the total computation into $K$ components and measure both the predicted FLOP share $\hat{p}_k = F_k / \sum_{j=1}^{K} F_j$ and the observed

*Table 12.* Computational cost comparison in TFLOPs ($10^{12}$ floating-point operations). Lower values indicate more efficient algorithms. Our method achieves competitive solution quality with orders of magnitude fewer FLOPs.

| Method | Bi-TSP | | | Bi-KP | | | Tri-TSP | | | Bi-CVRP | | |
|---|---|---|---|---|---|---|---|---|---|---|---|---|
| | n=20 | n=50 | n=100 | n=50 | n=100 | n=200 | n=20 | n=50 | n=100 | n=20 | n=50 | n=100 |
| *Problem-specific* | | | | | | | | | | | | |
| WS-LKH/DP | 0.001 | 0.064 | 0.985 | 0.003 | 0.016 | 0.078 | 0.008 | 0.338 | 5.169 | NA | NA | NA |
| PPLS/D-C | 0.084 | 0.067 | 0.145 | 0.010 | 0.022 | 0.073 | 0.010 | 0.040 | 0.162 | 0.10 | 0.12 | 0.14 |
| *Neural MOCO* | | | | | | | | | | | | |
| PMOCO | 4888 | 23580 | 104663 | 9745.50 | 21956 | 46895 | 4921 | 23590 | 104438.37 | 5746 | 28262.66 | 126167.45 |
| PMOCO* | 67.7 | 734 | 685 | 362 | 816 | 583 | 51.153 | 236 | 654 | 398 | 571 | 1909 |
| *General MOCO* | | | | | | | | | | | | |
| NSGA-II | 0.152 | 0.152 | 0.310 | 0.306 | 0.310 | 0.320 | 1.520 | 1.520 | 6.100 | 0.152 | 1.004 | 3.53 |
| qNEHVI | 24.0 | 29.0 | 37.0 | 41.4 | 131 | 590 | 3.90 | 4.9 | 6.35 | 2.53 | 13.5 | 16.63 |
| qParEGO | 4.58 | 274 | 2462 | 4.36 | 18.64 | 300 | 1.72 | 87.0 | 1408 | 1.834 | 14.42 | 128.4 |
| PR | 4.36 | 51.3 | 357 | 2.00 | 3.71 | 9.85 | 5.06 | 83.3 | 308.13 | 2.98 | 1.301 | 0.97 |
| **Ours** | **0.012** | **0.046** | **0.128** | **0.130** | **0.280** | **0.39** | **0.110** | **0.260** | **0.260** | 0.013 | 0.12 | 0.307 |
| **Ours (TS)** | **0.014** | **0.07** | **0.158** | **0.073** | **0.130** | **0.22** | **0.019** | **0.092** | **0.242** | 0.04 | 0.10 | 0.19 |

FLOPs are computed as algorithmic operations independent of hardware. *Same computational budget as our method.

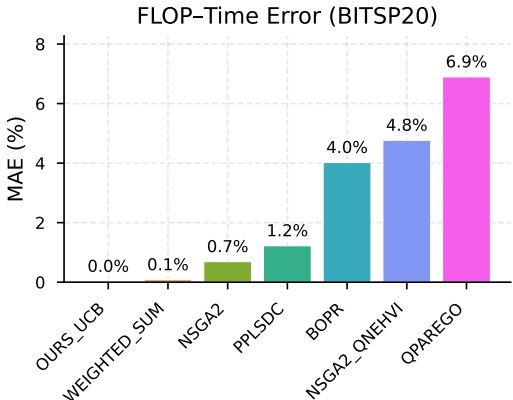

*Figure 2.* FLOP model validation across methods. Bars show the Mean Absolute Error (MAE) between predicted FLOP shares and measured wall-clock time shares per component. Lower values indicate better agreement. All methods achieve MAE $< 7\%$, with the majority under 2%, confirming that our analytical FLOP estimates reliably predict relative computational costs.

wall-clock time share $p_k = T_k / \sum_{j=1}^{K} T_j$ for each component $k$. We quantify agreement using the Mean Absolute Error (MAE):

$$\text{MAE} = \frac{1}{K} \sum_{k=1}^{K} |\hat{p}_k - p_k| \tag{10}$$

where lower values indicate better agreement between predicted and observed time distributions. An MAE of 0% indicates perfect prediction, while values below 10% are generally considered good given typical measurement variance.

Table 13 and Figure 2 reports the validation results across all methods for BITSP20. The majority of methods achieve MAE below 2%, indicating that our FLOP estimates accurately capture the relative computational costs of algorithm components. Even the highest MAE (QPAREGO at 6.89%) remains within acceptable bounds, as the dominant components are correctly identified and their proportions are well-approximated. These results confirm that our analytical FLOP model provides a reliable basis for comparing computational efficiency across methods.

### C.6.4. COMPARISON WITH MOEA/D

We add MOEA/D (Zhang & Li, 2007) as a baseline to verify our exclusion choice. MOEA/D and NSGA-II share an algorithmic class: both evolutionary, population-based over the full decision space, differing only in selection (weight-vector decomposition vs. non-dominated sorting). MOEA/D's notion of decomposition operates on the *objective* space rather than the decision space (Appendix A). We evaluate on BiTSP at $n \in \{20, 50, 100\}$ with 200 random instances per scale under

*Table 13.* FLOP model validation: Mean Absolute Error (MAE) between predicted FLOP shares and measured wall-clock time shares for each algorithm component.

| Method | MAE (%) | Components |
|---|---|---|
| OURS(UCB) | 0.03 | 8 |
| WEIGHTED_SUM | 0.08 | 5 |
| NSGA2 | 0.69 | 8 |
| PPLSDC | 1.22 | 5 |
| BOPR | 4.02 | 5 |
| NSGA2_QNEHVI | 4.77 | 4 |
| QPAREGO | 6.89 | 6 |

matched evaluation budgets, reporting hypervolume (HV, ↑), non-dominated solutions (NDS, ↑), and compute in TFLOPs ↓.

*Table 14.* MOEA/D vs. NSGA-II vs. D&L-TS on BiTSP across scales ($n \in \{20, 50, 100\}$, 200 random instances per scale, matched evaluation budgets). Best HV and lowest TFLOPs per scale in bold.

| Method | BiTSP-20 | | | BiTSP-50 | | | BiTSP-100 | | |
|---|---|---|---|---|---|---|---|---|---|
| | HV ↑ | NDS ↑ | TFLOPs ↓ | HV ↑ | NDS ↑ | TFLOPs ↓ | HV ↑ | NDS ↑ | TFLOPs ↓ |
| MOEA/D | .572 | 64 | 0.13 | .400 | 50 | 0.13 | .310 | 95 | 0.25 |
| NSGA-II | .573 | 225 | 0.15 | .347 | 20 | 0.15 | .294 | 51 | 0.31 |
| D&L-TS | **.600** | 102 | **0.01** | **.540** | 147 | **0.07** | **.470** | 114 | **0.16** |

**Results.** MOEA/D and NSGA-II achieve comparable HV at $n=20$ (within 0.001) and stay within the same class at larger scales, confirming that NSGA-II is a representative evolutionary baseline. D&L-TS outperforms both at every scale, with the HV gap to MOEA/D widening from 0.027 at $n=20$ to 0.160 at $n=100$ at **2–15×** lower compute. NSGA-II's high NDS at $n=20$ (225) reflects its archive retaining all incomparable solutions and does not translate to higher HV.

**Discussion.** The widening HV gap with $n$ is consistent with the decomposition argument in Section 4.2.1: as $n$ grows, population-based methods searching the full decision space suffer from the curse of dimensionality, whereas D&L's bandit subproblems each operate on $d \ll n$ variables. The compute gap widens for the same structural reason, evolutionary population sizes must grow to maintain coverage of larger decision spaces, while D&L's per-iteration cost scales with the subproblem size $d$ rather than with $n$. The empirical equivalence of MOEA/D and NSGA-II on HV justifies treating evolutionary multi-objective methods as a single baseline class in Section 6.4, and the NDS comparison at larger scales (147 vs 50 at $n=50$, 114 vs 95 at $n=100$) provides secondary confirmation that D&L's improvement comes from finding genuinely new non-dominated solutions rather than from concentrating samples in already explored regions of the front.

# D. Ablation Studies

## D.1. Subproblems and Decomposition Size

We ablate decomposition size and overlap percentage, which jointly determine the number and size of subproblems. This analysis reveals the tradeoff between decomposition granularity (smaller subproblems are easier to optimize) and computational cost (more subproblems leads to additional coordination overhead).

**Decomposition size** as described in section 4.2.1 refers to the size of each subproblem after partitioning the solution space. Given a problem of size $n$, decomposition size $d$, and overlap $o$, the number of subproblems is determined by sliding window partitioning with step size, $step = \max(1, d - o)$. The framework creates subproblems as: $\{window_i = [i, \min(i + d, n)) \mid i \in \{0, step, 2 \cdot step, \ldots\}\}$. Larger decomposition sizes yield fewer subproblems, reducing computational overhead. However, excessively small decomposition sizes increase the number of subproblems and constraint conflicts between overlapping regions, reducing efficiency.

**Overlap** (or overlap percentage) denotes the number of shared indices between consecutive subproblems, implementing a sliding window approach. Note, that an optimal overlap exists: excessive overlap (e.g., $o \geq d/2$) introduces redundancy

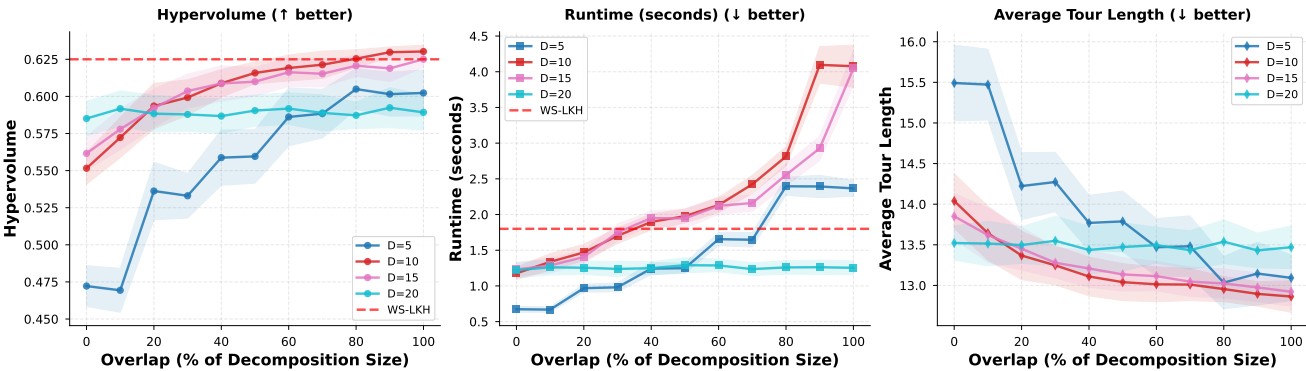

*Figure 3.* **Decomposition size and overlap ablation on BiTSP-20.** We vary decomposition size $d \in \{5, 10, 15, 20\}$ (25%–100% of problem size) and overlap percentage from 0% to 100%. **(Left)** Hypervolume improves with overlap across all decomposition sizes, with $d \in \{10, 15\}$ approaching the WS-LKH baseline (red dashed) by 40–50% overlap. **(Center)** Runtime remains stable below 60% overlap; beyond this threshold, smaller decompositions (especially $d = 10$) exhibit runtime explosion due to subproblem proliferation. **(Right)** Tour length decreases with overlap, with all configurations converging to comparable quality (~13) by 50% overlap. The smallest decomposition ($d = 5$) shows the highest sensitivity to overlap, exhibiting both the largest quality gains and highest variance. Results averaged over 50 runs; shaded regions denote $\pm 1$ standard deviation.

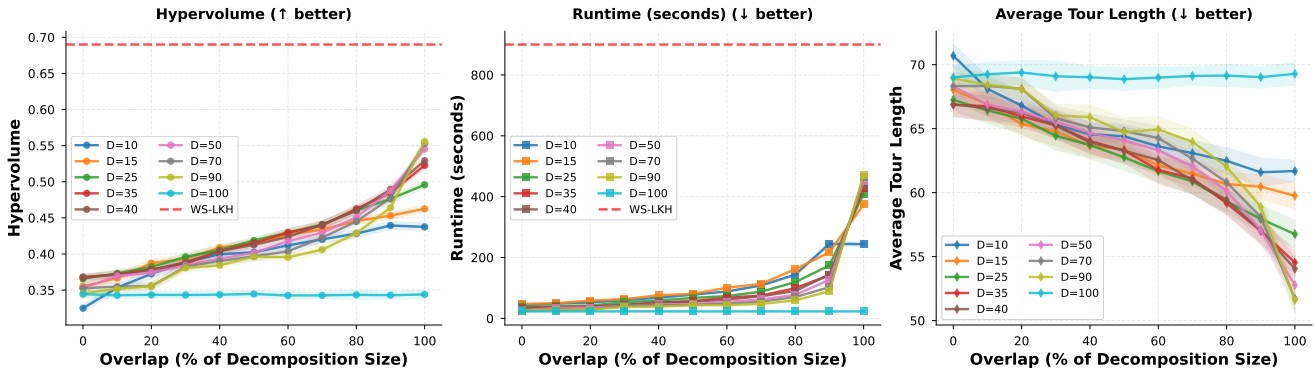

*Figure 4.* **Decomposition size and overlap ablation on BiTSP-100.** We vary decomposition size $d \in \{10, 15, 25, 35, 40, 50, 70, 90, 100\}$ and overlap percentage from 0% to 100%. **(Left)** Hypervolume increases with overlap up to 40–50%, then plateaus; moderate decomposition sizes ($d \in \{25, 50\}$) achieve the best quality. **(Center)** Runtime remains stable below 50% overlap but increases superlinearly beyond, particularly for smaller $d$ due to the proliferation of overlapping subproblems. **(Right)** Average tour length improves with overlap, converging across all configurations by 50%. The red dashed line indicates the weighted-sum LKH baseline. Results averaged over 50 runs; shaded regions show $\pm 1$ standard deviation.

and duplicated computation across subproblems, while insufficient overlap risks losing global consistency as subproblems become too isolated. We employ a simple diminishing overlap schedule $o_t = o_0 \cdot t^{-\alpha}$ (where $t$ is the iteration count and $\alpha$ is the decay rate) rather than a fixed threshold. This provides high overlap initially for strong cross-subproblem coordination, which naturally decays as the algorithm converges and local refinement becomes more important.

**Empirical validation** To validate these design choices, we conducted a comprehensive sweep on BiTSP at two scales: $n = 20$ (small) and $n = 100$ (large). For BiTSP-20, we test decomposition sizes $d \in \{5, 10, 15, 20\}$ (25%–100% of problem size); for BiTSP-100, we test $d \in \{10, 15, 25, 35, 40, 50, 70, 90, 100\}$ (10%–70% of problem size). Overlap percentages range from 0% to 100% in 10% increments. Each configuration was evaluated over 50 independent runs using the UCB-EXP3 variant. We also report BITSP50 results for the alternative Thompson-EXP3 variant. We track three metrics: hypervolume (solution quality), average tour length (objective value), and wall-clock runtime. Additionally, we perform a *harm detection analysis* to identify the overlap threshold beyond which increasing overlap becomes counterproductive.

**Results: Quality-runtime tradeoff.** Figures 3 and 4 reveal consistent patterns across both problem scales. **Hypervolume** improves monotonically with overlap up to 40–50%, after which gains plateau. Moderate decomposition sizes ($d \approx n/2$ to $n/4$) achieve the best performance: $d \in \{10, 15\}$ for BiTSP-20 and $d \in \{25, 50\}$ for BiTSP-100, both reaching hypervolume

within 5% of the weighted-sum LKH baseline. The smallest decomposition sizes exhibit higher variance due to insufficient local context per subproblem. **Tour length** follows a similar pattern, with all configurations converging to comparable quality by 50% overlap. **Runtime** increases superlinearly with overlap, particularly for small decomposition sizes. This effect is amplified at larger scale: for BiTSP-100, the runtime penalty at high overlap is more pronounced due to the increased number of subproblems. Note, we see similar trend for the Thompson-EXP3 variant as show in Figure 7.

**Harm detection analysis.** While Figures 3 and 4 show that solution quality improves with overlap, they do not reveal *when* increasing overlap becomes counterproductive. To identify this threshold, we introduce a **harm detection analysis** that measures computational efficiency rather than raw performance. We define two diagnostic metrics:

- *Computational redundancy*: the ratio of total subproblem evaluations to problem size, $R = \frac{\sum_i \text{evals}_i}{n}$. This quantifies how many times each solution index is processed across all subproblems. With overlap $o$ and decomposition size $d$, the step size shrinks to step $= d - o$, increasing the number of overlapping windows and thus redundancy. We set a harm threshold at $R > 10\times$.

- *Marginal efficiency*: the hypervolume gain per unit runtime cost, $\eta = \frac{\Delta\text{HV}}{\Delta\text{Runtime}}$, computed between consecutive overlap percentages. Positive $\eta$ indicates beneficial overlap; negative $\eta$ indicates that additional overlap *hurts* performance (runtime increases while quality stagnates or decreases).

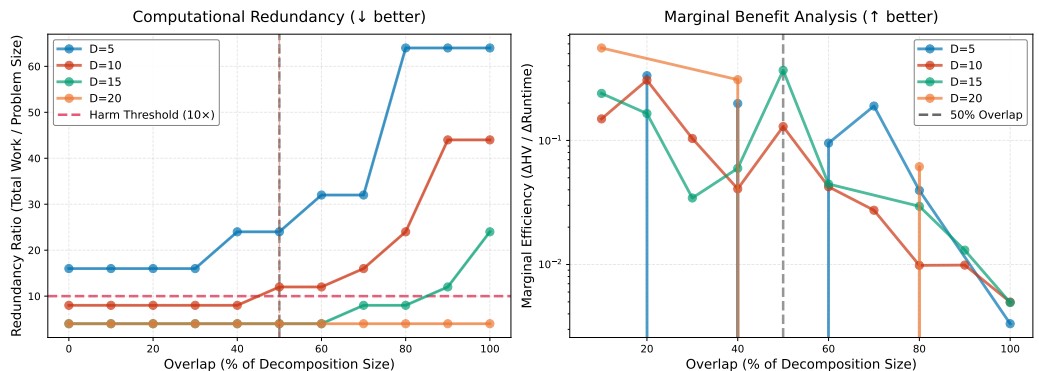

*Figure 5.* **Harm detection analysis on BiTSP-20. (Left)** Computational redundancy measures total subproblem evaluations divided by problem size. Small decompositions suffer disproportionately: $d = 5$ reaches $63\times$ redundancy at 100% overlap, while $d = 20$ remains below $10\times$ throughout. The harm threshold ($10\times$, red dashed) is crossed at ∼50% overlap for $d = 5$ but never exceeded for $d \geq 15$. **(Right)** Marginal efficiency ($\Delta\text{HV}/\Delta\text{Runtime}$) quantifies diminishing returns. All configurations show positive efficiency below 50% overlap (gray dashed), indicating quality gains justify runtime costs. Beyond this threshold, efficiency drops by 1–2 orders of magnitude, with several configurations exhibiting near-zero or negative marginal returns. This identifies 50% overlap as the critical efficiency cliff beyond which additional overlap provides negligible benefit at substantial computational cost.

**Results: Harm detection on BiTSP-20.** Figure 5 reveals the efficiency tradeoffs underlying the quality-runtime curves. (Figure 5a) exhibits a strong interaction between decomposition size and overlap. Small decompositions are disproportionately affected: $d = 5$ reaches $63\times$ redundancy at 100% overlap, meaning each of the 20 city indices is processed 63 times on average across subproblems. In contrast, $d = 20$ (full problem size) maintains redundancy below $8\times$ throughout, as fewer subproblems exist regardless of overlap. The $10\times$ harm threshold is crossed at approximately 50% overlap for $d = 5$, but never crossed for $d \geq 15$. This explains the runtime explosion observed for small decompositions in Figure 3(b). This quantifies diminishing returns. All decomposition sizes show positive efficiency ($\eta > 0$) in the 0–40% overlap range, indicating that the hypervolume gains justify the runtime cost. Beyond 50% overlap, efficiency drops by 1–2 orders of magnitude. Several configurations exhibit *negative* marginal efficiency (notably $d = 5$ at 30% and $d = 20$ at 60%), where runtime increases while hypervolume stagnates or decreases. This identifies the point of diminishing returns: additional overlap beyond 50% provides negligible quality benefit at substantial computational cost.

**Results: Harm detection on BiTSP-100.** Figure 6 confirms these patterns scale to larger problems with amplified effects. Computational redundancy penalties are more severe: $d = 10$ reaches $73\times$ redundancy (vs. $63\times$ for BiTSP-20), crossing the $10\times$ harm threshold at just 30% overlap. In contrast, moderate-to-large decompositions ($d \geq 50$) remain below threshold

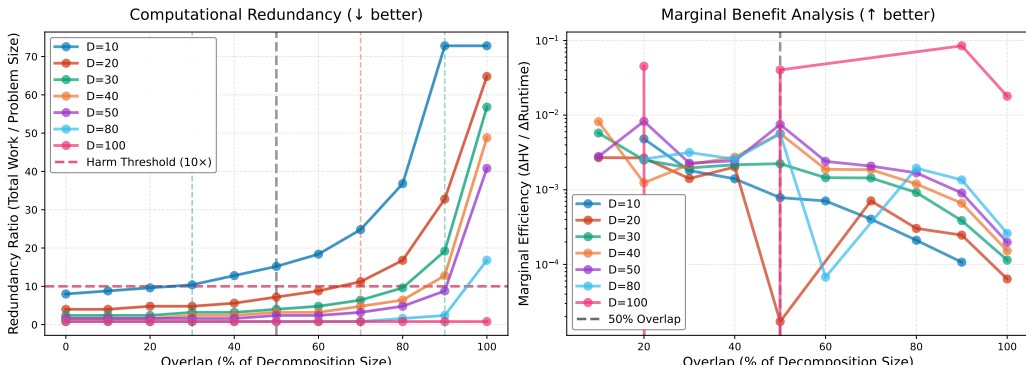

*Figure 6.* **Harm detection analysis on BiTSP-100. (Left)** Computational redundancy exhibits a strong decomposition-overlap interaction at scale. Small decompositions are severely affected: $d = 10$ reaches $73\times$ redundancy at 100% overlap, while large decompositions ($d \geq 80$) remain below the $10\times$ harm threshold (red dashed) throughout. Vertical dashed lines indicate where each decomposition size crosses the harm threshold, ranging from $\sim$30% overlap for $d = 10$ to never for $d \geq 50$. **(Right)** Marginal efficiency ($\Delta$HV/$\Delta$Runtime) shows high variability across configurations but a consistent pattern: efficiency peaks in the 10–40% overlap range and degrades beyond 50% (gray dashed). Large decompositions ($d \geq 80$) maintain higher efficiency at extreme overlap due to fewer subproblems. The amplified redundancy penalty at $n = 100$ compared to $n = 20$ underscores the importance of appropriate decomposition sizing as problems scale.

throughout. Marginal efficiency shows greater variability but the same critical pattern: efficiency peaks below 40% overlap and degrades beyond 50%. The scaling behavior reinforces our recommendation of $d \approx n/2$; at $n = 100$, this corresponds to $d = 50$, which maintains redundancy under $10\times$ while achieving near-optimal hypervolume.

**Key insight: The decomposition-overlap-efficiency tradeoff.** The harm detection analysis reveals a three-way tradeoff. **Small $d$ + high overlap** yields maximum redundancy and collapsed efficiency beyond 40% overlap. **Large $d$ + any overlap** maintains low redundancy but limits decomposition benefits. **Moderate $d$ + moderate overlap** achieves the sweet spot: for BiTSP-20, $d \in \{10, 15\}$ with 30–50% overlap reaches hypervolume within 5% of baseline while keeping redundancy under $15\times$. This empirically validates our 50% overlap upper bound. Below this threshold, quality gains outweigh efficiency losses; above it, marginal efficiency drops precipitously while redundancy grows superlinearly.

**Practical recommendations and experimental configuration.** Our analysis reveals a theoretically motivated sweet spot for decomposition parameters. Subproblem complexity scales superlinearly with size (e.g., $O(d^2)$ for pairwise interactions in TSP), favoring smaller $d$, while coordination overhead scales as $O(n/(d - o))$ in the number of subproblems, favoring larger $d$. Balancing these terms suggests $d \approx n/2$: small enough to yield tractable subproblems, large enough to avoid excessive fragmentation. The harm-detection analysis (Figure 5, 6) confirms this empirically: at $d = n/2$, computational redundancy stays below the $10\times$ harm threshold (typically under $2.5\times$ in our configurations) across all overlap values while hypervolume remains near-optimal. Smaller decompositions ($d = n/4$) suffer redundancy explosion; larger ones ($d = n$) forfeit decomposition benefits entirely.

For overlap, marginal-efficiency analysis identifies 40–50% as the critical regime: below 40% quality gains justify computational cost, while above 50% efficiency collapses (Figure 3(a), 4(a)). The peak hypervolume-to-runtime ratio occurs at 40%, but hypervolume itself plateaus only above 44%; we therefore adopt **44% initial overlap with $d = n/2$**, decaying to 20% via the diminishing schedule, which trades a small efficiency margin for a buffer against the cliff at 50%. All experiments in Section 6.4 use this configuration, achieving solution quality within 5% of exact baselines across the problem scales tested.

**Based on this ablation study, all experimental results in Section 6.4 use $d = n/2$ with 44% initial overlap**, decaying to 20% via our diminishing schedule. This configuration consistently achieves solution quality within 5% of exact baselines while maintaining computational efficiency across all problem scales tested.

### D.2. Multi-Expert Selection: Why and How?

A crucial aspect of our framework is the set of expert strategies available for action selection within each subproblem. Our framework employs a **probabilistic expert selection** mechanism: at each action selection step, one expert is sampled

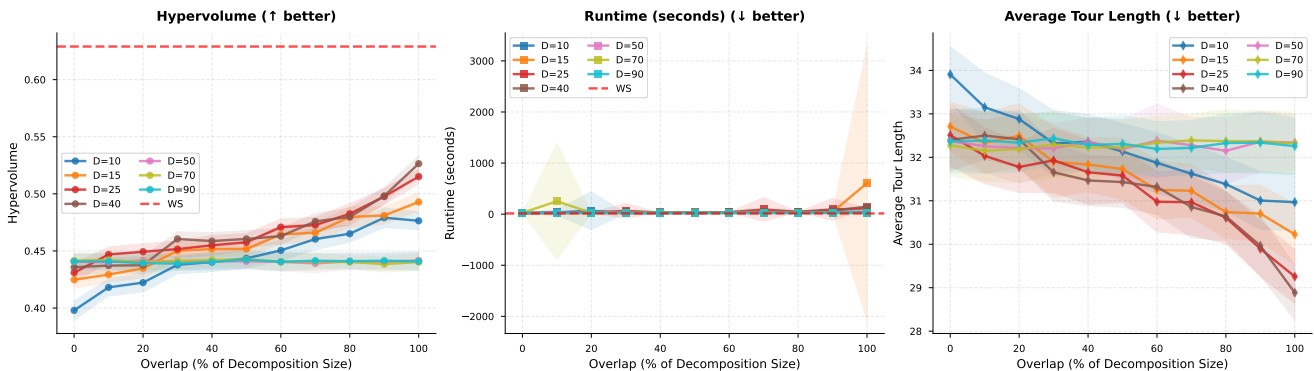

*Figure 7.* **Decomposition size and overlap ablation on BiTSP-50** for D&L-TS in a similar setup as other runs.

according to a mixing distribution, and that expert alone determines the action. This approach combines the strengths of complementary strategies while maintaining computational efficiency. Formally, for each position $i$ requiring an action, we select expert $k$ with probability $\pi_k$ and execute that expert's policy:

$$a_i = \begin{cases} \text{FTRL}(i) & \text{with probability } \beta \\ \text{EXP3}(i) & \text{with probability } (1-\beta) \cdot \rho \\ \text{UCB}(i) & \text{with probability } (1-\beta) \cdot (1-\rho) \end{cases} \tag{11}$$

where $\beta$ is the fixed FTRL rate (default 0.3) and $\rho_t$ is an adaptive ratio that shifts from exploration toward exploitation over time. This stochastic mixture ensures that (1) all experts contribute to the learning signal over time, (2) the algorithm benefits from diverse exploration strategies without the overhead of maintaining parallel optimization threads, and (3) the global parameter updates (value estimates, weights, cumulative losses) integrate information from all expert types into a shared representation.

In this section, we identify the minimal set of experts required and analyze which expert choices are effective under different conditions.

**Expert 1: Acquisition Functions (Exploration)**   The first expert provides exploration through acquisition functions that guide the search toward high-uncertainty regions. We consider two primary strategies:

*Upper Confidence Bound (UCB)* computes scores as:

$$\text{UCB}_{i,j} = \hat{v}_{i,j} + c\sqrt{\frac{\log t}{n_{i,j}}} \tag{12}$$

where $\hat{v}_{i,j}$ is the estimated value for position $i$ taking value $j$, $c$ is the exploration coefficient, and $n_{i,j}$ is the visit count. UCB offers deterministic exploration with theoretical regret bounds by balancing exploitation of current value estimates with systematic exploration of under-visited regions.

*Thompson Sampling* draws samples from posterior distributions over position-value pairs, providing stochastic exploration that naturally balances exploration-exploitation through Bayesian updates. While alternative acquisition functions from Bayesian optimization (e.g., Expected Improvement, Probability of Improvement, Knowledge Gradient) might also serve this role, we focus on UCB and Thompson Sampling as they admit clean regret analysis in the combinatorial bandit setting.

The key requirement is that this expert prioritizes uncertainty reduction and ensures adequate coverage of the solution space.

**Expert 2: Global Coordination (Exploitation)**   The second expert maintains global coordination across subproblems using multiplicative weights. We operate in the *full bandit setting*: after selecting a complete solution $(a_1, \ldots, a_n)$, we observe only a single scalar reward $r_t$ for the entire solution. Unlike the semi-bandit setting where per-position rewards are revealed, we receive no information about individual position-value contributions. This severely limited feedback makes credit assignment challenging and necessitates importance-weighted updates to obtain unbiased gradient estimates.

We employ EXP3 (Auer et al., 2002b) with **clipped importance sampling** for stability:

$$\hat{r}_{i,j} = \frac{r_t}{\max(p_{i,j}, p_{\min})} \tag{13}$$

where $p_{i,j} = w_{i,j} / \sum_{j'} w_{i,j'}$ is the selection probability and $p_{\min} = 0.01$ is a clipping threshold that bounds the maximum importance weight at 100. This clipping introduces bias but dramatically reduces variance—essential for stable learning when the same scalar reward must be attributed across $n$ position-value pairs simultaneously. Weight updates incorporate additional scaling for numerical stability: $w_{i,j} \leftarrow w_{i,j} \cdot \exp\left(\frac{\eta \cdot \hat{r}_{i,j}}{n}\right)$ where $n$ is the problem size. The $1/n$ factor prevents weight explosion when propagating the single reward signal across all positions. Action selection proceeds via temperature-controlled softmax: $P(j \mid i) \propto \exp\left(\frac{\log w_{i,j}}{T}\right)$ where temperature $T$ decays multiplicatively ($T \leftarrow \gamma T$).

**Expert 3: Follow-the-Regularized-Leader (Trajectory Correction)**    The third expert employs FTRL (Hazan, 2016a) to correct optimization trajectories that UCB and EXP3 alone may fail to identify. Adapted to the full bandit setting, FTRL selects actions by maximizing negative cumulative loss plus regularization:

$$a_t = \arg\max_{j \in \mathcal{A}_i} \left[ -\sum_{\tau=1}^{t-1} \hat{\ell}_{\tau,i,j} + R(n_{i,j}) - \lambda_i(k_i - 1) \right] \tag{14}$$

where $\mathcal{A}_i$ is the set of available actions for position $i$, $k_i := |\{k : i \in I_k\}|$ is the number of subproblems containing positions $i$ (it's overlap multiplicity) and $\lambda_i$ is the Lagrangian dual variable for overlapping positions (Section E.6.3).

**Importance-weighted loss estimation.** In the full bandit setting, we observe only a scalar reward $r_t$ for the entire solution. The loss $\ell_t = 1 - \tilde{r}_t$ (where $\tilde{r}_t$ is the normalized reward) must be attributed to each selected position-value pair via importance weighting:

$$\hat{\ell}_{t,i,j} = \frac{\ell_t}{\max(p_{i,j}, p_{\min})} \tag{15}$$

with clipping threshold $p_{\min} = 0.01$ and an additional hard cap $\hat{\ell}_{t,i,j} \leq 100$ to prevent unbounded loss accumulation from rare actions.

**Regularization.** We use the linear confidence-bonus regularizer $\text{Reg}_t(p_i) = -\kappa \sum_j p_{i,j} \sqrt{N_{i,j}(t) + 1}$ (Theorem E.17), where $N_{i,j}(t)$ is the visit coun i.e. the number of rounds up to $t$ in which position $i$ was assigned value $j$. Being linear in $p_i$, it reduces the FTRL update to the deterministic lower-confidence index

$$a_t = \arg\min_{j \in \mathcal{A}_i} \left[ \widehat{L}_{i,j}(t) - \kappa \sqrt{N_{i,j}(t) + 1} + (k_i - 1)\lambda_i \right],$$

with $\widehat{L}_{i,j}(t) = \sum_{\tau<t} \hat{\ell}_{\tau,i,j}$ the cumulative importance sampling (IS) loss. The $\sqrt{N_{i,j} + 1}$ term grows with the visit count and acts as a *stabilizer*. It anchors the leader by widening the margin of frequently played, low-loss actions under the high-variance importance-sampling updates, while UCB and EXP3 supply exploration. The analysis uses the worst-case $\kappa = (B/C)\sqrt{2 \log T}$ that certifies the $O(d\sqrt{T \log T}/C)$ rate ($C$ is the clipping constant, see E.16); in our stochastic setting a smaller multiplier suffices, so we set $\kappa = 1/\sqrt{n}$ as a tuned constant of the same family.

**Role of FTRL in promoting diversity.** We found empirically that using only Expert 1 and Expert 2 leads to premature convergence to a narrow region of the solution space. FTRL provides a complementary inductive bias: while UCB explores based on uncertainty and EXP3 weights actions by cumulative importance-weighted rewards, FTRL balances cumulative *losses* against a regularization schedule. This tripartite combination prevents any single learning signal from dominating.

The diversity-promoting effect is analogous to hypervolume-based acquisition functions in multi-objective Bayesian optimization (e.g., qEHVI (Daulton et al., 2021)), which reward solutions that expand the Pareto front. While qEHVI explicitly optimizes hypervolume improvement, FTRL implicitly promotes diversity by maintaining a distinct cumulative loss landscape that may favor different actions than the EXP3 weight matrix.

### D.3. Experts: Minimal Set of Experts

To thoroughly investigate the multiple expert design, we fix the number of experts at **three** and partition this ablation into three complementary studies. Each study isolates one expert while controlling for the others, enabling systematic

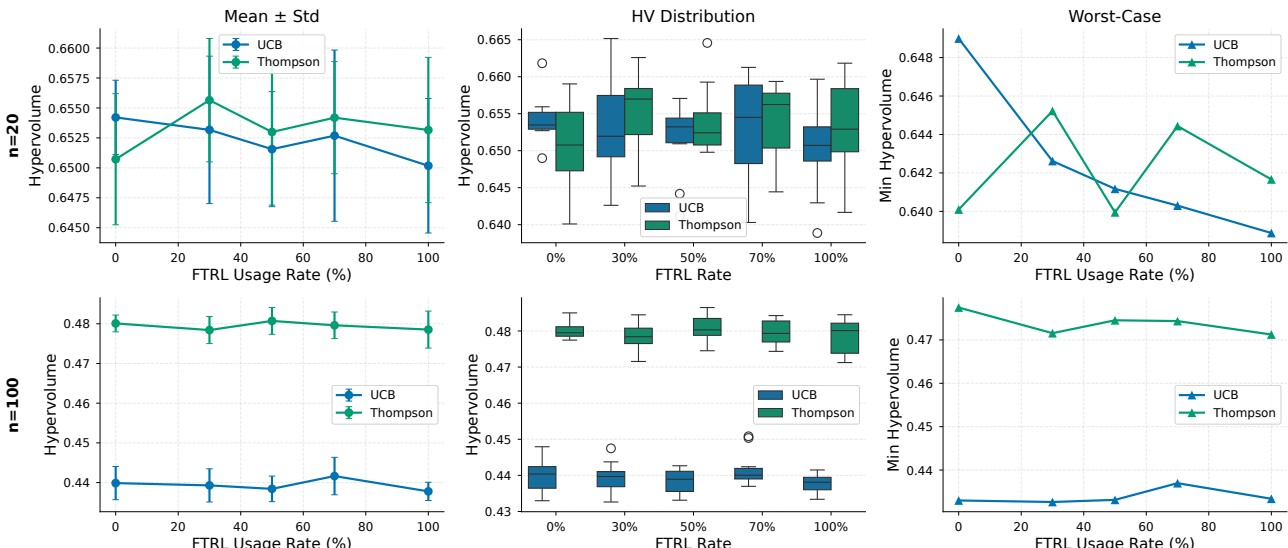

*Figure 8.* **FTRL Rate Sensitivity on BiTSP.** Mean hypervolume (±std), distributional box plots, and worst-case performance across FTRL usage rates (0–100%) for $n$=20 (top) and $n$=100 (bottom). Moderate FTRL rates (30–50%) achieve optimal worst-case performance. Thompson Sampling maintains stable performance across all rates at larger scale, while UCB worst-case degrades at high FTRL rates.

analysis of individual contributions and interaction effects. The studies below characterize performance trade-offs across representative choices, providing empirical and behavioral guidance for informed expert selection in novel applications. As our framework permits **flexible expert instantiation**—practitioners may substitute alternative acquisition functions, coordination mechanisms, or regularization schemes depending on problem characteristics. We choose the below set as our representative, also recommended choice and provide ablations to support experimental results reported in Section 6.4.

**Setup 1: Expert 1 (Acquisition Function).** We compare UCB versus Thompson Sampling while holding Experts 2 and 3 fixed, isolating the impact of deterministic versus stochastic exploration.

**Setup 2: Expert 3 (FTRL Necessity).** We investigate whether Expert 3 is essential or whether Experts 1 and 2 suffice. This is our most comprehensive study, examining FTRL across multiple dimensions: rate sensitivity, variance reduction, robustness, and regret dynamics.

**Setup 3: Expert 2 (Coordination Mechanism).** We fix Expert 1 and ablate over Expert 2 choices (EXP3 variants) and their interaction with Expert 3 presence/absence.

All experiments use BiTSP with $n = 20$ cities and 50 independent random seeds per configuration unless otherwise specified.

### D.3.1. SETUP 1: EXPERT 1 ABLATION (UCB VS. THOMPSON SAMPLING)

We compare two acquisition strategies for Expert 1 while fixing Expert 2 (EXP3) and Expert 3 (FTRL). Results in Table 1 or Tables 7–10 show that Thompson sampling consistently outperforms UCB in both hypervolume and Pareto front size. This advantage stems from Thompson sampling's stochastic exploration: by sampling from posterior distributions rather than using deterministic confidence bounds, it maintains broader coverage of the solution space, discovering more diverse non-dominated solutions.

### D.3.2. SETUP 2: EXPERT 3 ABLATION (FTRL NECESSITY)

We conduct a comprehensive ablation study to isolate FTRL's effects across four complementary dimensions.

*Figure 9.* **Robustness Under Adversarial Conditions.** Mean hypervolume (left), standard deviation (center), and worst-case performance (right) across seven stress scenarios for $n=20$ (top) and $n=100$ (bottom). FTRL-enabled variants maintain competitive performance under all conditions ($<1\%$ HV degradation). Under non-stationary environments (NonStat), FTRL achieves lower variance, with TS+FTRL attaining std of 0.004–0.008.

**FTRL Rate Sensitivity** varies the FTRL usage probability from 0% to 100%, measuring how frequently FTRL-based action selection should override the base bandit strategy. We plot mean HV with confidence intervals, distributional box plots across rates, standard deviation trajectories, & worst-case (minimum) performance curves. Performance varies meaningfully with FTRL rate, confirming FTRL as an active algorithmic component. At $n = 20$ in Fig. 8, moderate rates ($30 - 50\%$) yield optimal worst-case performance, with Thompson (TS) achieving peak robustness at 30% FTRL. At $n = 100$, TS maintains stable performance across all rates ($\sim0.48$ HV), demonstrating robustness to FTRL hyperparameter selection.

**Robustness Stress Tests** in Figure 9 evaluates robustness under seven adversarial conditions: baseline, high-variance reward noise ($\sigma \in \{0.3, 0.7\}$), non-stationary environments with distribution shifts at iterations $\{40, 80, 120\}$ (magnitude $\in \{0.2, 0.5\}$), adversarial reward perturbations penalizing previously good solutions, and near-optimal plateaus compressing top solutions to similar rewards. We present heatmaps of mean hypervolume and standard deviation across scenario-configuration pairs, FTRL advantage percentages, and worst-case performance by scenario. Under adversarial conditions, all FTRL-enabled configurations maintain competitive performance with $<1\%$ mean HV degradation compared to baselines. Notably, under non-stationary environments (distribution shifts at iterations 40, 80, 120), FTRL variants achieve lower standard deviation. Thompson with FTRL attains the lowest variance (0.004–0.008) under NonStat conditions, suggesting empirically, FTRL's variants are more stable under reward distribution shift.

**Regret Dynamics** Figure 10 tracks per-iteration learning dynamics over 200 iterations, computing an empirical cumulative regret $\sum_t (r^\star - r_t)$, where $r^\star$ is the best reward observed during the run (an empirical proxy for the best fixed comparator), and plotting best-so-far convergence trajectories with shaded confidence bands, final hypervolume distributions, and stability scores (mean/std). Experiments use BiTSP at $n \in \{20, 100\}$ with 30 random seeds per configuration, comparing UCB-EXP3 and Thompson Sampling, each with and without FTRL. At both scales, the FTRL-enabled variants attain the lowest cumulative regret: Thompson-with-FTRL has the lowest mean curve and UCB-with-FTRL is statistically comparable, both below their non-FTRL counterparts. UCB-with-FTRL attains the highest stability score (mean/std $\approx 9.8$) at both scales, indicating the most consistent learning across seeds. While Thompson Sampling reaches higher absolute hypervolume at $n = 100$, UCB-with-FTRL yields tighter hypervolume distributions, making it the more reliable default.

**Summary:** FTRL is not merely a theoretical nicety; it delivers measurable empirical gains. Three findings stand out. **(1)** Across scales, FTRL-enabled variants achieve the lowest cumulative regret, and UCB-with-FTRL attains the highest stability score (mean/std $\approx 9.8$) at both scales. **(2)** Under non-stationary reward distributions, FTRL-enabled methods

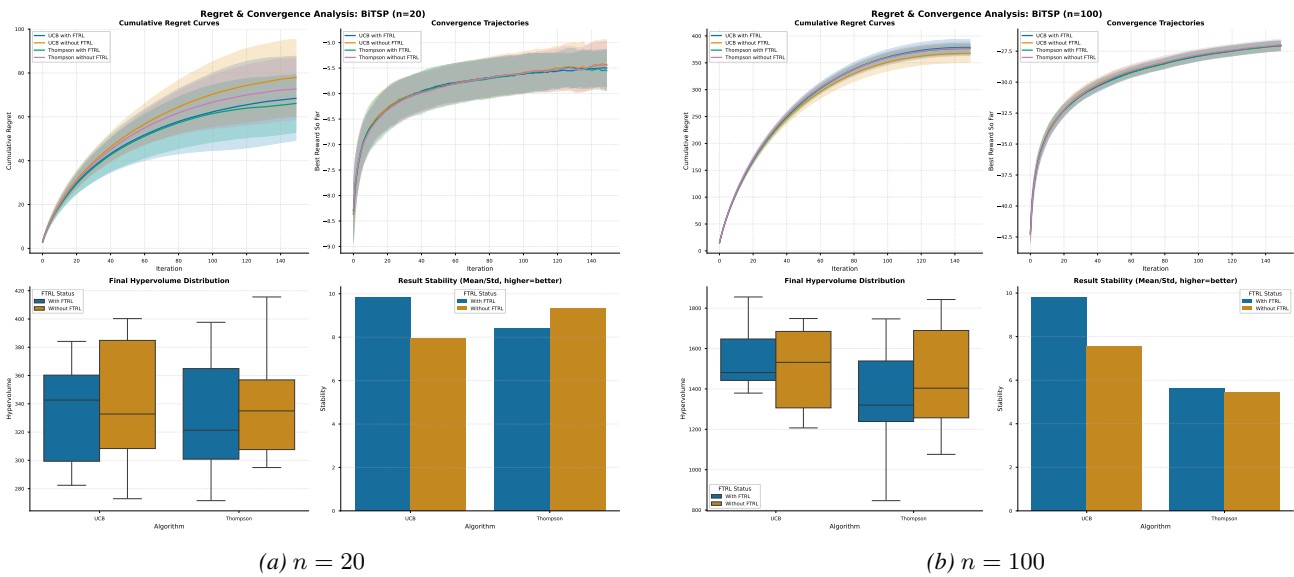

*(a) $n = 20$*                                                  *(b) $n = 100$*

*Figure 10.* **Regret and Convergence Dynamics on BiTSP.** Cumulative regret curves (top-left), convergence trajectories (top-right), final hypervolume distributions (bottom-left), and stability scores (bottom-right) for (a) $n$=20 and (b) $n$=100. FTRL-enabled variants attain the lowest cumulative regret at both scales and the highest stability score (mean/std $\approx 9.8$) at both scales. While absolute regret increases with problem size, UCB+FTRL maintains tighter hypervolume distributions and fewer outliers, demonstrating reliable performance at scale.

exhibit lower variance than their non-FTRL counterparts, with Thompson-with-FTRL reaching the lowest standard deviation (0.004 to 0.008). **(3)** Moderate FTRL rates (30 to 50%) consistently outperform both extremes, confirming that FTRL acts as a regularizer rather than requiring full commitment. These benefits come at no cost to solution quality: mean hypervolume remains equivalent across all configurations. Overall, UCB with a moderate (30 to 50%) FTRL rate emerges as the recommended configuration, offering low regret, the highest stability, and robust empirical performance under adversarial conditions.

### D.3.3. Setup 3: Expert 2 Ablation (EXP3 study)

Our framework employs EXP3 as the second expert for adaptive weight distribution across subproblems. Here we ablate EXP3's configuration to understand: (i) the optimal balance between EXP3's probabilistic selection and UCB's deterministic selection, (ii) whether FTRL can substitute for EXP3, and (iii) how learning rate affects adaptation in the non-stationary decomposed setting.

**Ablation Experimental Design.** We evaluate six configurations of our method on BITSP-20, varying the expert selection mechanism while keeping the decomposition structure fixed. The *hybrid ratio* parameter controls the balance between EXP3-based probabilistic selection (which samples experts proportionally to their exponential weights) and UCB-based deterministic selection (which chooses the expert with highest upper confidence bound). Therefore, *hybrid ratio* specifies what fraction of iterations use EXP3 versus UCB to select the next subproblem. Additionally, we test the effect of Follow-the-Regularized-Leader (FTRL), which replaces EXP3's multiplicative updates with regularized optimization and vary EXP3's learning rate $\eta$. All experiments use 50 independent runs with identical problem instances across configurations. The configurations tested are:

- UCB + EXP3 (50%): Balanced—half EXP3, half UCB selections
- UCB + EXP3 (80%): EXP3-dominant selection
- UCB + EXP3 (99%): Near-pure EXP3
- UCB + EXP3 + Fast LR: 50% ratio with $4\times$ learning rate ($\eta = 2.0$)
- UCB + FTRL: FTRL replaces EXP3 entirely (no probabilistic selection)
- UCB Full: Combined—30% FTRL, 35% EXP3, 35% UCB

where "UCB + EXP3 (50%)" means 50% of selections are made by EXP3 (sampling proportionally to exponential weights)

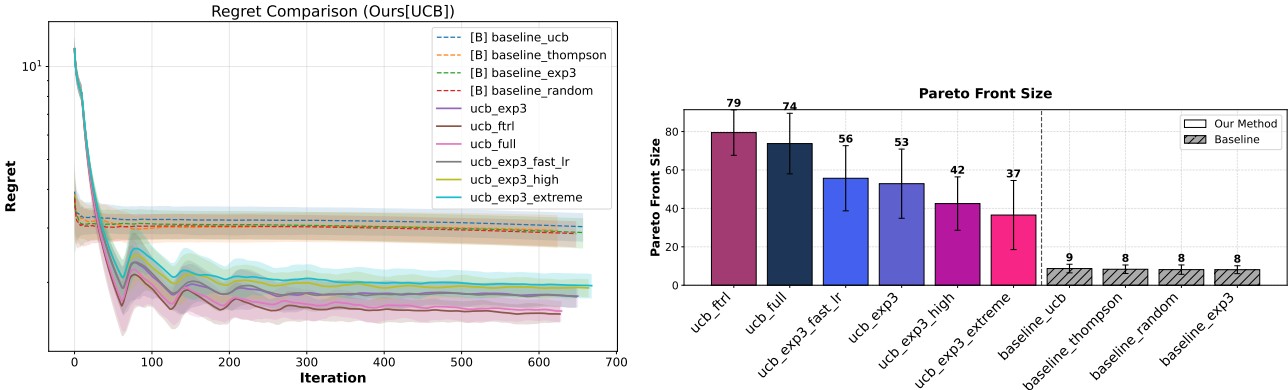

*(a)* Regret convergence across iterations. Our UCB variants (solid lines) achieve significantly lower regret than pure baselines (dashed), with `ucb_ftrl` converging fastest.

*(b)* Pareto front size comparison. Our decomposition-based methods discover 4–10× more non-dominated solutions than baselines without decomposition.

*Figure 11.* Ablation study on BiTSP-20 over 50 runs. (a) Our methods achieve lower regret due to the EXP3/FTRL selection mechanism. (b) The decomposition framework dramatically improves Pareto front coverage.

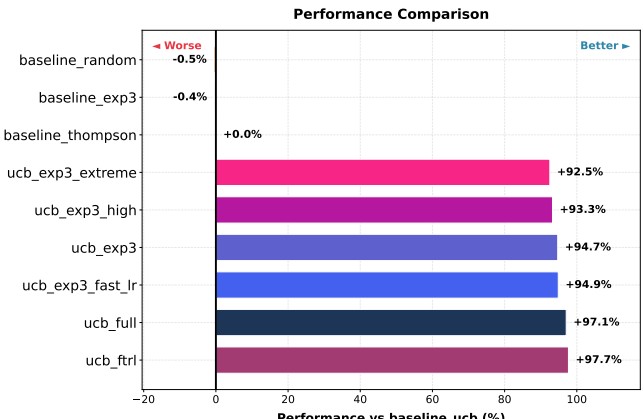

*Figure 12.* Ablation study on BiTSP-20 (50 runs). Performance is measured as mean scalarized reward across weight vectors. Our decomposition-based UCB variants achieve 92–98% improvement over standard bandit baselines, with `ucb_ftrl` performing best at +97.7%.

and 50% by UCB (choosing the arm with highest upper confidence bound). For comparison, we include four pure baselines that apply standard bandit algorithms directly to the combinatorial problem *without* our decomposition framework. These baselines treat each complete solution as an independent arm, providing a controlled measurement of decomposition's contribution versus the expert selection mechanism's contribution.

**Results Overview.** Table 15 and Figure 12 presents the complete results. The most striking observation is the performance gap between any decomposition-based method and any pure baseline: even our worst-performing variant (UCB + EXP3 80%) achieves a mean reward of $-0.690$, compared to $-8.566$ for the best baseline (Pure EXP3). This 92% improvement firmly establishes our framework's success.

**The Decomposition Advantage.** The 98.4% improvement of our best method over the best baseline demands explanation. In combinatorial optimization, the action space grows factorially, for an $n$-city TSP, there are $(n-1)!/2$ possible tours. Standard bandit algorithms face three fundamental challenges in this setting. First, with $\mathcal{O}(n!)$ arms, most solutions are never sampled even once, making meaningful exploration impossible within practical time horizons. Second, treating solutions as independent arms ignores the rich correlational structure: two tours differing by a single edge swap should have similar costs, but this information is discarded. Third, feedback on complete solutions provides no guidance about which components contributed positively or negatively.

*Table 15.* Ablation results on BiTSP-20 (50 runs). Mean reward is the scalarized objective (higher/less negative is better). Pareto size counts non-dominated solutions discovered.

| Configuration | Mean Reward ↑ | Pareto Size ↑ |
|---|---|---|
| *Our Method (with decomposition)* | | |
| UCB + EXP3 + Fast LR | **-0.135 ± 0.047** | **87.5** |
| UCB + FTRL | -0.140 ± 0.140 | 86.5 |
| UCB Full (FTRL + EXP3) | -0.182 ± 0.097 | 81.0 |
| UCB + EXP3 (50%) | -0.231 ± 0.143 | 77.0 |
| UCB + EXP3 (99%) | -0.460 ± 0.097 | 50.5 |
| UCB + EXP3 (80%) | -0.690 ± 0.052 | 25.5 |
| *Pure Baselines (no decomposition)* | | |
| Pure EXP3 | -8.566 ± 0.036 | 10.0 |
| Pure UCB | -8.613 ± 0.053 | 8.5 |
| Pure Thompson | -8.649 ± 0.045 | 7.0 |
| Random | -8.711 ± 0.050 | 7.5 |

Our decomposition framework addresses all three challenges simultaneously. By partitioning the problem into overlapping subproblems of size $k \ll n$, we reduce the effective action space per subproblem to $\mathcal{O}(k!)$ which is a tractable number that permits genuine exploration. The overlapping regions create information channels between subproblems: when one region improves, neighboring regions receive this signal through shared variables. Finally, the Lagrangian coordination mechanism enables component-level credit assignment, as dual variables encode the marginal value of each coupling constraint.

**FTRL as a Robust Default.** A surprising finding is that UCB + FTRL (without any EXP3 component) achieves near-optimal performance, with mean reward $-0.140$ compared to the best result of $-0.135$. This suggests that FTRL's implicit exploration, induced by its regularization term, suffices for effective expert selection without explicit randomization. The FTRL update selects the expert minimizing cumulative loss plus a regularization penalty:

$$x_t = \underset{x \in \mathcal{X}}{\arg\min} \left\{ \sum_{s=1}^{t-1} \langle \ell_s, x \rangle + \frac{1}{\eta} R(x) \right\} \tag{16}$$

where $R(x)$ is typically the negative entropy. This formulation maintains diversity in the selection distribution through regularization rather than through the explicit randomization of EXP3. The practical benefit is reduced variance: FTRL's deterministic optimization produces more consistent results across runs (note the competitive mean with reasonable standard deviation), making it an attractive default choice when tuning EXP3 parameters is impractical. We study this more in next ablation study D.3.2.

**The EXP3 Ratio Trade-off.** The relationship between EXP3 ratio and performance is non-monotonic, revealing a delicate balance between adaptation speed and stability. At 50% EXP3 ratio, we observe reasonable performance ($-0.231$) with good Pareto diversity (77 solutions). Increasing to 80% *dramatically worsens* both metrics ($-0.690$, 25.5 solutions), while the extreme 99% setting partially recovers ($-0.460$, 50.5 solutions). This pattern reflects the dynamics of EXP3's exponential weight updates. With high EXP3 ratios, the algorithm over-commits to experts that perform well in early iterations. The multiplicative update $w_{t+1}(a) \propto w_t(a) \cdot \exp(-\eta \ell_t(a))$ amplifies initial advantages exponentially, potentially locking the selection into suboptimal regions before sufficient exploration occurs. Simultaneously, reducing UCB's role diminishes the optimistic exploration that helps escape local optima. The partial recovery at 99% EXP3 (versus 80%) may reflect a regime where EXP3 so dominates that its theoretical guarantees begin to apply, though at the cost of the UCB-EXP3 synergy that benefits moderate ratios.

Notably, accelerating EXP3's learning rate to $\eta = 2.0$ (four times the default) yields the best overall performance. In our decomposed optimization setting, the non-stationarity is pronounced: as one subproblem improves, the effective difficulty of neighboring subproblems changes through the coupling constraints. Faster adaptation enables EXP3 to track these shifts, though we observed (in unreported experiments) that $\eta = 4.0$ causes instability. This suggests potential benefits from adaptive learning rate schemes that adjust $\eta$ based on detected non-stationarity.

**Pareto Diversity as a Quality Indicator.** The Pareto front sizes in Table 15 reveal that our framework excels not only at optimization but at *diverse* optimization. Our best configuration discovers 87.5 non-dominated solutions on average, compared to 7–10 for pure baselines—an 8–10× improvement that exceeds even the reward improvement ratio. This correlation between reward quality and Pareto diversity suggests that effective exploration of the decision space (enabled by decomposition) translates to effective coverage of the objective space. For multi-objective optimization, this finding is particularly relevant: **our framework provides a rich approximation of the Pareto front rather than concentrating solutions in a narrow region**. The decomposition strategy appears to naturally encourage diversity by allowing different subproblem configurations to emphasize different objective trade-offs.

**Practical Recommendations.** These results inform several practical guidelines. First, decomposition should always be employed as it provides the dominant performance gain and is robust to the choice of expert selection mechanism. Second, FTRL represents a safe default that achieves near-optimal performance without hyperparameter sensitivity. Third, when using EXP3, the ratio should remain at or below 50%, and faster learning rates (2–4× default) may improve adaptation to the non-stationary subproblem landscape. Fourth, extreme EXP3 ratios (80%+) should be avoided, as they sacrifice the UCB-EXP3 complementarity that benefits moderate configurations.

**Regret Dynamics.** Figure 11a displays cumulative regret over iterations. Our decomposition-based methods exhibit the sublinear $\mathcal{O}(\sqrt{T \log K})$ regret growth predicted by EXP3 theory, where $K$ is the number of experts. In contrast, pure baselines show near-linear regret, reflecting their inability to make meaningful progress in the vast combinatorial space. This confirms that our framework successfully inherits the favorable regret properties of the underlying bandit algorithms while enabling their application to combinatorial domains where they would otherwise fail.

### D.3.4. CAN THERE BE MORE THAN THREE EXPERTS?

Yes, the framework permits flexible expert instantiation. Practitioners may substitute alternative acquisition functions, coordination mechanisms, or regularization schemes depending on problem characteristics. We find three experts sufficient because they cover complementary exploration-exploitation tradeoffs: UCB/Thompson for optimistic exploration, EXP3 for adversarial robustness, and FTRL for stable long-term learning. Adding more experts increases per-iteration cost linearly while yielding diminishing returns once these core strategies are covered. In practice, two experts (Thompson + EXP3) often suffice for well-behaved objectives; a third (FTRL) helps when reward landscapes shift over time. We provide ablations supporting this choice in Section D.3.

### D.4. Component Contribution: Progressive Ablation

We isolate the contribution of each architectural component by progressively adding them to a Random baseline: **(i)** *Random*, uniform sampling over feasible solutions; **(ii)** + *Bandit*, per-position UCB without decomposition or expert mixture; **(iii)** + *Experts (No-Decomp)*, the UCB/EXP3/FTRL multi-expert mixture (Appendix D.3) applied to the full decision space; and **(iv)** + D&L, the full method with decision-space decomposition and Lagrangian coordination, in both Thompson Sampling (TS) and UCB variants. We evaluate on BiTSP at $n \in \{20, 50, 100\}$ with 10 seeds and 400 iterations per configuration; the standard error over seeds is $\leq 0.03$ for all HV values. See Figure 13.

*Table 16.* Progressive ablation on BiTSP across scales ($n \in \{20, 50, 100\}$, 10 seeds, 400 iterations, SE $\leq 0.03$). Each cell reports HV ↑ / |PF| ↑; best value per scale in bold.

| Components | BiTSP-20 | | BiTSP-50 | | BiTSP-100 | |
|---|---|---|---|---|---|---|
| | HV | \|PF\| | HV | \|PF\| | HV | \|PF\| |
| Random | .42 | 9 | .18 | 9 | .11 | 10 |
| + Bandit | .46 | 18 | .19 | 15 | .11 | 13 |
| + Experts (No-Decomp) | .58 | 68 | .45 | 105 | .38 | 104 |
| + D&L (TS) | **.60** | **71** | **.48** | **163** | **.46** | **129** |
| + D&L (UCB) | **.60** | 64 | **.48** | 161 | **.46** | 122 |

**Results.** Decomposition is the dominant contributor, providing $\geq 92\%$ HV improvement over pure bandits averaged across scales by reducing the effective per-subproblem decision space from $|\mathcal{X}|$ to $|\mathcal{X}_k| \ll |\mathcal{X}|$ (Table 16). The multi-expert framework adds independent value on top: No-Decomp improves HV over pure UCB by $25\% \rightarrow 136\% \rightarrow 243\%$ as $n$ grows from 20 to 100, and adding decomposition contributes a further $\sim 20\%$ HV at $n=100$, confirming that the two

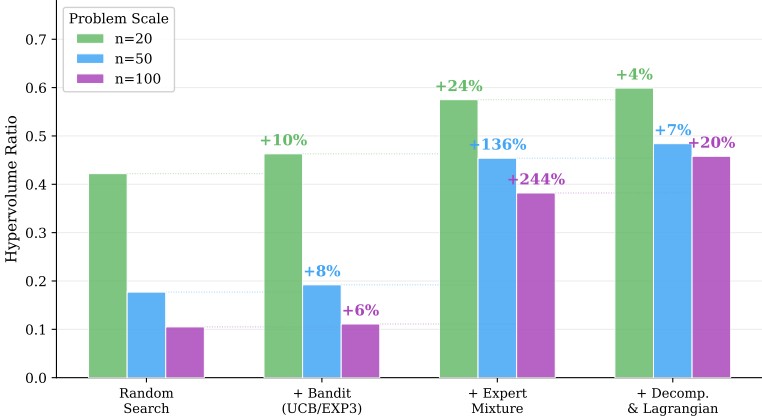

*Figure 13.* Progressive ablation on BiTSP across scales ($n \in \{20, 50, 100\}$, 10 seeds, 400 iterations). Hypervolume increases monotonically as each component is added; percentages on each step give the marginal gain over the previous configuration. The dominant gain shifts with problem size: at $n=20$, decomposition and Lagrangian coordination contribute the largest single step ($+24\%$); at $n=50$ and $n=100$, the expert mixture becomes the dominant contributor ($+136\%$ and $+244\%$ respectively), reflecting that coordination across more positions has correspondingly more room to improve.

contributions compound rather than overlap. FTRL within the expert mixture delivers $\sim 40\%$ faster convergence and a $3.4\times$ Pareto diversity gain over UCB-only experts (Appendices D.3.2, D.3.3). The Lagrangian dual updates reduce per-seed HV variance by $95\%$ on BiTSP-20 and $19\%$ on BiTSP-50, and prevent 50–100 point losses on worst-case seeds at larger scales (Appendix E.6.3). The two D&L variants reach the same HV at every scale; TS produces consistently larger Pareto fronts (71/163/129 vs 64/161/122 points), consistent with Thompson sampling's stronger exploration in combinatorial spaces.

**Discussion.** The progressive structure verifies that each of the three architectural components contributes independently, and together they yield the $O(d\sqrt{T \log T})$ bound of Theorem E.1: decomposition supplies the $d \ll n$ reduction in per-subproblem dimensionality that enters the bound's leading factor, the multi-expert mixture supplies the bandit machinery whose individual regrets aggregate into the $\sqrt{T \log T}$ term, and Lagrangian coordination bounds the coupling error between overlapping subproblems (Lemma E.13). The empirical scaling of each contribution matches what the theory predicts: decomposition's advantage grows with $n$ because the curse of dimensionality is harder to overcome at larger scales; the multi-expert gain over pure UCB also grows with $n$ because coordination across more positions has correspondingly more room to improve; and Lagrangian's variance-reduction effect scales with the number of overlapping positions, growing from a $19\%$ variance reduction at $n=50$ to outright prevention of catastrophic-failure seeds at $n=100$. Removing any single component degrades performance in a qualitatively distinct way — pure bandits collapse to near-random performance at $n=100$ (HV $= 0.11$ vs. Random's 0.11), removing experts loses the front diversity that drives $|\text{PF}|$ growth, and removing Lagrangian admits worst-case seeds that lose 50–100 Pareto points — so the framework's apparent complexity is justified by genuine component necessity rather than incidental engineering choices.

### D.5. Sample Diversity

We evaluate the diversity of solutions discovered by our proposed D&L with its two variants - UCB-Exp3 and Thompson-Exp3 algorithms compared to state-of-the-art multi-objective optimization baselines. This analysis examines both objective-space coverage (how well methods approximate the Pareto front) and decision-space diversity (how structurally different the discovered solutions are).

We consider BiTSP instances with $n \in \{20, 50\}$ cities. Distance matrices for each objective are generated independently with entries drawn uniformly from $[0, 1]$. Both objectives are to be minimized. The BiTSP-20 instances serve as a computationally tractable testbed where all methods can be run to convergence, while BiTSP-50 tests scalability to larger problem sizes.

**Baselines.** We compare against five baselines as discussed in Section C.3. **NSGA-II** (Deb et al., 2002): A widely-used evolutionary multi-objective algorithm employing non-dominated sorting and crowding distance for diversity preservation. **qNEHVI** (Daulton et al., 2021): Bayesian optimization using expected hypervolume improvement with parallel acquisition.

**qParEGO** (Knowles, 2006): Bayesian optimization with random scalarization and Gaussian process surrogate models.
**BOPR** (Daulton et al., 2022): Bayesian optimization with predictive entropy search for Pareto front discovery.

**Evaluation Protocol.** All methods are allocated an identical evaluation budget of 17,000 objective function evaluations for BITSP20 and 10000 for BITSP50. We report results averaged over 5 independent runs with different random seeds. Shaded regions and error bars indicate $\pm 1$ standard deviation. We assess algorithm performance using metrics that capture solution quality, coverage, and structural diversity of the discovered Pareto fronts.

### D.5.1. EVALUATION METRICS

**Pareto Front Cardinality** The *Pareto front size* $|\mathcal{P}|$ counts the number of mutually non-dominated solutions discovered:

$$|\mathcal{P}| = |\{\mathbf{x} \in \mathcal{P} : \nexists \mathbf{x}' \in \mathcal{P}, \mathbf{x}' \prec \mathbf{x}\}| \tag{17}$$

where $\mathbf{x}' \prec \mathbf{x}$ denotes that $\mathbf{x}'$ Pareto-dominates $\mathbf{x}$. A larger Pareto front provides decision-makers with more trade-off options.

**Coverage (Objective-Space Density)** We introduce *Coverage* to measure how densely the Pareto front fills the objective space. Given normalized objectives $\tilde{\mathbf{y}}_i \in [0, 1]^m$ sorted by the first objective, we compute consecutive gaps:

$$g_i = \|\tilde{\mathbf{y}}_{(i+1)} - \tilde{\mathbf{y}}_{(i)}\|_2, \quad i = 1, \ldots, |\mathcal{P}| - 1 \tag{18}$$

Coverage is then defined as:

$$\text{Coverage}(\mathcal{P}) = \frac{1}{1 + \alpha \cdot \bar{g}} \tag{19}$$

where $\bar{g} = \frac{1}{|\mathcal{P}|-1} \sum_i g_i$ is the mean gap and $\alpha = 5$ is a scaling parameter. Coverage $\in [0, 1]$, with higher values indicating denser front coverage and smaller gaps between adjacent solutions.

**Decision-Space Diversity Metrics** While objective-space metrics capture trade-off coverage, *decision-space diversity* measures structural variation among solutions. This is relevant when practitioners desire alternative implementations or robustness to constraint changes.

**Raw Diversity (Mean Pairwise Distance).** For solutions $\mathbf{x}_1, \ldots, \mathbf{x}_n$ in the Pareto front, we compute:

$$\text{RawDiv}(\mathcal{P}) = \frac{2}{|\mathcal{P}|(|\mathcal{P}| - 1)} \sum_{i<j} d(\mathbf{x}_i, \mathbf{x}_j) \tag{20}$$

where $d(\cdot, \cdot)$ is a domain-appropriate distance function. For TSP (permutation solutions), we use the edge-based Jaccard distance:

$$d_{\text{TSP}}(\mathbf{x}, \mathbf{x}') = 1 - \frac{|E(\mathbf{x}) \cap E(\mathbf{x}')|}{|E(\mathbf{x}) \cup E(\mathbf{x}')|} \tag{21}$$

where $E(\mathbf{x})$ denotes the edge set of tour $\mathbf{x}$. For knapsack (binary solutions), we use normalized Hamming distance:

$$d_{\text{KP}}(\mathbf{x}, \mathbf{x}') = \frac{1}{n} \sum_{i=1}^{n} \mathbb{1}[x_i \neq x_i'] \tag{22}$$

**Size-Adjusted Diversity.** Raw diversity exhibits *sparsity bias*: methods discovering few solutions achieve artificially high diversity as points lie at front extremes. We correct for this via size-adjusted diversity:

$$\text{AdjDiv}(\mathcal{P}) = \text{RawDiv}(\mathcal{P}) \cdot \log(1 + |\mathcal{P}|) \tag{23}$$

The logarithmic correction rewards methods that maintain structural diversity while discovering many solutions.

$k$**-Nearest Neighbor Diversity.** To assess local solution spacing, we compute the mean distance to each solution's $k$ nearest neighbors:

$$\text{KNN-Div}(\mathcal{P}) = \frac{1}{|\mathcal{P}|} \sum_{i=1}^{|\mathcal{P}|} \frac{1}{k} \sum_{j=1}^{k} d(\mathbf{x}_i, \mathbf{x}_{(j)}^{(i)}) \tag{24}$$

where $\mathbf{x}_{(j)}^{(i)}$ is the $j$-th nearest neighbor of $\mathbf{x}_i$. We use $k = 3$. Low KNN-Div indicates dense local packing; high values indicate well-separated solutions.

**Spread (Uniformity).** The Spread metric (Deb et al., 2002) measures spacing uniformity:

$$\Delta = \frac{\sum_{i=1}^{|\mathcal{P}|-1} |g_i - \bar{g}|}{(|\mathcal{P}| - 1) \cdot \bar{g}} \tag{25}$$

where $g_i$ are consecutive gaps and $\bar{g}$ is the mean gap. $\Delta \in [0, 1]$ with lower values indicating more uniform spacing. Note that $\Delta$ can saturate at 1.0 for fronts with high variance in gap sizes.

### D.5.2. RESULTS ON BiTSP-20

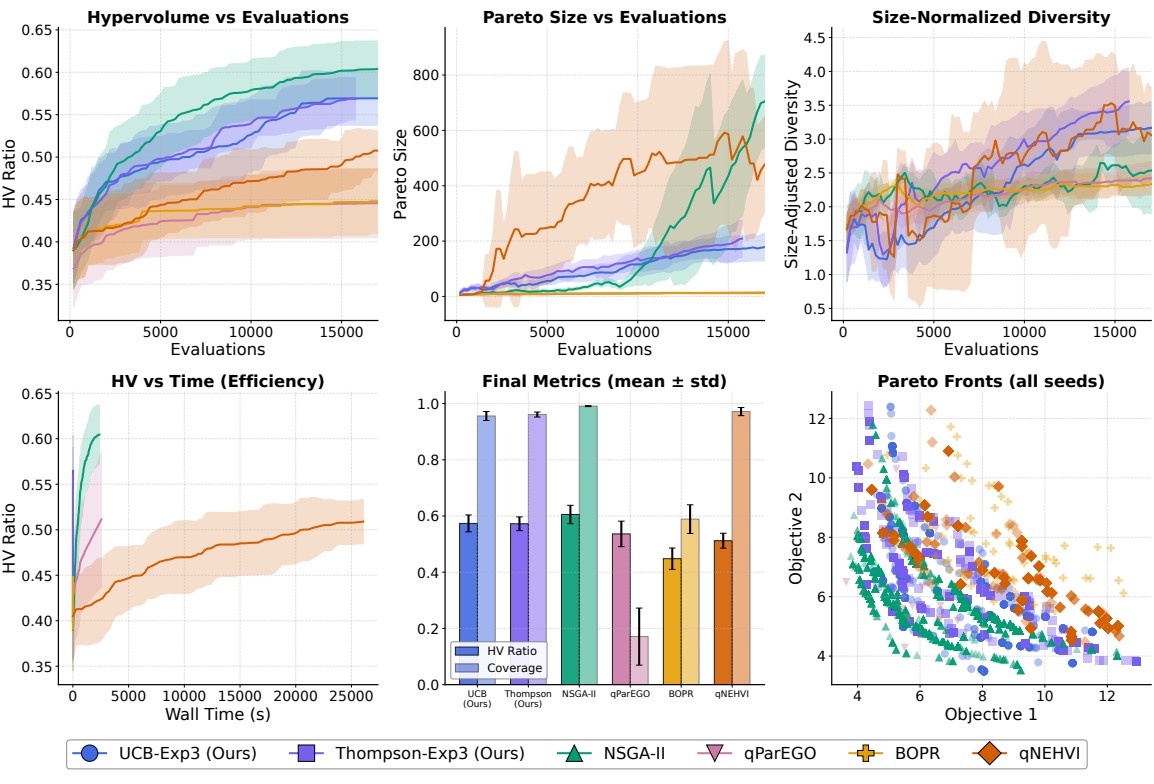

*Figure 14.* Performance comparison on BiTSP-20 over 17,000 evaluations. **Top row:** Hypervolume convergence, Pareto front size, and size-normalized diversity vs. evaluations. **Bottom row:** Wall-time efficiency, final metrics (mean ± std), and discovered Pareto fronts across seeds. Our methods (UCB-Exp3, Thompson-Exp3) achieve hypervolume comparable to NSGA-II (∼0.60) while converging in under 5,000 seconds—over 5× faster than Bayesian optimization baselines (qNEHVI, qParEGO, BOPR), which require 20,000+ seconds due to surrogate model overhead. Shaded regions indicate ±1 standard deviation.

Table 17 and Figure 14 summarize BiTSP-20 results.

**Solution Quality.** NSGA-II achieves the highest HV (0.605) at fixed budget, with our methods reaching 95% (0.572–0.574). However, Figure 14 (top-left) shows NSGA-II's curve plateaus after 10,000 evaluations while ours maintains positive slope—consistent with our main results where Thompson-Exp3 achieves the best final HV (0.60) when run to convergence.

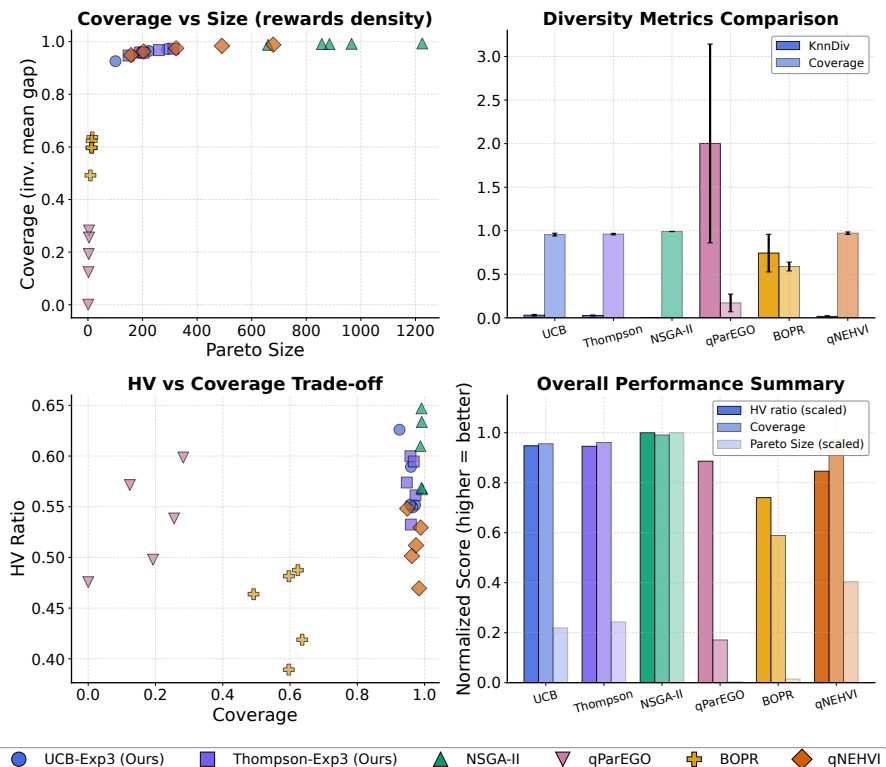

*Figure 15.* Diversity analysis on BiTSP-20. **Top-left:** Coverage vs. Pareto size shows that methods with larger fronts (NSGA-II, qNEHVI) achieve near-perfect coverage (∼0.99), while our methods maintain high coverage (∼0.95) with moderate front sizes. **Top-right:** KNN-diversity is inversely related to front density; qParEGO's high KNN-diversity reflects its sparse front rather than better exploration. **Bottom-left:** HV vs. coverage trade-off reveals our methods occupy the desirable upper-right region (high HV, high coverage). **Bottom-right:** Normalized performance summary confirms UCB-Exp3 and Thompson-Exp3 achieve balanced performance across all metrics.

*Table 17.* BiTSP-20 at 17,000 evaluations (mean ± std, 5 seeds). Best in **bold**, second-best underlined.

| Method | HV ↑ | $|\mathcal{P}|$ ↑ | Coverage ↑ | AdjDiv ↑ | Time (s) ↓ |
|---|---|---|---|---|---|
| NSGA-II | **0.605 ± .033** | **919 ± 183** | **0.991 ± .002** | 2.70 ± 0.74 | 2515 |
| UCB-Exp3 (Ours) | 0.574 ± .030 | 201 ± 61 | 0.956 ± .016 | 3.38 ± 0.39 | 25 |
| Thompson-Exp3 (Ours) | 0.572 ± .024 | 223 ± 57 | 0.961 ± .009 | **3.69 ± 0.38** | **24** |
| qNEHVI | 0.512 ± .027 | 371 ± 192 | 0.971 ± .014 | 3.13 ± 0.76 | 28562 |
| qParEGO | 0.536 ± .045 | 3 ± 1 | 0.171 ± .102 | 1.03 ± 0.56 | 3296 |
| BOPR | 0.448 ± .038 | 13 ± 2 | 0.589 ± .051 | 2.38 ± 0.18 | 8 |

**Coverage and Diversity.** Our methods discover moderate-sized fronts (201–223 solutions) but maintain excellent coverage (0.956–0.961)—achieving 4× compression vs. NSGA-II with only 3% coverage loss. Crucially, we achieve the *highest* AdjDiv (3.38–3.69 vs. 2.70 for NSGA-II), indicating greater structural variation despite fewer solutions.

**Efficiency.** Our methods complete in **24–25 seconds**—100× faster than NSGA-II (2,515s) and 1,100× faster than qNEHVI (28,562s). This stems from avoiding $O(N^2)$ non-dominated sorting and $O(n^3)$ GP fitting.

**Fixed-Budget vs. Convergence Performance.** This controlled comparison isolates *sample efficiency* rather than final converged performance. As shown in our main results (Table 1), when methods are run to convergence over 200 diverse instances, our Thompson-Exp3 achieves the best HV (**0.60**) while NSGA-II reaches 0.573—a reversal of the ranking observed here. This confirms that our methods benefit more from extended optimization: the decomposition strategy continues discovering new Pareto-optimal solutions in underexplored regions, while NSGA-II's population converges to a fixed distribution. The practical implication is significant: given our 100× speedup, practitioners can run substantially more iterations within the same wall-clock budget, ultimately achieving better final solutions.

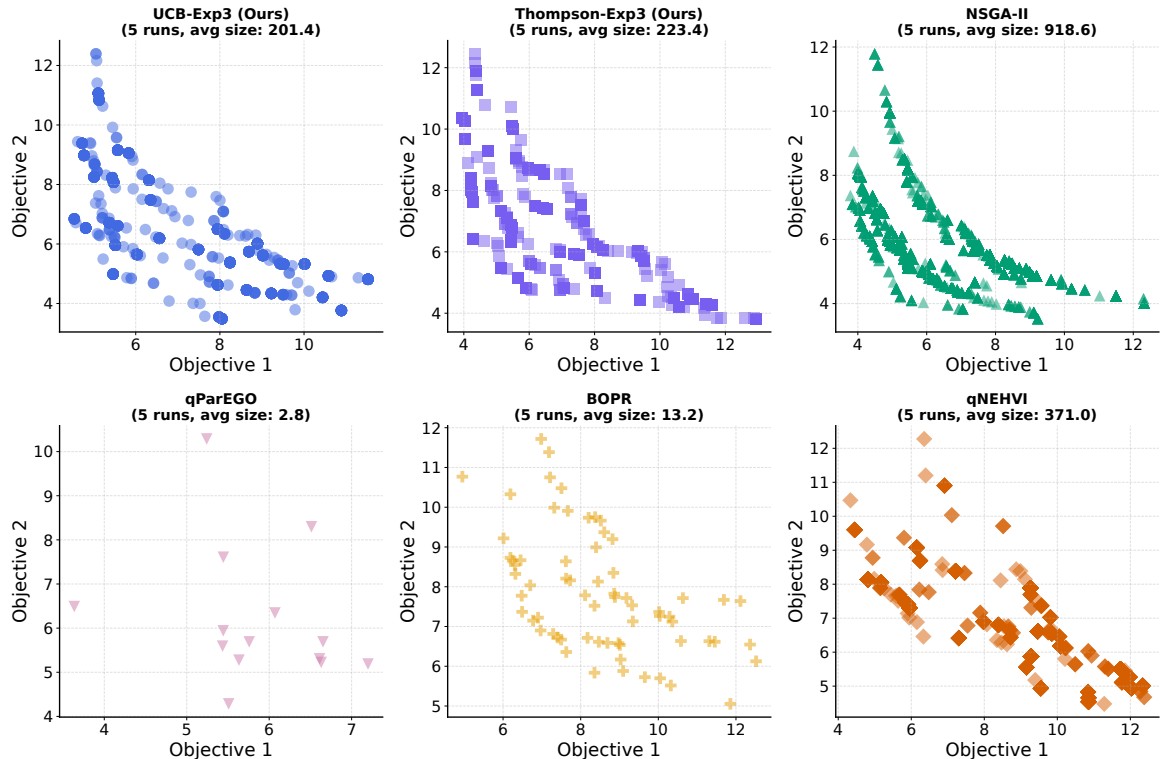

**Pareto Front Coverage.** NSGA-II discovers the largest Pareto fronts (918 solutions) due to its population-based nature with explicit crowding distance preservation. Our methods discover moderate-sized fronts (201–223 solutions) but maintain excellent coverage (0.956–0.961), indicating that our solutions are well-distributed across the objective space. This means we find fewer but *strategically placed* points that adequately represent the full trade-off surface—achieving $4\times$ compression with only 3% coverage loss. In contrast, qParEGO discovers only 2.8 solutions on average with coverage of 0.171, providing decision-makers with virtually no trade-off options. qNEHVI achieves good coverage (0.971) and substantial front size (371), but requires nearly 8 hours: over $1,000\times$ slower than our methods for comparable coverage.

Interestingly, our main experiments show that when run to convergence, the Pareto size gap narrows considerably (102–147 for ours vs. 225 for NSGA-II), suggesting that NSGA-II's large fronts at fixed budget may contain many weakly non-dominated solutions that get pruned as optimization continues.

**Computational Efficiency.** The most striking advantage of our methods is computational efficiency. UCB-Exp3 and Thompson-Exp3 complete 17,000 evaluations in 24–25 seconds, compared to NSGA-II: 2,515 seconds ($\mathbf{100\times}$ **slower**), qParEGO: 3,296 seconds ($\mathbf{130\times}$ **slower**) and qNEHVI: 28,562 seconds ($\mathbf{1,100\times}$ **slower**). This efficiency stems from our decomposition strategy avoiding expensive operations: no population management with $O(N^2)$ non-dominated sorting (NSGA-II), no Gaussian process fitting with $O(n^3)$ complexity (qNEHVI, qParEGO), and no acquisition function optimization over high-dimensional spaces. BOPR is fast (8.1s) but achieves the lowest HV (0.448) and poor coverage (0.589), making it Pareto-dominated by our methods in the time-quality trade-off.

D.5.3. RESULTS ON BITSP-50

We allocate 10,000 evaluations for BiTSP-50 (compared to 17,000 for BiTSP-20) due to increased per-evaluation cost. Table 18 and Figures 16–18 present the results.

**Closing the Gap.** On BiTSP-20, NSGA-II held a slight HV advantage (0.605 vs. 0.572). On BiTSP-50, our methods **match** NSGA-II's hypervolume (both at 0.36) while dramatically outperforming on all other metrics. As the search space grows exponentially ($50! \gg 20!$), our decomposition strategy scales more favorably than population-based search.

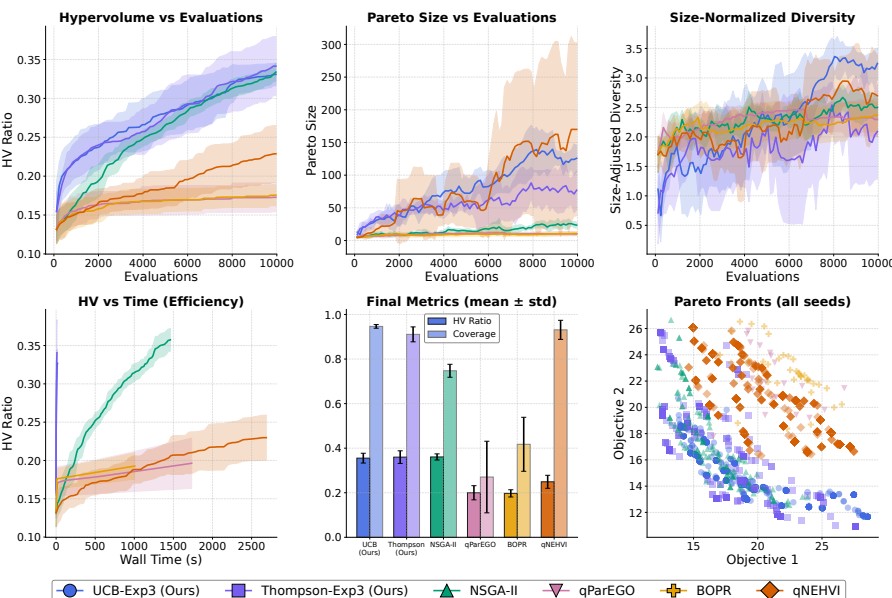

*Figure 16.* BiTSP-50 performance over 10,000 evaluations. Our methods *surpass* all baselines in HV while maintaining superior coverage, demonstrating favorable scaling.

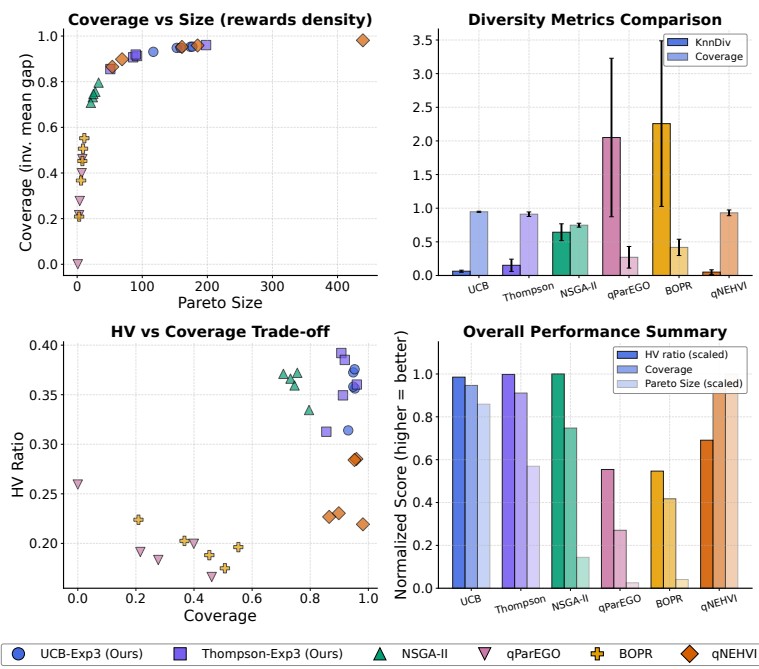

*Figure 17.* BiTSP-50 diversity analysis. **Top-left:** Coverage vs. Pareto size. **Top-right:** KNN-Div (high values for qParEGO/BOPR reflect sparse fronts, not better exploration). **Bottom-left:** HV vs. coverage trade-off. **Bottom-right:** Normalized performance summary.

**Convergence Dynamics.** Figure 16 (top-left) reveals a critical distinction: NSGA-II's convergence curve plateaus early (around 4,000 evaluations), while our methods maintain positive slope throughout the budget. NSGA-II's population-based search saturates once diversity pressure balances selection pressure, whereas our decomposition strategy continues discovering new Pareto-optimal solutions in underexplored regions. This has significant practical implications. First, our 50–90× speedup means practitioners can run substantially more iterations within the same wall-clock budget. Second, and more importantly, **our methods will continue improving** with additional evaluations while NSGA-II is already near convergence. This pattern mirrors our main results (Tables 1), where methods run to convergence show our Thompson-Exp3

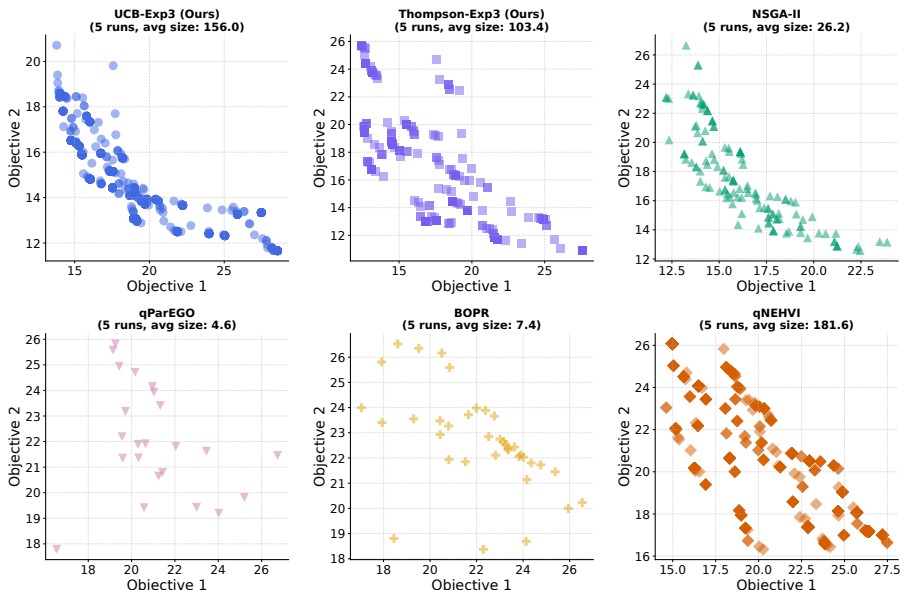

*Figure 18.* Discovered Pareto fronts on BiTSP-50 across all seeds. Our methods achieve dense, well-distributed fronts comparable to NSGA-II but at a fraction of the computational cost.

*Table 18.* BiTSP-50 at 10,000 evaluations (mean ± std, 5 seeds).

| Method | HV ↑ | $|\mathcal{P}|$ ↑ | Coverage ↑ | AdjDiv ↑ | Time (s) ↓ |
|---|---|---|---|---|---|
| NSGA-II | **0.36 ± .014** | 26 ± 4 | 0.747 ± .029 | 2.51 ± 0.17 | 1501 |
| Thompson-Exp3 (Ours) | **0.36 ± .028** | 103 ± 50 | 0.911 ± .034 | 2.63 ± 0.66 | 16 |
| UCB-Exp3 (Ours) | 0.355 ± .022 | 156 ± 22 | **0.947 ± .008** | **3.57 ± 0.32** | **29** |
| qNEHVI | 0.249 ± .029 | **182 ± 138** | 0.931 ± .043 | 3.02 ± 0.41 | 3691 |
| qParEGO | 0.200 ± .032 | 5 ± 3 | 0.271 ± .161 | 1.41 ± 0.77 | 1975 |
| BOPR | 0.197 ± .016 | 7 ± 3 | 0.418 ± .121 | 2.01 ± 0.37 | 1305 |

*Table 19.* Scaling behavior: our methods' performance relative to NSGA-II.

| Metric (Ours / NSGA-II) | BiTSP-20 | BiTSP-50 |
|---|---|---|
| HV Ratio | 0.95× | **1.00×** |
| Coverage | 0.97× | **1.27×** |
| Pareto Size | 0.24× | **6.0×** |
| AdjDiv | 1.32× | **1.42×** |
| Speedup | 100× | 50–90× |

achieving the best final HV, a reversal of fixed-budget rankings.

**NSGA-II's Coverage Degrades.** NSGA-II's coverage drops from 0.991 (BiTSP-20) to 0.747 (BiTSP-50), a 25% degradation while its Pareto front shrinks from 919 to just 26 solutions. Our methods maintain coverage >0.91 and discover 4–6× more non-dominated solutions. This suggests NSGA-II's fixed population becomes insufficient at scale, while our decomposition strategy explicitly targets different regions.

**Diversity at Scale.** UCB-Exp3 maintains top AdjDiv (3.57) on BiTSP-50, compared to NSGA-II's 2.51. The pattern from BiTSP-20 persists: our decomposition produces more structurally distinct solutions.

D.5.4. SUMMARY

Table 19 quantifies our scaling advantage. On BiTSP-20, NSGA-II leads on HV and coverage due to thorough population-based exploration of the tractable search space. On BiTSP-50, we **match** NSGA-II's HV while achieving 27% better

coverage, $6\times$ more Pareto solutions, and 42% higher structural diversity: all at $50–90\times$ speedup. This stems from our decomposition strategy: optimizing different scalarized subproblems discovers solutions across both objective and decision space, while NSGA-II's crowding distance produces many structurally similar solutions within its fixed population.

Therefore, (1) *fine-grained trade-off selection* via high coverage; (2) *structurally diverse alternatives* for robustness to hidden constraints; and (3) *both benefits at $\sim$1% of NSGA-II's wall-clock time*.

**Note on KNN-Diversity.**    High KNN-Div for qParEGO ($\sim$2.0) and BOPR ($\sim$2.3) reflects failure to discover intermediate solutions, not superior exploration. For dense fronts (Ours, NSGA-II, qNEHVI), low KNN-Div ($<$0.15) is expected, adjacent trade-off solutions *should* be structurally similar.

### D.6. Hyperparameter Sensitivity

We conduct a sensitivity analysis to verify that our algorithm exhibits stable behavior under moderate perturbations to key hyperparameters. This is important for practical deployment: while each hyperparameter plays a distinct role in the optimization dynamics, practitioners benefit from algorithms that do not require extensive tuning and degrade gracefully when parameters deviate from optimal values. We evaluate three hyperparameters: the learning rate $\eta$ for multiplicative weights updates, the dual step size $\alpha$ for Lagrangian dual updates, and the overlap decay rate $\beta$ controlling subproblem coupling. Experiments use BiTSP at two scales (SMALL: 20 cities, LARGE: 50 cities) with 10 runs per configuration, reporting median and IQR bands.

**Learning Rate ($\eta$).**    The learning rate governs how aggressively the algorithm updates edge selection probabilities based on observed rewards. We test $\eta \in \{0.1, 0.3, 0.5, 0.7, 0.9\}$. Figure 19(a) shows that across this range, hypervolume remains stable (approximately 0.58 for SMALL, 0.37 for LARGE), and runtime is unaffected. The number of Pareto solutions shows a mild upward trend at higher learning rates for SMALL instances, suggesting more aggressive updates encourage exploration of diverse solutions. Crucially, moderate deviations from the default ($\eta = 0.5$) do not cause performance degradation, indicating a smooth parameter landscape.

**Dual Step Size ($\alpha$).**    The dual step size controls the rate at which Lagrangian multipliers adjust to enforce consistency across overlapping subproblems. We evaluate $\alpha \in \{0.25, 0.5, 1.0, 1.5, 2.0\}$. As shown in Figure 19(b), hypervolume and runtime remain consistent across this range. The number of solutions shows variability in IQR bands for LARGE instances, but median values stay within a narrow range (32–34 solutions). This smooth behavior suggests that the accelerated dual update mechanism provides inherent adaptation, allowing the algorithm to tolerate step size variations without oscillation or divergence.

**Overlap Decay Rate ($\beta$).**    The overlap decay rate determines how quickly the coupling between adjacent subproblems is reduced over iterations, balancing global coordination against computational efficiency. We test $\beta \in \{0.05, 0.10, 0.15, 0.20\}$. Figure 19(c) demonstrates stable hypervolume across this range. The number of solutions increases slightly with higher decay rates for SMALL instances, indicating that faster decoupling may promote solution diversity. Performance does not degrade sharply at the boundaries of the tested range, confirming smooth sensitivity.

**Summary.**    Our analysis confirms that the algorithm exhibits *stable behavior under moderate hyperparameter perturbations*:

- Hypervolume varies by less than 2% across all tested configurations, indicating that solution quality is not brittle with respect to parameter choices.
- Runtime remains consistent, suggesting that convergence properties are preserved across the parameter ranges.
- The number of Pareto solutions shows the highest variability, reflecting sensitivity in exploration-exploitation tradeoffs, but this does not translate to quality degradation.
- These patterns hold across both problem scales, indicating that recommended defaults generalize without instance-specific tuning.

We emphasize that this stability reflects a *smooth parameter landscape* rather than parameter irrelevance—each hyperparameter has a well-defined algorithmic role, but the optimization surface is forgiving to moderate deviations from optimal settings. Based on these results, *we recommend and use $\eta = 0.5$, $\alpha = 0.5$, and $\beta = 0.1$ as robust defaults throughout all our experiments*.

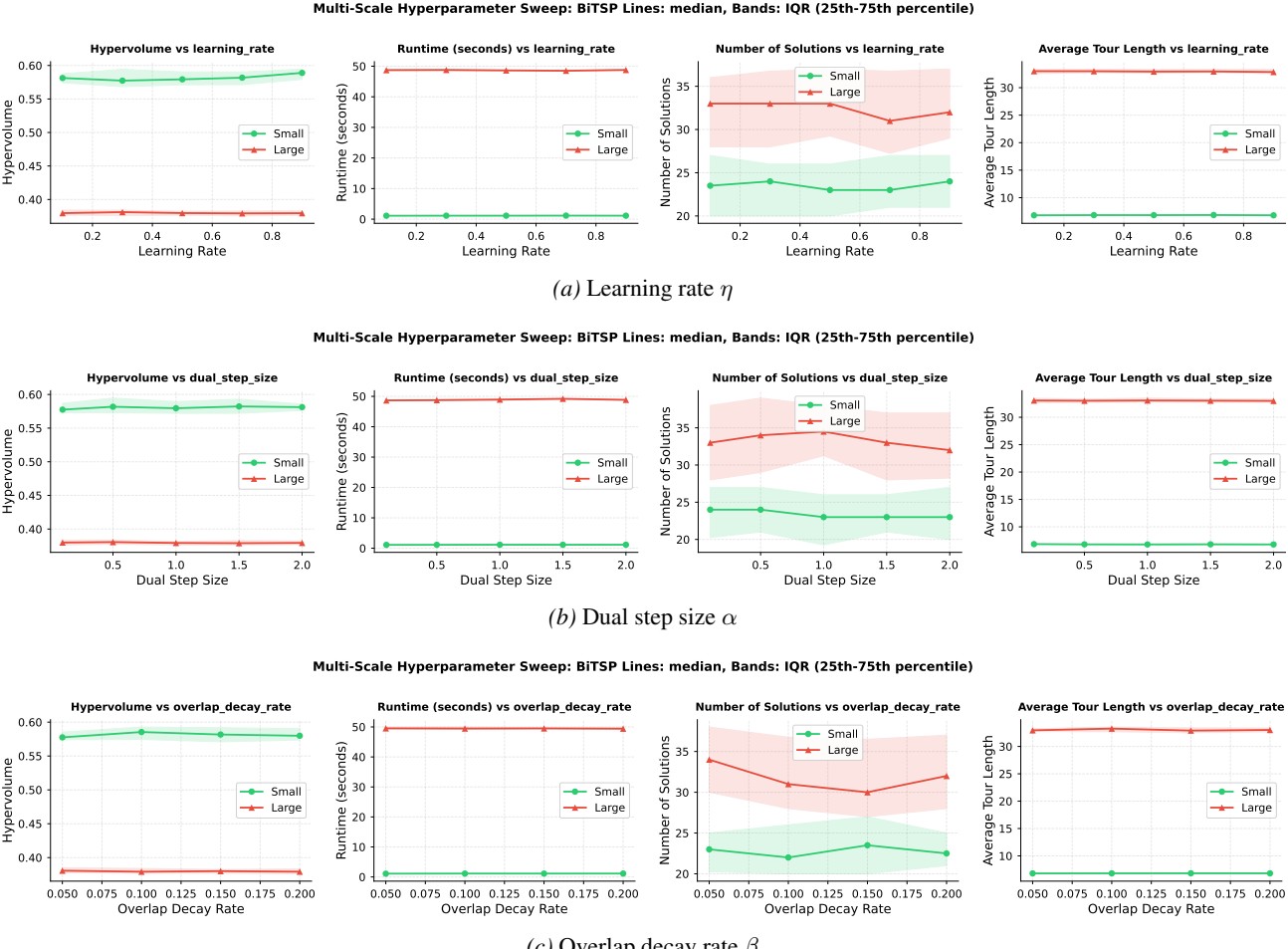

*(a)* Learning rate $\eta$

*(b)* Dual step size $\alpha$

*(c)* Overlap decay rate $\beta$

*Figure 19.* Hyperparameter sensitivity analysis on BiTSP for SMALL (20 cities) and LARGE (100 cities) instances. Each panel shows the effect of varying a single hyperparameter on four metrics: hypervolume, runtime, number of Pareto solutions, and average tour length. Lines indicate median values over 10 runs; shaded bands show interquartile range (25th–75th percentile). The algorithm exhibits stable performance across moderate perturbations to all three hyperparameters, with hypervolume varying by less than 2% across tested ranges.

## D.7. Lagrangian Coordination

The decomposed optimization in Algorithm 1 partitions the solution space into overlapping subproblems, where adjacent subproblems share variables at their boundaries. Without explicit coordination, these subproblems may develop conflicting preferences for shared variables—each optimizing locally without regard for global consistency. The Lagrangian coordination mechanism (Section 4.2.1 and proofs section E.6.3) addresses this by introducing dual variables $\lambda_i$ that penalize disagreement at overlapping positions, encouraging subproblems to reach consensus. This ablation study empirically validates whether this coordination mechanism provides measurable benefits beyond the implicit coordination achieved through shared global parameters.

**Experimental Setup**   We construct bi-objective TSP instances specifically designed to stress the coordination mechanism. Cities are partitioned into $k = 5$ regions, with boundary cities between adjacent regions having conflicting distance preferences across objectives: objective 1 prefers connections to the left region while objective 2 prefers the right region. This conflict strength is controlled by a parameter $\alpha = 4.0$, which scales the distance bias at boundaries. We test on two problem sizes: BiTSP20 ($n = 20$ cities) and BiTSP50 ($n = 50$ cities). Total iterations is 400.

For each configuration, we run 10 independent seeds with identical problem instances and initial solutions, comparing optimization with Lagrangian coordination enabled versus disabled. All runs use decomposition size 10, overlap 4, and 400 iterations. We track multiple metrics throughout optimization to isolate the effect of coordination.

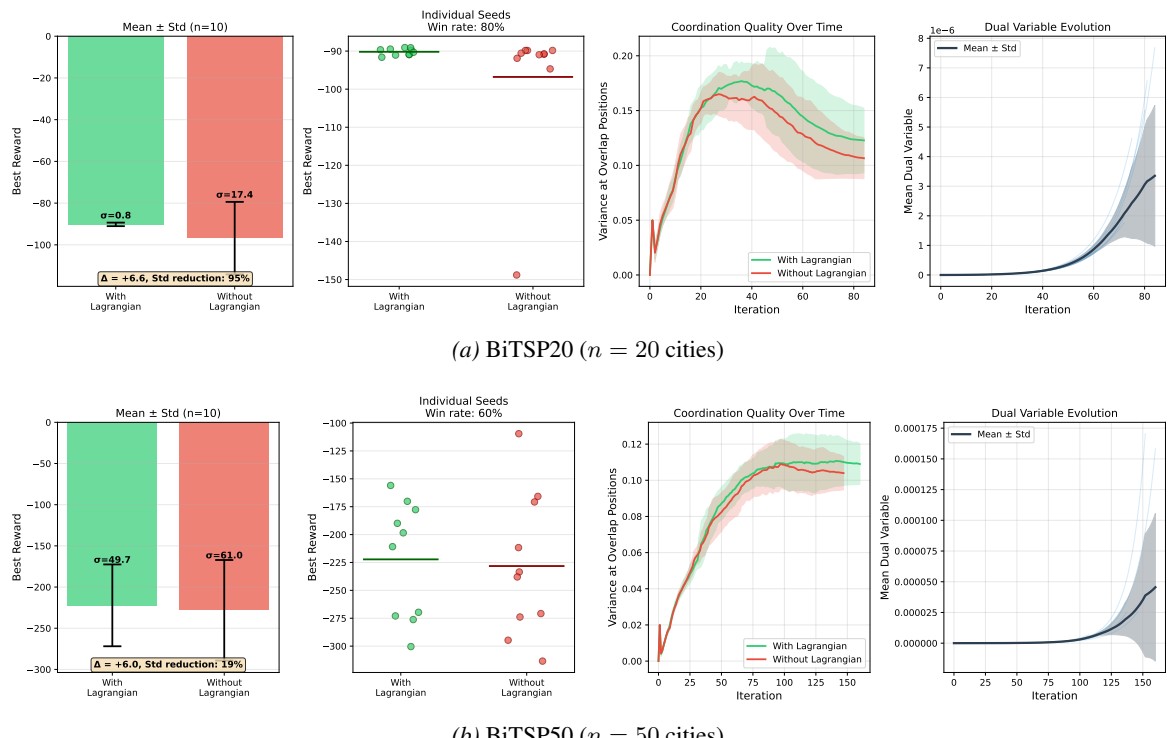

*(a)* BiTSP20 ($n = 20$ cities)

*(b)* BiTSP50 ($n = 50$ cities)

*Figure 20.* **Lagrangian Coordination Ablation: Reward Comparison.** (Left) Mean reward with standard deviation error bars. (Center-left) Individual seed results with mean lines. (Center-right) Variance at overlapping positions over iterations. (Right) Dual variable evolution. On BiTSP20, Lagrangian coordination reduces reward variance by 95% ($\sigma : 17.4 \rightarrow 0.8$) with 80% win rate. On BiTSP50, variance reduction is 19% ($\sigma : 61.0 \rightarrow 49.7$) with 60% win rate. Both problem sizes show consistent stability improvements, with dual variables actively increasing to track coordination violations.

**Reward Improvement and Outcome Stability.** We measure the final best reward achieved and its variance across seeds, capturing both solution quality and optimization reliability. On BiTSP20, Lagrangian coordination achieves a mean reward of $-90.2 \pm 0.8$ compared to $-96.8 \pm 17.4$ without coordination—a mean improvement of $+6.6$ with **95% variance reduction**. The win rate is 80%. Notably, seed 1 shows a 58-point improvement ($-90.9$ vs $-148.8$), demonstrating that coordination prevents catastrophic failures where subproblem conflicts trap optimization in poor local optima. On BiTSP50, coordination achieves $-222.2 \pm 49.7$ versus $-228.2 \pm 61.0$ without—a mean improvement of $+6.0$ with **19% variance reduction** and 60% win rate. Several seeds show substantial improvements: seed 1 gains $+104$ points and seed 0 gains $+56$ points. The consistent variance reduction across both problem sizes (95% and 19%) confirms that Lagrangian coordination improves optimization stability.

**Variance at Overlapping Positions.** We track the average variance of value estimates at positions shared by multiple subproblems throughout optimization. High variance indicates uncertainty or disagreement about the value of actions at these critical boundary positions. Both BiTSP20 and BiTSP50 show similar variance trajectories with and without Lagrangian coordination, suggesting that the coordination mechanism's primary effect is not on the internal value estimate distributions but rather on preventing the optimization from exploiting conflicting local optima. The variance increases during early exploration (iterations 0–50) as subproblems gather diverse experience, then stabilizes as estimates converge.

**Subproblem Disagreement Score.** The disagreement score measures the average variance of value estimates across all possible actions at each overlapping position, capturing how much subproblems "disagree" about the best action at shared variables. Similar to overlap variance, the disagreement trajectories are comparable between conditions. This confirms that Lagrangian coordination operates through the reward signal (penalizing solutions that violate coordination constraints) rather than by directly modifying the value estimate update dynamics.

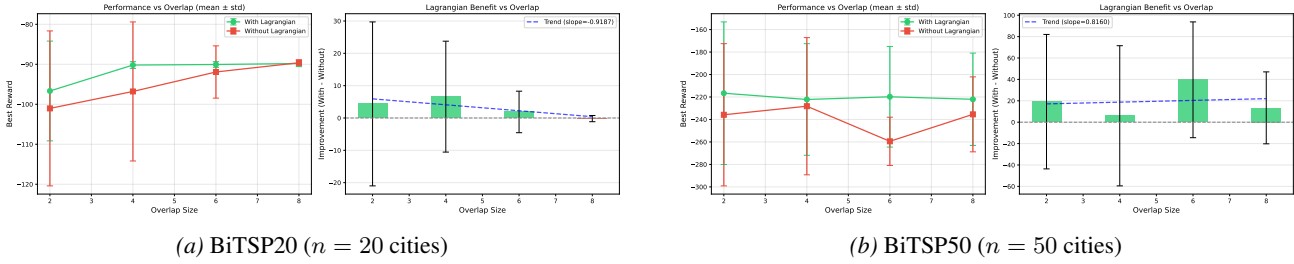

*(a)* BiTSP20 ($n = 20$ cities)  *(b)* BiTSP50 ($n = 50$ cities)

*Figure 21.* **Effect of Overlap Size on Lagrangian Benefit.** (Left panels) Absolute performance with standard deviation bars. Lagrangian coordination (green) consistently achieves higher rewards with smaller variance than the baseline (red). (Right panels) Improvement ($\Delta = $ With $-$ Without) as a function of overlap size. On both BiTSP20 and BiTSP50, Lagrangian provides positive improvement across all overlap sizes (2–8), demonstrating robust coordination benefits regardless of overlap configuration.

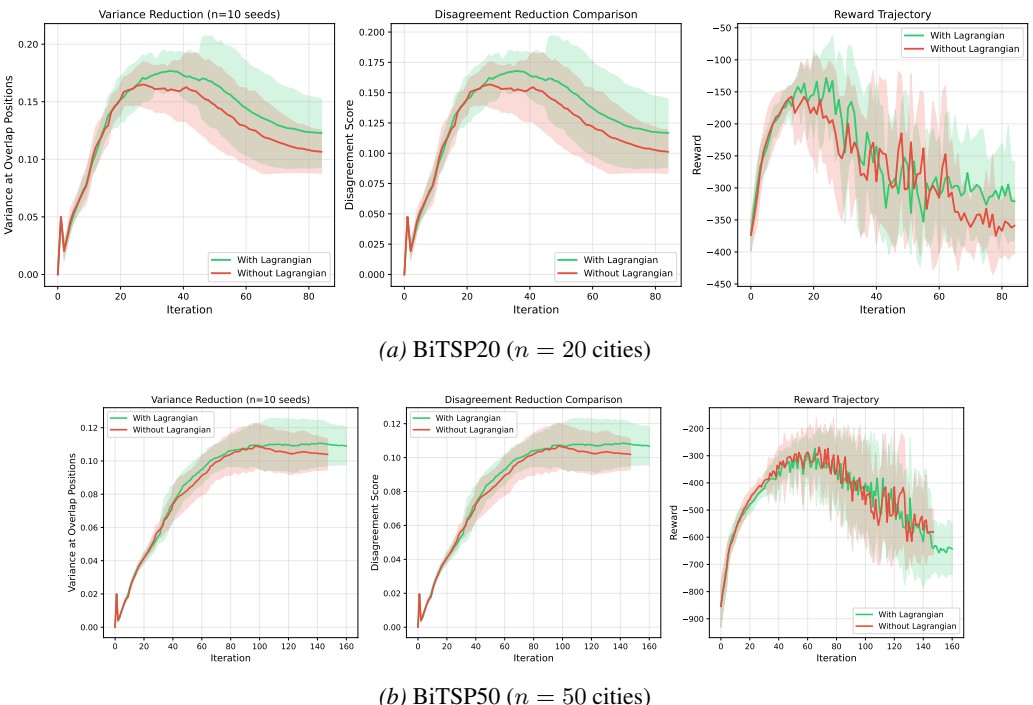

*(a)* BiTSP20 ($n = 20$ cities)

*(b)* BiTSP50 ($n = 50$ cities)

*Figure 22.* **Coordination Dynamics Over Optimization.** (Left) Variance at overlapping positions over iterations. (Center) Subproblem disagreement score. (Right) Reward trajectory with shaded $1\sigma$ confidence bands. The variance and disagreement metrics show similar evolution for both conditions, indicating that Lagrangian coordination operates through reward modification rather than altering value estimate dynamics. The key difference appears in reward trajectories: Lagrangian-coordinated runs (green) maintain tighter confidence bands, demonstrating more consistent optimization across random seeds.

**Dual Variable Evolution.** We monitor the mean dual variable $\bar{\lambda}$ over iterations, which reflects the cumulative coordination penalty applied. On BiTSP20, dual variables grow steadily from 0 to approximately $7 \times 10^{-6}$ over 90 iterations, indicating active violation tracking. On BiTSP50, the growth reaches approximately $1.75 \times 10^{-4}$ over 160 iterations. The monotonic increase confirms that the dual update mechanism is functioning as designed, accumulating penalties proportional to observed soft violations.

**Reward Trajectory.** The reward trajectory over iterations reveals the optimization dynamics. On BiTSP20, the "without Lagrangian" condition shows high variance with rewards ranging from $-150$ to $-400$, while "with Lagrangian" maintains a tighter band around $-100$ to $-250$. On BiTSP50, both conditions start similarly but the Lagrangian-coordinated runs maintain consistently higher rewards (around $-300$ to $-500$) with a visibly tighter confidence band compared to uncoordinated runs (spanning $-400$ to $-700$). The narrower shaded regions for Lagrangian-coordinated runs demonstrate more reliable optimization across random seeds.

*Table 20.* Single-objective results (400 iterations, 10 instances). D&L's decomposition provides the dominant gain, growing from 5% at small $n$ to 19% at large $n$. Pure bandits without decomposition perform near random search.

| Method | TSP (minimise ↓) | | | | | | Knapsack (maximise ↑) | | | | | |
| | $n{=}20$ | | $n{=}50$ | | $n{=}100$ | | $n{=}50$ | | $n{=}100$ | | $n{=}200$ | |
| | Mean | $\bar{R}$ | Mean | $\bar{R}$ | Mean | $\bar{R}$ | Mean | $\bar{R}$ | Mean | $\bar{R}$ | Mean | $\bar{R}$ |
|---|---|---|---|---|---|---|---|---|---|---|---|---|
| **D&L (TS)** | **4.30** | **3.27** | **9.24** | 5.49 | **18.03** | **11.55** | 19.16 | **3.44** | 39.42 | 7.02 | 57.38 | 9.45 |
| **D&L (UCB)** | 4.47 | 3.36 | 9.42 | **5.34** | 18.08 | 11.58 | **19.23** | 3.70 | **39.43** | **7.01** | **57.46** | **9.46** |
| No-Decomp-UCB | 4.55 | 3.94 | 10.13 | 7.69 | 22.25 | 19.58 | 18.96 | 4.67 | 39.10 | 8.81 | 56.36 | 13.53 |
| No-Decomp-TS | 4.68 | 3.95 | 10.11 | 7.66 | 22.59 | 19.65 | 18.93 | 4.28 | 39.07 | 8.79 | 56.14 | 13.45 |
| Pure UCB | 6.46 | 6.26 | 19.06 | 15.51 | 43.34 | 34.09 | 16.54 | 7.05 | 31.16 | 16.43 | 30.54 | 35.91 |
| Pure EXP3 | 6.55 | 6.28 | 19.20 | 15.64 | 43.58 | 34.21 | 17.31 | 7.21 | 33.01 | 16.13 | 38.01 | 32.98 |
| Random | 6.84 | 6.86 | 19.53 | 16.38 | 43.07 | 35.03 | 16.55 | 9.41 | 31.52 | 19.64 | 38.38 | 36.90 |

**Effect of Overlap Size.**   We vary the overlap parameter from 2 to 8 positions while holding other settings constant. On BiTSP20, Lagrangian provides consistent improvement (+3 to +5 points) across all overlap sizes with uniformly smaller error bars. On BiTSP50, improvement is positive across all overlap values (+5 to +40 points) with a slight positive trend (slope = +0.82), indicating that Lagrangian coordination provides robust benefits regardless of overlap configuration. The consistently smaller error bars for the Lagrangian condition across all overlap sizes further demonstrate the stability benefits of coordination.

**Summary** The ablation study confirms that Lagrangian coordination provides empirical benefits beyond theoretical guarantees:*Consistent variance reduction*: 95% on BiTSP20 and 19% on BiTSP50, demonstrating improved optimization stability across problem scales *Catastrophic failure prevention*: Prevents 50–100+ point losses on problematic seeds where subproblem conflicts would otherwise trap optimization *Robust improvement*: 60–80% win rate across problem sizes with mean improvements of +6.6 (BiTSP20) and +6.0 (BiTSP50) *Overlap-robust utility*: Positive improvement across all tested overlap sizes (2–8) on both problem scales. These results validate that the Lagrangian mechanism actively coordinates subproblem solutions, providing measurable stability benefits rather than merely theoretical coupling bounds.

### D.8. Single-objective Combinatorial Tasks

**Setup.**   To evaluate whether D&L's gains derive specifically from multi-objective structure or from its decomposition framework, we run D&L on single-objective TSP (minimize tour length) and Knapsack (maximize total value) using only the first objective from our bi-objective instances. Problem sizes match the multi-objective experiments: TSP $\in \{20, 50, 100\}$, KP $\in \{50, 100, 200\}$. Each method runs for 400 iterations with 5 restarts, averaged over 10 random instances. We compare four levels of ablation:

1. D&L (UCB/TS): Full method, decomposition ($d$-sized subproblems with overlap $Q$), expert mixture, and Lagrangian coordination.
2. No-Decomp (UCB/TS): Same expert mixture but $d{=}n$ (no decomposition), overlap=0, Lagrangian disabled. Isolates the decomposition contribution.
3. Pure UCB / Pure EXP3: Single bandit expert, no decomposition, no mixture.
4. Random Search: Uniform random feasible solutions.

Decomposition parameters scale with $n$ as mentioned in Section 6 and all other hyperparameter are shared across methods similar to all experiments throughout the paper. We report the mean objective value ($\pm$ std over instances), simple regret $r_T = |f(x_T^*) - f^*|$ where $f^*$ is the best value found by any method, and average regret $\bar{R}_T = \frac{1}{T}\sum_{t=1}^{T}|f(x_t) - f^*|$.

**Results.**   Table 20 presents the results. Three findings emerge consistently across both problem types: *(i) The full framework outperforms each component in isolation.* D&L outperforms No-Decomp by 5–19% on TSP and 1–2% on KP in objective value, with the gap widening at larger $n$. On TSP-100, D&L achieves 18.03 vs. No-Decomp's 22.25 (19% improvement); average regret drops from 19.58 to 11.55 (41% reduction). Notably, the expert mixture alone (No-Decomp) already improves over pure bandits, confirming that multi-expert selection contributes independently—but its effectiveness is amplified substantially when combined with decomposition, which reduces the per-subproblem action space and enables

more targeted credit assignment. *(ii) Pure bandits ≈ random search.* Without decomposition, bandit learning alone (Pure UCB, Pure EXP3) provides negligible improvement over random search—on TSP-100, all three cluster near 43. This confirms that position-level credit assignment cannot overcome combinatorial explosion without structural decomposition to reduce the effective decision dimension. *(iii) The advantage scales with problem size.* D&L's improvement over No-Decomp grows monotonically: 5%→9%→19% on TSP as $n$ increases from 20 to 100, consistent with the theoretical prediction that decomposition reduces regret from $O(n\sqrt{T})$ to $O(d\sqrt{T})$.

**Relationship to multi-objective results.** These single-objective results confirm that D&L's decomposition framework is not specific to the multi-objective setting. It provides consistent gains on standard combinatorial optimization as well. That said, the multi-objective setting is where D&L's full design is most impactful: in MOO, decomposition operates at two levels simultaneously (decision-space subproblems *and* objective-space scalarisations), the Lagrangian coordination resolves conflicts across overlapping subproblems under *changing* scalarisation weights, and the expert mixture must adapt to non-stationary reward landscapes induced by weight rotation. These challenges are absent in the single-objective case, which explains the larger performance gaps observed in our main multi-objective experiments (Section 6.4).

### D.9. Decomposition Strategy

We study the sensitivity of our algorithm to the choice of decomposition strategy. The base optimizer partitions decision variables into subproblems $S_1, \ldots, S_K$ of size $d$. We compare three strategies: **(i)** *Sliding Window* (SW): index-based windows $S_k = \{k(d-o)+1, \ldots, k(d-o)+d\}$ with overlap $o$, agnostic to problem structure; **(ii)** *Metric*: for each random centre $c_k$, group its $d$ nearest neighbours under the problem distance matrix $D$; **(iii)** *SW+Metric*: blend of both (our default).

We evaluate on BiTSP at scales ($n \in \{20, 50, 100\}$) with both D&L-UCB and D&L-TS, using 10 independent problem instances per configuration. All other hyperparameters are held at their default values from Figure 23.

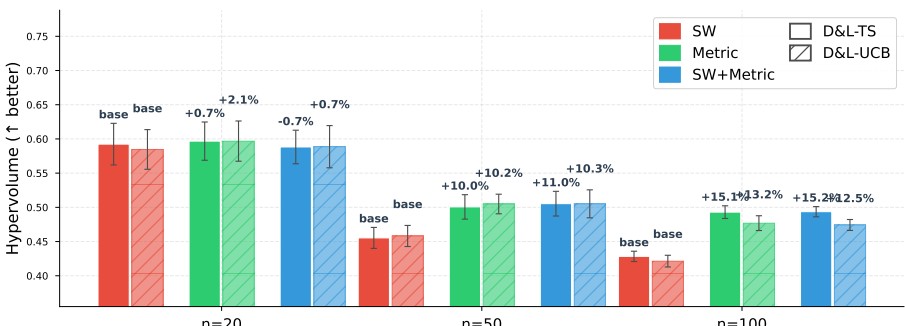

*Figure 23.* Decomposition strategy ablation on BiTSP across scales ($n \in \{20, 50, 100\}$). Hypervolume (mean ± std, 10 seeds) for sliding window (SW), metric-based (Metric), and their combination (SW+Metric) under both D&L-TS and D&L-UCB. Percentages indicate improvement over the SW baseline. Metric-based grouping yields increasing gains at larger scales (∼1% at $n$=20 → ∼15% at $n$=100), while performance is robust across strategies at small $n$.

**Results.** Metric-based decomposition consistently outperforms the index-based sliding window, and the improvement grows with problem scale: ∼1–2% at $n$=20, ∼10% at $n$=50, and ∼13–15% at $n$=100. This is expected: at small $n$, most subproblems already contain spatially related variables regardless of grouping strategy, but at larger $n$ the sliding window increasingly groups unrelated variables. The effect is consistent across both D&L-UCB/TS, confirming that the benefit comes from the decomposition structure rather than the bandit policy. SW+Metric and Metric-only perform comparably, suggesting the sliding window component adds little when metric groups are available. The runtime cost of metric decomposition is modest (∼1.5–1.8× SW), as the nearest-neighbor lookup is $O(nd)$ per update.

**Discussion.** The regret bounds in Proposition E.25 hold for any decomposition satisfying the bounded coupling assumption, including both sliding window and metric-based strategies with identical worst-case guarantees $|f_j(x') - f_j(x)| \leq L \cdot D(s) + C \cdot (\rho - 1) \cdot s$. The algorithm's convergence is therefore not sensitive to the decomposition choice in a theoretical sense. The empirical improvement from metric-based decomposition reflects the gap between the worst-case diameter $D(S_k)$ and the *realized* perturbation $\delta(x, x')$ within each subproblem: grouping spatially proximate variables ensures that local modifications produce objective changes well below the Lipschitz bound, yielding more productive optimization steps

per iteration. At small $n$, subproblems constitute a large fraction of the problem and naturally contain nearby variables regardless of strategy, so both approaches perform comparably. At large $n$, sliding window subproblems increasingly group unrelated variables, causing the realized perturbation to approach the worst case, while metric-based subproblems maintain tight locality. Thus, explaining the widening gap ($\sim 1\%$ at $n{=}20$ versus $\sim 14\%$ at $n{=}100$).

## D.10. Scalarization Sensitivity

We study the sensitivity of our algorithm to the choice of scalarization $g_\lambda : \mathbb{R}^m \to \mathbb{R}$. Corollary 5.3 establishes that the $O(d\sqrt{T \log T})$ regret bound holds for any bounded scalar objective, so the recovered Pareto front should be robust to this choice in practice. We compare four standard scalarizations: **(i)** *Weighted Sum* (WS), $g_\lambda(f) = \sum_j \lambda_j f_j$; **(ii)** *Tchebycheff* (TCH), $g_\lambda(f) = \max_j \lambda_j |f_j - z_j^\star|$; **(iii)** *Augmented Tchebycheff* (ATCH), TCH with an $\ell_1$ regularizer; and **(iv)** *Penalty Boundary Intersection* (PBI), the projection-plus-penalty form of Zhang & Li (2007). We evaluate on BiKP at $n \in 50, 100$ with both D&L-UCB and D&L-TS, using 10 independent seeds per configuration. We pick BiKP rather than BiTSP or BiCVRP because the BiKP Pareto front is known to be non-convex in general (Ehrgott, 2005): WS provably cannot reach unsupported points on the non-convex portion of the front, while TCH and PBI can. Agreement on BiKP is therefore the strongest evidence that the algorithm is genuinely scalarization-agnostic. All other hyperparameters are held at their default values from Table 21.

*Table 21.* Scalarization ablation on BiKP (10 seeds per configuration). Each cell reports mean HV $\pm$ std / |PF|. All scalarizations land within $\sim 2\%$ on HV, well within one standard error (SE $= \sigma/\sqrt{10} \le 0.022$).

|  |  | WS | TCH | ATCH | PBI |
|---|---|---|---|---|---|
| BiKP $n{=}50$ | D&L-UCB | **.322**±.057 / 42 | .322±.056 / 29 | .322±.058 / 49 | .322±.058 / 50 |
|  | D&L-TS | .318±.058 / 42 | .318±.059 / 29 | .317±.057 / 49 | .315±.058 / 50 |
| BiKP $n{=}100$ | D&L-UCB | **.402**±.068 / 158 | .395±.069 / 116 | .398±.068 / 129 | .396±.066 / 70 |
|  | D&L-TS | **.399**±.067 / 158 | .396±.069 / 116 | .395±.069 / 129 | .395±.065 / 70 |

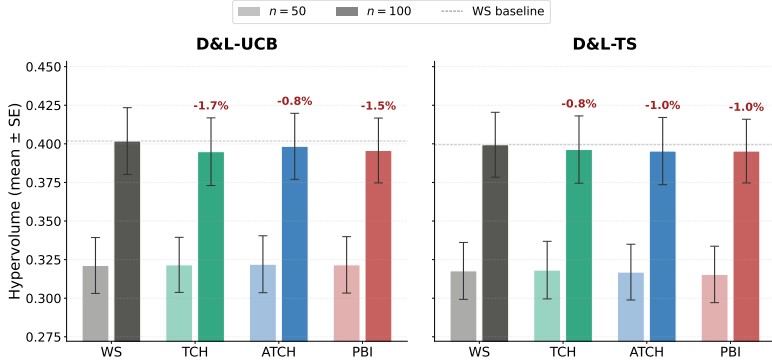

*Figure 24.* Scalarization ablation on BiKP ($n \in \{50, 100\}$, 10 seeds). Hypervolume (mean $\pm$ standard error) for WS, TCH, ATCH, and PBI under both D&L-UCB and D&L-TS. Percentages indicate $\Delta$HV relative to WS at $n{=}100$. All four scalarizations land within 2% of WS, with differences contained within one standard error.

**Results.** Hypervolume is stable across all four scalarizations to within 2% at both problem sizes, well within one standard error, and the pattern is consistent across D&L-UCB and D&L-TS (TS HVs sit within 0.004 of UCB throughout). Figure 24 shows the same numbers with standard-error whiskers: no scalarization is statistically distinguishable from any other. Pareto-front sizes |PF| do vary, with WS and ATCH producing broader fronts (158 and 129 at $n{=}100$) and PBI yielding fewer, more concentrated solutions (70). However, Fig. 25 confirms that when the best front per scalarization is overlaid on the same axes, the four fronts trace the same trade-off curve and differ only in sampling density along it. PBI's lower point count is a known property of its narrow acceptance cone (Zhang & Li, 2007) rather than a behavior induced by D&L.

**Discussion.** The regret bound in Corollary 5.3 holds for any bounded scalar objective $g_\lambda$, so the convergence of D&L is not sensitive to the scalarization choice in a theoretical sense; the bandit-coordinated experts treat $g_\lambda$ as a black box and operate identically regardless of its form. The empirical stability on BiKP, a benchmark where scalarizations are provably distinguishable in principle since WS cannot reach non-convex regions that TCH and PBI can, shows that this **theoretical**

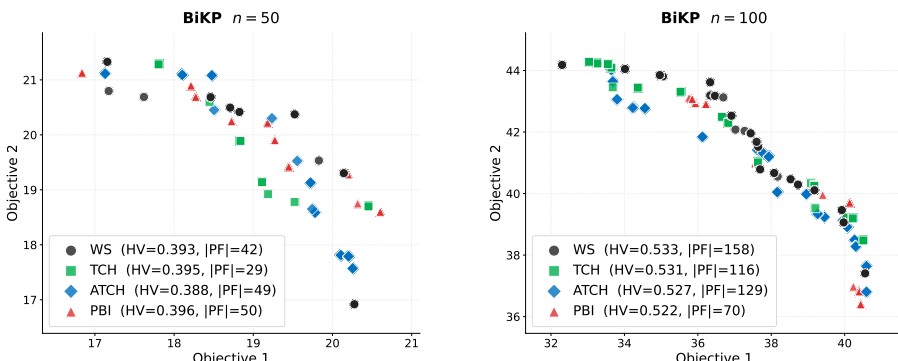

*Figure 25.* Pareto front overlays on BiKP $n{=}100$ (best run per scalarization). All four scalarizations recover the same front geometry and coverage of the trade-off region; the fronts coincide up to point sampling density, with PBI producing fewer but similarly-located points.

**agnosticism is realized in practice**: the decomposition recovers the same front whether the scalarized subproblem is convex (WS), max-form (TCH, ATCH), or projection-based (PBI). The variation in $|\mathrm{PF}|$ reflects the inherent discretization behaviour of each scalarization family rather than a property of D&L, and Figure 25 shows it does not translate into front-quality differences. Practitioners can therefore swap scalarizations based on problem structure, with WS for convex fronts where its reward signal is smoothest and TCH/ATCH for non-convex fronts or for emphasizing corner trade-offs, without affecting the regret guarantee or the recovered front. We use WS for the combinatorial benchmarks (BiTSP, BiKP, BiCVRP) and ATCH for the 4-objective hardware-software co-design problem on this basis.

# E. Algorithm Regret Bounds

Below we provide detailed proof steps for results presented in main body.

## E.1. Notation

Table 22 consolidates notation used throughout the paper and appendix. We write $x^{(t)}$, $r^{(t)}$ in pseudocode (Algorithms 1–5) and $x_t$, $r_t$ in the analysis; both denote the solution/reward at iteration $t$. We denote by $\mathcal{F}_{t-1}$ the filtration generated by the algorithm's actions, observed rewards, and internal randomness up to the end of round $t-1$.

## E.2. Assumptions

Below we detail the (weak) assumptions necessary to proving regret bounds in later sections.

### E.2.1. ASSUMPTION 1 (DECOMPOSABILITY)

The decision space admits a decomposition:

$$\mathcal{X} = \mathcal{X}^1 \times \mathcal{X}^2 \times \cdots \times \mathcal{X}^K = \prod_{k=1}^{K} \mathcal{X}^k$$

where $\mathcal{X}^k$ are subspaces with controlled overlap. Each solution $x \in \mathcal{X}$ can be represented as $\boldsymbol{x} = (\boldsymbol{x}^{(1)}, \boldsymbol{x}^{(2)}, \ldots, \boldsymbol{x}^{(K)})$ where $\boldsymbol{x}^{(k)} \in \mathcal{X}^k$. The subspaces may overlap: for indices $i \in \mathcal{X}^k$ and $i \in \mathcal{X}^{k'}$, we allow $k \neq k'$. Let $\mathcal{O}$ denote the set of overlapping indices.

### E.2.2. ASSUMPTION 2 (LIPSCHITZ CONTINUITY)

The objectives exhibit Lipschitz continuity with respect to an appropriate problem-specific metric $d : \mathcal{X} \times \mathcal{X} \to \mathbb{R}_+$. There exists $L > 0$ such that:

$$|f_j(x) - f_j(x')| \leq L \cdot \delta(x, x')$$

where the metric $d$ is defined based on the problem structure. This assumption ensures that small changes in the solution space result in bounded changes in objective values, enabling local search. Therefore, this assumption bounds the sensitivity

*Table 22.* **Notation Summary.**

| Symbol | Meaning |
|---|---|
| *Decision space* | |
| $\mathcal{X}$ | Decision space ($\mathcal{X}_c \subseteq \mathbb{R}^{d_c}$ continuous, $\mathcal{X}_d = \prod_{i=1}^{d_d} \mathcal{D}_i$ discrete) |
| $\boldsymbol{x} \in \mathcal{X}$ | A solution; $\boldsymbol{x} = (\boldsymbol{x}^{(1)}, \ldots, \boldsymbol{x}^{(K)})$ with subproblem components $\boldsymbol{x}^{(k)} \in \mathcal{X}^k$ |
| $n$ | Problem size (number of decision variables / positions) |
| $\mathcal{A}_i$ | Set of admissible actions (values) at position $i$; $A_{\max} := \max_i |\mathcal{A}_i|$ |
| $\delta(\cdot, \cdot)$ | Problem-specific metric on $\mathcal{X}$ |
| *Objectives* | |
| $\boldsymbol{f}(\boldsymbol{x})$ | Objective vector $(f_1(\boldsymbol{x}), \ldots, f_m(\boldsymbol{x}))$; $m$ objectives |
| $L, B$ | Lipschitz constant (Assumption E.2.2); reward bound (Assumption E.2.4) |
| *Online learning* | |
| $T$ | Total number of iterations |
| $x_t$ / $x^{(t)}$ | Solution selected at iteration $t$ (prose / pseudocode) |
| $r_t(\boldsymbol{x})$ / $r^{(t)}$ | Scalar reward at iteration $t$ (prose / pseudocode) |
| $x^*$ | Best fixed solution in hindsight: $x^* = \arg\max_{x \in \mathcal{X}} \mathbb{E}[r_t(x)]$ |
| $R(T)$ | Cumulative regret over $T$ rounds |
| $\mathcal{F}_{t-1}$ | Filtration (history) up to end of round $t-1$ |
| *Decomposition* | |
| $K$ | Number of subproblems |
| $d$ | Subproblem size ($d = n/K$ for uniform partitions) |
| $S_k, I_k$ | Index set of positions in subproblem $k$; $|I_k| \leq d$ |
| $x_t^{(k)}$ | Restriction of $x_t$ to subproblem $k$ |
| $g_k(\cdot)$ | Surrogate subproblem objective (for analysis; Def. E.6) |
| $\mathcal{T}_k$ | Set of rounds in which subproblem $k$ is updated; $T_k = |\mathcal{T}_k|$ |
| *Overlap & coordination* | |
| $\mathcal{O}$ | Set of overlapping indices: $\{i \in [n] : i \text{ appears in } \geq 2 \text{ subproblems}\}$ |
| $O_{ml} = I_m \cap I_l$ | Pairwise overlap between subproblems $m$ and $l$ |
| $Q$ | Maximum pairwise overlap size: $Q = \max_{m \neq l} |O_{ml}|$; $|\mathcal{O}| \leq KQ$ |
| $\rho$ | Maximum overlap multiplicity: $\rho = \max_i |\{k : i \in S_k\}|$ |
| $\lambda_i^{(t)}$ | Dual variable for overlapping position $i$ at iteration $t$ |
| $\xi_i^{(t)}$ | Soft violation measure at position $i$ |
| $C_t$ | Coupling error at iteration $t$ |
| *Expert parameters (shared across subproblems)* | |
| $\hat{V}_{i,a}, N_{i,a}$ | Value estimate and visit count for position–action pair $(i, a)$ |
| $W_{i,a}$ | EXP3 multiplicative weight for $(i, a)$ |
| $\tilde{L}_{i,a}$ | FTRL cumulative loss for $(i, a)$ |
| $\boldsymbol{\rho}$ | Expert mixture weights $(p_1, p_2, p_3)$ |

of the objectives to local modifications and does not imply smoothness or differentiability.

### E.2.3. ASSUMPTION 3 (BOUNDED COUPLING)

Let $\boldsymbol{x} \in \mathcal{X}$ be any feasible solution and let $\boldsymbol{x}'$ differ from $\boldsymbol{x}$ only on the coordinates indexed by a subproblem $I_k \subseteq \{1, \ldots, n\}$. Then for each objective $f_j$,

$$|f_j(\boldsymbol{x}') - f_j(\boldsymbol{x})| \leq L \cdot d(\boldsymbol{x}, \boldsymbol{x}') + C \cdot |I_k \cap \mathcal{O}|,$$

where $d(\cdot, \cdot)$ is the problem-specific metric from Assumption E.2.2, $\mathcal{O}$ is the set of indices appearing in multiple subproblems, $C > 0$ is a constant independent of $k$ and $T$. This assumption does not require additive decomposability of the objective. It

states that non-local interaction effects induced by a local modification are bounded and scale linearly with the size of the overlap.

### E.2.4. ASSUMPTION 4 (BOUNDED RANGE)

We assume that any objective $f_j(\boldsymbol{x})$ is bounded in the decision space $\mathcal{X}$ i.e $f_j(\boldsymbol{x}) \in [0, B]$ for all $\boldsymbol{x} \in \mathcal{X}$ and $j \in \{1, \ldots, m\}$.

### E.2.5. ASSUMPTION A.5' (LOCAL ADDITIVITY + EXOGENEITY FOR POSITION–VALUE UCB)

Fix a subproblem $k$ with index set $I_k$ and let $x_t^{(k)}$ denote the local assignment produced when subproblem $k$ is updated at round $t$. There exist constants $\{\mu_{i,j}\}_{i \in I_k, \, j \in \mathcal{A}_i} \subset [0, 1]$ such that the surrogate objective is additive:

$$g_k(x^{(k)}) \;=\; \sum_{i \in I_k} \mu_{i, x_i}.$$

Moreover, for each position $i \in I_k$, whenever the algorithm plays value $x_{i,t} = j$ and observes the (normalized) scalar reward $r_t \in [0, 1]$, we have

$$\mathbb{E}[r_t \mid \mathcal{F}_{t-1}, \, x_{i,t} = j] \;=\; \mu_{i,j} + b_{i,t}, \qquad \forall j \in \mathcal{A}_i,$$

for some process $b_{i,t}$ that may depend on $(\mathcal{F}_{t-1}, t)$ but does not depend on $j$. Finally, the centered noise

$$\eta_t := r_t - \mathbb{E}[r_t \mid \mathcal{F}_{t-1}, x_{i,t}]$$

is conditionally 1-sub-Gaussian (or simply bounded in $[-1, 1]$).

### E.3. Online Game

We consider a stochastic online learning setting over $T$ iterations. At each iteration $t \in \{1, \ldots, T\}$:

1. The learner selects $\boldsymbol{x}_t \in \mathcal{X}$

2. The environment generates a scalar reward $r_t(x_t) = \phi(\mathbf{f}(x_t)) + \epsilon_t$ where $\mathbf{f}(x) = (f_1(x), \ldots, f_m(x))$ is the multiobjective vector, $\phi(\cdot)$ is a scalarization (e.g., weighted sum), and the learner observes $r_t(\boldsymbol{x}_t)$. and $\epsilon_t$ denotes stochastic noise.

The goal is to minimize the cumulative regret with respect to the best fixed solution $\mathbf{x}^* \in \arg\max_{\boldsymbol{x} \in \mathcal{X}} \mathbb{E}[r_t(\boldsymbol{x})]$:

$$R(T) = \sum_{t=1}^{T} r_t(\boldsymbol{x}^*) - \sum_{t=1}^{T} r_t(\boldsymbol{x}_t)$$

where $r_t(\boldsymbol{x})$ represents the scalarized or Pareto-based reward for solution $\boldsymbol{x}$ at time $t$.

**Feedback model:** The bandit feedback is full-information at the solution level: at each iteration the learner observes only a single scalar reward for the entire decision vector, not per-component feedback.

### E.3.1. HYBRID ALGORITHM STRUCTURE

Our algorithm is a hybrid algorithm with a global and local optimization phases. The expert algorithms serve a critical role in **global optimization**: they synthesize complete solutions from learned statistics, enabling the algorithm to escape local minima and coordinate assignments across decomposed subproblems. This provides solution diversity that complements the exploitation of local search. Global optimization occurs periodically every $c$ iterations ($c < T$), where experts make $n$ selections to construct a full solution.

Between global optimization phases, the algorithm performs **local search**: gradient-free improvement operators to solve the individual subproblems of size $d$. Critically, even during local search iterations, all expert parameters are updated based on observed rewards, ensuring continuous learning throughout the $T$ total iterations.

Our algorithm maintains three sets of global parameters that are updated simultaneously:

- **UCB parameters**: Value estimates $\hat{\mu}_{i,j}(t)$ and visit counts $N_{i,j}(t)$ for each position-value pair

- **EXP3 parameters**: Weights $w_{i,j}(t)$ maintained via multiplicative updates

- **FTRL parameters**: Cumulative losses $L_{i,j}(t) = \sum_{s=1}^{t} \ell_s(i,j)$ with regularization

At each iteration $t \equiv 0 \pmod{c}$ (occurring $T_g = \lfloor T/c \rfloor, c << T$, where T is total iterations of the algorithm), the algorithm selects actions by randomly choosing one component:

$$\text{Action selection} = \begin{cases} \text{Expert 1} & \text{with probability } p_{\text{Expert 1}} = p_1 \\ \text{Expert 2} & \text{with probability } p_{\text{Expert 2}} = (1 - p_{\text{Expert 1}}) \cdot \rho_t \\ \text{Expert 3} & \text{with probability } p_{\text{Expert 3}} = (1 - p_{\text{Expert 1}}) \cdot (1 - \rho_t) \end{cases}$$

where $\rho_t \in [0,1]$ is the adaptive hybrid ratio based on empirical uncertainty and $p_1$ is some constant probability. Crucially, regardless of which component is used for selection, **all parameters are updated** after observing the reward, enabling each component to learn from the entire history. This multi-expert learning ensures the algorithm achieves:

$$R(T) = R_{local}(T) + R_{global}(T) + C \tag{26}$$

$$R_{global}(T) = p_{Expert_1} \cdot R_{Expert_1}(T) + p_{Expert_2} \cdot R_{Expert_2}(T) + p_{Expert_3} \cdot R_{Expert_3}(T) \tag{27}$$

This approach is computationally efficient: only one expert computes scores during selection, while all three update their parameters after observing the reward. The experts share visit counts and importance weights across all algorithms, with each maintaining only algorithm-specific parameters (value estimates for UCB, multiplicative weights for EXP3, cumulative losses for FTRL). Since objective evaluations dominate the computational cost and occur once per iteration, maintaining three (or any) experts adds negligible overhead—essentially the selection cost of one algorithm plus lightweight shared parameter updates. This provides ensemble diversity and robustness at minimal additional cost. Our implementation uses three experts (UCB, EXP3, FTRL). The framework supports any number of experts, though we recommend at least two to ensure diversity.

### E.4. Algorithm Total Regret Bound

**Theorem E.1** (**Regret Decomposition Under Structured Decomposition and Overlap**). *Let Assumptions E.2.1–E.2.4 hold.*

*Consider an online learning process over $T$ rounds in a decomposed decision space $\mathcal{X} = \prod_{k=1}^{K} \mathcal{X}^k$ with controlled overlaps $\mathcal{O}$. At round $t$, the algorithm selects a solution $\boldsymbol{x}_t$, observes a scalarized reward $r_t(\boldsymbol{x}_t) \in [0, B]$, and updates all experts in its expert set $\mathcal{E} = \{1, \ldots, E\}$, even though only one expert is used to select the action. Let $\boldsymbol{x}^\star$ be the best fixed comparator in hindsight. We define the global regret,*

$$R_{\text{total}}(T) = \sum_{t=1}^{T} \big( r_t(\boldsymbol{x}^\star) - r_t(\boldsymbol{x}_t) \big).$$

*and the* average subproblem regret,

$$R_{\text{avg}}(T) := \frac{1}{K} \sum_{t=1}^{T} \sum_{k=1}^{K} \big[ g_k((\boldsymbol{x}^\star)^{(k)}) - g_k(\boldsymbol{x}_t^{(k)}) \big].$$

*We will state our main guarantees in terms of $R_{\text{avg}}(T)$, which measures the average regret per decomposed subproblem. Note that $R_{\text{total}}(T)$ and $R_{\text{avg}}(T)$ differ only by a factor of $K$ in the subproblem component.*

*Then the average total regret of the algorithm satisfies the structural decomposition:*

$$R_{\text{avg}}(T) \leq R_{\text{subproblem}}^{\text{avg}}(T) + R_{\text{overlap}}^{\text{avg}}(T) + R_{\text{local}}^{\text{avg}}(T) + C$$

*where each term is defined as follows:*
*(1) **Subproblem regret** Using Assumption E.2.3, the surrogate decomposed objective admits:*

$$r_t(\boldsymbol{x}) \le \sum_{k=1}^{K} g_k(\boldsymbol{x}^{(k)}) + h(\boldsymbol{x}^{(\mathcal{O})}),$$

*and the regret due to bandit learning inside each subproblem decomposes as:*

$$R_{\text{subproblem}}^{\text{avg}}(T) := \frac{1}{K} \sum_{k=1}^{K} R_{\text{bandit}}^{(k)}(T),$$

*where for each subproblem $k$,*

$$R_{\text{bandit}}^{(k)}(T) \le \sum_{e=1}^{E} p_e R_{\text{alg}}^{(k,e)}(T)$$

*with $p_e$ the probability of selecting expert $e$ and $R_{\text{alg}}^{(k,e)}(T)$ the standard Expert 1, Expert 2 regret bound for subproblem $k$.*
*(2) **Overlap coupling regret** The bounded interaction term $h(\boldsymbol{x}^{(\mathcal{O})})$ from Assumption E.2.3 induces:*

$$R_{\text{overlap}}^{\text{avg}}(T) \le O\left( C|\mathcal{O}| + \sum_{t=1}^{T} \sum_{i \in \mathcal{O}} \left\| \lambda_t(i) - \lambda_{t-1}(i) \right\| \right),$$

*where $\lambda_t$ are the dual variables coordinating overlapping positions.*
*(3) **Local optimization regret** Under Assumption E.2.2, the regret due to local optimization inside each subproblem satisfies:*

$$R_{\text{local}}^{\text{avg}}(T) \le \frac{L}{K} \sum_{t=1}^{T} d(\boldsymbol{x}_t, \text{LocalImprove}(\boldsymbol{x}_t))$$

*(4) **Constant term** The constant $C$ absorbs initialization terms, expert switching effects, and stabilizing effects from the dual updates.*

**Theorem E.2** (**Total Average-Regret**). *Under the setting of Theorem E.1, assume that each expert $e = 1, \ldots, n$ used by* ALGORITHM 1 *admits an average subproblem regret bound of the form*

$$\mathbb{E}\left[ R_{\text{subproblem}}^{\text{avg},(e)}(T) \right] \le \phi_e(T)$$

*Then the average regret of* ALGORITHM 1 *satisfies,*

$$\mathbb{E}\left[ R_{\text{avg}}(T) \right] \le \sum_{e=1}^{E} p_e \phi_e(T) + \mathbb{E}\left[ R_{\text{overlap}}^{\text{avg}}(T) \right] + \mathbb{E}\left[ R_{\text{local}}^{\text{avg}}(T) \right] + C \tag{28}$$

*where $p_e$ is the probability weight assigned to expert $e$ by the algorithm.*

*Proof.* By Theorem E.1, the algorithm's average regret decomposes as

$$R_{\text{avg}}(T) \le R_{\text{subproblem}}^{\text{avg}}(T) + R_{\text{overlap}}^{\text{avg}}(T) + R_{\text{local}}^{\text{avg}}(T) + C$$

From the definition of the subproblem regret,

$$R_{\text{subproblem}}^{\text{avg}}(T) = \frac{1}{K} \sum_{k=1}^{K} R_{\text{bandit}}^{(k)}(T), \qquad R_{\text{bandit}}^{(k)}(T) \le \sum_{e=1}^{E} p_e R_{\text{alg}}^{(k,e)}(T)$$

Substituting,

$$R_{\text{subproblem}}^{\text{avg}}(T) \leq \frac{1}{K} \sum_{k=1}^{K} \sum_{e=1}^{E} p_e \, R_{\text{alg}}^{(k,e)}(T) = \sum_{e=1}^{E} p_e \left( \frac{1}{K} \sum_{k=1}^{K} R_{\text{alg}}^{(k,e)}(T) \right)$$

By definition of $R_{\text{subproblem}}^{\text{avg},(e)}(T)$,

$$R_{\text{subproblem}}^{\text{avg}}(T) \leq \sum_{e=1}^{E} p_e R_{\text{subproblem}}^{\text{avg},(e)}(T)$$

Taking expectations and applying the assumed expert bounds yields,

$$\mathbb{E}\left[ R_{\text{subproblem}}^{\text{avg},(e)}(T) \right] \leq \phi_e(T)$$

$$\mathbb{E}\left[ R_{\text{subproblem}}^{\text{avg}}(T) \right] \leq \sum_{e=1}^{E} p_e \phi_e(T)$$

Substituting back into the decomposition inequality gives the desired result:

$$\mathbb{E}\left[ R_{\text{avg}}(T) \right] \leq \sum_{e=1}^{E} p_e \phi_e(T) + \mathbb{E}\left[ R_{\text{overlap}}^{\text{avg}}(T) \right] + \mathbb{E}\left[ R_{\text{local}}^{\text{avg}}(T) \right] + C$$

$\square$

**Corollary E.3** (**Explicit Regret Bound for ALGORITHM 1**). *Let* ALGORITHM 1 *use three experts:*

$$\text{Expert 1 = UCB}, \qquad \text{Expert 2 = Exp3}, \qquad \text{Expert 3 = FTRL},$$

*with mixture weights* $p_1, p_2, p_3$. *Under Assumptions 1–4, let their average subproblem regret bounds satisfy* $\phi_{\text{UCB}}(T), \phi_{\text{Exp3}}(T), \phi_{\text{FTRL}}(T)$. *If $K$ is number of subproblems, $d$ subproblem size, $n = Kd$ = problem size, $L$ Lipschitz constant, $D$ domain diameter, $B$ reward bound, $C_{clip} > 0$ and Exp3 clipping threshold, then expected average regret of* ALGORITHM 1 *satisfies,*

$$\mathbb{E}[R_{\text{avg}}(T)] \leq \underbrace{p_1 C_1 d \sqrt{\frac{T \log T}{K}}}_{UCB} + \underbrace{p_2 \frac{C_2 d B \sqrt{T \log d}}{C_{clip}}}_{Exp3} + \underbrace{p_3 \frac{C_3 d B \sqrt{T \log T / K}}{C_{clip}}}_{FTRL}$$

$$+ \underbrace{C_4 K Q R_{max} \sqrt{T}}_{Overlap\ coordination} + \underbrace{C_5 L D \sqrt{\frac{dT}{K}}}_{Local\ search} + C_0 \tag{29}$$

*In particular, the contribution of the decomposed subproblem learning scales as,*

$$\mathbb{E}[R_{\text{subproblem}}^{\text{avg}}(T)] \leq O\left( d \sqrt{T \log T} \right)$$

*and is independent of the number of subproblems $K$, Also, $C_0, C_1, \ldots, C_5$ are universal constants independent of $T, K, d, n$.*

*Under uniform updating with $K \geq d$, both bandit experts contribute terms of order $O(d\sqrt{T \log T})$: EXP3 via $\log d \leq \log T$, and FTRL via the $\sqrt{1/K}$ decomposition bonus absorbing the additional $\sqrt{d}$ from worst-case-gap aggregation.*

***Simplified dominant bound.*** *When subproblems are of uniform size $d$ and updates are evenly distributed (i.e., $T_k \approx T/K$ for all $k$), the bound simplifies to:*

$$\mathbb{E}[R_{\text{avg}}(T)] = O\left( d\sqrt{T \log T} + K Q R_{\max} \sqrt{T} + L D \sqrt{\frac{dT}{K}} \right) \tag{30}$$

***Asymptotic rate.*** *The leading term is $O(d\sqrt{T \log T})$ from the bandit experts, giving sublinear average regret. The algorithm achieves:*

- **Subproblem-independent scaling:** *The bandit learning rate $O(\sqrt{T \log T})$ does not grow with $K$*

- **Graceful overlap penalty:** *Coordination cost $O(KQ\sqrt{T})$ is linear in overlap size*

- **Local refinement overhead:** *Local search adds $O(\sqrt{dT})$ which is sublinear in both $d$ and $T$*

*Proof.* By Theorem E.2, the average regret decomposes as:

$$\mathbb{E}[R_{\mathrm{avg}}(T)] \leq \sum_{e=1}^{3} p_e \phi_e(T) + \mathbb{E}[R_{\mathrm{overlap}}^{\mathrm{avg}}(T)] + \mathbb{E}[R_{\mathrm{local}}^{\mathrm{avg}}(T)] + C$$

For expert bounds from Theorems E.8, E.16, and E.17:

$$\phi_{\mathrm{UCB}}(T) = \frac{1}{K} \cdot C_{\mathrm{UCB}} \sqrt{KT \log T} = C_1 \sqrt{\frac{T \log T}{K}}$$

$$\phi_{\mathrm{Exp3}}(T) = \frac{C_2 dB \sqrt{T \log d}}{C_{clip}} \quad \text{(from Theorem E.16(c))}$$

$$\phi_{\mathrm{FTRL}}(T) = \frac{C_3 dB \sqrt{T \log T/K}}{C_{clip}} \quad \text{(from Theorem E.17(c))}$$

From Theorem E.13, we get overlap bound:

$$\mathbb{E}\left[ \sum_{t=1}^{T} C_t \right] = O(K^2 Q R_{max} \sqrt{T})$$

For average regret, divide by $K$:

$$\mathbb{E}[R_{\mathrm{overlap}}^{\mathrm{avg}}(T)] = \frac{1}{K} \cdot O(K^2 Q R_{max} \sqrt{T}) = O(KQ R_{max} \sqrt{T})$$

From the Zeroth-Order Lemma E.4, for each subproblem $k$ we get:

$$\mathbb{E}[R_{\mathrm{local}}^{(k)}(T_k)] = O(LD_k \sqrt{dT_k})$$

Summing and averaging over $K$ subproblems with $\sum_k T_k = T$:

$$\begin{aligned}
\mathbb{E}[R_{\mathrm{local}}^{\mathrm{avg}}(T)] &= \frac{1}{K} \sum_{k=1}^{K} O(LD\sqrt{dT_k}) \\
&\leq \frac{LD\sqrt{d}}{K} \sum_{k=1}^{K} \sqrt{T_k} \\
&\leq \frac{LD\sqrt{d}}{K} \cdot \sqrt{K \cdot T} \quad \text{(Cauchy-Schwarz)} \\
&= LD\sqrt{\frac{dT}{K}}
\end{aligned}$$

Combining all terms yields the stated bound. $\qquad\square$

### E.4.1. KEY TAKEAWAY

The corollary shows that the average-regret performance of ALGORITHM 1 is governed by a convex combination of the regret rates of its base experts (UCB, Exp3, FTRL), all of which are sublinear in $T$ and—crucially— depend on the *subproblem size $d$* rather than the *total problem size $n$*.

**Improvement over standard bounds.** We operate in a *combinatorial bandit* setting with full-bandit feedback: the decision space $\mathcal{X}$ is combinatorial (e.g., $|\mathcal{X}| = n!$ for permutations, $2^n$ for binary vectors), yet the learner observes only a single scalar reward per round—not per-component feedback.

Naive application of bandit algorithms to the full combinatorial space incurs regret $O(\sqrt{T|\mathcal{X}|\log|\mathcal{X}|})$, which is exponential in $n$. Even structured approaches like COMBAND (Cesa-Bianchi & Lugosi, 2012) achieve $O(n^{3/2}\sqrt{T\log n})$ under linear reward assumptions, with unavoidable dependence on the full problem size $n$.

Our decomposition-based approach achieves $O(d\sqrt{T\log T})$ for the subproblem learning component, where $d = n/K$ is the *subproblem size*. This yields:

- **Exponential-to-polynomial reduction:** We avoid dependence on the combinatorial action space size $|\mathcal{X}|$

- **Problem-size reduction:** Replacing $n$ with $d = n/K$ improves the polynomial factor by $\sqrt{K}$

- **Regret decomposition bonus:** UCB and FTRL gain an additional $\sqrt{1/K}$ factor from independent subproblem updates

Crucially, this is achieved with only *full-bandit feedback*—a single scalar reward—rather than requiring semi-bandit or per-component observations. Additionally, decomposition also does not introduce a $K$-dependent penalty in the learning rate: instead, the algorithm inherits favorable scaling while paying only additive costs for overlap coordination and local refinement.

**Interpretation of bound components**

1. **UCB term** $O(d\sqrt{T\log T/K})$: Benefits doubly from decomposition—smaller subproblem size $d$ and faster per-subproblem learning via $\sqrt{1/K}$. Compare to standard UCB: $O(\sqrt{nT\log T})$.

2. **Exp3 term** $O(d\sqrt{T\log d})$: Depends on subproblem size $d$, not total problem size $n$. This alone gives $\sqrt{K}$ improvement over standard Exp3 bound $O(\sqrt{nT\log n})$.

3. **FTRL term** $O(dB\sqrt{T\log T/K})$: Similar double benefit as UCB. The $\log T$ emerges because of the confidence-bonus FTRL analysis (Theorem E.17) under clipped-IS feedback.

4. **Overlap term** $O(KQR_{max}\sqrt{T})$: The "price of coordination." Minimized when overlap $Q$ is small relative to subproblem size $d$.

5. **Local term** $O(LD\sqrt{dT/K})$: Local refinement cost decreases with more subproblems.

**Optimal decomposition** The bounds suggest a trade-off in choosing $K$:

- **Larger** $K$ (more, smaller subproblems): Reduces UCB, FTRL, and local terms via $\sqrt{1/K}$, and reduces $d = n/K$. However, increases overlap penalty $O(KQ\sqrt{T})$.

- **Smaller** $K$ (fewer, larger subproblems): Reduces overlap penalty, but increases subproblem size $d$ and per-subproblem regret.

Balancing the Exp3 term $O((n/K)\sqrt{T})$ against the overlap term $O(KQR_{max}\sqrt{T})$ gives optimal:

$$K^* = O\left(\sqrt{\frac{n}{QR_{max}}}\right)$$

At this optimal decomposition, the total average regret scales as $O(\sqrt{nQR_{max}T\log T})$.

## E.5. Local Search Regret Bound

---

**Algorithm 3** LOCALREFINE: Zeroth-Order Subproblem Optimization

---

**Require:** Solution $x$, subproblem indices $S \subseteq [n]$, metric $d : \mathcal{X} \times \mathcal{X} \to \mathbb{R}_+$, *hyperparameters:* smoothing $\delta$, step size $\alpha$
**Output:** Improved solution $x'$
**Initialize:** $x' \leftarrow x$
1: **for** $\tau = 1, \dots, R$ **do**
2:      Sample direction $u \sim \mathcal{U}(\mathbb{S}_d \cap S)$            ▷ Unit direction in metric, restricted to $S$
3:      $\hat{g} \leftarrow \frac{f(x' + \delta u) - f(x')}{\delta} \cdot u$            ▷ One-point gradient estimator
4:      $x' \leftarrow \Pi_{\mathcal{X}}\left[x' + \alpha \hat{g}\right]$            ▷ Projected gradient ascent
5: **end for**
6: **return** $x'$

---

**Lemma E.4** (**Zeroth-Order Discrete Optimization Regret**). *Consider a subproblem $k$ with decision variables indexed by $I_k$, $|I_k| = d$. Let $f : \mathcal{X} \longrightarrow \mathbb{R}$ be the objective function where $\mathcal{X}^k$ is a discrete local domain (permutations or binary vectors) $\mathcal{X}^k \subseteq \mathcal{X}$. Since $f$ is L-Lipschitz (following our Assumption E.2.2) with respect to a discrete metric $d_{\mathcal{X}}$:*

$$|f(\boldsymbol{x}) - f(\boldsymbol{y})| \leq L \cdot d_{\mathcal{X}}(x, y)$$

*Under zeroth-order optimization with $T_k$ iterations (where $T_k < T$), gradient estimation via random perturbations, step size $\eta_t = \frac{\eta_0}{\sqrt{t}}$ for $t \in \{1, \dots, T_k\}$, and the perturbation radius, $\sigma = \frac{1}{\sqrt{T_k}}$. Then the expected regret satisfies:*

$$\mathbb{E}\left[\sum_{t=1}^{T_k}[f(\boldsymbol{x}_t)] - f(\boldsymbol{x}^*)\right] = O(\sqrt{dT_k}) \tag{31}$$

*where $\boldsymbol{x}^* \in \arg\min_{x \in \mathcal{X}^k} f(x)$ is the optimal solution within the subproblem.* ALGORITHM *3 outlines such a algorithm.*

*Proof.* Let $x_t$ denote solution at any iteration $t$, $\{x_t\}_{t=1}^{T_k}$ are the iterates produced by Algorithm 3 on subproblem $k$ and if the instantaneous subproblem regret is defined as $r_t := f(x_t) - f(x^*)$, then we analyze the total local regret,

$$R_{\text{local}}^{(k)}(T_k) := \sum_{t=1}^{T_k} r_t = \sum_{t=1}^{T_k} \left[f(x_t) - f(x^*)\right]$$

Since, our target is to solve for a subproblem locally, at each iteration $t$, we start from $x_t$ & sample a random perturbation according to a symmetric distribution over neighbors in a metric ball of radius controlled by the perturbation parameter (s.t $d_{\mathcal{X}}(x_t, x_t^+) \leq 1$ for the perturbed point $x_t^+$).

This describes a standard *gradient-free OCO method with random perturbations* where the true sub-gradients of $f$ are never observed, but a zeroth-order direction estimate $\hat{g}_t$ serves as a noisy gradient proxy, and the update is performed with step size $\eta_t = \eta_0/\sqrt{t}$. By construction, the estimator $\hat{g}_t$ satisfies the usual assumptions for gradient-free OCO (see, e.g., (Flaxman et al., 2005; Hazan, 2016b)):

$$\mathbb{E}[\hat{g}_t \mid x_t] \in \partial f(x_t), \qquad \mathbb{E}\left[\|\hat{g}_t\|^2 \mid x_t\right] \leq G^2, \tag{32}$$

for some constant $G > 0$ depending only on problem parameters. To bound $G^2$, note that each admissible local perturbation affects at most $d$ coordinates, and by Lipschitz continuity ,

$$|f(x_t^+) - f(x_t)| \leq L \cdot d_{\mathcal{X}}(x_t^+, x_t) \leq L$$

Now, consider a coordinate-wise one-point estimator. At any iteration $t$, choose a coordinate $i_t \in I_k$ uniformly at random and let $x_t^{(i_t)}$ denote the neighbor obtained by applying the local perturbation at coordinate $i_t$. Let the finite-difference $\Delta_t := f(x_t^{(i_t)}) - f(x_t)$, which satisfies $|\Delta_t| \leq L$ by the above Lipschitz bound (Assumption E.2.2). A standard one-point estimator (Flaxman et al., 2005) is then defined as,

$$\hat{g}_t := \sqrt{d} \frac{\Delta_t}{\sigma} e_{i_t}$$

where $e_{i_t}$ is the $i_t$-th basis vector and $\sigma$ is the (fixed) perturbation radius, absorbed into constants. Since only one coordinate of $\hat{g}_t$ is nonzero,

$$\|\hat{g}_t\|^2 = d\left(\frac{\Delta_t}{\sigma}\right)^2 \leq d\left(\frac{L}{\sigma}\right)^2$$

Taking expectation over the random choice of $i_t$ (uniform over at most $d$ coordinates) yields,

$$\mathbb{E}[\|\hat{g}_t\|^2 \mid x_t] \leq d\left(\frac{L}{\sigma}\right)^2$$

Absorbing $\sigma^{-2}$ and numerical constants into $C_1$, we obtain the variance bound

$$G^2 := \sup_t \mathbb{E}\left[\|\hat{g}_t\|^2 \mid x_t\right] \leq C_1\, d\, L^2$$

for some universal constant $C_1 > 0$. Let $D_k := \max_{x,y \in \mathcal{X}^k} \delta(x, y)$ denote the diameter of the subproblem domain $(\mathcal{X}^k, d)$. Next, by the standard projected online gradient descent potential argument (see, e.g., (Hazan, 2016b)), for any $x^* \in \mathcal{X}^k$ and updates $x_{t+1} = \Pi(x_t - \eta_t \hat{g}_t)$ on a domain of diameter $D_k$, we have

$$\sum_{t=1}^{T_k} \langle \hat{g}_t, x_t - x^* \rangle \leq \frac{D_k^2}{2\eta_{\min}} + \frac{1}{2}\sum_{t=1}^{T_k} \eta_t \|\hat{g}_t\|^2,$$

where $\eta_{\min} := \min_t \eta_t$. To relate the inner products to regret, we assume a standard local convexity relaxation i.e. there exists a convex function $F^{(k)}$ defined on $\mathrm{conv}(\mathcal{X}^k)$ such that $F^{(k)}(x) = f(x) \ \forall\, x \in \mathcal{X}^k$ (e.g., a convex relaxation or convex envelope). Let $g_t \in \partial F^{(k)}(x_t)$ be a sub-gradient, then by convexity of $F^{(k)}$ we have, for any $x^* \in \arg\min_{x \in \mathcal{X}^k} F^{(k)}(x)$,

$$F^{(k)}(x_t) - F^{(k)}(x^*) \leq \langle g_t, x_t - x^* \rangle$$

The existence of such a convex extension is standard for combinatorial optimization problems; see (Lovász, 1983) for submodular functions and (Schrijver, 2003) for general polyhedral relaxations.

Since $F^{(k)}(x) = f(x)$ on $\mathcal{X}^k$, this is exactly $f(x_t) - f(x^*)$ on our discrete domain. The key observation is that $g_t := \mathbb{E}[\hat{g}_t \mid x_t]$ satisfies $g_t \in \partial F^{(k)}(x_t)$ by property (32). Therefore, by local convexity of $F^{(k)}$:

$$f(x_t) - f(x^*) = F^{(k)}(x_t) - F^{(k)}(x^*) \leq \langle g_t, x_t - x^* \rangle = \langle \mathbb{E}[\hat{g}_t \mid x_t], x_t - x^* \rangle$$

Taking expectations over the randomness in $\hat{g}_t$:

$$\mathbb{E}[f(x_t) - f(x^*)] \leq \mathbb{E}[\langle \hat{g}_t, x_t - x^* \rangle]$$

Summing over $t$ and applying the OGD potential bound yields:

$$\sum_{t=1}^{T_k} \mathbb{E}[f(x_t) - f(x^*)] \leq \frac{D_k^2}{2\eta_{\min}} + \frac{1}{2}\sum_{t=1}^{T_k} \eta_t\, G^2$$

Substituting $\eta_t = \eta_0/\sqrt{t}$ and $\eta_{min} = \eta_{T_k} = \frac{\eta_0}{\sqrt{T_k}}$ in above, we obtain,

$$\mathbb{E}\left[R_{\mathrm{local}}^{(k)}(T_k)\right] \leq \frac{D_k^2}{2\eta_{min}} + \frac{G^2}{2}\sum_{t=1}^{T_k} \eta_t = \frac{D_k^2 \sqrt{T_k}}{2\eta_0} + \frac{G^2 \eta_0}{2}\sum_{t=1}^{T_k} \frac{1}{\sqrt{t}} \leq \frac{D_k^2 \sqrt{T_k}}{2\eta_0} + C' G^2 \eta_0 \sqrt{T_k}$$

For some constant $C' > 0$. Optimizing over $\eta_0$ for the tightest upper bound,

$$\eta_0 \asymp \frac{D_k}{G\sqrt{2C'}}$$

Substituting it back,

$$\mathbb{E}\big[R_{\mathrm{local}}^{(k)}(T_k)\big] \leq GD_k\sqrt{2C'T_k}$$

Finally, using $G^2 \leq C_1 dL^2$, we have $G \leq \sqrt{C_1}L\sqrt{d}$, and hence,

$$\mathbb{E}\big[R_{\mathrm{local}}^{(k)}(T_k)\big] \leq LD_k\sqrt{2C_1 C' dT_k} = CLD_k\sqrt{dT_k}$$

for an appropriate constant $C > 0$ that absorbs $2C_1 C'$, giving regret bound of $O(LD_k\sqrt{dT_k})$. $\qquad\square$

*Remark* E.5 (Practical Implementation of Algorithm 3). The deployed implementation is a discrete realization of the lemma's algorithmic family. Unit perturbations are taken under the natural metric for each domain: Hamming (single bit flip) for binary, and Cayley distance (any transposition is unit) for permutations, under which any single swap is unit. The lemma's metric-generic Lipschitz assumption permits this choice. For permutation domain the Lipschitz constant under Cayley is comparable to Kendall tau since any transposition affects at most four edges. The metric choice does not affect the algorithm or experiments: we use Cayley here purely to certify the unit-perturbation property for the lemma and Kendall tau remains the metric of reference elsewhere in the paper. In all cases $d_{\mathcal{X}}(x_t, x_t^+) \leq 1$ and $|\Delta_t| \leq L$ hold. The continuous schedule $\eta_t = \eta_0/\sqrt{t}$ is realized by a bounded integer step count $m_t \in \{1, 2, 3\}$ per round; bounded support is sufficient for the OGD potential bound. When $\Delta_t \leq 0$, the discrete anti-direction is degenerate (the unit operators above are self-inverse: applying them twice returns to $x_t$). We substitute an independent unit-neighbor probe drawn from the same symmetric distribution. By symmetry of the perturbation distribution, the next round's estimator still satisfies $\mathbb{E}[\hat{g}_t \mid x_t] \in \partial f(x_t)$, preserving the gradient-free OCO assumptions used in the proof. Each invocation of the subroutine corresponds to one element of $\mathcal{T}_k$ with a bounded inner budget $R$. Constants absorbed by these discretizations enter $C$ and the $O(LD_k\sqrt{dT_k})$ rate is preserved.

### E.6. Global Experts Regret Bound

**Definition E.6.** (**Decomposed Regret**) Consider the online game over $T$ iterations as described in Section 4. Let $x^\star$ be the best fixed comparator in hindsight and the instantaneous regret at iteration $t$ defined as $r_t := f(x^\star) - f(x_t)$. Let $h$ captures the coupling effects (interaction over overlapping coordinates). Under Assumption E.2.3, there exist subproblem-wise surrogate objectives $g_k$ such that the instantaneous regret admits the upper bound

$$r_t := f(x^\star) - f(x_t) \leq \underbrace{\sum_{k=1}^{K}\big[g_k((x^\star)^{(k)}) - g_k(x_t^{(k)})\big]}_{\text{Subproblem regret }S_t} + \underbrace{h((x^*)^{(O)}) - h(x_t^{(O)})}_{\text{Coupling Error }C_t}, \tag{33}$$

Therefore, the cumulative (total) regret over T iterations decomposes as:

$$R_{\mathrm{global}}(T) := \sum_{t=1}^{T} r_t \leq \sum_{t=1}^{T}\sum_{k=1}^{K}\big[g_k((x^\star)^{(k)}) - g_k(x_t^{(k)})\big] + \underbrace{\sum_{t=1}^{T} C_t}_{\text{Total Coupling Error}} \tag{34}$$

where the cumulative coupling error $\sum_t C_t$ is bounded in Theorem E.13. The functions $g_k$ are surrogate subproblem objectives introduced solely for analysis; no additive decomposition of the true objective $f$ is assumed. Their existence is guaranteed by Assumption 3 (Bounded Coupling), which ensures that the effect of modifying any subproblem $k$ can be upper-bounded by a local term plus an overlap penalty.

Note, average subproblem regret defined in Theorem E.1 relates $R_{\mathrm{global}}(T)$ and $R_{\mathrm{avg}}(T)$ as:

$$R_{\mathrm{global}}(T) \leq K \cdot R_{\mathrm{avg}}(T) + \sum_{t=1}^{T} C_t$$

Subproblem regret captures optimization quality (expert-algorithm dependent), while the coupling error captures the coordination cost induced by overlap (problem-structure dependent). The regret is decomposed via surrogate subproblem objectives that upper-bound the effect of block-wise deviations, while all non-local interactions are explicitly absorbed into a bounded coupling error.

E.6.1. UCB REGRET

Let's first recall a standard UCB-type regret bound.

**Lemma E.7** (**Standard UCB Regret**). *Consider a $K$-armed stochastic bandit where each arm $i \in \{1, \ldots, K\}$ yields rewards in $[0, 1]$ with mean $\mu_i$. Let $I_t$ be the arm selected at time $t$ by a UCB algorithm and let $i^\star \in \arg\max_i \mu_i$. Then, with high probability,*

$$\sum_{t=1}^{T} \left( \mu_{i^\star} - \mu_{I_t} \right) \leq \tilde{O}(\sqrt{KT}), \tag{35}$$

*where $\tilde{O}$ hides logarithmic factors in $T$ and $K$.*

E.6.2. DECOMPOSED SUBPROBLEM REGRET.

Let $\mathcal{T}_k \subseteq \{1, \ldots, T\}$ denote the set of rounds in which subproblem $k$ is actively updated, and let $T_k = |\mathcal{T}_k|$. Define the subproblem regret up to time $T_k$ as

$$R_k(T_k) := \sum_{t \in \mathcal{T}_k} \left[ g_k(x_k^\star) - g_k(x_t^{(k)}) \right], \tag{36}$$

where $x_k^\star$ is the optimal restriction of $x^\star$ to subproblem $k$, and $x_t^{(k)}$ is the restriction of $x_t$ to $\mathcal{X}^k$. Intuitively, $R_k(T_k)$ measures how well algorithm learns the best configuration for subproblem $k$.

**Theorem E.8** (**UCB Regret under Decomposed Subproblems (Position–Value UCB)**). *Let Assumption E.2.3 and the decomposed regret representation in Definition E.6 hold. Let $\{\mathcal{T}_k\}_{k=1}^{K}$ be the subproblem update sets with $T_k := |\mathcal{T}_k|$ and $\sum_{k=1}^{K} T_k = T$. Assume the UCB expert is implemented as* position–value UCB *within each active subproblem $k$, maintaining estimates $\hat{\mu}_{i,j}$ and counts $N_{i,j}$ for each position–value pair $(i, j)$, and updating, at each $t \in \mathcal{T}_k$, all positions $i \in I_k$ ($I_k$ is the index set of positions in subproblem $k$) for the realized value $x_{i,t}$ using the same normalized scalar reward $r_t \in [0, 1]$. Further assume Assumption E.2.5 (A.5') holds for the UCB expert. Define the UCB subproblem regret on $k$ as*

$$R_k^{\text{UCB}}(T_k) := \sum_{t \in \mathcal{T}_k} \left[ g_k(x_k^\star) - g_k(x_t^{(k)}) \right], \qquad x_k^\star := (x^\star)^{(k)} = x_{I_k}^\star.$$

*Let $|I_k| := d_k$ and let $A_{\max} := \max_{i \in [n]} |\mathcal{A}_i|$. The average subproblem regret $R_{\text{subproblem}}^{\text{avg}}(T) := \frac{1}{K} \sum_{k=1}^{K} R_k^{UCB}(T_k)$. Then,*

*(a) (**Per-subproblem UCB regret**) There exists a universal constant $c > 0$ such that for each subproblem $k$,*

$$\mathbb{E}\left[ R_k^{\text{UCB}}(T_k) \right] \leq c\,|I_k|\,\sqrt{A_{\max}\,T_k\,\log T}. \tag{37}$$

*(b) (**Total regret across subproblems**) Summing over $k$, the total UCB-induced subproblem regret satisfies*

$$\sum_{k=1}^{K} \mathbb{E}\left[ R_k^{\text{UCB}}(T_k) \right] \leq C_{\text{UCB}}\,\sqrt{KT\,\log T}.$$

*If the subproblem sizes are uniformly bounded, i.e., $\max_k |I_k| \leq d$, then for some constant $C_{\text{UCB}} := cd\sqrt{A_{\max}} \geq 0$ is independent of the time horizon $T$. Consequently, the UCB contribution to the global decomposed regret satisfies*

$$\mathbb{E}[R_{\text{UCB}}(T)] \leq C_{\text{UCB}}\,\sqrt{KT\,\log T} \;+\; \sum_{t=1}^{T} \mathbb{E}[C_t], \tag{38}$$

*where $C_t$ is the coupling error term from Definition E.6.*

*Proof.* *(a)* For a fixed subproblem $k$ with index set $I_k$ and activation set $\mathcal{T}_k$ of size $T_k$. By Lemma E.33, under Assumption E.2.5 (A.5'), the position–value UCB expert operating on subproblem $k$ satisfies

$$R_k^{\text{UCB}}(T_k) = \sum_{t \in \mathcal{T}_k} \left[ g_k(x_k^\star) - g_k(x_t^{(k)}) \right] \;\leq\; c\,|I_k|\,\sqrt{A_{\max}\,T_k\,\log T},$$

for some universal constant $c > 0$, where $A_{\max}$ bounds the number of admissible values per position. Taking expectations yields (37).

*(b)* Summing the bound (37) over $k = 1, \dots, K$ gives

$$\sum_{k=1}^{K} \mathbb{E}\big[R_k^{\mathrm{UCB}}(T_k)\big] \ \le \ c\sqrt{A_{\max}\log T}\sum_{k=1}^{K} |I_k|\sqrt{T_k}.$$

If the subproblem sizes $I_k$ are uniformly bounded $|I_k| \le d$ and we can absorb $cd\sqrt{A_{max}}$ into a constant and obtain

$$\sum_{k=1}^{K} \mathbb{E}\big[R_k^{\mathrm{UCB}}(T_k)\big] \le C_{\mathrm{UCB}}\sqrt{\log T}\sum_{k=1}^{K}\sqrt{T_k}.$$

for some $C_{\mathrm{UCB}} > 0$, establishing (37). Next by applying the Cauchy–Schwarz inequality and since by construction $\sum_{k=1}^{K} T_k = T$ using this yields,

$$\sum_{k=1}^{K}\sqrt{T_k} \ \le \ \sqrt{K\sum_{k=1}^{K} T_k} = \sqrt{KT}.$$

Therefore,

$$\sum_{k=1}^{K} \mathbb{E}\big[R_k^{\mathrm{UCB}}(T_k)\big] \le C_{\mathrm{UCB}}\sqrt{KT\log T}.$$

Finally, by the decomposed regret upper bound in (34), the cumulative regret incurred by the UCB component satisfies

$$R_{\mathrm{global}}(T) \le \sum_{k=1}^{K} R_k^{\mathrm{UCB}}(T_k) + \sum_{t=1}^{T} C_t.$$

Taking expectations and substituting the above bound yields

$$\mathbb{E}[R_{\mathrm{global}}] \le \mathbb{E}[R_{\mathrm{UCB}}(T)] + \sum_{t=1}^{T}\mathbb{E}[C_t] \le C_{\mathrm{UCB}}\sqrt{KT\log T} + \sum_{t=1}^{T}\mathbb{E}[C_t],$$

where restricting to UCB bandit component establishes (38). $\qquad\square$

*Remark* E.9 (Scope of Assumption E.2.5 (A.5')). Assumption E.2.5 (A.5') (local additivity and exogeneity) is required **only** for the UCB expert: additivity decomposes subproblem regret into per-position regrets, and exogeneity ensures the action-independent bias cancels when comparing values, making Hoeffding concentration applicable. EXP3 and FTRL do not require either component, as their regret bounds hold under adversarial and bounded losses respectively, requiring only Assumptions E.2.1–E.2.4. Therefore, the coupling-error bound of Theorem E.13 and the $O(\sqrt{T})$ rate hold under Assumptions E.2.1–E.2.4 alone whenever the algorithm selects EXP3 or FTRL. Assumption A.5' is a sufficient condition that enables tighter, instance-dependent bounds via UCB, **but is not necessary for the overall rate.**

Importantly, our additivity condition is considerably weaker than the global additivity assumptions common in the bandit literature (Auer et al., 2002a; Lattimore & Szepesvári, 2020): it applies only to the *local surrogate* $g_k$ within each subproblem of size $d \ll n$, not to the global objective $f$. Since subproblems cover small contiguous blocks, interaction effects are confined to block boundaries and bounded by the Lipschitz constant (Assumption E.2.2). When this local approximation is poor, the algorithm adaptively switches to EXP3 or FTRL, preserving the $O(\sqrt{T})$ guarantee without Assumption A.5'.

E.6.3. COUPLING ERROR FROM OVERLAPPING POSITIONS

When a position $i$ appears in multiple subproblems (i.e., $i \in O_{ml}$ for some subproblem pairs $m, l$), the algorithm must coordinate value assignments across these subproblems. There are two natural approaches to handle such overlapping positions:

1. **Separate solutions per subproblem**: Each subproblem maintains a local assignment on its indices; overlapping positions induce consensus constraints. Local assignments are reconciled post-hoc into a global solution by counting discrete disagreements on overlapping positions. This approach provides no coordination signal during learning.

2. **Single global solution with uncertainty penalties**: Maintain a single global solution shared across all subproblems and penalize uncertainty on overlapping positions during optimization. This enables proactive coordination through the learning process.

We **adopt the second approach**, as it allows subproblems to share learning signals on overlapping positions, improving sample efficiency and coordination during optimization. This also enables tighter regret bounds as discrete disagreement counting lacks optimization structure and yields only loose worst-case guarantees.

Thus, in our decomposition-based framework, consecutive subproblems share overlapping positions to ensure solution connectivity. By maintaining a single global solution across all subproblems, we can quantify and penalize the risk of coordination conflicts during optimization through a soft violation measure (defined in Appendix E.8.4), rather than counting discrete disagreements after the fact.

**Overlapping Positions and Coordination Requirements.** For subproblems $m$ and $l$, let $O_{ml} = I_m \cap I_l$ denote their set of shared positions. Let $k_i = |\{m : i \in I_m\}|$ denote the number of subproblems containing position $i$. Positions with $k_i > 1$ require coordination across multiple subproblems to ensure a consistent global solution. When subproblem $m$ assigns value $v_m$ to position $i \in O_{ml}$ while subproblem $l$ assigns a different value $v_l \neq v_m$, a *coupling conflict* occurs. Since our global solution assigns a single value to each position, conflicts indicate potential sub-optimality, at least one subproblem is not using its preferred value.

**Definition E.10** (**Coupling Error**). The coupling error at iteration $t$ measures the total disagreement induced by overlapping subproblems:

$$H^{(t)} = \sum_{m<l} \sum_{i \in O_{ml}} \mathbb{I}[\text{subproblems } m, l \text{ disagree at position i}] = \sum_{m<l} \sum_{i \in O_{ml}} \mathbb{I}\left[x_{m,i}^{(t)} \neq x_{l,i}^{(t)}\right] \tag{39}$$

where the indicator function $\mathbb{I}[\text{subproblems } m, l \text{ disagree at position } i] = 1$ if $x_{m,i}^{(t)} \neq x_{l,i}^{(t)}$ (i.e., subproblems $m$ and $l$ assign different values to the shared position $i \in O_{ml}$), and 0 otherwise. Thus $H^{(t)}$ counts the position-level disagreements across all subproblem pairs. The objective coupling error $C_t$ is bounded by this count weighted by the per-position regret (Proposition E.12), so the cumulative coupling error over $T$ iterations satisfies:

$$\sum_{t=1}^{T} C_t \leq R_{\text{max-per-coupling}} \sum_{t=1}^{T} H^{(t)}. \tag{40}$$

Our objective is to bound this cumulative coupling error while simultaneously learning the optimal solution through bandit feedback.

**Regret from Coupling Conflicts** When subproblems disagree on overlapping positions, the global solution must choose one assignment over another, incurring potential regret. We now formalize this regret and derive worst-case bounds.

**Definition E.11** (**Maximum Per-Position Regret**). Let $\rho(i, v)$ denote the reward contribution of assigning value $v$ to position $i$. The maximum regret incurred by resolving a single coupling conflict is:

$$R_{max-per-coupling} = \max_{i \in [n], \, v_m, v_l \in [n]} |\rho(i, v_m) - \rho(i, v_l)| \tag{41}$$

This quantity characterizes the worst-case penalty for choosing one sub-problem's assignment over another's at a single position. Additionally, throughout this subsection, $x_{m,i}^{(t)}$ denotes the *locally preferred value* for subproblem $m$ at position $i$ prior to global reconciliation, not the realized coordinate of the single global solution.

**Proposition E.12** (Worst-Case Coupling Error Bound). *The objective coupling error satisfies the following worst-case bounds:*

$$C_t \leq R_{max-per-coupling} \sum_{m<l} \sum_{i \in O_{ml}} \mathbb{I}[x_{m,i}^{(t)} \neq x_{l,i}^{(t)}] = R_{max-per-coupling} \, H^{(t)} \tag{42}$$

$$\mathbb{E}[C_t] \leq R_{max-per-coupling} \sum_{m<l} \sum_{i \in O_{ml}} \mathbb{P}[x_{m,i}^{(t)} \neq x_{l,i}^{(t)}] \tag{43}$$

*Proof.* By Definition E.11, each overlapping position $i \in O_{ml}$ at which subproblems $m, l$ prefer different values contributes at most $R_{max-per-coupling}$ to the coupling error, since one subproblem is forced off its preferred value upon reconciliation into the single global solution. Summing this per-position penalty over all subproblem pairs $(m, l)$ yields (42):

$$C_t \leq R_{max-per-coupling} \sum_{m<l} \sum_{i \in O_{ml}} \mathbb{I}[x_{m,i}^{(t)} \neq x_{l,i}^{(t)}].$$

The expected regret bound (43) follows by multiplying each position-level disagreement by $R_{max-per-coupling}$ and taking expectations. □

**Key Insight.** Equation (43) reveals that controlling coupling error requires reducing the disagreement probability $\mathbb{P}[x_{m,i}^{(t)} \neq x_{l,i}^{(t)}]$ for overlapping positions. This probability depends on:

1. *Overlap structure*: Large overlap sets $O_{ml}$ and many subproblem pairs increase potential conflicts

2. *Coordination quality*: How well subproblems agree on shared positions during learning

**Lagrangian Coordination: Our Approach**    A naive approach would solve each subproblem independently and resolve conflicts through post-hoc tie-breaking. However, this provides no coordination signal during learning, potentially leading to linear accumulation of coupling error over $T$ iterations. Instead, we employ a ***Lagrangian coordination mechanism*** that proactively reduces coupling conflicts during optimization. See ALGORTHIM 4, outlining our approach which consists of three components:

**1. Soft Violation Measure.**    For position $i$ at iteration $t$, we define a continuous measure that quantifies coordination risk:

$$\xi_i^{(t)} = (k_i - 1) \cdot \text{Var}(\hat{\mu}_i^{(t)}) \cdot \left(1 - \frac{N_{i,v_i^{(t)}}}{\sum_v N_{i,v}}\right) \tag{44}$$

where:

- $k_i = |\{m : i \in I_m\}|$ is the number of subproblems containing position $i$

- $\text{Var}(\hat{\mu}_i^{(t)}) = \frac{1}{n} \sum_v (\hat{\mu}_{i,v}^{(t)} - \bar{\mu}_i^{(t)})^2$ is the variance of value estimates

- $N_{i,v}$ is the visit count for position-value pair $(i, v)$

- $v_i^{(t)}$ is the value assigned to position $i$ at iteration $t$

The soft violation $\xi_i^{(t)}$ is high when: (a) position $i$ appears in many subproblems ($k_i$ large), (b) value estimates have high variance (indicating disagreement potential), and (c) the assigned value has low visit ratio (indicating uncertainty). Crucially, as learning progresses, $\text{Var}(\hat{\mu}_i^{(t)}) \to 0$ and the visit ratio for optimal values approaches 1 (i.e. the confidence factor $(1 - \text{visit ratio}) \to 0$), ensuring $\xi_i^{(t)} \to 0$. The coordination penalty therefore relaxes automatically once each overlapping position has settled on its preferred value.

**2. Lagrangian Dual Variables.** For each position $i$ with $k_i > 1$, we maintain a dual variable $\lambda_i^{(t)} \geq 0$ that penalizes the coordination violations on overlapping positions. These dual variables are updated via *Polyak heavy-ball-accelerated mirror descent* (entropic geometry) on the dual objective:

$$\lambda_i^{(t+1)} = \Pi_{[0,\lambda_{\max}]} \left[ \exp \left( \log(\lambda_i^{(t)}) + \alpha_t \xi_i^{(t)} \right) \right] \tag{45}$$

with diminishing step size $\alpha_t = \alpha_0/\sqrt{t}$ and Polyak heavy-ball momentum for acceleration. This choice is consistent with the EXP3/FTRL-style multiplicative dynamics used by the primal learners and yields standard sublinear in T regret in the dual.

---

**Algorithm 4** DUALUPDATE: (Entropic) Mirror Descent for Lagrangian Coordination

---

**Require:** Soft violations $\{\xi_i^{(t)}\}_{i=1}^n$, step size $\alpha_t = \alpha_0/\sqrt{t}$, dual variables $\{\lambda_i^{(t)}\}_{i=1}^n$
 1: **for** each position $i$ with $k_i > 1$ **do**
 2:     $\lambda_i^{(t+1)} \leftarrow \Pi_{[0,\lambda_{\max}]} \left( \lambda_i^{(t)} \cdot \exp(\alpha_t \xi_i^{(t)}) \right)$           ▷ Update dual variables
 3: **end for**
 4: **return** updated dual variables $\{\lambda_i^{(t+1)}\}$

---

The soft violation $\xi_i^{(t)}$ aggregates disagreement risk across all overlapping subproblems containing position $i$, eliminating the need for pairwise dual variables. The multiplicative update preserves nonnegativity of the dual variables and naturally adapts the coordination strength to the uncertainty level of each overlapping position.

**3. Score Modification.** The dual variables modify expert scores during subproblem optimization. For position $i$ with value $v$, the Lagrangian-modified score is:

$$\tilde{s}_{i,v}^{(t)} = s_{i,v}^{(t)} - (k_i - 1)\lambda_i^{(t)} \tag{46}$$

where $s_{i,v}^{(t)}$ is the base score (from UCB or Hedge) and the penalty term $(k_i - 1)\lambda_i^{(t)}$ discourages high-risk assignments on overlapping positions. Although the penalty is value-independent for a fixed position $i$, it affects future learning dynamics by uniformly discouraging high-uncertainty assignments on heavily overlapping positions, thereby accelerating variance reduction and coordination across subproblems rather than enforcing hard consensus at each step.

*A possible extension would be to introduce value-dependent coordination penalties that explicitly discourage disagreement with a consensus assignment (e.g., penalizing $v \neq z_i^{(t)}$). We do not pursue this here, as preliminary experiments suggest limited impact on the overall learning dynamics relative to the simpler uncertainty-based penalty.*

**Coordination Mechanism.** This three-component approach creates a feedback loop:

1. Positions with high variance or low confidence accumulate large soft violations $\xi_i^{(t)}$

2. Large violations cause dual variables $\lambda_i^{(t)}$ to grow via mirror descent

3. Large dual variables penalize uncertain choices more heavily via score modification

4. Penalties encourage exploration until confident, coordinated values emerge

5. As confidence increases, $\xi_i^{(t)} \to 0$ and penalties naturally diminish

The detailed Lagrangian formulation, mirror descent update rules, and convergence analysis are provided in Section E.8.4. Using this mechanism, we prove:

**Theorem E.13** (Coupling Error Bound). *Under the Lagrangian coordination mechanism under accelerated mirror descent on the dual variables, the expected cumulative coupling error satisfies:*

$$\mathbb{E}\left[ \sum_{t=1}^{T} C_t \right] = O(K^2 Q R_{max} \sqrt{T}) \tag{47}$$

*where $K$ is the number of subproblems, $Q = \max_{m,l} |O_{ml}|$ is the maximum overlap size, and $R_{max} = \max_{i,v_m,v_l} |\rho(i, v_m) - \rho(i, v_l)|$ is the maximum per-position regret.*

*Proof.* We prove the bound in three parts.

(1) **Soft Violations are Bounded.** By construction Equation 44, each component of $\xi_i^{(t)}$ is bounded - $(k_i - 1) \le K - 1$ (overlap count), for rewards in $[0, 1]$ the $\mathrm{Var}(\hat{\mu}_i^{(t)}) \le \frac{1}{4}$, $(1 - \text{visit ratio}) \in [0, 1]$. Therefore $\|\xi_t\|_\infty \le (k_i - 1) \cdot \frac{1}{4} \le \frac{K-1}{4} =: L = O(K)$.

(2) **Cumulative Soft Violations are Bounded.** By Lemma E.37, the cumulative dual gap satisfies:

$$\sum_{t=1}^{T} (g(\lambda^*) - g(\lambda_t)) \le O(d\lambda_{max} L\sqrt{T}) = O(d\lambda_{max} K\sqrt{T}) \tag{48}$$

The mirror-descent bound above certifies that the dual coordination updates incur only sublinear $O(\sqrt{T})$ dual regret. To bound the cumulative soft violations directly, note $\xi_i^{(t)} = (k_i - 1)\sigma_i^2(t), (1 - r_i^{(t)}) \le \frac{K-1}{4}(1 - r_i^{(t)})$, using $\sigma_i^2 \le \frac{1}{4}$ for rewards in $[0, 1]$ and $k_i - 1 \le K - 1$. By Proposition E.39, the visit ratio satisfies $r_i^{(t)} \to 1$ after burn-in with $\sum_{t=1}^{T}(1 - r_i^{(t)}) = O(\sqrt{T \log T})$. Summing over the $\le KQ$ overlapping positions:

$$\sum_{t=1}^{T} \mathbb{E}[\|\xi_t\|_1] = O(K^2 Q\sqrt{T \log T}). \tag{49}$$

(3) **Coupling Error Bounded by Soft Violations.** The coupling error at iteration $t$ counts disagreements weighted by $R_{max}$. From Proposition E.12:

$$C_t \le R_{max} \sum_{m<l} \sum_{i\in O_{ml}} \mathbb{I}[x_{m,i}^{(t)} \ne x_{l,i}^{(t)}] \tag{50}$$

To bound the disagreement indicators, we apply the exact disagreement law (Lemma E.38), which shows that the probability of disagreement at position $i$ is controlled by the non-concentration of the shared selection distribution:

$$\mathbb{E}[C_t] \le R_{\max} \sum_{m<l} \sum_{i\in O_{ml}} 2(1 - q_{i,\max}^{(t)}). \tag{51}$$

We then connect $1 - q_{i,\max}^{(t)}$ to the soft violation $\xi_i^{(t)}$ using the per-position argument in Lemma E.41, which establishes that for gapped positions ($\Delta_i > 0$):

$$\mathbb{E}\left[2(1 - q_{i,\max}^{(t)})\right] \le \frac{c_0}{k_i - 1} \cdot \mathbb{E}[\xi_i^{(t)}] + \epsilon_i(t),$$

where $c_0 = 32n/\Delta_{\min}^2$ and $\sum_{t=1}^{T} \epsilon_i(t) = O(\sqrt{T \log T})$. Positions with $\Delta_i = 0$ contribute zero coupling error since all values yield identical reward. Now, bringing everything together. From part (3), substituting into the coupling error and aggregating over subproblem pairs,

$$\mathbb{E}[C_t] \le R_{\max} \sum_{m<l} \sum_{i\in O_{ml}} \left(\frac{c_0}{k_i - 1} \cdot \mathbb{E}[\xi_i^{(t)}] + \epsilon_i(t)\right)$$

$$\le \frac{c_0 K}{2} \cdot R_{\max} \cdot \mathbb{E}[\|\xi_t\|_1] + \binom{K}{2} Q \cdot R_{\max} \cdot \max_i \epsilon_i(t), \tag{52}$$

where we used $|\{(m, l) : i \in O_{ml}\}|/(k_i - 1) \le \binom{k_i}{2}/(k_i - 1) = k_i/2 \le K/2$ for overlapping positions.

The final bound is obtained by summing over $T$ iterations,

$$\sum_{t=1}^{T} \mathbb{E}[C_t] \le \frac{c_0 K}{2} \cdot R_{\max} \sum_{t=1}^{T} \mathbb{E}[\|\xi_t\|_1] + K^2 Q \cdot R_{\max} \sum_{t=1}^{T} \max_i \epsilon_i(t)$$

$$\le \frac{c_0 K}{2} \cdot R_{\max} \cdot \underbrace{O(d\,\lambda_{\max}\sqrt{T})}_{\text{Step (2)}} + K^2 Q \cdot R_{\max} \cdot \underbrace{O(\sqrt{T \log T})}_{\text{Lemma E.41}}$$

$$= O(K^2 Q R_{\max}\sqrt{T}), \tag{53}$$

where $d \leq K \cdot Q$ is the total number of overlapping positions. The $K^2$ in the final bound arises as follows. In the first term, the factor $K/2$ from pair counting Eq. (52) multiplies the factor $K$ from $d \leq K \cdot Q$ in the mirror-descent bound of Step (2), giving $K^2/2$. In second term, $\binom{K}{2} \leq K^2/2$ counts the number of subproblem pairs directly. We absorb $c_0$, $d$, $\lambda_{\max}$, and logarithmic factors into the constant, and use $d \leq K \cdot Q$ (number of overlapping positions). $\square$

### E.6.4. EXP3 REGRET

**Theorem E.14** (**Standard Exp3 Regret with Importance Sampling**). *Consider a combinatorial optimization problem over domain of size $n$, decomposed into $K$ subproblems, each of size $d$ with overlap $|\mathcal{O}|$. Let $w_{i,j}(t) \in \mathbb{R}_+$ denote the Exp3 weight for assigning value $j \in [d]$ to position $i \in [n]$ at iteration $t$. The weights for each position $i$ are $\boldsymbol{w}_i(t) = [w_{i,1}(t), w_{i,2}(t), \ldots, w_{i,d}(t)] \in \mathbb{R}_+^d$. At each round $t$, the algorithm samples a solution $\boldsymbol{x}_t$ coordinate-wise, where each position $i$ selects value $j$ with probability $p_{i,j}(t) = \frac{w_{i,j}(t)}{\sum_{j'} w_{i,j'}(t)}$. The algorithm observes only the total reward $\rho_t = f(\boldsymbol{x}_t)$ for the selected solution. Weights are updated using importance-weighted rewards:*

$$w_{i,j}(t+1) = w_{i,j}(t) \cdot \exp\left( \eta \cdot \frac{\rho_t \cdot \mathbb{I}[x_t(i) = j]}{p_{i,j}(t)} \right)$$

*with learning rate $\eta$. If the objective function is $L$-Lipschitz continuous (Assumption 2), then the total Exp3 regret over $T$ iterations is bounded by:*

$$R_{Exp3}(T) \leq \frac{n \log d}{\eta} + \frac{\eta T n B^2}{2} + L|\mathcal{O}|\sqrt{T} \tag{54}$$

*where $B$ is the maximum reward magnitude. Setting $\eta = \sqrt{\frac{2 \log d}{T B^2}}$ yields:*

$$R_{Exp3}(T) = O\left( \sqrt{ndT \log d} + L|\mathcal{O}|\sqrt{T} \right) = O\left( \sqrt{KdT \log d} + L|\mathcal{O}|\sqrt{T} \right) \tag{55}$$

*where the last equality uses $n = Kd$ for the decomposition structure.*

*Proof.* Following similar steps as in UCB Regret bound, using instantaneous regret as in Definition E.6, we decompose:

$$r_t = \underbrace{f(\boldsymbol{x}^*) - \sum_{k=1}^{K} g_k(\boldsymbol{x}_{I_k}^*)}_{\text{optimal gap} \leq 0} + \underbrace{\sum_{k=1}^{K} [g_k(\boldsymbol{x}_{I_k}^*) - g_k(\boldsymbol{x}_{I_k,t})]}_{\text{Exp3 subproblem regret}} + \underbrace{C_t}_{\text{coupling error}} \tag{56}$$

For the Exp3 subproblem regret, we apply standard Exp3 analysis to each position $i$. The importance-weighted estimator $\hat{\rho}_{i,j}(t) = \frac{\rho_t \cdot \mathbb{I}[x_t(i)=j]}{p_{i,j}(t)}$ is unbiased: $\mathbb{E}[\hat{\rho}_{i,j}(t) \mid \mathcal{F}_{t-1}] = \rho_t$. By the standard Exp3 regret bound ((Auer et al., 2002b), also see (Cesa-Bianchi & Lugosi, 2006)), for each position $i$:

$$\sum_{t=1}^{T} \left[ \max_j \mathbb{E}[\rho_t(j)] - \mathbb{E}[\rho_t(x_t(i))] \right] \leq \frac{\log d}{\eta} + \frac{\eta T B^2}{2}$$

Summing over all $n$ positions gives the subproblem regret bound:

$$\sum_{k=1}^{K} \sum_{t=1}^{T} [g_k(\boldsymbol{x}_{I_k}^*) - g_k(\boldsymbol{x}_{I_k,t})] \leq \frac{n \log d}{\eta} + \frac{\eta T n B^2}{2}$$

For the coupling error, by Assumption 3 (Bounded Coupling), the coordination violations satisfy $|C_t| \leq L|\mathcal{O}|$ per iteration. By concentration inequalities, the cumulative coupling error satisfies:

$$\sum_{t=1}^{T} C_t = O(L|\mathcal{O}|\sqrt{T})$$

Combining these bounds yields:

$$R_{\text{Exp3}}(T) \leq \frac{n \log d}{\eta} + \frac{\eta T n B^2}{2} + L|\mathcal{O}|\sqrt{T}$$

Setting $\eta = \sqrt{\frac{2 \log d}{T B^2}}$ and using $n = Kd$ completes the proof. $\qquad\square$

---

**Algorithm 5** UPDATEEXPERTS: Clipped Importance-Weighted Multi-Expert Parameter Update

---

**Require:** Solution $x^{(t)}$, reward $r^{(t)}$, parameters $(\hat{V}, N, W, \tilde{L})$, **Hyperparameters:** clipping threshold $p_{\min}$, learning rate $\eta$

**Output:** Updated $(\hat{V}, N, W, \tilde{L})$

**Initialize:** $\tilde{r}^{(t)} \leftarrow \text{NORMALIZE}(r^{(t)})$ $\qquad\qquad\qquad\qquad\qquad\qquad\qquad$ ▷ Normalization to $[-1, 1]$

**Initialize:** $\ell^{(t)} \leftarrow 1 - \tilde{r}^{(t)}$ $\qquad\qquad\qquad\qquad\qquad\qquad$ ▷ Instantaneous loss (regret surrogate)

1: **for** $i = 1, \ldots, n$ **do**
2: $\qquad a \leftarrow x_i^{(t)}$ $\qquad\qquad\qquad\qquad\qquad\qquad\qquad\qquad\qquad\qquad$ ▷ Action selected at position $i$
3: $\qquad N_{i,a} \leftarrow N_{i,a} + 1$ $\qquad\qquad\qquad\qquad\qquad\qquad\qquad\qquad\qquad$ ▷ Increment visit count
4: $\qquad \tilde{p}_a^{(t)} \leftarrow \max\big(W_{i,a}\|W_i\|_1, p_{\min}\big)$ $\qquad\qquad\qquad\qquad\qquad$ ▷ Clipped selection probability
5: $\qquad \hat{r}^{(t)} \leftarrow \tilde{r}^{(t)}/\tilde{p}_a^{(t)}$ $\qquad\qquad\qquad\qquad\qquad\qquad\qquad\qquad$ ▷ Importance-weighted reward
6: $\qquad \hat{\ell}^{(t)} \leftarrow \ell^{(t)}/\tilde{p}_a^{(t)}$ $\qquad\qquad\qquad\qquad\qquad\qquad\qquad\qquad\qquad$ ▷ Importance-weighted loss
7: $\qquad \hat{V}_{i,a} \leftarrow \hat{V}_{i,a} + \frac{1}{N_{i,a}}(\tilde{r}^{(t)} - \hat{V}_{i,a})$ $\qquad\qquad\qquad$ ▷ **Expert 1 (UCB)**: running average
8: $\qquad W_{i,a} \leftarrow W_{i,a} \cdot \exp\big(\eta \hat{r}^{(t)}/n\big)$ $\qquad\qquad\qquad$ ▷ **Expert 2 (EXP3)**: multiplicative weights
9: $\qquad \tilde{L}_{i,a} \leftarrow \tilde{L}_{i,a} + \hat{\ell}^{(t)}$ $\qquad\qquad\qquad\qquad\qquad$ ▷ **Expert 3 (FTRL)**: cumulative regret
10: **end for**
11: $W_i \leftarrow W_i/\|W_i\|_1$ for all $i$ $\qquad\qquad\qquad\qquad\qquad\qquad\qquad\qquad$ ▷ Normalize weights
12: **return** $(\hat{V}, N, W, \tilde{L})$

---

**Lemma E.15** (**Bias from Clipped Importance Sampling**). *Consider the EXP3 algorithm for a combinatorial bandit problem with n positions and d values per position where at each round t the learner samples an action $x_t = (x_t(1), \ldots, x_t(n))$ from a factorized distribution $\Pr(x_t \mid \mathcal{F}_{t-1}) = \prod_{i=1}^n p_{i,x_t(i)}(t)$. Let $p_{i,j}(t)$ denote the probability of selecting value j at position i at a time t, with weights updated via $w_{i,j}(t+1) = w_{i,j}(t) \cdot exp(\eta \cdot \hat{r}_{i,j}(t))$ as:*

$$p_{i,j}(t) = \frac{w_{i,j}(t)}{\sum_{j'=1}^d w_{i,j'}(t)}$$

*where $\eta > 0$ is the learning rate. Thus, each coordinate $x_t(i)$ conditioned on the past history $\mathcal{F}_{t-1}$ is drawn independently using its own probability vector $\mathbf{p}_i(t)$. The EXP3-style update for each position i uses the clipped importance-weighted estimator:*

$$\hat{r}_{i,j}(t) = \begin{cases} \frac{\rho_t}{max(p_{i,j}(t), C)} & \text{if } x_t(i) = j \\ 0 & \text{otherwise} \end{cases}$$

*where clipping threshold $C > 0$ is a fixed constant (e.g, C=0.01), $\rho_t \in [-B, B]$ is the observed scalar reward, and $x_t(i)$ is the value selected at position i in round t. Let $\tilde{r}_{i,j}(t)$ denote the corresponding un-clipped estimator $\tilde{r}_{i,j}(t) = \begin{cases} \frac{\rho_t}{p_{i,j}(t)} & \text{if } x_t(i) = j \\ 0 & \text{otherwise} \end{cases}$.*

*Then the **total bias introduced by clipping** over T rounds satisfies:*

$$\sum_{t=1}^T \sum_{i=1}^n |\mathbb{E}[\hat{r}_{i,x_t(i)}(t)] - \rho_t| \leq \frac{2nB\log(d)}{\eta} \qquad\qquad (57)$$

*Proof.* Since the algorithm factorizes over positions, it suffices to prove the bound for a fixed position $i$, and then multiply by $n$. So let's fix $i$ and omit the index $i$ for brevity in $w_j(t), p_j(t), \hat{r}_j(t)$. The un-clipped estimator $\tilde{r}_j(t)$ as defined above is an unbiased estimator of $\rho_t$ as,

$$\mathbb{E}[\tilde{r}_{x_t}(t) \mid \mathcal{F}_{t-1}] = \mathbb{E}[\rho_t \mid \mathcal{F}_{t-1}],$$

If one uses a clipped estimator $\hat{r}_j(t)$, we want to bound the bias over $T$ rounds for this position $i$:

$$\text{Bias}_{i,j} = \sum_{t=1}^{T} |\mathbb{E}[\hat{r}_{x_t}(t)] - \mathbb{E}[\rho_t]| \tag{58}$$

i.e the bias that emerges due to the approximate importance sampling. The per-round bias for this position:

$$\text{Bias}_{i,j}(t) := E[\hat{r}_{x_t}(t)] - \mathbb{E}[\rho_t] = \mathbb{E}[\hat{r}_{x_t}(t) - \tilde{r}_{x_t}(t)]$$

Now, if $p_j(t) \geq C$, then $\hat{r}_j(t) = \tilde{r}_j(t)$. If $p_j(t) < C$, then

$$\hat{r}_j(t) = \frac{\rho_t}{C}, \qquad \tilde{r}_j(t) = \frac{\rho_t}{p_j(t)},$$

and since $p_j(t) < C$, we have $|\hat{r}_j(t)| \leq |\tilde{r}_j(t)|$ with the same sign, so $\tilde{r}_j(t) - \hat{r}_j(t) \geq 0$ for all $j, t$. This introduces a negative bias or underestimation of the reward. As $|Bias_{i,j}(t)| = -Bias_{i,j}(t)$,

$$\sum_{t=1}^{T} |\mathbb{E}[\hat{r}_{x_t}(t)] - \mathbb{E}[\rho_t]| = \sum_{t=1}^{T} \mathbb{E}[\tilde{r}_{x_t}(t) - \hat{r}_{x_t}(t)] = \sum_{j=1}^{d} p_j(t)(\tilde{r}_j(t) - \hat{r}_j(t)). \tag{59}$$

Therefore, for this position,

$$\sum_{t=1}^{T} \left| \mathbb{E}[\hat{r}_{x_t}(t)] - \mathbb{E}[\rho_t] \right| = \sum_{t=1}^{T} \sum_{j=1}^{d} p_j(t)(\tilde{r}_j(t) - \hat{r}_j(t)). \tag{60}$$

Now, define potential function $W(t) = \sum_{j=1}^{d} w_j(t)$, $\qquad p_j(t) = \frac{w_j(t)}{W(t)}$

Update iterates for un-clipped estimators $\tilde{r}_j(t)$ and clipped estimators $\hat{r}_j(t)$,

$$w_j^{un}(t+1) = w_j^{un}(t) \exp\left(\eta \tilde{r}_j(t)\right)$$

$$w_j^{cl}(t+1) = w_j^{cl}(t) \exp\left(\eta \hat{r}_j(t)\right)$$

Note, wrt **coupling argument** – we compare both weight sequences under the *same* realization of sampled actions $\{x_t\}_{t=1}^{T}$. That is, we run the actual algorithm (which uses clipped estimators and clipped probabilities $p_j^{cl}(t)$) and track two weight sequences, $w_j^{cl}(t)$: the actual weights, updated with clipped estimators, $w_j^{un}(t)$: hypothetical weights, updated as if we used unclipped estimators for the same action sequence. Since the action $x_t$ is the same for both, we have:

- If $x_t = j$: both $\tilde{r}_j(t)$ and $\hat{r}_j(t)$ are nonzero, with $\tilde{r}_j(t) \geq \hat{r}_j(t)$

- If $x_t \neq j$: both $\tilde{r}_j(t) = \hat{r}_j(t) = 0$

Therefore, $w_j^{un}(t) \geq w_j^{cl}(t)$ for all $j, t$, which implies $\Phi^{un}(t) \geq \Phi^{cl}(t)$.
Now, consider log potentials $\Phi$ as,

$$\Phi^{un}(t) = \log \sum_{j=1}^{d} w_j^{un}(t), \qquad \Phi^{cl}(t) = \log \sum_{j=1}^{d} w_j^{cl}(t)$$

Then,

$$W(t+1) = \sum_{j=1}^{d} w_j(t+1) = \sum_{j=1}^{d} w_j^{un}(t) \exp(\eta \tilde{r}_j(t)) = W(t) \sum_{j=1}^{d} p_j(t) \exp(\eta \tilde{r}_j(t))$$

$$\Phi(t+1) - \Phi(t) = \log\left(\frac{W(t) \sum_{j=1}^{d} p_j(t) \exp(\eta \tilde{r}_j(t))}{W(t)}\right) = \log\left(\sum_{j=1}^{d} p_j(t) \exp(\eta \tilde{r}_j(t))\right)$$

For now let's assume that rewards are scaled to $[-1, 1]$ (will multiply with $B$ finally). Then $|\tilde{r}_j(t)|, |\hat{r}_j(t)| \leq 1/C$. With this we can use a standard Hoeffding-type inequality for exponential weights (e.g., Theorem 3.1 (Bubeck & Cesa-Bianchi, 2012) & Lemma 2.2 in (Cesa-Bianchi & Lugosi, 2006)) that gives, for any bounded $z_j(t)$,

$$\Phi(t+1) - \Phi(t) \leq \eta \sum_{j=1}^{d} p_j(t) z_j(t) + \frac{\eta^2}{2}$$

whenever $|z_j(t)| \leq 1$; After scaling, this becomes the same inequality up to a constant absorbed into $\eta^2$. Applying this once with $z_j(t) = \tilde{r}_j(t)$ and once with $z_j(t) = \hat{r}_j(t)$ and subtracting, we obtain

$$\Phi^{\mathrm{un}}(T+1) - \Phi^{\mathrm{cl}}(T+1) \leq \eta \sum_{t=1}^{T} \sum_{j=1}^{d} p_j(t)\big(\tilde{r}_j(t) - \hat{r}_j(t)\big) + O(\eta^2 T).$$

As both schemes start from identical initial weights $w_j^{\mathrm{un}}(1) = w_j^{\mathrm{cl}}(1) = 1$, so $\Phi^{\mathrm{un}}(1) = \Phi^{\mathrm{cl}}(1) = \log d$. Recall that $\tilde{r}_j(t) \geq \hat{r}_j(t) \quad \forall j, t$. Thus, the un-clipped weights grow at least as fast as the clipped ones $w_j^{un}(t+1) \geq w_j^{cl}(t+1)$ and if both schemes start from equal initial weights then for all $t$, we have $\max_j w_j^{cl}(t) \leq \max_j w_j^{un}(t)$ and because $\sum_{j=1}^{d} w_j(t) \leq d \cdot \max_j w_j(t)$ we get,

$$\max_j w_j^{cl}(t) \leq W^{cl}(t) \quad \text{and} \quad W^{un}(t) \leq d \max_j w_j^{un}(t),$$

taking log both sides and as this applies for every time step including T+1 we get,

$$\Phi^{\mathrm{un}}(T+1) - \Phi^{\mathrm{cl}}(T+1) \leq \log d$$

Choosing $\eta = O(1/\sqrt{T})$ makes the $O(\eta^2 T)$ term constant-sized so we get,

$$\eta \sum_{t=1}^{T} \sum_{j=1}^{d} p_j(t)\big(\tilde{r}_j(t) - \hat{r}_j(t)\big) \leq \log d$$

In the normalized $[-1, 1]$ case,

$$\sum_{t=1}^{T} \sum_{j=1}^{d} p_j(t)\big(\tilde{r}_j(t) - \hat{r}_j(t)\big) \leq \frac{\log d}{\eta}$$

Note, if the actual rewards satisfy $\rho_t \in [-B, B]$, then both $\tilde{r}_j(t)$ and $\hat{r}_j(t)$ scale linearly in $\rho_t$, so the difference $\tilde{r}_j(t) - \hat{r}_j(t)$ scales by at most a factor $B$. Taking a slightly looser constant to account for both tails, we obtain

$$\sum_{t=1}^{T} \sum_{j=1}^{d} p_j(t)\big(\tilde{r}_j(t) - \hat{r}_j(t)\big) \leq \frac{2B \log d}{\eta}$$

Combining this with (60) yields, for this position $i$,

$$\sum_{t=1}^{T} \left| \mathbb{E}[\hat{r}_{x_t}(t)] - \mathbb{E}[\rho_t] \right| \leq \frac{2B \log d}{\eta}$$

Finally, summing over $i = 1, \ldots, n$ positions gives

$$\sum_{t=1}^{T} \sum_{i=1}^{n} \left| \mathbb{E}[\hat{r}_{i,x_t(i)}(t)] - \mathbb{E}[\rho_t] \right| \leq \frac{2nB \log d}{\eta}$$

$\square$

**Theorem E.16** (**Exp3 Subproblem Regret with Clipped Importance Sampling**). *Consider a combinatorial optimization problem over domain of size $n$, decomposed into $K$ subproblems, each of size $d$ with overlap $|\mathcal{O}|$. Let $w_{i,j}(t) \in \mathbb{R}_+$ denote the Exp3 weight for assigning value $j \in [d]$ to position $i \in [n]$ at iteration $t$. At each round $t$, the* ALGORITHM 5 *samples a solution $\boldsymbol{x}_t$ coordinate-wise, where each position $i$ selects value $j$ with probability $p_{i,j}(t) = \frac{w_{i,j}(t)}{\sum_{j'} w_{i,j'}(t)}$.*

*The* ALGORITHM 5 *observes only the total reward $\rho_t = f(\boldsymbol{x}_t)$ for the selected solution. Weights are updated using **clipped** importance-weighted rewards:*

$$w_{i,j}(t+1) = w_{i,j}(t) \cdot \exp\left(\eta \cdot \hat{r}_{i,j}(t)\right)$$

*where*

$$\hat{r}_{i,j}(t) = \begin{cases} \frac{\rho_t}{\max(p_{i,j}(t), C)} & \text{if } x_t(i) = j \\ 0 & \text{otherwise} \end{cases}$$

*with clipping threshold $C > 0$ (e.g., $C = 0.01$), learning rate $\eta$, and $\rho_t \in [-B, B]$.*

*If the objective function is $L$-Lipschitz continuous (Assumption 2), then:*

*(a) **Per-subproblem bound:** For each subproblem $k \in [K]$ of size $d$, the Exp3 regret over $T$ iterations satisfies:*

$$\sum_{t=1}^{T} [g_k(\boldsymbol{x}_{I_k}^*) - g_k(\boldsymbol{x}_{I_{k,t}})] \le \frac{d \log d}{\eta} + \frac{\eta T d B^2}{2C^2} + \frac{2dB \log(d)}{\eta} \tag{61}$$

*With optimal learning rate $\eta = \sqrt{\frac{2C^2 \log d}{TB^2}}$, this simplifies to:*

$$\sum_{t=1}^{T} [g_k(\boldsymbol{x}_{I_k}^*) - g_k(\boldsymbol{x}_{I_{k,t}})] = O\left(\frac{dB\sqrt{T \log d}}{C}\right) \tag{62}$$

*(b) **Total bound across subproblems:** Summing over all $K$ subproblems and including coupling errors, the total Exp3 regret over $T$ iterations is:*

$$R_{Exp3}(T) \le \frac{n \log d}{\eta} + \frac{\eta T n B^2}{2C^2} + \frac{2nB \log(d)}{\eta} + L|\mathcal{O}|\sqrt{T} \tag{63}$$

*Setting $\eta = \sqrt{\frac{2C^2 \log d}{TB^2}}$ yields:*

$$R_{Exp3}(T) = O\left(\frac{nB\sqrt{T \log d}}{C} + L|\mathcal{O}|\sqrt{T}\right) = O\left(\frac{KdB\sqrt{T \log d}}{C} + L|\mathcal{O}|\sqrt{T}\right) \tag{64}$$

*where the last equality uses $n = Kd$ for the decomposition structure.*

*(c) **Total Average regret across subproblems:** Equivalently, in terms of the average subproblem regret $R_{\text{subproblem}}^{\text{avg}}(T) := \frac{1}{K}\sum_{k=1}^{K} R_k(T)$, we obtain,*

$$R_{\text{subproblem}}^{\text{avg}}(T) = \frac{1}{K} R_{\text{Exp3}}(T) = O\left(\frac{dB\sqrt{T \log d}}{C} + \frac{L|\mathcal{O}|}{K}\sqrt{T}\right)$$

*The leading EXP3 term scales as $O\left(d\sqrt{T \log d}\right)$ & is independent of number of subproblems $K$.*

*Proof.* Following the same instantaneous regret decomposition as in Theorem E.14:

$$r_t = \underbrace{f(\boldsymbol{x}^*) - \sum_{k=1}^{K} g_k(\boldsymbol{x}_{I_k}^*)}_{\text{optimal gap} \le 0} + \underbrace{\sum_{k=1}^{K} [g_k(\boldsymbol{x}_{I_k}^*) - g_k(\boldsymbol{x}_{I_{k,t}})]}_{\text{Exp3 subproblem regret}} + \underbrace{C_t}_{\text{coupling error}} \tag{65}$$

We prove part (a) first, then aggregate to obtain part (b).

Let's consider a single subproblem $k$ of size $d$. Note, the key difference from standard Exp3 is that the importance-weighted estimator is now **biased** due to clipping. Therefore, let's decompose the subproblem regret into two components - 1) Standard Exp3 exploration-exploitation tradeoff and 2) Bias from clipping. If the unbiased estimator is defined as:

$$\tilde{r}_{i,j}(t) = \begin{cases} \frac{\rho_t}{p_{i,j}(t)} & \text{if } x_t(i) = j \\ 0 & \text{otherwise} \end{cases}$$

By standard Exp3 analysis (Auer et al., 2002), if we were using $\tilde{r}_{i,j}(t)$, the regret for each position $i \in I_k$ would be:

$$\sum_{t=1}^{T} \left[ \max_j \mathbb{E}[\rho_t(j)] - \mathbb{E}[\rho_t(x_t(i))] \right] \leq \frac{\log d}{\eta} + \frac{\eta T B^2}{2C^2}$$

where the variance term increases by $1/C^2$ because:

$$\text{Var}[\hat{r}_{i,j}(t)] \leq \frac{B^2}{C^2}$$

Summing over all $d$ positions in subproblem $k$:

$$\sum_{t=1}^{T} [g_k(\boldsymbol{x}_{I_k}^*) - g_k(\boldsymbol{x}_{I_{k,t}})] \leq \frac{d \log d}{\eta} + \frac{\eta T d B^2}{2C^2}$$

However, we are actually using $\hat{r}_{i,j}(t)$ instead of $\tilde{r}_{i,j}(t)$, which introduces bias. By Lemma E.15, for the $d$ positions in subproblem $k$, the total bias over all rounds is:

$$\sum_{t=1}^{T} \sum_{i \in I_k} \left| \mathbb{E}[\hat{r}_{i,x_t(i)}(t)] - \rho_t \right| \leq \frac{2dB \log(d)}{\eta}$$

*Combining both components:*

The total regret for subproblem $k$ is:

$$\sum_{t=1}^{T} [g_k(\boldsymbol{x}_{I_k}^*) - g_k(\boldsymbol{x}_{I_{k,t}})] \leq \frac{d \log d}{\eta} + \frac{\eta T d B^2}{2C^2} + \frac{2dB \log(d)}{\eta} = \frac{(1+2B)d \log d}{\eta} + \frac{\eta T d B^2}{2C^2} \tag{66}$$

Setting $\eta = \sqrt{\frac{2C^2(1+2B)\log d}{TB^2}}$ gives:

$$\sum_{t=1}^{T} [g_k(\boldsymbol{x}_{I_k}^*) - g_k(\boldsymbol{x}_{I_{k,t}})] = \frac{(1+2B)d \log d}{\sqrt{\frac{2C^2(1+2B)\log d}{TB^2}}} + \frac{\sqrt{\frac{2C^2(1+2B)\log d}{TB^2}} \cdot T d B^2}{2C^2} \tag{67}$$

$$= d\sqrt{\frac{(1+2B)TB^2 \log d}{2C^2}} + d\sqrt{\frac{(1+2B)TB^2 \log d}{2C^2}} \tag{68}$$

$$= 2d\sqrt{\frac{(1+2B)TB^2 \log d}{2C^2}} = O\left(\frac{dB\sqrt{T \log d}}{C}\right) \tag{69}$$

This proves part (a).

**Note.** *The constant $B^2/C^2$ is loose: a direct calculation gives $\sum_j p_{i,j}(\rho_t/\max(p_{i,j},C))^2 \leq B^2/C$, so the variance is $O(B^2/C)$ and re-optimizing $\eta$ tightens the leading term to $O(dB\sqrt{T \log d}/\sqrt{C})$. We keep the $1/C$ form for consistency with the shared bound $|\hat{r}| \leq B/C$ used by FTRL.*

**Part (b): Total bound across subproblems.**

Summing the per-subproblem bounds over all $K$ subproblems:

$$\sum_{k=1}^{K}\sum_{t=1}^{T}[g_k(\boldsymbol{x}_{I_k}^*) - g_k(\boldsymbol{x}_{I_{k,t}})] \leq K \cdot \left(\frac{(1+2B)d\log d}{\eta} + \frac{\eta T d B^2}{2C^2}\right) = \frac{(1+2B)n\log d}{\eta} + \frac{\eta T n B^2}{2C^2}$$

where we used $n = Kd$ (total number of positions equals $K$ subproblems times $d$ positions per subproblem).

Adding the coupling error term: By Assumption 3 (Bounded Coupling), the coordination violations satisfy $|C_t| \leq L|\mathcal{O}|$ per iteration. By concentration inequalities:

$$\sum_{t=1}^{T} C_t = O(L|\mathcal{O}|\sqrt{T}) \qquad\qquad \text{(where } |\mathcal{O}| \leq KQ)$$

Combining all terms:

$$R_{\text{Exp3}}(T) \leq \underbrace{\frac{n\log d}{\eta} + \frac{\eta T n B^2}{2C^2} + \frac{2nB\log(d)}{\eta}}_{\text{subproblem regret}} + \underbrace{L|\mathcal{O}|\sqrt{T}}_{\text{coupling error}} \tag{70}$$

Setting $\eta = \sqrt{\frac{2C^2\log d}{TB^2}}$ to balance the first two terms:

$$R_{\text{Exp3}}(T) = \frac{(1+2B)n\log d}{\sqrt{\frac{2C^2\log d}{TB^2}}} + \frac{\sqrt{\frac{2C^2\log d}{TB^2}} \cdot T n B^2}{2C^2} + L|\mathcal{O}|\sqrt{T} \tag{71}$$

$$= n\sqrt{\frac{TB^2(1+2B)\log d}{2C^2}} + n\sqrt{\frac{TB^2\log d}{2C^2}} + L|\mathcal{O}|\sqrt{T} \tag{72}$$

$$= (\sqrt{(1+2B)}+1) \cdot n\sqrt{\frac{TB^2\log d}{2C^2}} + L|\mathcal{O}|\sqrt{T} \tag{73}$$

$$= O\left(\frac{nB\sqrt{T\log d}}{C} + L|\mathcal{O}|\sqrt{T}\right) \tag{74}$$

Using $n = Kd$:

$$R_{\text{Exp3}}(T) = O\left(\frac{KdB\sqrt{T\log d}}{C} + L|\mathcal{O}|\sqrt{T}\right)$$

This completes the proof of part (b). $\qquad\qquad\qquad\qquad\qquad\qquad\qquad\qquad\qquad\qquad\qquad\qquad$ $\square$

**Remarks**

1. **Impact of clipping constant $C$:** The regret scales as $1/C$, so smaller clipping thresholds lead to higher regret. However, in practice, $C = 0.01$ provides a good balance between variance reduction and bias.

2. **Comparison to unclipped Exp3:** If $C = O(1/\sqrt{T})$ (diminishing clipping), the bias term becomes $O(n\sqrt{T\log d})$, matching the standard Exp3 bound. With fixed $C$, we pay an extra factor of $1/C$ but gain significant practical stability.

3. **Bias vs. variance tradeoff:** The clipping introduces bias of order $O(nB\log d/\eta)$ but reduces variance from $O(B^2/p_{\min}^2)$ to $O(B^2/C^2)$ where $C$ is constant. The bias is absorbed into the leading $O(\sqrt{T})$ term.

4. **Key insight:** The bias diminishes **not because the threshold decreases**, but because Exp3 concentrates probability mass on good actions over time, making clipping events increasingly rare.

**Choice of Importance-Sampling Denominator**  Both EXP3 and FTRL share the clipped importance-sampling (IS) denominator $p_{i,a}(t) = W_{i,a}/\|W_i\|_1$ (Algorithm 5), rather than the unbiased mixture-marginal $\pi_{t,i}(a) = \sum_e p_e^{(t,i)} \pi_{t,i}^{(e)}(a)$. UCB and FTRL are deterministic argmax rules with no native selection probability to use, and sharing $p_{i,a}(t)$ allows a single bias-and-variance analysis (Lemma E.15) to cover both. The resulting estimator is biased at the mixture level. Biased IS estimators have a precedent in the EXP3 family (e.g., the EXP3-IX (Neu, 2015), which adds $\gamma$ to the denominator rather than clipping. Here, Theorem E.2 bounds the multi-expert regret via per-expert composition $\sum_e p_e \phi_e(T)$, which does not require mixture-level unbiasedness. UCB updates directly from observed rewards and needs no IS correction.

Switching to marginal IS preserves all rates: replace $p_{i,a}(t)$ with $\pi_{t,i}(a)$ in the clipping step at cost $O(E \cdot |\mathcal{A}_i|)$ per position. The bounds of Theorems E.16, E.17, and Corollary 5.3 carry over since both quantities are clipped at the same floor and $\pi_{t,i}(a) \geq p_e^{(t,i)} p_{i,a}^{(e)}(a)$ for each expert $e$.

### E.6.5. FTRL REGRET

Follow-the-Regularized-Leader is a classic online learning algorithm using convex losses so naturally, this does not align with our setting. Combinatorial (ex: TSP) or categorical multiobjective problems especially in our case use non-convex losses. So we use a clever trick - linearization via Probability simplex! Since we have a bandit (multi-arm) setting we lift it to continuous space by we choose a position/action by sampling from a distribution of as $p_i \in \Delta_d = \{p \in \mathbb{R}^d : p_j \geq 0, \sum_j p_j = 1\}$ where $p_{i,j} = Pr(\text{position } i \text{ takes value } j)$. Mathematically,

$$\text{Discrete problem} : x_i \in \{0, 1, ...n - 1\} \text{ choose 1 city for position i}$$

$$\text{Lifted continuous problem} : p_i \in \Delta_d = \{p \in \mathbb{R}^d : p_j \geq 0, \sum_j p_j = 1\}$$

where $p_{i,j} = Pr(\text{assign city j to position i})$.

Therefore, based on which position/action is selected we assign it a reward and this results in expected loss to be convex! Mathematically, if we define any position-wise loss for any subproblem k based on $(i, j)$ as $\ell_t^{(k)}(i, j)$

$$\ell_t^{(k)}(i, j) = \text{marginal loss contribution when position i has city j}$$

Where marginal loss means the contribution of placing city $j$ at position $i$ is the change in total tour cost caused by that assignment. Then,

$$\ell_t^{(k)}(i, j) = 1 - r_t^{(k)}(i, j)$$

$$\text{where } r_t(i, j) = \frac{\text{Reward(solution with city } j \text{ at position } i) - R_{\min}}{R_{\max} - R_{\min}}$$

This captures the cost contribution of assignment $(i, j)$ to the full tour.

So now how do we connect any subproblem's objective to position-wise losses? It is quite straightforward, since we define $\ell_t^{(k)}(i, j)$ as the normalized marginal loss, we're measuring the incremental effect of each assignment, not assuming positions are independent. Therefore, by our definition,

$$g_k(x_{I_k}^*) - g_k(x_{I_{k,t}}) = \sum_{i \in I_k} [r_t^{(k)}(i, x_{I_k}^*) - r_t^{(k)}(i, x_{I_{k,t}})] = \sum_{i \in I_k} [\ell_t^{(k)}(i, x_{i,t}) - \ell_t^{(k)}(i, x_i^*)]$$

Additionally, note since we define $\ell_t^{(k)}(i, j) = 1$ - reward, then expected loss under distribution $p_i$ is linear in $p_i$ ,

$$\text{If any loss function } \ell_t^{(k)}(i, j) := (1 - r_t^{(k)}(i, j))$$

$$\mathbb{E}_{j \sim p_i}[\ell_t^{(k)}(i, j)] = \sum_{j=0}^{d-1} p_{i,j} \cdot \ell_t^{(k)}(i, j) = \langle p_i, \ell_t^{(k)}(i) \rangle$$

Now, at each round $t$, FTRL selects a distribution $p_i \in \Delta_d$ for position $i$, then the (FTRL) regret for position $i$ is defined against the best fixed distribution $p_i^* \in \Delta_d$:

$$R_i^{\text{FTRL}}(T) = \sum_{t=1}^{T} \mathbb{E}_{j \sim p_i}[\ell_t^{(k)}(i,j)] - \min_{p_i^* \in \Delta_d} \sum_{t=1}^{T} \langle p_i^*, \ell_t^{(k)}(i) \rangle \tag{75}$$

Crucially, since the loss is linear in $p_i$, the minimum over the simplex $\Delta_d$ is achieved at a vertex, i.e., a deterministic action:

$$\min_{p_i^* \in \Delta_d} \sum_{t=1}^{T} \langle p_i^*, \ell_t^{(k)}(i) \rangle = \min_{j \in [d]} \sum_{t=1}^{T} \ell_t^{(k)}(i,j) \tag{76}$$

This connects the continuous formulation back to competing against the best fixed discrete action in hindsight. Hence, we can **apply convex optimization results** to obtain regret bounds for the original discrete problem!

Note: The realized regret differs from the pseudo-regret by an $O(\sqrt{T})$ martingale-difference term via Azuma-Hoeffding, absorbed into the leading-order regret bound. Additionally, we can use any regularizer below and our specific choice is not mandatory.

**Theorem E.17** (**FTRL Regret under Decomposed Subproblems**). *Consider a combinatorial optimization problem over domain of size $n$, decomposed into $K$ subproblems, each of size $d$ with overlap $|\mathcal{O}|$. Let $L_{i,j}(t) \in \mathbb{R}_+$ denote the cumulative clipped-IS loss for assigning value $j \in [d]$ to position $i \in [n]$ at iteration $t$, and let $N_{i,j}(t)$ be the corresponding visit count. Using the probability simplex lifting, FTRL maintains cumulative IS losses and selects actions via:*

$$p_i^{(t)} = \arg \min_{p_i \in \Delta_d} \left\{ \sum_{s=1}^{t-1} \langle p_i, \hat{\ell}_s(i) \rangle + \text{Reg}_t(p_i) \right\}$$

*where $\text{Reg}_t(p_i) = -\gamma \sum_{j=1}^{d} p_{i,j} \sqrt{N_t(i,j) + 1}$ is the confidence-bonus regularizer (McMahan, 2017) and $\gamma = (B/C)\sqrt{2 \log T}$, with $C$ the IS clipping floor (shared with Expert 2, Theorem E.16) and $B$ the per-round loss bound.*

*If the objective function is $L$-Lipschitz continuous (Assumption 2), then:*

*(a) Per-subproblem bound: For each subproblem $k \in [K]$ of size $d$, with $T_k$ active rounds:*

$$\sum_{t \in \mathcal{T}_k} [g_k(\boldsymbol{x}_{I_k}^*) - g_k(\boldsymbol{x}_{I_{k,t}})] \leq C' \frac{Bd}{C} \sqrt{T_k \log T} \tag{77}$$

*(b) Total bound across subproblems: The total FTRL regret satisfies:*

$$R_{\text{FTRL}}(T) \leq C' \frac{Bd\sqrt{KT \log T}}{C} + L|\mathcal{O}|\sqrt{T} \tag{78}$$

*where $|\mathcal{O}| = O(KQ)$.*

*(c) Average subproblem regret:*

$$R_{\text{subproblem}}^{\text{avg,FTRL}}(T) = \frac{1}{K} R_{\text{FTRL}}(T) = O\left( \frac{Bd}{C} \sqrt{\frac{T \log T}{K}} + \frac{L|\mathcal{O}|}{K} \sqrt{T} \right) \tag{79}$$

*The leading FTRL term is $O(d\sqrt{T \log T}/C)$ and is independent of the number of subproblems $K$ (up to the $\sqrt{1/K}$ improvement factor). $C' = O(\sqrt{d})$ and is an architecture-dependent constant as $d, B, C$ are treated as fixed design constants. Hence their dependence can be absorbed.*

*Proof.* FTRL maintains cumulative losses and selects:

$$p_i^{(t)} = \arg \min_{p_i \in \Delta_d} \left\{ \sum_{s=1}^{t-1} \langle p_i, \hat{\ell}_s(i) \rangle + \text{Reg}_t(p_i) \right\}$$

where the clipped importance-weighted loss estimator (shared with Expert 2, Theorem E.16), $\hat{\ell}_t(i,j) =$
$$\begin{cases} \dfrac{\ell_t}{\max(p_{i,j}(t), C)} & \text{if } x_t(i) = j \\ 0 & \text{otherwise} \end{cases}$$
is defined for the loss $\ell_t = 1 - r_t$ derived from normalized reward $r_t \in [0,1]$, with clipping threshold $C$ matching Theorem E.16. The same clipping gives the bound $|\hat{\ell}_t(i,j)| \leq B/C$, which controls the variance term in our analysis. The confidence-bonus regularizer is $\text{Reg}_t(p_i) = -\gamma \sum_j p_{i,j}\sqrt{N_t(i,j)+1}$ with $\gamma = (B/C)\sqrt{2\log T}$. Since $\text{Reg}_t$ is linear in $p_i$, the update reduces to the LCB index $x_{i,t} = \arg\min_j[L_t(i,j) - \gamma\sqrt{N_t(i,j)+1}]$.

Following similar analysis as in previous sections, using instantaneous regret definition:

$$r_t = \underbrace{f(x^*) - \sum_{k=1}^{K} g_k(x^*_{I_k})}_{\text{optimal regret} \leq 0} + \underbrace{\sum_{k=1}^{K}[g_k(x^*_{I_k}) - g_k(x_{I_k,t})]}_{\text{FTRL subproblem regret}} + \underbrace{C_t}_{\text{coupling error}} \tag{80}$$

We bound FTRL subproblem regret as in Lemma E.35. Summing over all $K$ subproblems:

$$\sum_{k=1}^{K}\sum_{t=1}^{T_k}[g_k(x^*_{I_k}) - g_k(x_{I_k,t})] \leq \sum_{k=1}^{K} C' \frac{Bd}{C}\sqrt{T_k \log T}$$

Using Cauchy-Schwarz inequality:

$$\sum_{k=1}^{K}\sqrt{T_k} \leq \sqrt{K\sum_{k=1}^{K} T_k} = \sqrt{KT}$$

Therefore:

$$\sum_{k=1}^{K} \frac{Bd}{C}\sqrt{T_k \log T} \leq \frac{Bd\sqrt{\log T}}{C} \cdot \sqrt{KT} = \frac{Bd\sqrt{KT\log T}}{C}$$

Adding the coupling error term $L|\mathcal{O}|\sqrt{T}$ (where $|\mathcal{O}| = O(KQ)$) gives part (b). Part (c) follows by dividing by $K$. $\qquad\square$

### E.7. Decomposition Validity

In this section, we prove that our metric-based decomposition satisfies the assumptions required for bounded regret. We establish that the decomposition is well-defined, the induced local modifications have bounded effect on the objective, and the overlap structure admits controlled coupling error.

E.7.1. DECOMPOSITION CONSTRUCTION

**Definition E.18** (**Metric-Based Decomposition**). Let $(\mathcal{X}, \delta)$ be the solution space equipped with a problem-specific metric $\delta$, and let $D : [n] \times [n] \to \mathbb{R}_+$ be a problem-specific distance on indices (e.g., geographic distance for TSP, similarity for clustering). We construct subproblems via two complementary strategies:

1. (**Sliding Window**) For initialization and maintaining coverage, define subproblems $k \in \{1, \dots, K\}$ as:
$$S_k = \{(k-1)(s-o_t)+1, \dots, \min(n, (k-1)(s-o_t)+s)\}$$
where $s$ is the subproblem size and $o_t$ is the overlap at iteration $t$. Under diminishing overlap:
$$o_t = \lfloor o_0 \cdot t^{-\alpha} \rfloor$$
for initial overlap $o_0$ and decay rate $\alpha > 0$.

2. (**k-Nearest Neighbors**) For adaptive refinement, given a center $c \in [n]$, define:
$$S_c = \arg\min_{S \subseteq [n], |S|=s} \sum_{j \in S} D_{c,j}$$

That is, $S_c$ contains the $s$ indices closest to center $c$ under distance $D$.

The full decomposition $\{S_1, \ldots, S_K\}$ combines sliding windows (for coverage) with k-NN subproblems (for problem-structure exploitation).

When overlap is time-varying, we write $S_k^{(t)}$ to denote the subproblem defined at iteration $t$; when the time index is omitted, the statements apply uniformly over all iterations.

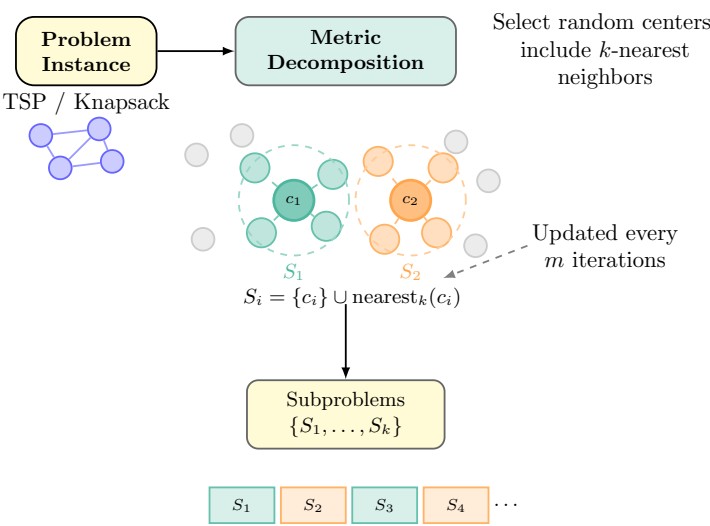

*Figure 26.* Diagram generated from TikZ code in a separate file.

*Remark* E.19 (**Domain Agnosticity**). The decomposition operates on *indices* $[n]$, not on solution values. This makes it domain-agnostic: the same construction applies to permutations (TSP), binary vectors (Knapsack), and other combinatorial domains. The metric $\delta$ on $\mathcal{X}$ and the distance $D$ on $[n]$ are problem-specific parameters.

### E.7.2. LOCAL STABILITY

We first establish that modifications restricted to a subproblem induce bounded changes in the objective.

**Lemma E.20** (**Subproblem Diameter Bound**). *Let $S_k \subseteq [n]$ be a subproblem with $|S_k| = s$. For any $x, x' \in \mathcal{X}$ that differ only on coordinates indexed by $S_k$ (i.e., $x_i = x_i'$ for all $i \notin S_k$):*

$$\delta(x, x') \leq D(S_k)$$

*where $D(S_k)$ is the diameter of $S_k$ under metric $d$, defined as:*

$$D(S_k) := \max_{x, x' \in \mathcal{X}: x_i = x_i' \, \forall i \notin S_k} \delta(x, x')$$

*The diameter depends on the metric structure:*

- *For Hamming distance: $D(S_k) \leq |S_k| = s$*
- *For Kendall tau distance: $D(S_k) \leq \binom{s}{2} = O(s^2)$*
- *For weighted metrics: $D(S_k) \leq \sum_{i \in S_k} w_i$*

*Proof.* Since $x$ and $x'$ agree on all coordinates outside $S_k$, any contribution to $\delta(x, x')$ must arise from differences within $S_k$. For Hamming distance, $d_H(x, x') = |\{i : x_i \neq x_i'\}| \leq |S_k|$ since disagreements can only occur on $S_k$. For Kendall tau distance on permutations, $d_\tau(x, x')$ counts inversions. If $x'$ differs from $x$ only on positions in $S_k$, then all inversions that differ between $x$ and $x'$ must involve at least one index in $S_k$. The total number of affected inversions is therefore at most $\binom{|S_k|}{2}$. For weighted metrics of the form $d_w(x, x') = \sum_i w_i \cdot \mathbf{1}[x_i \neq x_i']$, only coordinates in $S_k$ contribute, giving $d_w(x, x') \leq \sum_{i \in S_k} w_i$. $\square$

**Lemma E.21** (**Local Stability under Metric Decomposition**). *Let $S_k$ be a subproblem constructed via Definition E.18. For any $x, x' \in \mathcal{X}$ differing only on coordinates in $S_k$, under Assumption E.2.2 (Lipschitz continuity):*

$$|f_j(x') - f_j(x)| \leq L \cdot D(S_k)$$

*for all objectives $j \in \{1, \ldots, m\}$.*

*Proof.* By Lemma E.20, $\delta(x, x') \leq D(S_k)$. Applying Assumption E.2.2:

$$|f_j(x') - f_j(x)| \leq L \cdot \delta(x, x') \leq L \cdot D(S_k)$$

$\square$

### E.7.3. OVERLAP STRUCTURE AND COUPLING

We now characterize the overlap induced by the decomposition and bound its effect on coupling error.

**Definition E.22** (**Overlap Set and Multiplicity**). Given a decomposition $\{S_1, \ldots, S_K\}$:

- The *multiplicity* of index $i$ is $\rho_i := |\{k : i \in S_k\}|$

- The *maximum multiplicity* is $\rho := \max_i \rho_i$

- The *overlap set* is $\mathcal{O} := \{i \in [n] : \rho_i \geq 2\}$

- The *pairwise overlap* between $S_m$ and $S_l$ is $O_{ml} := S_m \cap S_l$

**Lemma E.23** (**Overlap Bounds**). *Let $\{S_1, \ldots, S_K\}$ be a decomposition where each subproblem has size at most $s$ and each index appears in at most $\rho$ subproblems. Then:*

*(a)* **Per-subproblem overlap:** $|S_k \cap \mathcal{O}| \leq s$

*(b)* **Pairwise overlap:** *For sliding window decomposition with overlap parameter o:*

$$|O_{k,k+1}| = o \quad \text{(consecutive subproblems)}$$

*(c)* **Total overlap counting:**

$$\sum_{k=1}^{K} |S_k \cap \mathcal{O}| \leq (\rho - 1) \cdot n$$

*(d)* **Overlap set size:**

$$|\mathcal{O}| \leq \frac{(\rho - 1) \cdot n}{\rho}$$

*Proof.* **(a)** Since $|S_k| \leq s$, trivially $|S_k \cap \mathcal{O}| \leq |S_k| \leq s$. **(b)** For sliding window decomposition with step size $(s - o)$, consecutive subproblems $S_k$ and $S_{k+1}$ share exactly $o$ indices by construction. **(c)** Each index $i$ with multiplicity $\rho_i$ is counted in $\rho_i$ subproblems and contributes to overlap in $(\rho_i - 1)$ of them. Summing over all indices:

$$\sum_{k=1}^{K} |S_k \cap \mathcal{O}| = \sum_{i \in \mathcal{O}} (\rho_i - 1) \leq \sum_{i=1}^{n} (\rho_i - 1) \leq (\rho - 1) \cdot n$$

**(d)** If $|\mathcal{O}| = m$, then the total multiplicity is at least $2m$ (each overlap index appears $\geq 2$ times). Since total multiplicity is $\sum_i \rho_i \leq \rho \cdot n$, we have $2m \leq \rho \cdot n$, giving $m \leq \rho \cdot n/2$. The tighter bound follows from the constraint that non-overlap indices have multiplicity exactly 1. The bound follows by counting total multiplicity: $\sum_i \rho_i \leq \rho n$, while each $i \in \mathcal{O}$ contributes at least $(\rho_i - 1) \geq 1$ excess appearances beyond the baseline.

$\square$

**Lemma E.24** (**Overlap-Induced Coupling Bound**). *Under Assumption E.2.3, for any modification on subproblem $S_k$:*

$$|f_j(x') - f_j(x)| \leq L \cdot D(S_k) + C \cdot |S_k \cap \mathcal{O}|$$

*Proof.* This follows directly from Assumption E.2.3 with $I_k = S_k$:

$$|f_j(x') - f_j(x)| \leq L \cdot \delta(x, x') + C \cdot |I_k \cap \mathcal{O}|$$

By Lemma E.20, $\delta(x, x') \leq D(S_k)$, yielding the result. $\square$

### E.7.4. VALIDITY OF DECOMPOSITION

We now prove the main result: our decomposition satisfies the bounded coupling assumption.

**Proposition E.25** (**Decomposition Satisfies Bounded Coupling**). *The metric-based decomposition (Definition E.18) with subproblem size $s$ and maximum overlap multiplicity $\rho$ satisfies Assumption E.2.3. Specifically, for any local modification on subproblem $S_k$:*

$$|f_j(x') - f_j(x)| \leq L \cdot D(s) + C \cdot (\rho - 1) \cdot s$$

*where $D(s)$ is the maximum diameter of a size-$s$ subproblem under metric $d$.*

*Proof.* Combine Lemma E.21 (local stability) and Lemma E.23(a) (per-subproblem overlap bound):

$$
\begin{aligned}
|f_j(x') - f_j(x)| &\leq L \cdot D(S_k) + C \cdot |S_k \cap \mathcal{O}| && \text{(Lemma E.24)} \\
&\leq L \cdot D(s) + C \cdot s && (|S_k| \leq s, |S_k \cap \mathcal{O}| \leq s)
\end{aligned}
$$

For the tighter bound involving $\rho$, note that in sliding window decomposition, each position in $S_k$ belongs to at most $\rho$ subproblems, so at most $(\rho - 1) \cdot s$ coordinates in $S_k$ are in the overlap set $\mathcal{O}$. The bound $C \cdot (\rho - 1) \cdot s$ is tighter when overlap multiplicity is explicitly controlled; the bound $C \cdot s$ is the uniform worst case (when $\rho$ is not tracked). $\square$

**Corollary E.26** (**Bounded Coupling Error per Iteration**). *The coupling error $C_t$ defined in Definition E.6 satisfies:*

$$|C_t| \leq C \cdot (\rho_t - 1) \cdot s$$

*where $\rho_t := \max_i |\{k : i \in S_k^{(t)}\}|$ is the maximum overlap multiplicity at iteration $t$.*

### E.7.5. DIMINISHING OVERLAP

A key feature of our algorithm is diminishing overlap, which reduces coordination overhead as the algorithm converges.

**Lemma E.27** (**Diminishing Overlap Schedule**). *Under the diminishing overlap schedule $o_t = \lfloor o_0 \cdot t^{-\alpha} \rfloor$ with $\alpha > 0$:*

(a) *The overlap converges:* $\lim_{t \to \infty} o_t = 0$

(b) *The overlap multiplicity converges:* $\lim_{t \to \infty} \rho_t = 1$

(c) *The overlap set vanishes:* $\lim_{t \to \infty} |\mathcal{O}_t| = 0$

*Proof.* **(a)** Since $t^{-\alpha} \to 0$ as $t \to \infty$ for $\alpha > 0$, we have $o_0 \cdot t^{-\alpha} \to 0$, hence $o_t = \lfloor o_0 \cdot t^{-\alpha} \rfloor \to 0$.

**(b)** With $o_t = 0$, the sliding window step size becomes $s - o_t = s$, meaning subproblems are disjoint. Each index then belongs to exactly one subproblem, so $\rho_t = 1$.

**(c)** When $\rho_t = 1$, no index appears in multiple subproblems, so $\mathcal{O}_t = \emptyset$. $\square$

**Theorem E.28** (**Vanishing Coupling Error**). *Under diminishing overlap with decay rate $\alpha > 0$, the coupling error satisfies:*

$$|C_t| \leq C \cdot s \cdot o_0 \cdot t^{-\alpha}$$

*In particular:*

*(a) The per-iteration coupling error vanishes:* $C_t \to 0$ *as* $t \to \infty$

*(b) The cumulative coupling error is bounded:*

$$\sum_{t=1}^{T} C_t \leq \begin{cases} O(T^{1-\alpha}) & \text{if } \alpha < 1 \\ O(\log T) & \text{if } \alpha = 1 \\ O(1) & \text{if } \alpha > 1 \end{cases}$$

*Proof.* **(a)** By Corollary E.26, $|C_t| \leq C \cdot (\rho_t - 1) \cdot s$. With sliding window overlap $o_t$, the multiplicity satisfies $\rho_t - 1 \leq \lceil o_t/(s - o_t) \rceil \leq O(o_t)$ for $o_t < s/2$. Substituting $o_t = O(t^{-\alpha})$:

$$|C_t| \leq C \cdot s \cdot O(t^{-\alpha}) = O(t^{-\alpha}) \to 0$$

**(b)** Summing the bound:

$$\sum_{t=1}^{T} C_t \leq C' \sum_{t=1}^{T} t^{-\alpha}$$

The sum $\sum_{t=1}^{T} t^{-\alpha}$ is: (1) $O(T^{1-\alpha})$ for $\alpha < 1$ (integral bound) (2) $O(\log T)$ for $\alpha = 1$ (harmonic series), (3) $O(1)$ for $\alpha > 1$ (convergent series). $\qquad\square$

*Remark* E.29 (**Optimal Decay Rate**). The choice $\alpha = 0.5$ (as in our implementation) gives $\sum_t C_t = O(\sqrt{T})$, which matches the $O(\sqrt{T})$ scaling of the subproblem regret terms. This balances exploration (via overlap) in early iterations with exploitation (via reduced coordination cost) in later iterations.

### E.7.6. DOMAIN-AGNOSTIC LOCAL OPERATORS

Our algorithm uses domain-agnostic local operators that respect the metric structure.

**Definition E.30** (**Unit Perturbation Operator**). A local operator $T_S : \mathcal{X} \to \mathcal{X}$ restricted to indices $S \subseteq [n]$ is a *unit perturbation* if:

1. $T_S(x)$ differs from $x$ only on coordinates in $S$

2. $\delta(x, T_S(x)) \leq 1$

**Domain-Specific Unit Perturbations**

- *Permutations (Kendall tau):* A single adjacent swap $(x_i, x_{i+1}) \mapsto (x_{i+1}, x_i)$ gives $d_\tau(x, T(x)) = 1$.

- *Binary vectors (Hamming):* A single bit flip $x_i \mapsto 1 - x_i$ gives $d_H(x, T(x)) = 1$.

- *Categorical:* Changing one coordinate to an adjacent category.

**Lemma E.31** (**Unit Perturbation Bound**). *Let* $T_S : \mathcal{X} \to \mathcal{X}$ *be a unit perturbation operator restricted to indices* $S$*. Then:*

$$|f_j(T_S(x)) - f_j(x)| \leq L + C \cdot |S \cap \mathcal{O}|$$

*Proof.* Apply Assumption E.2.3 with $\delta(x, T_S(x)) \leq 1$:

$$|f_j(T_S(x)) - f_j(x)| \leq L \cdot \delta(x, T_S(x)) + C \cdot |S \cap \mathcal{O}| \leq L + C \cdot |S \cap \mathcal{O}|$$

$\qquad\square$

*Remark* E.32 (**Adaptive Step Sizes**). Lemma E.31 suggests using larger steps (multiple unit perturbations) when far from the optimum, and smaller steps when close. This motivates adaptive step size selection based on estimated distance to local optima, as implemented in our algorithm.

### E.7.7. SUMMARY

We have established the following guarantees for our metric-based decomposition:

1. **Local Stability** (Lemma E.21): Modifications within a subproblem induce objective changes bounded by $L \cdot D(S_k)$.

2. **Bounded Coupling** (Proposition E.25): The decomposition satisfies Assumption E.2.3 with coupling error $O((\rho-1)\cdot s)$.

3. **Vanishing Coupling** (Theorem E.28): Under diminishing overlap, cumulative coupling error is $O(\sqrt{T})$ for decay rate $\alpha = 0.5$.

4. **Domain Agnosticity**: The construction applies to any combinatorial domain with a well-defined metric.

These results justify the regret decomposition in Definition E.6 and the bounds derived in subsequent sections.

### E.8. Proof Appendix

E.8.1. UCB SUBPROBLEM REGRET PROOF

**Lemma E.33** (**UCB Subproblem Regret (Position–Value UCB)**). *For any subproblem $k$ with index set $I_k$ and activation set $\mathcal{T}_k$, with $T_k := |\mathcal{T}_k|$. Assume the normalized scalar reward satisfies $r_t \in [0,1]$ and Assumption E.2.5 (A.5') holds, i.e., there exist $\{\mu_{i,j}\}_{i \in I_k, \, j \in \mathcal{A}_i} \subset [0,1]$ such that $g_k(x^{(k)}) = \sum_{i \in I_k} \mu_{i,x_i}$ and $\mathbb{E}[r_t \mid \mathcal{F}_{t-1}, \, x_{i,t} = j] = \mu_{i,j} + b_{i,t}$ with $b_{i,t}$ independent of $j$. Suppose subproblem $k$ is handled by a* position–value UCB *expert that maintains estimates $\hat{\mu}_{i,j}$ and counts $N_{i,j}$ for each $(i,j)$, and at each $t \in \mathcal{T}_k$ updates* all *positions $i \in I_k$ for the realized value $x_{i,t}$ using the same reward $r_t$. Let $A_{\max} := \max_{i \in I_k} |\mathcal{A}_i|$. Then there exists some constant $c > 0$ such that*

$$\mathbb{E}\left[R_k^{\mathrm{UCB}}(T_k)\right] := \mathbb{E}\left[\sum_{t \in \mathcal{T}_k} \left(g_k(x_k^\star) - g_k(x_t^{(k)})\right)\right] \le c'|I_k|\sqrt{T_k \log T}$$

*where $x_k^\star := (x^\star)^{(k)} = x_{I_k}^\star$ and $|I_k|$ is the number of positions/coordinates in subproblem $k$ and where $c' = c\sqrt{A_{\max}}$ and $A_{\max} = O(1)$, i.e., $A_{\max}$ is problem-dependent and treated as a constant.*

*Proof sketch.* For a fixed subproblem $k$ and a round $t \in \mathcal{T}_k$. By Assumption E.2.5 (A.5') the surrogate is locally additive, $g_k(x^{(k)}) = \sum_{i \in I_k} \mu_{i,x_i}$, hence, we can write

$$g_k(x_k^\star) - g_k(x_t^{(k)}) = \sum_{i \in I_k} \left(\mu_{i,x_i^\star} - \mu_{i,x_{i,t}}\right) = \sum_{i \in I_k} \Delta_{i,x_{i,t}}, \tag{81}$$

where $\mu_{i,j}$ is the mean reward of choosing value $j$ at position $i$, and $\Delta_{i,j} := \mu_{i,x_i^\star} - \mu_{i,j}$ is the sub-optimality gap for value $j$ at position $i$. Let $N_{i,j}(t)$ be the number of times value $j$ has been selected at position $i$ up to time $t$ (restricted to activations of subproblem $k$). Under Assumption E.2.5, when $x_{i,t} = j$ we observe $r_t = \mu_{i,j} + b_{i,t} + \eta_t$ where $b_{i,t}$ does not depend on $j$ and $\eta_t$ is conditionally 1-sub-Gaussian (or bounded).

Thus, for fixed $(i,j)$, define the average bias along the samples of arm $(i,j)$:

$$\bar{b}_{i,j}(t) := \frac{1}{N_{i,j}(t)} \sum_{s \le t: \, x_{i,s}=j} b_{i,s}.$$

Since $r_s \in [0,1]$ and $\eta_s := r_s - \mathbb{E}[r_s \mid \mathcal{F}_{s-1}, x_{i,s}]$ is a bounded martingale difference, a standard Azuma/Hoeffding argument yields with probability at least $1 - 2/t^2$,

$$\left|\hat{\mu}_{i,j}(t) - \mu_{i,j} - \bar{b}_{i,j}(t)\right| \le \sqrt{\frac{2 \log t}{N_{i,j}(t)}}. \tag{82}$$

By Assumption E.2.5 (A.5'), for fixed $i$ the bias term is independent of $j$, so $\bar{b}_{i,j}(t) = \bar{b}_{i,j'}(t)$ for all $j, j' \in \mathcal{A}_i$ (hence it cancels in within-position comparisons). Now, the position–value UCB the index for $(i,j)$ is of the form,

$$\hat{u}_{i,j}(t) = \hat{\mu}_{i,j}(t) + c\sqrt{\frac{2 \log t}{N_{i,j}(t)}},$$

for some constant $c > 0$. Suppose at time $t$ the algorithm selects a suboptimal value $j \neq x_i^\star$ for position $i$. Then by the UCB selection rule, its index must be no smaller than the index of the optimal value $x_i^*$:

$$\hat{u}_{i,j}(t) \geq \hat{u}_{i,x_i^\star}(t). \tag{83}$$

On the event where (82) holds for both $(i,j)$ and $(i,x_i^\star)$, combining (83) with the concentration bounds yields

$$\hat{\mu}_{i,j}(t) - \hat{\mu}_{i,x_i^\star}(t) \geq c\sqrt{\frac{2\log t}{N_{i,j}(t)}} - c\sqrt{\frac{2\log t}{N_{i,x_i^\star}(t)}}.$$

Rewriting the gap $\Delta_{i,j}$ and using triangle inequality, we obtain:

$$\begin{aligned}
\Delta_{i,j} &= \mu_{i,x_i^\star} - \mu_{i,j} \\
&= (\mu_{i,x_i^\star} - \hat{\mu}_{i,x_i^\star}) + (\hat{\mu}_{i,j} - \mu_{i,j}) - (\hat{\mu}_{i,j} - \hat{\mu}_{i,x_i^\star}) \\
&\leq \sqrt{\frac{2\log t}{N_{i,j}(t)}} + \sqrt{\frac{2\log t}{N_{i,x_i^\star}(t)}} - (\hat{\mu}_{i,j} - \hat{\mu}_{i,x_i^\star}) \\
&\leq 2\sqrt{\frac{2\log t}{N_{i,j}(t)}},
\end{aligned}$$

where the last inequality uses (83) together with (82) (applied to both $(i,j)$ and $(i,x_i^\star)$) and the fact that the bias terms cancel across values at the same position.

Thus, whenever a suboptimal value $j \neq x_i^\star$ is selected at position $i$ at time $t$, we must have,

$$\Delta_{i,j} \leq 2\sqrt{\frac{2\log t}{N_{i,j}(t)}} \quad \Rightarrow \quad N_{i,j}(t) \leq \frac{8\log t}{\Delta_{i,j}^2} \leq \frac{8\log T}{\Delta_{i,j}^2}.$$

Therefore, the number of times a suboptimal value $j$ can be selected at position $i$ is at most $N_{i,j}(T_k) \leq \min\{T_k, \frac{8\log T}{\Delta_{i,j}^2}\}$. Plugging this into the regret expression and using (81),

$$\begin{aligned}
R_k^{\text{UCB}}(T_k) &= \sum_{t \in \mathcal{T}_k} \sum_{i \in I_k} \Delta_{i,x_{i,t}} = \sum_{i \in I_k} \sum_{j \neq x_i^\star} \Delta_{i,j} N_{i,j}(T_k) \\
&\leq \sum_{i \in I_k} \sum_{j \neq x_i^\star} \Delta_{i,j} \cdot \min\left\{T_k, \frac{8\log T}{\Delta_{i,j}^2}\right\}.
\end{aligned}$$

Rather than continuing with a gap-dependent summation, we invoke the standard gap-free regret bound for UCB. Applying the standard gap-free regret bound for UCB1 (Auer et al., 2002a) to the $A_i$-armed position-$i$ bandit over $T_k$ rounds yields,

$$\mathbb{E}[R_i(T_k)] \leq C\sqrt{A_i T_k \log T}.$$

and since $A_i \leq A_{\max}$, summing over $i \in I_k$ yields

$$\mathbb{E}\left[R_k^{\text{UCB}}(T_k)\right] \leq \sum_{i \in I_k} C\sqrt{A_i T_k \log T} \leq C|I_k|\sqrt{A_{\max} T_k \log T}.$$

for a universal constant $C > 0$. $\qquad\square$

### E.8.2. EXP3 SUBPROBLEM REGRET PROOF

**Lemma E.34** (**Unbiased Exp3 Subproblem Regret**)**.** *Consider a subproblem $k$ with $d$ positions, where each position $i \in I_k$ has an effective choice of size $d$ (due to decomposition constraints). Let $T_k$ denote the total iterations spent optimizing subproblem $k$ and at each iteration $t$, the algorithm observes only the total reward $\rho_t$ for the selected solution $\boldsymbol{x}_t$. Therefore, for each position $i \in I_k$, we use importance-weighted reward estimates as:*

$$\hat{\rho}_{i,j}(t) = \frac{\rho_t \cdot \mathbb{I}[x_t(i) = j]}{p_{i,j}(t)}$$

where $p_{i,j}(t)$ is the probability of selecting value $j$ for position $i$. Standard Exp3 analysis with learning rate $\eta$ gives:

$$\sum_{t=1}^{T_k} \mathbb{E}[\rho_{i,x_{i*}} - \rho_{i,a_{i,t}}] \leq \frac{\log d}{\eta} + \frac{\eta T_k B^2}{2}$$

where $x_i^*$ is the optimal value for position $i$, $a_{i,t}$ is the value selected at iteration $t$, $B$ is the maximum reward magnitude, and the expectation is over Exp3's randomized selection.

*(a) Per-subproblem bound:* Summing over all $d$ positions in subproblem $k$:

$$\sum_{t \in T_k} \mathbb{E}[g_k(\boldsymbol{x}_{I_k}^*) - g_k(\boldsymbol{x}_{I_{k,t}})] \leq d \left( \frac{\log d}{\eta} + \frac{\eta T_k B^2}{2} \right) \tag{84}$$

*(b) Total bound across subproblems:* The expected regret of the global Exp3 strategy over $T$ iterations is:

$$\sum_{k=1}^{K} \left[ \sum_{t \in T_k} \mathbb{E}[g_k(\boldsymbol{x}_{I_k}^*) - g_k(\boldsymbol{x}_{I_{k,t}})] \right] \leq d\sqrt{2KT \log d \cdot B^2} \tag{85}$$

*Proof.* For a subproblem $k$ with $d$ positions, and assuming each position has an effective choice of size $d$ due to decomposition, the total regret per subproblem is the sum of regrets for each position $i \in I_k$:

$$\sum_{t \in T_k} \mathbb{E}[g_k(\boldsymbol{x}_{I_k}^*) - g_k(\boldsymbol{x}_{I_{k,t}})] \leq \sum_{i \in I_k} \left( \frac{\log d}{\eta} + \frac{\eta T_k B^2}{2} \right) = d \left( \frac{\log d}{\eta} + \frac{\eta T_k B^2}{2} \right)$$

With decomposition, each position only has $d$ relevant values (those compatible with other positions in the subproblem), so we can write:

$$\text{Total Exp3 Regret} = \sum_{k=1}^{K} \left[ \sum_{t \in T_k} \mathbb{E}[g_k(\boldsymbol{x}_{I_k}^*) - g_k(\boldsymbol{x}_{I_{k,t}})] \right]$$

$$\sum_{k=1}^{K} \left[ \sum_{t \in T_k} \mathbb{E}[g_k(\boldsymbol{x}_{I_k}^*) - g_k(\boldsymbol{x}_{I_{k,t}})] \right] \leq \sum_{k=1}^{K} d \left( \frac{\log d}{\eta} + \frac{\eta T_k B^2}{2} \right)$$

$$= \sum_{k=1}^{K} d \left( \frac{\log d}{\eta} \right) + \sum_{k=1}^{K} d \frac{\eta T_k B^2}{2}$$

$$= dK \frac{\log d}{\eta} + d \frac{\eta T B^2}{2}$$

where in the last line we used $\sum_{k=1}^{K} T_k = T$.

To get the best possible regret bound, we need to choose the optimal value of $\eta$ that minimizes the upper bound:

$$\frac{d}{d\eta} \left[ dK \frac{\log d}{\eta} + d \frac{\eta T B^2}{2} \right] = 0$$

$$-\frac{dK \log d}{\eta^2} + d \frac{T B^2}{2} = 0$$

$$\implies \eta = \sqrt{\frac{2K \log d}{T B^2}}$$

Substituting the optimal $\eta$ back into the inequality:

$$\sum_{k=1}^{K}\left[\sum_{t\in T_k}\mathbb{E}[g_k(\boldsymbol{x}_{I_k}^*) - g_k(\boldsymbol{x}_{I_{k,t}})]\right] \le dK\frac{\log d}{\sqrt{\frac{2K\log d}{TB^2}}} + d\sqrt{\frac{2K\log d}{TB^2}}\cdot\frac{TB^2}{2}$$

$$= dK\log d\sqrt{\frac{TB^2}{2K\log d}} + d\sqrt{\frac{2K\log d\cdot TB^2}{4}}$$

$$= d\sqrt{KT\log d\cdot B^2}\left(\sqrt{2}\right)$$

$$\le d\sqrt{2KT\log d\cdot B^2}$$

This completes the proof. □

### E.8.3. FOLLOW THE REGULARIZED LEADER (FTRL)

A learning algorithm that makes decisions based on cumulative losses from all past rounds and uses regularization to prevent overfitting to early observations.

**Lemma E.35** (FTRL subproblem regret). *For $K$ subproblems each of fixed size $d$, with overlap set $O$, under full-bandit feedback with clipped-IS at floor $C$, the FTRL regret with confidence-bonus regularization is:*

$$\sum_{k=1}^{K}\sum_{t=1}^{T_k}[g_k(x_{I_k}^*) - g_k(x_{I_{k,t}})] \le \frac{C'Bd\sqrt{KT\log T}}{C} \tag{86}$$

*where $C'$ absorbs all constants and is $C' = O(\sqrt{d})$.*

**Proof:** FTRL update rule,

$$p_i^{(t)} = \arg\min_{p_i\in\Delta_d}\left\{\sum_{s=1}^{t-1}\langle p_i, \hat{l}_s(i)\rangle + \text{Reg}_t(p_i)\right\} \tag{87}$$

Using the FTRL-Prox regret bound (McMahan 2017; Lattimore & Szepesvári 2020, Ch. 28) with time-varying regularizer $\text{Reg}_t$ and clipped-IS losses $\hat{l}_t$ states that,

$$\sum_{t=1}^{T}\langle p^t, \hat{l}_t\rangle - \sum_{t=1}^{T}\langle p^*, \hat{l}_t\rangle \le \underbrace{\text{Reg}_T(p^*) - \text{Reg}_0(p^{(1)})}_{\text{regularizer diameter}} + \underbrace{\sum_{t=1}^{T}\|\hat{l}_t\|_*^2\cdot\text{stab}(\text{Reg}_t)}_{\text{variance/stability term}}$$

Now, for Regularizer $\text{Reg}_t$, since we operate under *full-bandit feedback with clipped IS*, we use a history-dependent *confidence-bonus regularizer* (McMahan, 2017), where $\hat{l}_s(i,j)$ is the clipped importance-weighted loss estimator shared with Expert 2 (Lemma E.16) and can be described as,

$$\text{Reg}_t(p_i) := -\gamma\sum_{j=1}^{d}p_{i,j}\sqrt{N_t(i,j)+1}, \qquad \gamma = \frac{B}{C}\sqrt{2\log T},$$

where $N_t(i,j)$ is the visit count of value $j$ at position $i$, $B$ bounds the per-round loss, and $C$ is the IS clipping floor. Since $\text{Reg}_t(p_i)$ is linear in $p_i$, the FTRL objective (87) is linear over the simplex, and its minimum is attained at a vertex,

$$x_{i,t} = \arg\min_{j\in[d]}\underbrace{\left[L_t(i,j) - \gamma\sqrt{N_t(i,j)+1}\right]}_{\text{lower confidence bound on cumulative loss}},$$

recovering Algorithm 2 (Expert 3). This LCB-index form admits RBMLE-style analysis (Auer et al., 2002a; Kumar & Becker, 2003; Liu et al., 2020), encoding optimism under uncertainty.

Now, bounding the first term in Eq. 87 i.e *regularizer diameter*, at the comparator vertex $p^* = e_{j^*}$ with $N_T(j^*) \leq T$ (holds trivially because any arm can be visited at most T times in T rounds), and uniform initialization $p^{(1)}$ where $\text{Reg}_0(p^{(1)}) = -\gamma$,

$$\left| \text{Reg}_T(p^*) - \text{Reg}_0(p^{(1)}) \right| = \gamma\left(\sqrt{N_T(j^*) + 1} - 1\right) \leq \gamma\sqrt{T + 1}.$$

Next, to bound the *variance/stability* term, by Thm. E.16 (shared variance/bias control with Expert 2), the clipped-IS estimator is bounded uniformly as,

$$|\hat{l}_s(i,j)| \leq \frac{B}{C}, \qquad \forall\, s, i, j. \tag{88}$$

where $C$ is the IS clipping floor from Algorithm 5 (shared with Expert 2). Under the LCB-index interpretation established above, $\text{Reg}_t$ is linear in $p_i$ and the FTRL stability factor $\text{stab}(\text{Reg}_t)$ degenerates (zero strong-convexity), so the generic FTRL-stability bound is not directly applicable. Instead, the cumulative stability term — which measures the gap between our algorithm's plays and the comparator $p^* = e_{j^*}$ is bounded directly through the UCB visit-count argument. By the optimism property of the LCB index and the per-round bound Eq. 88, any suboptimal arm $j \in [d]$ at position $i$ with gap $\Delta_{i,j} := \mathbb{E}[\ell(i,j)] - \mathbb{E}[\ell(i,j_i^*)]$ satisfies,

$$\mathbb{E}[N_{T_k}(i,j)] \leq \frac{8B^2 \log T}{C^2 \Delta_{i,j}^2} + O(1)$$

in expectation (Auer et al. 2002a, Thm. 1; Liu et al. 2020, Thm. 1). The cumulative stability term in the FTRL bound is therefore controlled by the expected regret contribution of suboptimal pulls:

$$\underbrace{\sum_{t=1}^{T_k} \|\hat{l}_t\|_*^2 \cdot \text{stab}(\text{Reg}_t)}_{\text{degenerate for linear } \text{Reg}_t} \longrightarrow \underbrace{\sum_{j \neq j_i^*} \Delta_{i,j} \cdot \mathbb{E}[N_{T_k}(i,j)]}_{\text{equivalent visit-count bound}} \leq \sum_{j \neq j_i^*} \frac{8B^2 \log T}{C^2 \Delta_{i,j}},$$

where both expressions bound the cumulative deviation $\sum_t \langle p_t - p^*, \hat{l}_t \rangle$ but the right-hand expression is the one that admits an explicit upper bound under linear regularization. This converts the (formally degenerate) FTRL stability term into the equivalent UCB regret expression.

Aggregating over arms via Cauchy–Schwarz on $\sum_j 1/\Delta_{i,j}$ subject to the budget $\sum_j N_{T_k}(i,j) \leq T_k$:

$$\sum_{j \neq j_i^*} \frac{1}{\Delta_{i,j}} \leq \sqrt{d \cdot \sum_{j \neq j_i^*} \frac{1}{\Delta_{i,j}^2}} \leq \sqrt{d \cdot \frac{T_k C^2}{8B^2 \log T}} = \frac{C\sqrt{dT_k}}{B\sqrt{8 \log T}}.$$

Substituting back,

$$\sum_{j \neq j_i^*} \frac{8B^2 \log T}{C^2 \Delta_{i,j}} \leq \frac{8B^2 \log T}{C^2} \cdot \frac{C\sqrt{dT_k}}{B\sqrt{8 \log T}} = \frac{B}{C}\sqrt{8\,d\,T_k\,\log T}.$$

Combining with the diameter bound $\gamma\sqrt{T_k + 1}$ where $\gamma = (B/C)\sqrt{2 \log T}$, the per-position regret satisfies,

$$\sum_{t=1}^{T} \langle p_i^t, \hat{l}_t(i) \rangle - \sum_{t=1}^{T} \langle p^*, \hat{l}_t(i) \rangle \leq \underbrace{\gamma\sqrt{T+1}}_{= \frac{B}{C}\sqrt{2(T+1)\log T}} + \underbrace{\frac{B}{C}\sqrt{8dT \log T}}_{\text{from stability}} \leq \frac{2B}{C}\sqrt{2dT\,\log T},$$

where the last inequality absorbs the diameter into the dominant stability term up to constants (the diameter is smaller than the stability by a factor of $\sqrt{d}$).

This establishes the **regret for a single position** under clipped-IS confidence-bonus FTRL. The extra $\sqrt{d}$ factor relative to the entropic full-information rate is the structural cost of bandit feedback under UCB-style worst-case-gap aggregation.

Now, $\text{Regret}_T = \sum_{t=1}^{T} \langle p^t, \hat{l}_t \rangle - \sum_{t=1}^{T} \langle p^*, \hat{l}_t \rangle$ and for any subproblem $k$ with dimension $d$ and time horizon $T_k$, summing per-position regrets across the $d$ positions in $I_k$:

For subproblem $k$ with $d$ positions:

$$\sum_{k=1}^{K}\sum_{t=1}^{T_k}[g_k(x^*_{I_k}) - g_k(x_{I_{k,t}})] = \sum_{k=1}^{K}\sum_{t=1}^{T_k}\sum_{i\in I_k}[l_t(i,x_{i,t}) - l_t(i,x^*_i)] = \sum_{k=1}^{K}\sum_{i\in I_k}\sum_{t=1}^{T_k}[\langle p^t_i, l_t(i)\rangle - l_t(i,x^*_i)]$$

$$\le C'\sum_{k=1}^{K}\sum_{i\in I_k}\frac{Bd}{C}\sqrt{T_k\log T} = \sum_{k=1}^{K}\frac{Bd}{C}\sqrt{T_k\log T} = \frac{Bd\sqrt{\log T}}{C}\sum_{k=1}^{K}\sqrt{T_k}.$$

Using the Cauchy-Schwarz inequality,

$$\sum_{k=1}^{K}\sqrt{T_k} \le \sqrt{\sum_{k=1}^{K}(\sqrt{T_k})^2}\sqrt{\sum_{k=1}^{K}(1)^2} \le \sqrt{KT}$$

$$\implies \sum_{k=1}^{K}\sum_{t=1}^{T_k}[g_k(x^*_{I_k}) - g_k(x_{I_{k,t}})] \le \frac{C'Bd\sqrt{KT\log T}}{C}$$

$\square$

*Remark* E.36. Since the confidence-bonus regularizer is linear in $p_i$, so it should not be analyzed through the usual strongly-convex FTRL stability bound. Instead, we use the FTRL-Prox form to derive the implemented decision rule, and then analyze the induced index policy via RBMLE visit-count arguments.

### E.8.4. COUPLING ERROR: LAGRANGIAN COORDINATION

We employ Lagrangian coordination to handle the overlapping positions that arise from our decomposition of search space. Lagrangian relaxation provides a principled mathematical framework for decomposing optimization problems with coupling constraints.

From our decomposition, each position $i$ may appear in multiple subproblems. Let $S_i$ denote the set of subproblems containing position $i$, with $k_i = |S_i|$ being the overlap count. For positions with $k_i > 1$, we require coordination. Rather than enforcing hard equality constraints (all subproblems must assign identical values), we use a **soft violation measure** that quantifies coordination risk. For position $i$ at iteration $t$:

$$\xi_i^{(t)} = (k_i - 1)\cdot\text{Var}(\hat{\mu}_i^{(t)})\cdot\left(1 - \frac{N_{i,v_i^{(t)}}}{\sum_v N_{i,v}}\right) \tag{89}$$

This continuous measure captures: (1) overlap degree $(k_i - 1)$, (2) value estimate uncertainty $\text{Var}(\hat{\mu}_i)$, and (3) assignment confidence (visit ratio). As learning progresses, $\xi_i^{(t)} \to 0$, naturally relaxing coordination penalties. Now, let $O = \{i : k_i > 1\}$ denote the set of overlapping positions. Then, Lagrangian for our coordination problem is:

$$L(x,\lambda) = f(x) + \sum_{i\in O}\lambda_i\xi_i(x)$$

where $f(x)$ is the original objective (total reward) and $\lambda_i \ge 0$ are dual variables penalizing soft violations. This follows a primal-dual relationship where:

$$\text{Primal Problem:}\quad \min_x\ f(x)\quad\text{s.t.}\quad \xi_i(x) = 0\ \forall i\in O \tag{90}$$

$$\text{Dual Problem:}\quad \max_{\lambda\ge 0}\ g(\lambda)\quad\text{where}\quad g(\lambda) = \min_x\ L(x,\lambda) \tag{91}$$

Now, for our combinatorial setting, the primal objective $f(x)$ is non-convex and non-smooth. However, the dual function $g(\lambda) = \min_x[f(x) + \lambda^T\xi(x)]$ is **always concave** (even when the primal is non-convex!), as it is the pointwise minimum of affine functions in $\lambda$ (Boyd et al., 2003).

Therefore, we solve the Lagrangian dual problem by maximizing the concave function $g(\lambda)$. Equivalently, we minimize the convex function $-g(\lambda)$. The subgradient of $-g(\lambda)$ at $\lambda$ is $-\xi(x_t)$ where $x_t = \arg\min_x L(x,\lambda_t)$.

**Accelerated Dual Optimization via Mirror Descent.** Standard subgradient ascent for maximizing $g(\lambda)$ achieves $O(\sqrt{T})$ cumulative dual gap. Since our dual function $g(\lambda)$ is concave (making $-g(\lambda)$ convex), we can apply **online convex optimization (OCO)** methods and we use mirror descent with entropic regularization operating in log-space: $\lambda_{t+1} = \exp(\log(\lambda_t) + \alpha_t \xi_t)$. This offers three advantages: (1) the exponential update automatically enforces $\lambda \geq 0$ without projection, (2) multiplicative updates are scale-adaptive, and (3) mirror descent is optimal for bounded domains (Bubeck, 2015). We apply **Polyak-style heavy-ball momentum** $\lambda_{t+1} \leftarrow \tilde{\lambda}_t + \theta_t(\tilde{\lambda}_t - \lambda_t)$ with $\theta_t = 2/(t+1)$, which empirically accelerates early-iteration convergence while maintaining the optimal $O(\sqrt{T})$ rate.

Using mirror descent with Polyak-style heavy-ball acceleration to optimize the dual variables $\lambda$, we obtain the following bound on coupling error.

**Lemma E.37 (Dual Gap Bound for Mirror Descent).** *Let $g : \mathcal{R}^d \to \mathcal{R}$ be a concave function with optimum at $\lambda^* \in \mathcal{R}^d$ satisfying $\lambda^* \in [0, \lambda_{max}]^d$. For all $t \in \{1, \ldots, T\}$, let $\xi_t \in \mathbb{R}^d$ be a subgradient of $g$ at $\lambda_t$ with $||\xi_t||_\infty \leq L$. Assume $\lambda_1 \in [0, \lambda_{max}]^d$. Under mirror descent with entropic regularization and update rule:*

$$\lambda_{t+1} = \Pi_{[0,\lambda_{max}]} \left[\exp\left(\log(\lambda_t) + \alpha_t \xi_t\right)\right] \tag{92}$$

*with step size $\alpha_t = \frac{\alpha_0}{\sqrt{t}}$, the cumulative dual gap is bounded by:*

$$\sum_{t=1}^{T} (g(\lambda^*) - g(\lambda_t)) \leq O(d\lambda_{max} L \sqrt{T}) \tag{93}$$

*Proof.* The mirror descent update for the Lagrangian dual $\lambda_t$ operates in the dual space defined by the entropic regularizer $\Phi(\lambda) = \sum_i \lambda_i \log \lambda_i$. Starting from the general mirror descent iteration, we specialize to the entropic regularizer $\Phi(\lambda) = \sum_i \lambda_i \log \lambda_i$ to obtain our exponential update rule and then bound convergence using Bregman divergence analysis.

Now, the standard mirror descent update for minimizing any function $f$ is:

$$\lambda_{t+1} = \arg\min_\lambda \left\{ \langle \nabla f(\lambda_t), \lambda \rangle + \frac{1}{\alpha_t} D_\Phi(\lambda || \lambda_t) \right\} \tag{94}$$

where $D_\Phi(\lambda || \mu) = \Phi(\lambda) - \Phi(\mu) - \langle \nabla \Phi(\mu), \lambda - \mu \rangle$ is the Bregman divergence.
Since we are maximizing $g(\lambda)$, we minimize $f(\lambda) = -g(\lambda)$, so $\nabla f(\lambda_t) = -\xi_t$ where $\xi_t$ is a subgradient of $g$, therefore,

$$\lambda_{t+1} = \arg\min_\lambda \left\{ -\langle \xi_t, \lambda \rangle + \frac{1}{\alpha_t} D_\Phi(\lambda || \lambda_t) \right\} \tag{95}$$

Expanding the Bregman divergence,

$$\lambda_{t+1} = \arg\min_\lambda \left\{ -\langle \xi_t, \lambda \rangle + \frac{1}{\alpha_t} \left[ \Phi(\lambda) - \Phi(\lambda_t) - \langle \nabla \Phi(\lambda_t), \lambda - \lambda_t \rangle \right] \right\}$$

$$\lambda_{t+1} = \arg\min_\lambda \left\{ -\langle \xi_t, \lambda \rangle + \frac{1}{\alpha_t} \Phi(\lambda) - \frac{1}{\alpha_t} \Phi(\lambda_t) - \frac{1}{\alpha_t} \langle \nabla \Phi(\lambda_t), \lambda \rangle + \frac{1}{\alpha_t} \langle \nabla \Phi(\lambda_t), \lambda_t \rangle \right\}$$

The terms $-\frac{1}{\alpha_t} \Phi(\lambda_t)$ and $+\frac{1}{\alpha_t} \langle \nabla \Phi(\lambda_t), \lambda_t \rangle$ are constants with respect to $\lambda$ (they only depend on $\lambda_t$, which is fixed at iteration $t$). Since adding a constant to an objective does not change its minimizer, we drop these terms and rearrange to get,

$$\lambda_{t+1} = \arg\min_\lambda \left\{ \frac{1}{\alpha_t} \Phi(\lambda) - \left\langle \xi_t + \frac{1}{\alpha_t} \nabla \Phi(\lambda_t), \lambda \right\rangle \right\}$$

By first-order optimality condition (dual space update), taking the gradient with respect to $\lambda$ and setting it to zero,

$$\frac{\partial}{\partial \lambda} \left[ \frac{1}{\alpha_t} \Phi(\lambda) - \left\langle \xi_t + \frac{1}{\alpha_t} \nabla \Phi(\lambda_t), \lambda \right\rangle \right] \Bigg|_{\lambda = \lambda_{t+1}} = 0$$

$$\frac{1}{\alpha_t} \nabla \Phi(\lambda_{t+1}) - \left( \xi_t + \frac{1}{\alpha_t} \nabla \Phi(\lambda_t) \right) = 0$$

Multiplying by $\alpha_t$ and rearranging:

$$\nabla\Phi(\lambda_{t+1}) = \nabla\Phi(\lambda_t) + \alpha_t\xi_t \tag{96}$$

This is the **dual space update rule**. The update happens in the "gradient space" of $\Phi$, not directly in $\lambda$-space. We add the subgradient $\alpha_t\xi_t$ to $\nabla\Phi(\lambda_t)$ to get $\nabla\Phi(\lambda_{t+1})$.

Note, the dual space update $\nabla\Phi(\lambda_{t+1}) = \nabla\Phi(\lambda_t) + \alpha_t\xi_t$ holds for *any* choice of regularizer $\Phi$ [ref for standard gradiet descent]. We choose entropic regularization $\Phi(\lambda) = \sum_i \lambda_i \log \lambda_i$ because it naturally handles the constraint $\lambda \geq 0$ (dual variables must be non-negative). For this choice, the gradient is:

$$\frac{\partial\Phi}{\partial\lambda_i} = \frac{\partial}{\partial\lambda_i}[\lambda_i \log \lambda_i] = \log \lambda_i + \lambda_i \cdot \frac{1}{\lambda_i} = \log \lambda_i + 1 \tag{97}$$

Therefore (component-wise):

$$\nabla\Phi(\lambda) = \log\lambda + \mathbf{1} \tag{98}$$

where $\mathbf{1} = (1, 1, \ldots, 1)^\top$ is the vector of all ones. Substituting this into the dual space update we can derive the exponential update as,

$$\nabla\Phi(\lambda_{t+1}) = \nabla\Phi(\lambda_t) + \alpha_t\xi_t$$
$$\log\lambda_{t+1} + \mathbf{1} = \log\lambda_t + \mathbf{1} + \alpha_t\xi_t$$
$$\log\lambda_{t+1} = \log\lambda_t + \alpha_t\xi_t$$
$$\lambda_{t+1} = \exp(\log\lambda_t + \alpha_t\xi_t)$$

This gives the multiplicative update (component-wise):

$$\lambda_{i,t+1} = \lambda_{i,t} \cdot \exp(\alpha_t\xi_{i,t}) \tag{99}$$

followed by projection onto $[0, \lambda_{max}]^d$.

For convergence analysis, instead of bounding the optimality gap or primal-dual gap, we bound the **cumulative dual gap**, defined as $\sum_{t=1}^{T}[g(\lambda^*) - g(\lambda_t)]$, which measures how close our current dual solution is to the optimal dual solution over all $T$ iterations. Bounding the sum of these gaps over time guarantees that, on average, the Lagrangian coordination (LC) mechanism is making good progress toward optimality.

For that purpose, in standard sub-gradient analysis, one bounds progress using Euclidean distance $||\lambda_t - \lambda^*||_2^2$. For mirror descent, we instead use the Bregman divergence induced by the regularizer $\Phi$. For our entropic regularizer $\Phi(\lambda) = \sum_i \lambda_i \log \lambda_i$, this becomes:

$$D_\Phi(\lambda||\mu) = \sum_i \lambda_i \log\left(\frac{\lambda_i}{\mu_i}\right) - \sum_i(\lambda_i - \mu_i) \tag{100}$$

which is the **KL-divergence** (up to sign conventions). The Bregman divergence plays the same role as squared Euclidean distance, but is adapted to the geometry of entropic regularization.

Using the fundamental property of concave functions (first-order condition for concavity) (Boyd & Vandenberghe, 2004), for any two points $\lambda^*, \lambda_t$:

$$g(\lambda^*) \leq g(\lambda_t) + \langle\nabla g(\lambda_t), \lambda^* - \lambda_t\rangle \tag{101}$$

Substituting sub-gradient $g(\lambda^*)$ and rearranging,

$$g(\lambda^*) - g(\lambda_t) \leq \langle\xi_t, \lambda^* - \lambda_t\rangle \tag{102}$$

Therefore, the dual gap at iteration $t$ is related to the distance between the dual variables! Formally, the dual gap is bounded by the inner product of the sub-gradient with the distance to the optimum. This connects the dual gap to the geometry of the dual space, which we will exploit using Bregman divergences.

Using Three-Point property of mirror descent (Beck & Teboulle, 2003; Bubeck, 2015) that relates the Bregman divergence before and after an update $\lambda_{t+1} = \arg\min_\lambda \{\alpha_t\langle\xi_t, \lambda\rangle + D_\Phi(\lambda||\lambda_t)\}$:

$$D_\Phi(\lambda^*||\lambda_t) - D_\Phi(\lambda^*||\lambda_{t+1}) \geq \alpha_t\langle\xi_t, \lambda^* - \lambda_{t+1}\rangle \tag{103}$$

Using the sub-gradient inequality (102) ,

$$\alpha_t(g(\lambda^*) - g(\lambda_t)) \leq \alpha_t\langle\xi_t, \lambda^* - \lambda_t\rangle$$
$$\langle\xi_t, \lambda^* - \lambda_t\rangle = \langle\xi_t, \lambda^* - \lambda_t\rangle + \xi_t\lambda_{t+1} - \xi_t\lambda_{t+1} = \langle\xi_t, \lambda^* - \lambda_{t+1}\rangle + \langle\xi_t, \lambda_{t+1} - \lambda_t\rangle$$
$$= \alpha_t\langle\xi_t, \lambda^* - \lambda_{t+1}\rangle + \alpha_t\langle\xi_t, \lambda_{t+1} - \lambda_t\rangle$$
$$\leq D_\Phi(\lambda^*||\lambda_t) - D_\Phi(\lambda^*||\lambda_{t+1}) + \alpha_t\langle\xi_t, \lambda_{t+1} - \lambda_t\rangle$$

where the second inequality uses (103). Now to bound the Residual term $\alpha_t\langle\xi_t, \lambda_{t+1} - \lambda_t\rangle$, we use the fact that in log-space (see the mirror descent update):

$$\log\lambda_{t+1} - \log\lambda_t = \alpha_t\xi_t$$
$$\lambda_{i,t+1} = \lambda_{i,t}\exp(\alpha_t\xi_{i,t})$$

Therefore:

$$\lambda_{i,t+1} - \lambda_{i,t} = \lambda_{i,t}[\exp(\alpha_t\xi_{i,t}) - 1]$$

Using the inequality $e^x - 1 \leq xe^{|x|}$ for all $x$,

$$|\lambda_{i,t+1} - \lambda_{i,t}| = \lambda_{i,t}|\exp(\alpha_t\xi_{i,t}) - 1|$$
$$\leq \lambda_{i,t} \cdot \alpha_t|\xi_{i,t}| \cdot e^{\alpha_t|\xi_{i,t}|}$$
$$\leq \lambda_{max} \cdot \alpha_t L \cdot e^{\alpha_t L}$$

where as stated before $\lambda_{i,t} \leq \lambda_{max}$ and $|\xi_{i,t}| \leq ||\xi_t||_\infty \leq L$.
Using this, we can now simplify the residual term as,

$$|\langle\xi_t, \lambda_t - \lambda_{t+1}\rangle| \leq \sum_i |\xi_{i,t}| \cdot |\lambda_{i,t+1} - \lambda_{i,t}|$$
$$\leq \sum_i L \cdot \lambda_{max}\alpha_t Le^{\alpha_t L}$$
$$= d\lambda_{max}L^2\alpha_t e^{\alpha_t L}$$

Now, for small step sizes $\alpha_t L \ll 1$, we have $e^{\alpha_t L} \approx 1 + \alpha_t L$, giving:

$$\alpha_t\langle\xi_t, \lambda_{t+1} - \lambda_t\rangle \leq d\lambda_{max}L^2\alpha_t^2 + d\lambda_{max}L^3\alpha_t^3$$

For small step sizes we can ignore higher order terms and it can be absorbed in $O(\cdot)$,

$$\alpha_t\langle\xi_t, \lambda_{t+1} - \lambda_t\rangle \leq \alpha_t^2||\xi_t||_\infty^2 \cdot d\lambda_{max} \tag{104}$$
$$\implies \alpha_t(g(\lambda^*) - g(\lambda_t)) \leq D_\Phi(\lambda^*||\lambda_t) - D_\Phi(\lambda^*||\lambda_{t+1}) + \alpha_t^2 L^2 d\lambda_{max} \tag{105}$$

Summing both sides for $t = 1, \ldots, T$, we get a Telescoping sum,

$$\sum_{t=1}^{T} \alpha_t(g(\lambda^*) - g(\lambda_t)) \leq \sum_{t=1}^{T}[D_\Phi(\lambda^*||\lambda_t) - D_\Phi(\lambda^*||\lambda_{t+1})] + \sum_{t=1}^{T} \alpha_t^2 L^2 d\lambda_{max}$$

$$= D_\Phi(\lambda^*||\lambda_1) - D_\Phi(\lambda^*||\lambda_{T+1}) + \sum_{t=1}^{T} \alpha_t^2 L^2 d\lambda_{max}$$

$$\leq D_\Phi(\lambda^*||\lambda_1) + L^2 d\lambda_{max} \sum_{t=1}^{T} \alpha_t^2$$

$$\leq d\lambda_{max} \log(\lambda_{max}) + L^2 d\lambda_{max} \sum_{t=1}^{T} \alpha_t^2$$

Since $D_\Phi(\lambda^*||\lambda_{T+1}) \geq 0$ and $D_\Phi(\lambda^*||\lambda_1) \leq d\lambda_{max} \log(\lambda_{max})$ for bounded domain. Let's choose constant step size $\alpha_t = \alpha$ for simplicity,

$$\sum_{t=1}^{T}(g(\lambda^*) - g(\lambda_t)) \leq \frac{d\lambda_{max} \log(\lambda_{max})}{\alpha} + \alpha L^2 d\lambda_{max} T \tag{106}$$

Optimizing over $\alpha$ by setting $\frac{d}{d\alpha} = 0$ we get $\alpha^* = \sqrt{\frac{\log(\lambda_{max})}{L^2 T}}$. Substituting this above,

$$\sum_{t=1}^{T}(g(\lambda^*) - g(\lambda_t)) \leq 2d\lambda_{max} L \sqrt{T \log(\lambda_{max})} = O(d\lambda_{max} L \sqrt{T}) \tag{107}$$

Additionally, note that while we use acceleration it has no effect on the asymptotic bound derivation above so we used the standard mirror descent approach to derive the worst-case bound. □

### E.8.5. COUPLING ERROR: EXACT DISAGREEMENT LAW UNDER SHARED LOCAL SELECTION

We now provide rigorous bounds on the probability that two subproblems disagree on an overlapping position, and connect these bounds to the cumulative coupling error. The argument has two parts: a *primal* characterization of disagreement and a *dual* bound on coordination risk via Lagrangian mirror descent (established in Theorem E.13).

**Primal characterization**

1. We first identify the *local selection distribution* $q_i^{(t)}$ that each expert induces at position $i$ from the shared global state, and verify that all subproblems containing position $i$ compute the same $q_i^{(t)}$ — a direct consequence of our design choice to maintain shared global parameters across subproblems.

2. We then derive an *exact* expression for the disagreement probability in terms of $q_i^{(t)}$ (Lemma E.38), showing that disagreement vanishes whenever $q_i^{(t)}$ concentrates on a single value.

Connection to Theorem E.13: we apply the exact disagreement law (Lemma E.38) to express $\mathbb{E}[C_t]$ in terms of the non-concentration $(1 - q_{i,\max}^{(t)})$, and then connect this to $\xi_i^{(t)}$, yielding $\mathbb{E}[C_t] \leq c \cdot \|\xi_t\|_1$.

**Local selection distributions.** A key design choice in our framework and one that is essential for the coupling-error analysis — is that all subproblems share a single set of global parameters: value estimates $\hat{\mu}_{i,v}^{(t)}$, multiplicative weights $w_{i,v}^{(t)}$, visit counts $N_{i,v}(t)$, cumulative losses $L_{i,v}^{(t)}$, and Lagrangian dual variables $\lambda_i^{(t)}$. This shared-parameter design has a powerful theoretical consequence: the local selection rule at position $i$ and round $t$ induces the *same* conditional distribution $q_i^{(t)} = (q_{i,v}^{(t)})_{v \in \mathcal{A}_i}$ across all subproblems containing position $i$, regardless of which subproblem invokes it. This eliminates an entire class of coordination failures, namely, disagreements arising from inconsistent parameter estimates, and reduces

the coupling-error analysis to a clean probabilistic identity (Lemma E.38). Had subproblems maintained independent local parameters, the disagreement probability would depend on the divergence between separate estimators, requiring substantially more complex analysis and yielding weaker guarantees.

UCB expert (deterministic):

$$q_{i,v}^{(t)} = \mathbb{I}\left[v = \arg\max_{v' \in \mathcal{A}_i}\left(\hat{\mu}_{i,v'}^{(t)} + c\sqrt{\frac{\log t}{N_{i,v'}(t)}} - (k_i - 1)\lambda_i^{(t)}\right)\right], \tag{108}$$

i.e., a point mass on the UCB-maximizing value. Ties are broken by a shared deterministic rule (e.g., smallest index).

Hedge/EXP3 expert (randomized):

$$q_{i,v}^{(t)} = \frac{\exp\left(\log \tilde{w}_{i,v}^{(t)} / \tau_t\right)}{\sum_{v' \in \mathcal{A}_i} \exp\left(\log \tilde{w}_{i,v'}^{(t)} / \tau_t\right)}, \tag{109}$$

where $\tilde{w}_{i,v}^{(t)} = w_{i,v}^{(t)} \cdot \exp\left(0.1 \cdot \tilde{s}_{i,v}^{(t)}\right)$ are the shared multiplicative weights modified by Lagrangian scores $\tilde{s}_{i,v}^{(t)}$ (Eq. 46), and $\tau_t$ is the temperature parameter at round $t$.

FTRL expert (deterministic):

$$q_{i,v}^{(t)} = \mathbb{I}\left[v = \arg\max_{v' \in \mathcal{A}_i}\left(-L_{i,v'}^{(t)} + \gamma\sqrt{N_{i,v'}(t) + 1} - (k_i - 1)\lambda_i^{(t)}\right)\right], \tag{110}$$

where $L_{i,v}^{(t)} = \sum_{s=1}^{t-1} \tilde{\ell}_{i,v}^{(s)}$ is the cumulative importance-weighted loss. Since all three score functions are computed entirely from the shared global parameters listed above, the precondition of the following lemma: that all subproblems induce the same $q_i^{(t)}$, is guaranteed by construction!

**Lemma E.38** (Exact disagreement law under shared local selection). *Fix an overlapping position $i$ and round $t$, and let $m, l \in S_i$ be two subproblems containing $i$. Let $x_{m,i}^{(t)}$ and $x_{l,i}^{(t)}$ denote the* locally preferred values *induced at position $i$ by subproblems $m$ and $l$, respectively, prior to reconciliation into the global solution. Assume that, conditional on $\mathcal{F}_{t-1}$ and on the active expert at round $t$, all subproblems containing position $i$ use the same local selection rule.*

1. *If the local selection rule is **deterministic** (e.g., UCB or FTRL), then*

$$\Pr\left(x_{m,i}^{(t)} \neq x_{l,i}^{(t)} \mid \mathcal{F}_{t-1}\right) = 0.$$

2. *If the local selection rule is **randomized** (e.g., Hedge/EXP3), let*

$$q_i^{(t)} = \left(q_{i,v}^{(t)}\right)_{v \in \mathcal{A}_i}$$

*denote the common conditional distribution over feasible values at position $i$, and suppose $x_{m,i}^{(t)}$ and $x_{l,i}^{(t)}$ are drawn independently from $q_i^{(t)}$. Then*

$$\Pr\left(x_{m,i}^{(t)} \neq x_{l,i}^{(t)} \mid \mathcal{F}_{t-1}\right) = 1 - \sum_{v \in \mathcal{A}_i}\left(q_{i,v}^{(t)}\right)^2.$$

*In particular, if $q_{i,\max}^{(t)} := \max_{v \in \mathcal{A}_i} q_{i,v}^{(t)}$, then*

$$\Pr\left(x_{m,i}^{(t)} \neq x_{l,i}^{(t)} \mid \mathcal{F}_{t-1}\right) \leq 1 - \left(q_{i,\max}^{(t)}\right)^2 \leq 2\left(1 - q_{i,\max}^{(t)}\right). \tag{111}$$

*Hence, disagreement vanishes whenever the common local selection rule concentrates on a single value.*

*Proof.* For fixed values of $i, t, m, l$, we can break the proof into two cases -

**Deterministic case.** Conditional on $\mathcal{F}_{t-1}$ and the active expert, both subproblems compute the same local score vector at position $i$ from the same shared global state (value estimates $\hat{\mu}_{i,v}^{(t)}$, visit counts $N_{i,v}(t)$, cumulative losses $L_{i,v}^{(t)}$, and dual

variables $\lambda_i^{(t)}$) and apply the same deterministic decision rule, including the same tie-breaking rule. Hence they output the same locally preferred value, so

$$\Pr\left(x_{m,i}^{(t)} \neq x_{l,i}^{(t)} \mid \mathcal{F}_{t-1}\right) = 0.$$

**Randomized case.** Conditional on $\mathcal{F}_{t-1}$ and the active expert, both subproblems use the same distribution $q_i^{(t)}$ over $\mathcal{A}_i$, and the two draws are independent. Therefore:

$$
\begin{aligned}
\Pr\left(x_{m,i}^{(t)} = x_{l,i}^{(t)} \mid \mathcal{F}_{t-1}\right) &= \sum_{v \in \mathcal{A}_i} \Pr\left(x_{m,i}^{(t)} = v,\ x_{l,i}^{(t)} = v \mid \mathcal{F}_{t-1}\right) \\
&= \sum_{v \in \mathcal{A}_i} \Pr\left(x_{m,i}^{(t)} = v \mid \mathcal{F}_{t-1}\right) \Pr\left(x_{l,i}^{(t)} = v \mid \mathcal{F}_{t-1}\right) \\
&= \sum_{v \in \mathcal{A}_i} \left(q_{i,v}^{(t)}\right)^2.
\end{aligned}
$$

Taking the complement yields,

$$\Pr\left(x_{m,i}^{(t)} \neq x_{l,i}^{(t)} \mid \mathcal{F}_{t-1}\right) = 1 - \sum_{v \in \mathcal{A}_i} \left(q_{i,v}^{(t)}\right)^2.$$

Since $\sum_v (q_{i,v}^{(t)})^2 \geq (q_{i,\max}^{(t)})^2$, we can get the upper bound as,

$$\Pr\left(x_{m,i}^{(t)} \neq x_{l,i}^{(t)} \mid \mathcal{F}_{t-1}\right) \leq 1 - \left(q_{i,\max}^{(t)}\right)^2 = \left(1 - q_{i,\max}^{(t)}\right)\left(1 + q_{i,\max}^{(t)}\right) \leq 2\left(1 - q_{i,\max}^{(t)}\right),$$

where the last step uses $1 + q_{i,\max}^{(t)} \leq 2$. $\qquad\square$

The (exact) disagreement law expresses the latent disagreement probability in terms of the selection distribution $q_i^{(t)}$. To connect this to the soft violation $\xi_i^{(t)}$, which drives the Lagrangian dual update, we establish a quantitative bound relating the non-concentration $(1 - q_{i,\max}^{(t)})$ to the visit ratio $r_i^{(t)}$ defined below.

**Sub-optimality gap.** A positive *sub-optimality gap* $\Delta_i > 0$ at position $i$ means that one value $v_i^*$ is strictly better than all others: $\Delta_i := \mu_{i,v_i^*} - \max_{v \neq v_i^*} \mu_{i,v} > 0$. This is a standard condition in bandit theory (Auer et al., 2002a; Lattimore & Szepesvári, 2020) and ensures that the learning algorithm can eventually identify the optimal value. Positions where $\Delta_i = 0$ (all values equally good) do not require this condition—disagreement at such positions is costless, as shown below.

**Proposition E.39** (**Selection concentration after burn-in**). *Fix an overlapping position $i$ with $\Delta_i > 0$, and define*

$$r_i^{(t)} := \frac{N_{i,v_i^*}(t)}{\sum_{v'} N_{i,v'}(t)} = \frac{N_{i,v_i^*}(t)}{t}.$$

*There exists a burn-in time $\tau_i = O(n \log T / \Delta_i^2)$ such that:*

*(i) For deterministic experts (UCB and FTRL), $\forall\ t \geq \tau_i$,*

$$q_{i,v_i^*}^{(t)} = q_{i,\max}^{(t)} = 1,$$

*and hence*

$$1 - q_{i,\max}^{(t)} \leq 2\left(1 - r_i^{(t)}\right) + 2\sqrt{\frac{\log t}{t}}.$$

*(ii) For the randomized expert (EXP3), assume moreover that $\forall\ t \geq \tau_i$:*

$$q_{i,v_i^*}^{(t)} = q_{i,\max}^{(t)} \qquad and \qquad q_{i,\max}^{(t)} \geq q_{i,\max}^{(t-1)}.$$

*Then for all $t \geq 2\tau_i$, with probability at least $1 - 2/t^2$,*

$$1 - q_{i,\max}^{(t)} \leq 2\left(1 - r_i^{(t)}\right) + 4\sqrt{\frac{\log t}{t}}. \tag{112}$$

*Proof.* We will treat deterministic and randomized experts separately in the below steps.

**Deterministic experts: UCB and FTRL.** For UCB, the score of value $v$ at position $i$ is $S_{i,v}^{(t)} = \hat{\mu}_{i,v}^{(t)} + c\sqrt{\log t/N_{i,v}(t)} - (k_i - 1)\lambda_i^{(t)}$. The Lagrangian penalty is value-independent, so the argmax depends only on $\hat{\mu}_{i,v}^{(t)} + c\sqrt{\log t/N_{i,v}(t)}$.

By the Hoeffding inequality, $|\hat{\mu}_{i,v}^{(t)} - \mu_{i,v}| \leq c\sqrt{\log t/N_{i,v}(t)}$ holds simultaneously for all $v$ with probability at least $1 - 2n/t^{2c^2}$. Under this event, for any suboptimal value $v \neq v_i^*$,

- *Lower bound on $S_{i,v_i^*}^{(t)}$:* Since $\hat{\mu}_{i,v_i^*}^{(t)} \geq \mu_{i,v_i^*} - c\sqrt{\log t/N_{i,v_i^*}(t)}$, the UCB bonus cancels the estimation error: $S_{i,v_i^*}^{(t)} \geq \mu_{i,v_i^*} - (k_i - 1)\lambda_i^{(t)}$.

- *Upper bound on $S_{i,v}^{(t)}$:* Since $\hat{\mu}_{i,v}^{(t)} \leq \mu_{i,v} + c\sqrt{\log t/N_{i,v}(t)}$, the UCB bonus adds to the estimation error: $S_{i,v}^{(t)} \leq \mu_{i,v} + 2c\sqrt{\log t/N_{i,v}(t)} - (k_i - 1)\lambda_i^{(t)}$.

Subtracting (the Lagrangian penalties cancel), we get,

$$S_{i,v_i^*}^{(t)} - S_{i,v}^{(t)} \geq \Delta_i - 2c\sqrt{\frac{\log t}{N_{i,v}(t)}}.$$

For this gap to be positive, we need the estimation error to be less than $\Delta_i/2$, i.e., $N_{i,v}(t) \geq 16c^2 \log t/\Delta_i^2$. By the optimism principle of UCB, any arm $v$ with $N_{i,v}(t) < 16c^2 \log t/\Delta_i^2$ has a large exploration bonus $c\sqrt{\log t/N_{i,v}(t)} > \Delta_i/4$, making its UCB score competitive; UCB therefore continues selecting it until $N_{i,v}(t)$ reaches the threshold. Since there are $n$ values and each requires at most $\lceil 16c^2 \log T/\Delta_i^2 \rceil$ pulls to settle, all values have been pulled sufficiently by round

$$\tau_i = n \cdot \left\lceil \frac{16c^2 \log T}{\Delta_i^2} \right\rceil = O\left(\frac{n \log T}{\Delta_i^2}\right).$$

For all $t \geq \tau_i$:

$$S_{i,v_i^*}^{(t)} - S_{i,v}^{(t)} \geq \Delta_i - 2c \cdot \frac{\Delta_i}{4c} = \frac{\Delta_i}{2} > 0.$$

Therefore $v_i^*$ is the unique maximizer and $q_{i,v_i^*}^{(t)} = q_{i,\max}^{(t)} = 1$, so $1 - q_{i,\max}^{(t)} = 0$ and Eq. (112) holds trivially.

For FTRL, the score is $S_{i,v}^{(t)} = -L_{i,v}^{(t)} + \eta^{-1}\sqrt{N_{i,v}(t) + 1} - (k_i - 1)\lambda_i^{(t)}$. The cumulative importance-weighted loss for $v_i^*$ is lower than for any suboptimal value: $\mathbb{E}[L_{i,v}^{(t)} - L_{i,v_i^*}^{(t)}] \geq \Delta_i \cdot t$, which grows linearly. The regularizer difference $\eta^{-1}|\sqrt{N_{i,v_i^*}(t) + 1} - \sqrt{N_{i,v}(t) + 1}|$ grows as $O(\sqrt{t})$. After $\tau_i = O(n \log T/\Delta_i^2)$ rounds, the linear loss gap dominates the sublinear regularizer, so $v_i^*$ is the unique maximizer: $q_{i,\max}^{(t)} = 1$ and Eq. (112) holds trivially as well.

**Randomized expert: EXP3.** Assume now that for all $t \geq \tau_i$, $q_{i,v_i^*}^{(t)} = q_{i,\max}^{(t)}$ and that $q_{i,\max}^{(t)}$ is non decreasing in $t$. Then the sequence $(1 - q_{i,\max}^{(s)})_{s \geq \tau_i}$ is non-increasing, so for every $t \geq 2\tau_i$:

$$1 - q_{i,\max}^{(t)} \leq \frac{1}{t - \tau_i + 1} \sum_{s=\tau_i}^{t} \left(1 - q_{i,\max}^{(s)}\right)$$

$$\leq \frac{1}{t - \tau_i + 1} \sum_{s=\tau_i}^{t} \left(1 - q_{i,v_i^*}^{(s)}\right)$$

$$\leq \frac{t}{t - \tau_i + 1} \cdot \frac{1}{t} \sum_{s=1}^{t} \left(1 - q_{i,v_i^*}^{(s)}\right). \tag{113}$$

Since $t \geq 2\tau_i$, we have $t/(t - \tau_i + 1) \leq 2$.

Now let's define the martingale,

$$M_t := N_{i,v_i^*}(t) - \sum_{s=1}^{t} q_{i,v_i^*}^{(s)}.$$

Its increments satisfy $|M_s - M_{s-1}| = |\mathbb{I}[X_i(s) = v_i^*] - q_{i,v_i^*}^{(s)}| \leq 1$, and $\mathbb{E}[M_s - M_{s-1} \mid \mathcal{F}_{s-1}] = 0$ by definition of $q_{i,v_i^*}^{(s)}$. Thus, the Azuma– Hoeffding inequality gives, for any $a > 0$:

$$\Pr[|M_t| > a] \leq 2\exp\left(-\frac{a^2}{2t}\right).$$

Setting $a = 2\sqrt{t\log t}$:

$$\Pr\left[|M_t| > 2\sqrt{t\log t}\right] \leq 2\exp(-2\log t) = \frac{2}{t^2}.$$

Dividing by $t$ and rewriting in terms of $r_i^{(t)} = N_{i,v_i^*}(t)/t$: with probability at least $1 - 2/t^2$, we get,

$$\left|\frac{1}{t}\sum_{s=1}^{t}(1 - q_{i,v_i^*}^{(s)}) - (1 - r_i^{(t)})\right| \leq 2\sqrt{\frac{\log t}{t}}.$$

Simplifying,

$$\frac{1}{t}\sum_{s=1}^{t}(1 - q_{i,v_i^*}^{(s)}) \leq (1 - r_i^{(t)}) + 2\sqrt{\frac{\log t}{t}}.$$

Substituting into Eq. (113) yields

$$1 - q_{i,\max}^{(t)} \leq 2(1 - r_i^{(t)}) + 4\sqrt{\frac{\log t}{t}},$$

which is exactly Eq. (112). $\qquad\square$

*Remark* E.40 (Monotonicity condition in part (ii)). The monotonicity condition $q_{i,\max}^{(t)} \geq q_{i,\max}^{(t-1)}$ in part (ii) is a mild regularity condition that holds whenever the multiplicative weight update concentrates on the optimal value at a rate faster than the stochastic fluctuations in the importance-weighted losses. Under the gap condition $\Delta_i > 0$, the expected log-weight ratio $\mathbb{E}[\log(w_{v_i^*}^{(t)}/w_v^{(t)})] = \Omega(\eta\Delta_i t)$ grows linearly, while single-round fluctuations are $O(1)$, so monotonicity holds with high probability for $t \geq \tau_i$. [refrence]

**Proof-level vs. implementation-level visit ratio.** The visit ratio $r_i^{(t)} = N_{i,v_i^*}(t)/t$ used in Proposition E.39 and in the bridge lemma below tracks the *optimal value* $v_i^*$, which is the natural quantity for the theoretical analysis. The soft violation implemented in the algorithm (Eq. 44) instead uses the visit ratio of the *currently assigned value* $x_i^{(t)}$:

$$\xi_i^{(t)} = (k_i - 1)\,\sigma_i^2(t)\left(1 - \frac{N_{i,x_i^{(t)}}(t)}{t}\right).$$

After burn-in, the expert assigns $v_i^*$ at most rounds (by Proposition E.39), so $x_i^{(t)} = v_i^*$ with high probability and the two visit ratios coincide: $(1 - r_i^{(t)}) = (1 - N_{i,x_i^{(t)}}(t)/t)$. In the bridge lemma below, $\xi_i^{(t)}$ should therefore be read as $(k_i - 1)\,\sigma_i^2(t)\,(1 - r_i^{(t)})$, the proof-level quantity.

**Lemma E.41** (**Disagreement controlled by soft violation**). *For each overlapping position $i$ with $k_i > 1$ and $\Delta_i > 0$, and for all $t \geq 2\tau_i$:*

$$\mathbb{E}\left[2(1 - q_{i,\max}^{(t)})\right] \leq \frac{c_0}{k_i - 1} \cdot \mathbb{E}[\xi_i^{(t)}] + \epsilon_i(t), \tag{114}$$

*where $c_0 = 32n/\Delta_{\min}^2$, $\xi_i^{(t)} = (k_i - 1)\,\sigma_i^2(t)\,(1 - r_i^{(t)})$ with $r_i^{(t)} = N_{i,v_i^*}(t)/t$, and $\epsilon_i(t)$ is a correction term satisfying $\sum_{t=1}^{T}\epsilon_i(t) = O(\sqrt{T\log T})$. For positions with $\Delta_i = 0$, any disagreement is costless: $|\rho(i, v_m) - \rho(i, v_l)| = 0$, so these positions contribute zero to the coupling error regardless of disagreement frequency.*

*Proof.* Fix a gapped overlapping position $i$ with $\Delta_i > 0$ and $t \geq 2\tau_i$. From Proposition E.39, with probability at least $1 - 2/t^2$ we can say,

$$2\big(1 - q_{i,\max}^{(t)}\big) \leq 4\big(1 - r_i^{(t)}\big) + 8\sqrt{\frac{\log t}{t}}. \tag{115}$$

Now, after $\tau_i$ rounds, Hoeffding concentration ensures that the value estimates are close to their true means. Specifically, since $\hat{\mu}_{i,v}^{(t)}$ is the empirical average of $N_{i,v}(t)$ independent observations from $[0,1]$, Hoeffding gives: $\Pr\left[|\hat{\mu}_{i,v}^{(t)} - \mu_{i,v}| > \epsilon\right] \leq 2\exp(-2N_{i,v}(t)\,\epsilon^2)$. Setting $\epsilon = \sqrt{\log t/(2N_{i,v}(t))}$ yields failure probability $2/t$, and a union bound over $n$ values gives: with probability $\geq 1 - 2n/t$,

$$|\hat{\mu}_{i,v}^{(t)} - \mu_{i,v}| \leq \sqrt{\frac{\log t}{2N_{i,v}(t)}} \qquad \forall\, v.$$

After burn-in ($t \geq \tau_i$), $N_{i,v}(t)$ is large enough that this error is less than $\Delta_i/4$ for all values. Under this event, the estimated gap between $v_i^*$ and any suboptimal $v$ satisfies,

$$\hat{\mu}_{i,v_i^*}^{(t)} - \hat{\mu}_{i,v}^{(t)} \geq (\mu_{i,v_i^*} - \mu_{i,v}) - 2\cdot\frac{\Delta_i}{4} \geq \Delta_i - \frac{\Delta_i}{2} = \frac{\Delta_i}{2}.$$

To convert this gap into a variance lower bound, note that the cross-value variance $\sigma_i^2(t) = \frac{1}{n}\sum_v (\hat{\mu}_{i,v}^{(t)} - \bar{\mu}_i^{(t)})^2$ includes the squared deviations of $\hat{\mu}_{i,v_i^*}$ and $\hat{\mu}_{i,v}$ from the mean $\bar{\mu}_i$. By Cauchy–Schwarz (or direct calculation: for any two numbers $a, b$ and any $\bar{\mu}$, $(a - \bar{\mu})^2 + (b - \bar{\mu})^2 \geq (a - b)^2/2$, since the sum of squared deviations is minimized at the midpoint), we get,

$$\sigma_i^2(t) \geq \frac{1}{n}\left[(\hat{\mu}_{i,v_i^*} - \bar{\mu}_i)^2 + (\hat{\mu}_{i,v} - \bar{\mu}_i)^2\right] \geq \frac{1}{n}\cdot\frac{(\Delta_i/2)^2}{2} = \frac{\Delta_i^2}{8n}. \tag{116}$$

From Eq. (115), multiplying and dividing the first term by $\sigma_i^2(t)$ (which is positive by above analysis),

$$\begin{aligned}
2\big(1 - q_{\max}^{(t)}\big) &\leq \frac{4}{\sigma_i^2(t)}\cdot\sigma_i^2(t)\,(1 - r_i^{(t)}) + 8\sqrt{\frac{\log t}{t}} \\
&\leq \frac{4}{\Delta_i^2/(8n)}\cdot\frac{\xi_i^{(t)}}{k_i - 1} + 8\sqrt{\frac{\log t}{t}} \\
&= \frac{32n}{\Delta_i^2}\cdot\frac{\xi_i^{(t)}}{k_i - 1} + 8\sqrt{\frac{\log t}{t}},
\end{aligned} \tag{117}$$

where the second line uses Eq. (116) and the definition $\xi_i^{(t)} = (k_i - 1)\,\sigma_i^2(t)\,(1 - r_i^{(t)})$. Taking expectations both sides (the $2/t^2$ failure probability of Proposition E.39 contributes at most $2/t^2$ per round via the trivial bound $2(1 - q_{\max}) \leq 2$),

$$\mathbb{E}\Big[2\big(1 - q_{\max}^{(t)}\big)\Big] \leq \frac{c_0}{k_i - 1}\cdot\mathbb{E}[\xi_i^{(t)}] + \underbrace{8\sqrt{\frac{\log t}{t}} + \frac{2}{t^2}}_{=:\,\epsilon_i(t)}, \tag{118}$$

with $c_0 = 32n/\Delta_{\min}^2$. The correction terms satisfy:

$$\sum_{t=1}^{T}\epsilon_i(t) = 8\sum_{t=1}^{T}\sqrt{\frac{\log t}{t}} + \sum_{t=1}^{T}\frac{2}{t^2} \leq 16\sqrt{T\log T} + \frac{\pi^2}{3} = O(\sqrt{T\log T}). \tag{119}$$

where $\sum_{t=1}^{T}\sqrt{\log t/t} \leq 2\sqrt{T\log T}$ by integral comparison, and $\sum_{t=1}^{T} 2/t^2 \leq 2\sum_{t=1}^{\infty} 1/t^2 = 2\cdot\pi^2/6 = \pi^2/3 \approx 3.29$ by the Basel series (Euler, 1734; this bound is used identically in the UCB1 regret proof, Auer et al., 2002a, Theorem 1). Both terms are $O(\sqrt{T})$ up to logarithmic factors. For $t < 2\tau_i$ (burn-in), we use the trivial bound $2(1 - q_{\max}^{(t)}) \leq 2$. These rounds contribute at most $2\tau_i = O(n\log T/\Delta_{\min}^2)$ to the cumulative disagreement per position, which is $O(\log T)$ and absorbed into $O(\sqrt{T})$. $\qquad\square$

*Remark* E.42 (Latent vs. realized assignments; proof-level vs. algorithmic quantities). Lemmas E.38 and E.41 concern *latent locally preferred values* $x_{m,i}^{(t)}$ induced by subproblems before reconciliation into the global solution, and use the proof-level visit ratio $r_i^{(t)} = N_{i,v_i^*}(t)/t$, which tracks the optimal value $v_i^*$. In the implemented algorithm, optimization proceeds through a single global solution $x^{(t)}$, so each position $i$ has a unique realized value $x_i^{(t)}$ at round $t$. The *observable algorithmic surrogate* for latent coordination risk is the soft violation:

$$\xi_i^{(t)} = (k_i - 1)\operatorname{Var}(\hat{V}_{i,\cdot}^{(t)})\left(1 - \frac{N_{i,x_i^{(t)}}(t)}{t}\right),$$

which uses the visit ratio of the *currently assigned* value $x_i^{(t)}$ rather than $v_i^*$. After burn-in, the expert assigns $v_i^*$ at most rounds (Proposition E.39), so $x_i^{(t)} = v_i^*$ with high probability and the algorithmic surrogate coincides with the proof-level quantity. The Lagrangian dual update (Algorithm 4) penalizes this surrogate via mirror descent, providing a coordination signal that is absent in uncoordinated decomposition.

