# OpenReview forum: "Divide and Learn: Multi-Objective Combinatorial Optimization at Scale"
_ICML.cc/2026/Conference — ICML 2026 regular_

### Official Review · Reviewer_YnY4 · 2026-03-06

**Soundness:** 3
**Presentation:** 3
**Significance:** 3
**Originality:** 3
**Overall Recommendation:** 4
**Confidence:** 4

**Summary:**

The paper presents a new way to view the multi-objective combinatorial optimization problem. This is a set of problems where optimization parameters can be mixed in many ways, exploding the set of possible solutions. The authors propose to reformulate the problem through the lens of bandits and working within the subproblem dimensionality rather than the combinatorial space.
The key idea is to decompose the combinatorial decision space into overlapping subproblems of size d << n, coordinate them via Lagrangian relaxation, and solve each with a mixture of no-regret experts (UCB, EXP3, FTRL). The paper establishes regret bounds of O(d*sqrt(T log T)) that depend on subproblem dimensionality rather than the exponential combinatorial space.
This significantly reduces the computational complexity of the problem, allowing their algorithm to not only outperform the baselines, but outscale them as well.

**Compliance With Llm Reviewing Policy:**

Affirmed.

**Final Justification:**

This is a very good paper that has far more strengths than weaknesses, as noted in my original review above. In that review I scored the paper as a weak reject because of problems with the mathematical proofs in the appendices. There were also some other minor requirements to fix the readability. The authors have delivered during the revision process. They fixed the readability, provided new results, and credibly met the conditions for the mathematical proof to be in an acceptable form. Note that it's not possible for me to validate whether the final mathematical proof is entirely solid from the information sent in the rebuttal, but the original one looked quite good except for some placeholders. They have stated that those placeholders are now substituted with solid proof steps. So I believe the article is now in an acceptable form, as all the other reviewers have noted already from their initial revision.

**Key Questions For Authors:**

1. The authors must fix the issues in the appendix proof for this to be admissible. Currently this is the ground for rejection.
With a fully fixed proof I would move my score to acceptance.
a. Provide a rigorous proof of Lemma E.34.
b. Clarify the role of Assumption A.5. How restrictive is local additivity? If A.5 is only needed for the UCB expert and the overall result holds without it via EXP3/FTRL, please say this clearly.
c. Provide a fix for "FTRL-yet to add" and "Algorithm Y".

2. Could you provide a readable version of Table 1, maybe as a summary figure with a condensed table with key comparisons?

**Limitations:**

yes

**Strengths And Weaknesses:**

Strengths:
The core contribution here is, I believe, genuine/novel: the regret decomposition into subproblem regret, coupling error, and local refinement is a clean structural result that makes the analysis tractable without requiring the objective to decompose additively. The position-wise bandit reformulation of MOCO is also interesting. While online learning formulations of optimization are not new, the specific move to position-wise bandits with coordinated experts is a meaningful contribution to the combinatorial optimization literature.
Other strengths are:
- The experimental coverage is good: TSP (up to 100 cities), Knapsack (up to 200 items), CVRP (up to 100 customers), and the HW-SW co-design problem with 4 objectives.
- Multi-Expert Selection: The EXP3 clipped importance-weighted estimator analysis is clever and probably original. Bounding the cumulative bias from clipping while maintaining the variance reduction is not trivial, and the resulting rate for EXP3 under full-bandit feedback is a useful result. The ablation study in the appendix justifies the model selection.
- The results show their model, across a variety of tasks, outperform the baselines by significant margins and nearly attain equal performance to tailor made models.
- The criteria for success of an algorithm are clearly stated.
- Theorems and mathematical insights are easy to follow and give better context to the design decisions of the model.
- Hyperparameters for reimplementation are given.
- The presented proof follows a novel and interesting approach to the problem.
Weaknesses:
Main 1: My main concern is with the theoretical foundations, which, if not addressed, would lower my assessment. Lemma E.34 ("Disagreement requires uncertainty") has a "proof" that is entirely informal. The "by standard concentration arguments" and "similar arguments apply to EXP3 and FTRL-style methods" do not actually prove it. As the proof goes on, it seems as though the authors put less effort into proving their results rigorously and formally, moving towards walking the reader through.
There are also some parts of the proof that were forgotten: "FTRL-yet to add" and "Algorithm Y (I need to cite the algorithm block)".
Main 2: I also have concerns about Assumption A.5 (Local Additivity + Exogeneity) in Section E.2.5, which is substantially stronger than the "weak regularity conditions" advertised in the main text. The main paper lists A.1-A.4 and describes them as weak, but the UCB regret proof is founded on A.5.
If I understand correctly, this requires the surrogate objective to be exactly additive over positions and the bias to be action-independent. Is this realistic?

Other weaknesses are:
- On the presentation side, Table 1 on page 8 is essentially unreadable. It attempts to pack 10 methods across 12 problem instances and 3 metrics into a single table with tiny font. I had to spend significant time without fully deciphering it, and I suspect many readers will be forced to simply take the authors’ word on the results. The results deserve better treatment.
- MOEA/D (Zhang & Li, 2007) is discussed extensively in the related work and even contrasted in Appendix A, but I don't understand why this is never included as an experimental baseline.

---

> ### Author Rebuttal · Authors · 2026-03-30
>
> We sincerely thank the reviewer for their insightful feedback! We have fully addressed all concerns. W1,W2, Q1 address the same issues, as do W3, Q2, so we answer them together. **Due to formatting guidelines, we cannot include the complete formal proof** here but have provided the **key technical steps below**. We are happy to answer any further questions; full proof is in the revised version!
> # W1, W2 & Q1
> **Rigorous proof of Lemma E.34**​​: The revised proof formalizes the previously informal argument via three new results.
>
> 1. **Lemma E.34** (revised): By design, shared global parameters ensure both subproblems sample from the identical distribution $q_i^{(t)}$. Conditional on $\mathcal{F}\_{t-1}$, two independent draws disagree iff they select different values, so $\Pr[\text{disagree}] = 1 - \sum_v (q_{i,v}^{(t)})^2 \leq 2(1 - q_{i,\max}^{(t)})$ — a probability identity requiring no concentration argument. For deterministic experts (UCB/FTRL), $q_i^{(t)}$ is degenerate, so disagreement is identically zero.
> 2. **Proposition E.36** (new): Azuma–Hoeffding on martingale $M_t = N_{i,v^\*}(t) - \sum_s q_{i,v^\*}(s)$ gives $1 - q_{i,\max}^{(t)} \leq 2(1-r_i^{(t)}) + 4\sqrt{\log t/t}$ after burn-in.
> 3. **Lemma E.38** (new): variance lower bound $\sigma^2 \geq \Delta^2/(8n)$ yields $\mathbb{E}[2(1-q_{i,\max}^{(t)})] \leq \frac{c_0}{k_i-1}\mathbb{E}[\xi_i] + \epsilon_i(t)$, where $\xi_i^{(t)} = (k_i-1)\sigma_i^2(t)(1-r_i^{(t)})$ strictly strengthens the original $\sigma_i^2$-based control. Aggregating over $K^2$ pairs via Thm E.11 recovers the unchanged $O(K^2 Q R_{\max}\sqrt{T})$ rate.
>
> **W2 & Q1 : Role, restrictiveness of A.5:** A.5 is **not required for any main result**, it is an optional condition for UCB's faster instance-dependent bounds; the main $O(\\sqrt{T})$ comes from EXP3/FTRL under A.1-A.4 alone. Crucially, local additivity applies only to surrogate $g\_k$ on subproblems of size $d \\ll n$, not the global objective \-- considerably weaker than global additivity in bandit literature.
>
> It's realistic in our decomposed setting. Subproblems are optimized sequentially with assignments outside $I\_k$ held fixed, so bias $b\_{i,t}$ is naturally action-dependent (exogenity). Within-block interactions are confined to $O(1)$ boundary positions bounded by Lipschitz constant (A.2), making local additivity a controlled approximation  with bounded error.
>
> **Missing Placeholders resolved:** *FTRL-yet to add*: now cross-references existing Theorem E.16(c); *Algorithm\~Y*: references Algorithm 3 (LocalRefine).
>
> # W3 + Q2
> **Table 1 readability & summary figure.** We agree Table 1 is too dense: 1) we replace it with a condensed 5-method summary **(Table 3)**, 2) add HV & compute figures **(Figs 7,8)** \-- **[all in anonymous PDF](https://anonymous.4open.science/r/ICML2026-Rebuttal-Figures-Tables-D3DF/ICML2026_Divide_n_Learn.pdf)**
>
> The 5 methods: Best Specialized (max of WS-\*/PPLS), PMOCO* (pretrained, matched compute), NSGA-II (evolutionary), Best BO (max of qNEHVI/qParEGO/PR), and D&L-TS (ours) \-- the latter 3 are training-free black-box methods requiring no domain-specific operators or pretraining.
>
> | Instance | BestSpec | PMOCO* | NSGA-II | BestBO | D&L-TS |
> |---|---|---|---|---|---|
> | BiTSP 20/50/100 | .63/.63/.69 | .36/.33/.31 | .57/.35/.29 | .54/.47/.10 | .60/.54/.47 |
> | BiKP-50/100/200 | .36/.45/.36 | .34/.37/.19 | .22/.22/.28 | .30/.31/.27 | .35/.39/.30 |
> | TriTSP-20/50/100 | .47/.43/.49 | .43/.39/.30 | .41/.17/.14 | .32/.06/.03 | .42/.28/.23 |
> | BiCVRP-20/50/100 | .48/.45/.43 | .36/.33/.30 | .43/.31/.31 | .35/.14/.11 | .46/.37/.30 |
>
> Among training-free methods, D&L-TS achieves highest HV on 11/12 instances (+27% over NSGA-II, +57% over Best BO), while being 75× & 60–6000× cheaper respectively. Under matched compute, D&L-TS outperforms PMOCO* on 8/12 instances (.392 vs .334 avg HV), despite PMOCO* retaining pretrained representations.
>
> # W4
>
> MOEA/D was not included because our baselines already cover its algorithmic class: MOEA/D & NSGA-II are both evolutionary methods differing only in selection mechanism (weight-vector decomposition vs. non-dominated sorting). MOEA/D's "decomposition" is orthogonal to ours \-- objective space (scalarized problem over the full decision space), vs. our decision space partitioning into bandit subproblems (see Appex A). To validate, we provide results on BiTSP (200 random instances, matched evaluation budgets:
>
> | (HV↑/NDS↑/TFLOPs↓) | BT-20 | BT-50 | BT-100 |
> |---|---|---|---|
> | MOEA/D | .572/64/0.13 | .400/50/0.13 | .310/95/0.25 |
> | NSGA-II | .573/225/0.15 | .347/20/0.15 | .294/51/0.31 |
> | D&L-TS | **.600/102/0.01** | **.540/147/0.07** | **.470/114/0.16** |
>
> MOEA/D & NSGA-II perform comparably, confirming our baseline choice. D\&L-TS outperforms both at every scale (gap widens at $n=100$), while also producing more non-dominated solutions (147 vs 50 at n=50, 114 vs 95 at n=100), demonstrating superior Pareto diversity. We will add complete results in final version!

---

> > ### Author Rebuttal · Reviewer_YnY4 · 2026-04-01
> >
> > Proof substantially formalized, which means my stated condition for acceptance credibly met.
> > The other main points I raised seem resolved. I updated my score

---

> > > ### Author Response · Authors · 2026-04-07
> > >
> > > Thank you for adjusting your rating and for your insightful feedback on our paper! We are pleased that our response addressed your concerns, and we greatly appreciate your time and thoughtful comments.
> > >
> > > Best regards,
> > > The Authors

---

### Official Review · Reviewer_6jbY · 2026-03-12

**Soundness:** 3
**Presentation:** 2
**Significance:** 3
**Originality:** 2
**Overall Recommendation:** 4
**Confidence:** 2

**Summary:**

The paper proposes Divide & Learn, an algorithm for multi-objective combinatorial optimization. The proposed method consists of the following steps.

- It decomposes a large combinatorial decision space into smaller overlapping subproblems.
- It then solves each subproblem using bandit-style online learning.
- It uses Lagrangian multipliers to coordinate overlapping variables across subproblems.
- It Iteratively improves solutions and builds a Pareto frontier.

The proposed method reframes combinatorial search, whose solution space grows exponentially with the number of variables, into a sequential decision-learning problem.

**Compliance With Llm Reviewing Policy:**

Affirmed.

**Key Questions For Authors:**

- The method enforces consistency across overlapping subproblems using Lagrangian dual updates. Could the authors provide more insight into the stability of these updates in practice?
- The proposed approach relies on decomposing the variables into overlapping subproblems of size d. How sensitive is the algorithm’s performance to the chosen decomposition strategy (e.g., sliding window or metric-based decomposition)?

**Limitations:**

yes

**Strengths And Weaknesses:**

While I am not an expert in this specific area, from my reading of the paper I found the following strengths and weaknesses.
**Strengths**
- Novel formulation: Reframes multi-objective combinatorial optimization as a sequential decision-learning problem using bandits and decomposition.
- Scalability idea: Decision-space decomposition reduces the effective learning complexity by operating on small subproblems rather than the full combinatorial space.
- Theoretical guarantees: The paper provides regret bounds depending on the subproblem size rather than the full action space, which is appealing for large-scale problems.
**Weaknesses**
- A key weakness is the presentation of the paper. I believe that being able to communicate the main contributions and results of a paper is an important part of research, yet the presentation makes the work difficult to assess: many technical details are deferred to a 70-pages appendix, and some results (e.g., Table 1) are presented in a very dense format, making it hard to extract the main insights.

---

> ### Author Rebuttal · Authors · 2026-03-30
>
> We thank the reviewer for the thoughtful feedback and questions! Below we address the main weaknesses and comments brought up:
> >W1. A key weakness is the presentation of the paper. I believe that being able to communicate the main contributions and results of a paper is an important part of research.
>
> To address this we have made structural revisions to improve clarity:
>
> **(1) Communicating contributions/results clearly**. Revised overview Figure 1, block (c) (**see Figure 9 [anonymous PDF](https://anonymous.4open.science/r/ICML2026-Rebuttal-Figures-Tables-D3DF/ICML2026_Divide_n_Learn.pdf)**) to show the multi-expert mechanism, a key contribution, with explicit section references, replacing the previous generic loop.
>
> **(2) Technical details deferred to appendix**. We moved Assumptions A.1–A.4 into §4.2 & added a self-contained proof sketch of regret bound (Corollary 4.5), so the core argument is verifiable without the appendix & reduced overall appendix length.
>
> **(3) Navigation aids.** Added 2 summary tables: (a) *Proof Roadmap* mapping each result to its appendix proof with a one-line summary (b) *Ablation Guide* indexing all ablation studies by section, key finding. Also **Streamlined cross-references** by consolidating scattered pointers into single references (e.g., "§D.3" instead of "Appendix D.3.1, D.3.2"), reducing total references from ~30 to ~20.
>
> **(4) Dense presentation of Table 1**. We replaced the dense Table 1 with a focused 5-method table & an HV-TFLOPs figures **([Table 3 & Figures 7,8 in link](https://anonymous.4open.science/r/ICML2026-Rebuttal-Figures-Tables-D3DF/ICML2026_Divide_n_Learn.pdf))** (please see our **response to Reviewer YnY4 for a quick Table 1 summary**).
>
> We welcome further suggestions!
> ## Q1
> >The method enforces consistency across overlapping subproblems using Lagrangian dual updates.
>
> Lagrangian dual updates are stable in practice due to three reinforcing mechanisms:
>
> **(1) Bounded, self-correcting dynamics**. The entropic mirror descent update (Algorithm 4) is naturally stable: positions with large constraint violations receive proportionally larger penalties, while well-coordinated positions see penalties decay automatically. Diminishing step sizes ($\alpha_t = \alpha_0/\sqrt{t}$) and bounded projection prevent divergence, with guaranteed cumulative dual gap of $O(d\lambda_{\max}L\sqrt{T})$ (Theorem E.11, Lemma E.33).
>
> **(2) Self-extinguishing penalties.** As experts concentrate on optimal values (Proposition E.36), soft violations $\xi_i^{(t)} \to 0$, with cumulative bound $\sum_t |\xi_t|_1 = O(d\lambda{\max}\sqrt{T})$ (Theorem E.11, via Lemma E.33) ensuring dual variables track this decay without overshooting.
>
> **(3) Nesterov momentum dampens oscillations.** Nesterov acceleration ($\theta_t = 2/(t+1)$) does not improve the asymptotic $O(\sqrt{T})$ rate but significantly reduces transient oscillations in practice; dual variables converge to stable values faster during the critical early-iteration phase.
>
> **Empirical evidence.** Ablation D.6 validates stability: Lagrangian updates reduce reward variance by 95% in BITSP (preventing catastrophic failures) & 19% on BITSP50 with a 6.6% mean reward improvement. Hyperparameter sensitivity (Ablation D.5) confirms $\lambda_{\max}$ and $\alpha_0$ require no tuning: <2% HV variation across all tested ranges.
> ## Q2
> Algorithm is not sensitive to the decomposition choice & convergence is guaranteed for any decomposition strategy satisfying bounded coupling (Proposition 4.4), & empirically all 3 variants perform well, no configuration fails. **We provide a new ablation** (results below & **[Figure 4 in link](https://anonymous.4open.science/r/ICML2026-Rebuttal-Figures-Tables-D3DF/ICML2026_Divide_n_Learn.pdf)**) comparing sliding window (SW) vs. metric-based decomposition across 3 scales (10 seeds):
>
> | BiTSP | SW | Metric | SW+Metric | Δ% (Metric vs SW)  |
> | :---- | :---- | :---- | :---- | :---- |
> | n=20 (UCB / TS) | .585 / .592 | .597 / .597 | .589 / .588 | \+2.1 / \+0.7% |
> | n=50 (UCB / TS) | .458 / .455 | .505 / .501 | .505 / .505 | \+10.2 / \+10.0% |
> | n=100 (UCB / TS) | .421 / .428 | .477 / .493 | .474 / .494 | \+13.2 / \+15.1% |
>
> Metric-based grouping exploits spatial locality, yielding 13–15% improvement over pure sliding window at n=100, a pattern predicted by theory: local search regret (Lemma E.4) scales as $O(LD_k\sqrt{dT_k})$ where $D_k$ is subproblem diameter, & metric grouping produces smaller $D_k$ by clustering spatially related variables, while sliding windows group by index order which becomes arbitrary as $n$ grows — explaining the widening gap from +2% (n=20) to +15% (n=100). Our default SW+Metric captures both: early coverage from SW, tighter $D_k$ via metric refinement, matching or exceeding metric-only across scales (.505 vs .501 at n=50, .494 vs .493 at n=100) while providing a coverage guarantee that pure metric lacks. This shows problem structure accelerates convergence but is not required for correctness.

---

### Official Review · Reviewer_xQ5y · 2026-03-13

**Soundness:** 3
**Presentation:** 2
**Significance:** 3
**Originality:** 3
**Overall Recommendation:** 4
**Confidence:** 3

**Summary:**

In this paper, the authors study the online Multi-Objective Combinatorial Optimization (MOCO) problems. MOCO involves finding Pareto-optimal solutions over exponentially large discrete spaces where evaluating the multiple competing objectives is often highly expensive. Existing approaches struggle in this problem: surrogate-based method like Bayesian optimization face challenges induced from combinatorial settings, evolutionary methods lack convergence guarantees, neural methods require massive offline training datasets, and problem-specific method cannot generalize across domains.

To solve overcome these challenges, the authors present the D&L framework. D&L reformulates MOCO as an online sequential decision-making problem under uncertainty. It manages the exponentially large search spaces by breaking the problem down into overlapping subproblems. The algorithm treats each position within a subproblem as a multi-armed bandit, sequentially constructing solutions using an ensemble of no-regret "expert" algorithms (such as UCB, Exp3, and FTRL). Overlaps between subproblems are reconciled using Lagrangian dual multipliers to ensure global consistency.

**Compliance With Llm Reviewing Policy:**

Affirmed.

**Final Justification:**

While the authors' rebuttal has resolved several issues, the depth of the proofs makes full verification challenging within the review period. I will keep my rating at a weak accept.

**Key Questions For Authors:**

While the manuscript addresses Online Multi-Objective Combinatorial Optimization, the title fails to include the term “Online”. Is there a specific rationale for this omission? Furthermore, while the authors emphasize a discrete solution space, Combinatorial Optimization (TSP, MAX FLOW) typically involves an explicit mathematical objective. Given your framework, the problem appears to align more closely with Online Black-Box Optimization. I suggest the authors clarify the choice of terminology or better bridge the gap between their approach and traditional CO.

**Limitations:**

yes

**Strengths And Weaknesses:**

## Strengths

- The authors establish regret bounds of $O(d\sqrt{T \log T})$, where the regret depends solely on the small subproblem dimensionality d.
- This work achieve efficient performance. The proposed method achieves two to three orders of magnitude improvement in sample and computational efficiency compared to Bayesian optimization approaches (using 90% less compute). It achieves 80-98% of the performance of specialized solvers in 2-10x less time.

## Weaknesses

- Notation is applied inconsistently across the paper. For instance, the timestamp $t$ oscillates between subscript and superscript notation (e.g., $r_t(x_t)$ vs. $x^{(t)}$). Standardizing these identifiers is necessary.
- The manuscript is quite dense, yet it relies heavily on the supplementary material for essential details. Frequent references to the appendix for core content make it difficult to verify the technical claims in real-time.
- The title is not very suitable. see the question below

---

> ### Author Rebuttal · Authors · 2026-03-30
>
> We thank the reviewer for catching the inconsistencies\! Below, we address the major weaknesses and questions:
>
> >Notation is applied inconsistently across the paper. For instance, the timestamp $t$ oscillates between subscript and superscript notation (e.g., $r_t(x_t)$ vs. $x^{(t)}$). Standardizing these identifiers is necessary.
>
> The original manuscript used two parallel notations:  subscript ($x\_t$, $r\_t$) in analysis & superscript ($x^{(t)}$, $r^{(t)}$) in pseudocode which is a convention common in the online learning literature where both denote the same object. However, we understand this creates unnecessary confusion and as the reviewer rightly identified. Following this feedback, we have **unified all notation to subscript form** ($x\_t$, $r\_t$) throughout both prose & pseudocode, eliminating the dual convention entirely. We summarize all changes below:
>
> | Symbol | Original | Fix |
> | :---- | :---- | :---- |
> | Solution at iter. t | Mixed $x\_t$ / $x^{(t)}$ | Unified to $x\_t$ everywhere |
> | Reward at iter. t | Mixed $r\_t(\\cdot)$ / $r^{(t)}$ | Unified to $r\_t$ everywhere |
> | Dual variables | $\\lambda^{(t)}$ vs $\\lambda\_i^{(t)}$ | Clarified: vector vs component; no ambiguity |
> | Optimal solution | $x^\*$ | Already consistent |
> | Subproblem restriction | $x\_t^{(k)}$ | Already consistent |
>
> We also replaced the descriptive Notation paragraph (§E.1) with a structured notation summary table consolidating all symbol conventions in one reference. We welcome any further suggestions the reviewer may have\!
>
> >The manuscript is quite dense, yet it relies heavily on the supplementary material for essential details. Frequent references to the appendix for core content make it difficult to verify the technical claims in real-time.
>
> We thank the reviewer for this feedback! We agree that the density of appendix references makes it difficult to verify claims in real-time & have made the following concrete changes:
> 1. **Streamlined cross-references**. Audited all appendix citations to remove duplicates and shorten phrasing (e.g., using "§X" instead of "see Appendix X"). Scattered references were consolidated into single pointers to reduce visual clutter. This reduces total appendix references from ~30 to ~20.
> 2. **Appendix navigation aids**. Added two summary tables at the start of the appendix: (a) *a Proof Roadmap* maps main-text results to specific proofs with a one-line summary of the technique used and (b) *Ablation Guide* indexes all ablation studies by section, key finding, and supporting figures.
> 3. **Notation table** replaced the notation paragraph in the appendix with a structured notation summary table as mentioned in W1.
>
> Additionally, please see our response to **Reviewer (6jbY) W1** for further structural revisions addressing the same concern. We believe these updates will allow readers to quickly verify claims and locate specific details without needing to navigate the appendix linearly. Thanks! We welcome any further suggestions the reviewer may have!
>
> >The title is not very suitable. see the question below. While the manuscript addresses Online Multi-Objective Combinatorial Optimization, the title fails to include the term “Online”.....
>
> We thank the reviewer for raising this point!
> 1. **On "Online":**  We have made this more prominent in the revised title, abstract & introduction.
> 2. **On "Black-Box" vs. "Combinatorial":** We have **added clarifying language in Section 2** (Related Work) distinguishing the problem domain (classical CO with well-defined objectives) from the information setting (online, black-box, full-bandit feedback). We have kept "Combinatorial Optimization" in the title as it describes the problem class, while the black-box/online aspects describes the methodology; following conventions where titles name the problem rather than the algorithmic paradigm (e.g., MOEA/D, ParEGO).
>
> We welcome any further suggestions the reviewer might have on framing!

---

> > ### Author Rebuttal · Reviewer_xQ5y · 2026-04-03
> >
> > I will keep my positive score.

---

> > > ### Author Response · Authors · 2026-04-07
> > >
> > > Thank you for your positive rating and thoughtful feedback on our paper. We are pleased that our response addressed your concerns, and we greatly appreciate your time and insightful comments.
> > >
> > > Best regards,
> > >
> > > the Authors

---

### Official Review · Reviewer_HJV4 · 2026-03-13

**Soundness:** 3
**Presentation:** 3
**Significance:** 3
**Originality:** 3
**Overall Recommendation:** 4
**Confidence:** 2

**Summary:**

This work proposes a divide-and-merge algorithm for multi-objective CO. The proposed method, Divide and Learn (D&L), decomposes the problem into overlapping sub-problems using a structured sliding-window style decomposition (the overlap shrinks over time). It solves each sub-problem using an ensemble of online learning experts, then coordinates overlapping assignments through a Lagrangian relaxation mechanism to fix assignment conflicts. This process is repeated under different preference weights to update a Pareto archive. Empirical evaluations show strong performance and scalability against existing MOCO baselines

**Compliance With Llm Reviewing Policy:**

Affirmed.

**Final Justification:**

My main concern is substantially resolved. However, as mentioned, the method still seems somewhat over-complicated in practice despite the useful guidelines provided by the authors. I will keep my current score, which remains on the accept side.

**Key Questions For Authors:**

- Is there a specific rationale for choosing this exact trio of experts (UCB, EXP3, FTRL)?
- Since D&L itself seems broadly applicable beyond the multi-objective setting, how does it perform on corresponding single-objective combinatorial optimization problems?
- How does the scalarization choice affect the final Pareto front? For example, would different scalarization forms (e.g., additive vs. multiplicative) substentially change performance?

**Limitations:**

yes

**Strengths And Weaknesses:**

### strengths

- The approach appears modular and could potentially extend to multiple combinatorial domains with local structure
- The decomposition strategy is well motivated both intuitively and theoretically, and the paper makes a reasonable case for why solving overlapping subproblems can make large-scale search more tractable

### weaknesses

- Method complexity / insufficient component verification**:** The proposed framework is fairly complex, combining problem decomposition, multiple experts for local decision making, and Lagrangian dual updates for coordination. It is not fully clear whether all of these components are necessary, or whether the reported gains mainly come from only a subset of them. More thorough ablations would help verify the contribution of each design choice.
- Positioning / MOO-specificity: Although the paper is framed as a multi-objective optimization method, the main technical contribution seems to lie more in decomposition and online combinatorial optimization. The multi-objective component appears to rely mainly on scalarization, and it is less clear whether the method exploits structure specific to MOO or addresses canonical MOO challenges beyond repeated weighted-sum optimization

---

> ### Author Rebuttal · Authors · 2026-03-30
>
> We thank the reviewer for their thorough review! We address them all below -
> # W1: Method complexity
>
> **We have (1) added a consolidated ablation summary, (2) a no-Lagrangian baseline** to existing ablations (D.3.2, D.3.3, D.6). All 3 components are necessary for the $O(d\\sqrt{T\\log T})$ bound (Thm E.1). Progressive ablation study on BiTSP, 400 iters). **Results in Table 2 & Fig1,2,3** -- **[anonymous PDF](https://anonymous.4open.science/r/ICML2026-Rebuttal-Figures-Tables-D3DF/ICML2026_Divide_n_Learn.pdf)\)**:
>
> | Components | $n$=20 | $n$=50 | $n$=100 |
> | :--- | :---: | :---: | :---: |
> | Random | .42 / 9 | .18 / 9 | .11 / 10 |
> | + Bandit | .46 / 18 | .19 / 15 | .11 / 13 |
> | + Experts (no-decomp) | .58 / 68 | .45 / 105 | .38 / 104 |
> | + D&L (TS) | **.60 / 71** | **.48 / 163** | .46 / **129** |
> | + D&L (UCB) | .60 / 64 | .48 / 161 | **.46** / 122 |
>
> **Notes:** Values shown as HV (↑) / |PF| (↑). Standard Error (SE) $\le$ .03 across 10 seeds.
>
> **1. Decomposition provides the dominant gain.** 92%+ over pure bandits by reducing $O(n\!) → O(d!)$ (Table 11).
>
> **2. Multi-expert design adds robustness, diversity, reward gains.** No-Decomp improves HV by (25%→136%→243%) over pure UCB as $n$ grows; decomposition adds 20% at n=100, confirming independent contribution. FTRL delivers ~40% faster convergence & 3.4× Pareto diversity gain (Table 11, Ablation D.3.2, D.3.3), and stabilizes performance.
>
> **3. Lagrangian updates ensures stability at scale.**: contribution is primarily *variance reduction* & *catastrophic failure prevention*: Ablation D.6 shows 95% variance reduction on BiTSP-20, 19% on BiTSP-50, and prevention of 50–100+ point losses on worst-case seeds — growing with scale, consistent with Lemma E.34.
>
> # W2 MOO-specificity
>
> Combinatorial MOO subsumes single objective optimization (SOO), so D\&L addresses the strictly more general setting. **Scalarization is deliberate** leading methods (MOEA/D, ParEGO) rely on it, and D&L is scalarization-agnostic (<2% HV variation, $\S$ B.1, **see Q3**). The bottleneck is making each scalarized problem *tractable* which D&L solves with $O(d\sqrt{T\log T})$ regret. **MOO-specific gains**: In 3-objective settings, BO methods collapse while D&L finds 2–3× more non-dominated solutions. Lagrangian updates matters *more* in MOO: each weight induces different optima at shared positions, amplifying conflict. $\S$ D.6 shows 95% variance reduction & +6.6% reward from resolving these conflicts.
>
> # Q1.
>
> The multi-expert framework requires only no-regret experts ($\S$ 4.1.2). The rationale: competing experts with complementary strengths collectively cover failure modes that any single algorithm cannot & each serves a distinct role:
> 1. Expert 1 (UCB/TS) drives exploration under uncertainty via confidence bounds or posterior sampling;
> 2. Expert 2 (EXP3) handles exploitation and global coordination across subproblems;
> 3. Expert 3 (FTRL) provides variance reduction and trajectory correction via regularized cumulative loss minimization.
>
> Ablations (D.3.1–D.3.3) confirm each role is necessary: removing FTRL causes premature convergence & up to 50% higher variance ($\S$D.2, Fig. 10); removing EXP3 loses robustness (Table 11). No single expert dominates — trio is a recommended default, not rigid.
>
> # Q2.
>
> D\&L applies directly to SOO by fixing the weight vector =(1.0). New ablation on single-objective TSP (average regret, same setup as W1 MOO):
>
> | Components | n=20↓ | n=50↓ | n=100↓ |
> | :--- | :---: | :---: | :---: |
> | Random | 6.86 | 16.38 | 35.03 |
> | + Bandit | 6.26 | 15.51 | 34.09 |
> | + Experts (no-decomp) | 3.94 | 7.69 | 19.58 |
> | **+ D&L-TS** | **3.27** | **5.49** | **11.55** |
> *AvgReg (↓); UCB similar. Mean & SE in* **[Table 1 link](https://anonymous.4open.science/r/ICML2026-Rebuttal-Figures-Tables-D3DF/ICML2026_Divide_n_Learn.pdf)**
>
> Clear hierarchy on TSP-100: pure bandits perform near random, expert mixture halves regret (~19.6), full D\&L reaches 11.55 -- 41% lower than No-Decomp.  Gap grows with $n$ (17%→29%→41%), consistent with O(d√T). Since MOO is strictly harder, requiring adaptation to changing weights & Lagrangian updates, strong SOO performance follows naturally, validating D\&L's architectural gains.
>
> # Q3
>
> Our regret bound is scalarization-agnostic \& holds for any bounded scalar objective. We validate via 4-way ablation: Weighted Sum(WS), Tchebycheff(TCH), Augmented TCH (ATCH), PBI on BiKP (non-convex PF, $n \\in \\{50, 100\\}$):
>
> | D&L-UCB BiKP | WS | TCH | ATCH | PBI |
> | :--- | :---: | :---: | :---: | :---: |
> | **n=50** | **.322** / 42 | .322 / 29 | .322 / 49 | .322 / 50 |
> | **n=100** | **.402** / 158 | .395 / 116 | .398 / 129 | .396 / 70 |
> *HV / \|PF\| (TS within .004). SE ≤ .07 (10 seeds).*
>
> All HVs within 2%, within one SE. $\| PF \|$ variation reflects known scalarization characteristics & PF plots confirm similar geometry in **[(Figure 5,6 in link)](https://www.overleaf.com/project/69275b072f8a67143d61dd37)**. D&L's value is in tractable scalarized subproblems.

---

> > ### Author Rebuttal · Reviewer_HJV4 · 2026-04-03
> >
> > Thanks for clarifying the questions, and my main concern is substantially resolved. However, the method’s practical robustness still depends on the choice of expert composition and mixing ratios, as shown in Table 11, even though the authors provide some useful guidelines. I will therefore keep my current score, which remains on the accept side.

---

> > > ### Author Response · Authors · 2026-04-07
> > >
> > > Thank you for your thoughtful re-evaluation and for confirming that the main concerns are resolved. We appreciate the feedback on expert composition — we will make the default configuration guidelines more prominent in the final version. We are grateful for your time and constructive comments throughout the review process.
> > >
> > > Best regards,
> > >
> > > The Authors

---

### Decision · Program_Chairs · 2026-04-30

**Decision:**

Accept (regular)

**Comment:**

The authors propose a solution strategy to solve large-scale multi-objective optimization problems by decomposing the decision space and reformulation as online-learning problem. The reviewers generally liked the approach and while some weaknesses remain, the strengths outweigh. One of reviewers increased their score, as part of the rebuttal process, so that we have a uniform (4,4,4,4).